# Decentralized Sporadic Federated Learning: A Unified Algorithmic Framework with Convergence Guarantees

**Shahryar Zehtabi**
Purdue University
szehtabi@purdue.edu

**Dong-Jun Han**
Yonsei University
djh@yonsei.ac.kr

**Rohit Parasnis**
MIT
rohit100@mit.edu

**Seyyedali Hosseinalipour**
University at Buffalo (SUNY)
alipour@buffalo.edu

**Christopher G. Brinton**
Purdue University
cgb@purdue.edu

## Abstract

Decentralized federated learning (DFL) captures FL settings where both (i) model updates and (ii) model aggregations are exclusively carried out by the clients without a central server. Existing DFL works have mostly focused on settings where clients conduct a fixed number of local updates between local model exchanges, overlooking heterogeneity and dynamics in communication and computation capabilities. In this work, we propose Decentralized Sporadic Federated Learning (DSpodFL), a DFL methodology built on a generalized notion of *sporadicity* in both local gradient and aggregation processes. DSpodFL subsumes many existing decentralized optimization methods under a unified algorithmic framework by modeling the per-iteration (i) occurrence of gradient descent at each client and (ii) exchange of models between client pairs as arbitrary indicator random variables, thus capturing *heterogeneous and time-varying* computation/communication scenarios. We analytically characterize the convergence behavior of DSpodFL for both convex and non-convex models and for both constant and diminishing learning rates, under mild assumptions on the communication graph connectivity, data heterogeneity across clients, and gradient noises. We show how our bounds recover existing results from decentralized gradient descent as special cases. Experiments demonstrate that DSpodFL consistently achieves improved training speeds compared with baselines under various system settings.

## 1 Introduction

Traditional works in federated learning (FL) have focused on a conventional "star topology" configuration where clients are connected directly to a central server (Konečný et al., 2016; Bonawitz et al., 2019). In this setup (Fig. 1a), FL iterates between (i) client-side local model updates, typically via stochastic gradient descent (SGD) on local datasets, and (ii) server-side model aggregations. However, a central server may not always be present/feasible for synchronization, e.g., in the growing body of direct peer-to-peer networks (Brinton et al., 2024). To address this, recent research has proposed decentralized federated learning (DFL) (Koloskova et al., 2020), replacing the server's role in FL aggregations with distributed optimization techniques (Nedić et al., 2018). This introduces a new challenge in DFL as clients need to reach consensus while optimizing their local models via gradient descent (Figs. 1b-1d). Towards this end, clients exchange models with their neighbors over the decentralized topology to form aggregations through gossip protocols (Huang et al., 2022).

FL settings are often dominated by heterogeneity and dynamics in various dimensions, including client processing capabilities, communication capabilities, and local dataset statistics varying across clients and over time (Li et al., 2020). This causes (i) computing gradients at every iteration to be costlier (e.g., in terms of delay) at clients with weaker/slower processing units, and (ii) higher transmission delays for clients with low-quality communication links (e.g., lower available bandwidth or transmit power), among other impacts (Wang et al., 2021). Existing works in centralized FL have addressed these issues by letting the number of local SGD steps between aggregations vary across

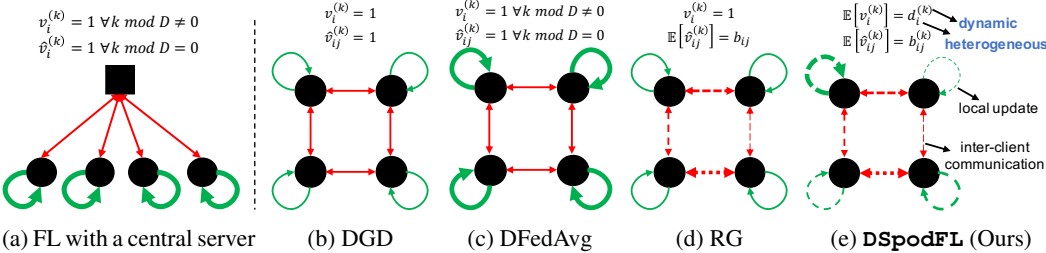

Figure 1: Illustrations of centralized FL (Fig. 1a) and different consensus-based decentralized optimization algorithms (Figs. 1b-1e). In decentralized gradient descent (DGD, Fig. 1b), local updates and inter-client communications occur at every iteration of training. Fig. 1c depicts decentralized local SGD, or DFedAvg, where communications occur only every $D$-th iteration. Communication and computation operations are carried out in a deterministic pattern (solid lines, thickness representing relative frequency) in Figs. 1b and 1c. Randomized gossip (RG, Fig. 1d) adopts sporadic communications for aggregations. DSpodFL in Fig. 1e considers sporadicity in both communications and computations (dashed lines), where the number of local SGDs and the period of model aggregations are heterogeneous across clients and vary over time.

clients (Maranjyan et al., 2022), and also over training rounds (Yang et al., 2022), i.e., to deal with differing and/or varying capabilities while maintaining convergence guarantees.

**Motivation and key challenges.** In the fully decentralized setting, by contrast, there has not yet been a comprehensive study of how different forms of heterogeneity and dynamics jointly impact the FL performance. Integrating these factors into DFL makes the analysis challenging because there are multiple client aggregators without a central coordinator which must reach consensus under the following conditions: (i) the aggregation periods across the system become heterogeneous as they depend on the number of local SGDs conducted by each client's neighbors prior to sharing; and (ii) these periods become time-varying depending on dynamics in communication/computation resource availability of clients/links. Most existing DFL algorithms (Koloskova et al., 2020; Sun et al., 2022; Mishchenko et al., 2022) have not taken these factors into account, resulting in longer times to achieve a target accuracy in the presence of heterogeneous and time-varying resources. We thus aim to answer the following key question in this paper:

*How can we integrate heterogeneity and dynamics of local SGDs and aggregations into decentralized FL to capture the impact of resource availability while maintaining convergence guarantees?*

**Contributions.** We answer this question by developing a generalized algorithmic framework for DFL that allows local model aggregations and transmissions to happen after any arbitrary number of local updates. We refer to this as *sporadicity* in client participation, which enables capturing the impacts of *heterogeneity and dynamics* in DFL. Our methodology, Decentralized Sporadic Federated Learning (DSpodFL), encapsulates the joint effects of (i) sporadicity in local client computations and (ii) sporadicity in inter-client communications arising from resource variations over clients and time. In doing so, DSpodFL captures the impact of different numbers of local SGDs and different aggregation periods across clients, and allows for these values to vary over the training process, not constraining them to any prefixed deterministic pattern. We make the following novel contributions:

- *Sporadic DFL framework capturing resource heterogeneity and dynamics*: We formulate DSpodFL by modeling the occurrences of (i) a local SGD step at each client and (ii) an exchange of models between a pair of clients in each training iteration as arbitrary indicator random variables. This enables clients to conduct these processes intermittently according to their available resources without delaying DFL training. As illustrated in Fig. 1, DSpodFL (Fig. 1e) subsumes multiple decentralized optimization methods from existing research (Fig. 1b-1d), which can be seen as handling special cases of our generalized notion of sporadicity.

- *Convergence analysis under mild assumptions for convex and non-convex settings*: We analytically characterize the convergence behavior of DSpodFL for both strongly-convex and non-convex loss functions, under mild assumptions on the network graph, data heterogeneity, and gradient noises. We conduct our analysis for both constant (Thms. 4.11, 4.12) and diminishing learning rates (App. I, G), revealing conditions under which zero optimality/stationarity gap can be achieved. The introduction of sporadicity to DFL makes the analysis challenging, since both local SGDs and model aggregations occur without any predetermined pattern. We show how our results recover the convergence rates of existing DFL algorithms under special cases of sporadicity.

| Paper | Properties of Algorithmic Framework | | | | Assumptions and Theoretical Results | | | |
|---|---|---|---|---|---|---|---|---|
| | Fully Decen. | Sporadic SGDs | Sporadic Aggr. | Dynamic Resource Het. | General. Data Het.[a] | Loose Graph Conn.[b] | Last Iterates Conv.[c] | Convex & Non-Convex Analysis |
| Koloskova et al. (2020) | ✓ | | ✓ | | ✓ | ✓ | | ✓ |
| Maranjyan et al. (2022) | | ✓ | | | | | ✓ | |
| Yang et al. (2022) | | ✓ | ✓ | ✓ | | | | |
| Sun et al. (2022) Mishchenko et al. (2022) | ✓ | | | | | | ✓ | |
| **Ours** | ✓ | ✓ | ✓ | ✓ | ✓ | ✓ | ✓ | ✓ |

[a]In Assumptions 4.1-(c) and 4.2-(b), we consider a more general/milder data heterogeneity assumption based on two parameters $\delta$ and $\zeta$. These assumptions are not restricting the gradient norms to a constant bound.
[b]In Assumption 4.4, we neither require the underlying network graph to be static, nor $B$-connected.
[c]Discussed in Sec. 4.5, and Appendix F.4(proof of Theorem 4.11 on convergence of convex models).

Table 1: Summary of eight key properties of our paper compared to representative related works.

- *Experiments in heterogeneous and time-varying DFL settings*: Our numerical experiments demonstrate that `DSpodFL` reaches target accuracies with significantly smaller delays compared to DFL baselines. Further, we find that `DSpodFL` consistently outperforms the baselines as the degrees of data heterogeneity, resource heterogeneity and dynamics, and the network properties vary.

## 2 RELATED WORKS

Table 1 summarizes key contributions of our work relative to closely related literature in centralized and decentralized FL. To the best of our knowledge, our work is the first to consider sporadic SGDs and aggregations simultaneously, capturing heterogeneous and time-varying resources in the fully decentralized setting. Below, we discuss related works along DFL's two key processes.

**Local SGDs.** Several works in centralized FL (Li et al., 2019; Lin et al., 2019; Karimireddy et al., 2020; Woodworth et al., 2020; Mishchenko et al., 2022) proposed algorithms with multiple local updates between consecutive model aggregations, assuming fixed number of local SGDs across clients. In Maranjyan et al. (2022), an FL method is proposed where at each round of training, the number of SGD steps differs for each client considering resource heterogeneity. However, the varying number of local SGD steps across clients remains fixed throughout the training process. To alleviate this issue, Anarchic FL was proposed Yang et al. (2022); similar to our notion of sporadicity, each client chooses when to conduct computations/communications freely on its own throughout the training process, generalizing all prior work discussed above. Compared to these works in centralized FL, we focus on sporadicity in the decentralized setting. This introduces new challenges to our analysis, including dealing with multiple client aggregators and the consensus process among the clients.

A few recent works have also considered decentralized counterparts of fixed local SGD methods (Wang & Joshi, 2018; Sun et al., 2022; Nguyen et al., 2023; Liu et al., 2024), without considering the heterogeneity and dynamics of client resources. Compared to these works, our focus is on the sporadic case in DFL, modeling heterogeneous number of local SGD steps for different clients and allowing them to be time-varying as well. As a result, `DSpodFL` subsumes prior methods in decentralized fixed local SGD as a special case. We will further show in Sec. 4 how our convergence results (Thms. 4.11, 4.12) recover DGD-like methods when there is no sporadicity in SGDs. Finally, we note that our contribution takes the consideration of sporadic SGDs a step further, by analyzing the joint effects of sporadic SGDs and sporadic aggregations in DFL.

**Consensus strategies.** Sporadicity in communications for distributed consensus formation has been studied in randomized gossip (RG) algorithms. Several works Boyd et al. (2006); Even et al. (2021); Pu & Nedić (2021) study gossip algorithms with two clients conducting consensus at each iteration, while (Koloskova et al., 2019; Kong et al., 2021; Chen et al., 2021; Zhu et al., 2022) allow more general mixing matrices. Saha et al. (2024) further deals with privacy constraints while implementing gossip communications. In another direction, Srivastava & Nedic (2011); Lian et al. (2018); Bornstein et al. (2022) have studied asynchronous DFL, where the communication delay between inter-client model exchanges can be modeled as sporadic aggregations. The authors of Koloskova et al. (2020) unify several existing DGD algorithms, under similar generalized data heterogeneity and graph connectivity assumptions that we consider in our analysis. However, none of these works in sporadic aggregations have considered sporadic SGDs in their methodology, making our work among the first to analyze the efficacy of the joint consideration of sporadic SGDs and aggregations. A contemporary of our work Even et al. (2024) has also unified several DFL algorithms, but

for the asynchronous setting without consideration for time variations in resource heterogeneity. In this respect, compared to all prior works, our modeling using general indicator random variables (Sec. 3.1) enables us to jointly incorporate *heterogeneous* and *time-varying* resource availability of clients. This approach leads to more generalized results accompanied by new challenges in studying convergence, making both our algorithmic framework and analysis unique.

# 3 DECENTRALIZED SPORADIC FEDERATED LEARNING

In this section, we formalize our `DSpodFL` algorithmic framework and the notion of sporadicity. A summary of notation used throughout this paper can be found in Appendix A.

## 3.1 `DSpodFL`: DECENTRALIZED FL WITH SPORADICITY

We consider a DFL system with $m$ clients $\mathcal{M} := \{1, \ldots, m\}$ and a series of training iterations $k = 1, ..., K$. The clients are connected through a time-varying communication graph $\mathcal{G}^{(k)} = (\mathcal{M}, \mathcal{E}^{(k)})$, where $(i, j) \in \mathcal{E}^{(k)}$ if clients $i$ and $j$ can directly communicate at iteration $k$. Define $\mathcal{G} = (\mathcal{M}, \mathcal{E})$ as a graph such that $\mathcal{E}^{(k)} \subset \mathcal{E}$ for all $k \geq 0$. The goal of DFL is for clients to discover the globally optimal model $\theta^\star = \arg\min_{\theta \in \mathbb{R}^n} F(\theta)$ for a given global loss function $F(\theta)$. To do so, each client $i \in \mathcal{M}$ updates its own version $\theta_i$ of the model by (i) conducting SGDs on its local loss $F_i(\theta)$ and (ii) mixing its model with those received from its neighbors. We have

$$F(\theta) = \frac{1}{m} \sum_{i \in \mathcal{M}} F_i(\theta), \quad F_i(\theta) = \sum_{(\mathbf{x}, y) \in \mathcal{D}_i} \ell_{(\mathbf{x}, y)}(\theta), \tag{1}$$

in which $\mathcal{D}_i$ is the local dataset of client $i \in \mathcal{M}$, $(\mathbf{x}, y)$ denotes a data point with features $\mathbf{x}$ and label $y$, and $\ell_{(\mathbf{x}, y)}(\theta)$ is the loss incurred by ML model $\theta$ on a data point $(\mathbf{x}, y)$.

**Goal and motivation.** Eq. 1 is the conventional objective function of FL, which we consider optimizing for the decentralized setting under sporadicity. Specifically, we aim for each client to arrive at $\theta_1 = \cdots = \theta_m = \theta^\star$. This means that the clients need to reach consensus through the local exchange process alongside implementing local SGD (Nedic, 2020). Due to heterogeneity and time variance in communication/computation resources, we allow for autonomy in the number of SGDs conducted and in the periods between model sharing across client pairs. This will make the process particularly challenging to analyze too.

**Algorithmic framework.** `DSpodFL` achieves the above goal by modeling *sporadicity* in client participation for DFL, decoupling the number of SGD iterations conducted by a client from the time between model exchanges with its neighbors and the resulting consensus mixing process. Specifically, at each iteration $k$, client $i$'s update is modeled in the following generalized manner:

$$\theta_i^{(k+1)} = \theta_i^{(k)} + \underbrace{\sum_{j \in \mathcal{M}} r_{ij} \left( \theta_j^{(k)} - \theta_i^{(k)} \right) \hat{v}_{ij}^{(k)}}_{\text{Sporadic aggregation}} - \underbrace{\alpha^{(k)} \mathbf{g}_i^{(k)} v_i^{(k)}}_{\text{Sporadic SGD}}, \tag{2}$$

where $\theta_i^{(k)}$ is the vector of model parameters of client $i$ at iteration $k$, and $\mathbf{g}_i^{(k)} = \nabla F_i(\theta_i^{(k)}) + \epsilon_i^{(k)}$ is the local stochastic gradient with SGD noise $\epsilon_i^{(k)}$. In Eq. 2, $v_i^{(k)} \in \{0, 1\}$ is a random indicator variable, capturing the sporadicity in SGD iterations, which is 1 if the client performs SGD in that iteration. Similarly, $\hat{v}_{ij}^{(k)} \in \{0, 1\}$ is a binary random variable capturing the sporadicity in model aggregations, which indicates whether the link $(i, j)$ is being used for communications at iteration $k$ or not ($\hat{v}_{ij}^{(k)} = \hat{v}_{ji}^{(k)}$ and $\hat{v}_{ii}^{(k)} = 0$). $r_{i,j} \in [0, 1]$ is the mixing weight assigned to link $(i, j)$, for which the only requirement is that the mixing matrix $\mathbf{R} = [r_{ij}]_{1 \leq i, j \leq m}$ is doubly stochastic. For example, with the Metropolis-Hastings heuristic (Boyd et al., 2004), $r_{ij} = 1/(1 + \max\{|\mathcal{N}_i|, |\mathcal{N}_j|\})$ when $j \in \mathcal{N}_i$, and 0 if $j \notin \mathcal{N}_i$, in which $\mathcal{N}_i$ is the set of neighbors of client $i$ in the physical graph $\mathcal{G} = (\mathcal{M}, \mathcal{E})$. Setting $r_{ii} = 1 - \sum_{j \in \mathcal{M}} r_{ij}$ and noting that $r_{ij} = r_{ji}$, i.e., $\mathbf{R}$ being symmetric, completes the design of a doubly stochastic mixing matrix $\mathbf{R}$. The full pseudocode of `DSpodFL` implementing Metropolis-Hastings mixing weights is given in Appendix B.

The update rule of Eq. 2 with two different *sporadicity terms* $v_i^{(k)}$ and $\hat{v}_{ij}^{(k)}$, each capturing both heterogeneous and time-varying characteristics, has not been considered in the DFL literature.

### 3.2 KEY TAKEAWAYS FROM DSPODFL

**Interpreting sporadicity.** The novelty of DSpodFL in the integration of the two sporadicity terms (i.e, $v_i^{(k)}$ and $\hat{v}_{ij}^{(k)}$) to model the impacts of resource heterogeneity and dynamics in DFL. Specifically, client $i$ may set $v_i^{(k)} = 0$ for computation efficiency in iterations where computing a new SGD is not feasible or does not significantly benefit the *statistical/inference* performance of the decentralized system. Similarly, a pair of clients can set $\hat{v}_{ij}^{(k)} = 0$ for communication efficiency when using link $(i, j)$ is too costly at iteration $k$ relative to local resource availability and/or expected performance impact. These two key parameters $v_i^{(k)}$ and $\hat{v}_{ij}^{(k)}$ can vary arbitrarily over the training process according to decisions made by clients independently over time. This incorporation of sporadicity increases the degrees of freedom DSpodFL accounts for, thereby distinguishing it from the literature in Sec. 2 (See Appendix P.2 for further remarks).

**Unifying existing work.** By introducing these two sporadicity terms, DSpodFL subsumes other decentralized learning algorithms, including those shown in Fig. 1. Specifically, DSpodFL reduces to DGD (Fig. 1b), DFedAvg (Fig. 1c), and RG (Fig. 1d) for specific configurations of $v_i^{(k)}$ and $\hat{v}_{ij}^{(k)}$. However, these two sporadicity terms introduce several novel challenges when analyzing convergence due to their creation of uncorrelated aggregation periods in DFL, which we address in Sec. 4.

### 3.3 MATRIX FORM OF UPDATES IN DSPODFL

To facilite our convergence analysis, we can rewrite the update rule given in Eq. 2 compactly as

$$\mathbf{\Theta}^{(k+1)} = \mathbf{P}^{(k)}\mathbf{\Theta}^{(k)} - \alpha^{(k)}\mathbf{V}^{(k)}\mathbf{G}^{(k)}, \tag{3}$$

where $\mathbf{\Theta}^{(k)}$ and $\mathbf{G}^{(k)}$ are matrices with their rows comprised of $(\theta_i^{(k)})^T$ and $(\mathbf{g}_i^{(k)})^T$, respectively, and $\mathbf{V}^{(k)}$ is a diagonal matrix with $v_i^{(k)}$ as its diagonal entries, for clients $1 \leq i \leq m$. Here, $\mathbf{G}^{(k)} = \nabla^{(k)} + \mathbf{E}^{(k)}$, where $\nabla^{(k)}$ (respectively, $\mathbf{E}^{(k)}$) is the matrix whose $i$-th row is $(\nabla F_i(\theta_i^{(k)}))^T$ (respectively, $(\epsilon_i^{(k)})^T$) for $1 \leq i \leq m$. The elements of $\mathbf{P}^{(k)} = [p_{ij}^{(k)}]_{1 \leq i,j \leq m}$ are defined as

$$p_{ij}^{(k)} = r_{ij}\hat{v}_{ij}^{(k)}; \quad i \neq j, \qquad p_{ij}^{(k)} = 1 - \sum_{j \in \mathcal{M}} r_{ij}\hat{v}_{ij}^{(k)}; \quad i = j. \tag{4}$$

Note that the random matrix $\mathbf{P}^{(k)}$, by definition, is *doubly stochastic and symmetric* with non-negative entries, i.e., $\mathbf{P}^{(k)}\mathbf{1} = \mathbf{1}$ and $(\mathbf{P}^{(k)})^T = \mathbf{P}^{(k)}$. Finally, in our analysis, we will find it useful to define a row vector $\bar{\theta}^{(k)}$ as the average of model vectors $\theta_1^{(k)}, ..., \theta_m^{(k)}$ across clients. Using Eq. 3,

$$\bar{\theta}^{(k+1)} = \bar{\theta}^{(k)} - \alpha^{(k)}\overline{\mathbf{g}v}^{(k)}, \tag{5}$$

where $(\overline{\mathbf{g}v}^{(k)})^T = (1/m)\sum_{i \in \mathcal{M}} \mathbf{g}_i^{(k)}v_i^{(k)}$.

## 4 CONVERGENCE ANALYSIS

### 4.1 DEFINITIONS AND ASSUMPTIONS

**Assumption 4.1 (Convex loss functions)** *For analysis in the strongly convex case, we assume the local loss function $F_i$ at each client $i \in \mathcal{M}$ is (a) $\beta_i$-smooth and (b) $\mu_i$-strongly convex. Also, (c) the gradient diversity is measured via $\delta_i > 0$ and $\zeta_i \geq 0$ as $\|\nabla F(\theta) - \nabla F_i(\theta)\| \leq \delta_i + \zeta_i \|\theta - \theta^\star\|$, for all $\theta \in \mathbb{R}^n$. We define $\beta = \max_{i \in \mathcal{M}} \beta_i$, $\mu = \min_{i \in \mathcal{M}} \mu_i$, $\delta = \max_{i \in \mathcal{M}} \delta_i$ and $\zeta = \max_{i \in \mathcal{M}} \zeta_i$.*

For the non-convex analysis in Sec. 4.5, we will replace Assumptions 4.1-(b)&(c) with the following:

**Assumption 4.2 (Non-convex loss functions)** *The local loss function $F_i$ at each client $i \in \mathcal{M}$ is (a) $\beta_i$-smooth. Also, (b) the gradient diversity across clients is measured via $\delta_i > 0$ and $\zeta_i \geq 0$ as $\|\nabla F_i(\theta)\| \leq \delta_i + \zeta_i \|\nabla F(\theta)\|$, for all $\theta \in \mathbb{R}^n$, $i \in \mathcal{M}$. We let $\beta = \max_{i \in \mathcal{M}} \beta_i$, $\delta = \max_{i \in \mathcal{M}} \delta_i$ and $\zeta = \max_{i \in \mathcal{M}} \zeta_i$.*

**Assumption 4.3 (Random variables)** *For all $i \in \mathcal{M}$ and all $k \geq 0$, (a) The gradient noise $\epsilon_i^{(k)}$ of each client is zero mean with bounded variance $\sigma_i^2$. We also let $\sigma^2 = \max_{i \in \mathcal{M}} \sigma_i^2$. (b) Gradient noise vectors $\epsilon_i^{(k)}$ are uncorrelated across the clients, as are the indicator variables $v_i^{(k)}$. The indicator variables $\hat{v}_{ij}^{(k)}$ are also uncorrelated among the network links. (c) Random variables $\epsilon_i^{(k)}$ and $v_i^{(k)}$ are uncorrelated for all clients .*

**Assumption 4.4 (Asymptotic graph connectivity)** *Denote the asymptotic graph union of underlying time-varying communication network graphs by $\mathcal{G} = \left(\mathcal{M}, \lim_{K\to\infty} \cup_{k=0}^{K} \mathcal{E}^{(k)}\right)$. We assume that $\mathcal{G}$ is connected, and for every edge $(i,j) \in \mathcal{G}$, $(i,j) \in \mathcal{E}^{(k)}$ for infinitely many iterations $k$.*

More detailed mathematical expositions of our assumptions are provided in Appendix C. In Assumptions 4.1-(c)&4.2-(b), we do not make the stricter assumption of $\zeta = 0$ found in some works on DFL (Sun et al., 2022; Mishchenko et al., 2022). The addition of this proximal term makes our analytical bounds tighter as they only require a constant bound at optimal/stationary points (Lin et al., 2021). Assumption 4.4 is milder than similar assumptions made in prior works, e.g., static connected graphs (Sun et al., 2022; Mishchenko et al., 2022; Wang & Nedić, 2023) or $B$-connected graphs (Nedic & Ozdaglar, 2009). Table 1 summarizes this comparison along these and other dimensions.

**Definition 4.5** *We define $\boldsymbol{\Xi}^{(k)}$ as the collection of random variables $v_i^{(r)}$, $\hat{v}_{ij}^{(r)}$ and $\epsilon_i^{(r)}$ for all $i \in \mathcal{M}$, $(i,j) \in \mathcal{E}^{(r)}$, and $0 \leq r \leq k$. With this, the expected consensus rate (Koloskova et al., 2020) can be characterized via $\tilde{\rho}^{(k)}$, the spectral radius of the expected mixing matrix, as $\mathbb{E}_{\boldsymbol{\Xi}^{(k)}}[\|\mathbf{P}^{(k)}\boldsymbol{\Theta}^{(k)} - \mathbf{1}_m\bar{\theta}^{(k)}\|^2] \leq \tilde{\rho}^{(k)}\mathbb{E}_{\boldsymbol{\Xi}^{(k-1)}}[\|\boldsymbol{\Theta}^{(k)} - \mathbf{1}_m\bar{\theta}^{(k)}\|^2]$, which we present/prove in Lemma D.4-(c) and Appendix E.4-(c), respectively.[1]*

**Definition 4.6 (Indicator variables)** *The expected values of variables $v_i^{(k)}$ and $\hat{v}_{ij}^{(k)}$ are defined as*

$$\mathbb{E}_{v_i^{(k)}}\left[v_i^{(k)}\right] = d_i^{(k)}, \qquad \mathbb{E}_{\hat{v}_{ij}^{(k)}}\left[\hat{v}_{ij}^{(k)}\right] = \mathbb{E}_{\hat{v}_{ji}^{(k)}}\left[\hat{v}_{ji}^{(k)}\right] = b_{ij}^{(k)} = b_{ji}^{(k)},[2]$$

*in which $d_i^{(k)} \in (0,1]$ captures client $i$'s probability of conducting SGD, and $b_{ij}^{(k)} \in (0,1]$ captures the probability of link $(i,j)$ being used for communication, at iteration $k$. We also define $d_{\max}^{(k)} = \max_{i\in\mathcal{M}} d_i^{(k)}$ and $d_{\min}^{(k)} = \min_{i\in\mathcal{M}} d_i^{(k)}$. Note that the probability distributions of these indicator variables can be time-varying, allowing for an arbitrary range of profiles for $v_i^{(k)}$ and $\hat{v}_{ij}^{(k)}$.*

## 4.2 AVERAGE MODEL ERROR AND CONSENSUS ERROR

To characterize the convergence behavior of DSpodFL, we first provide an upper bound on the average model error $\mathbb{E}_{\boldsymbol{\Xi}^{(k)}}[\|\bar{\theta}^{(k+1)} - \theta^\star\|^2]$ (Lemma 4.7), and also upper bound the consensus error $\mathbb{E}_{\boldsymbol{\Xi}^{(k)}}[\|\boldsymbol{\Theta}^{(k+1)} - \mathbf{1}_m\bar{\theta}^{(k+1)}\|^2]$ (Lemma 4.8), at each iteration $k$.

**Lemma 4.7 (Average model error)** *(See Appendix F.1 for the proof.) Let Assumptions 4.1 and 4.3 hold. For each iteration $k \geq 0$, we have the following bound on the expected average model error:*

$$\mathbb{E}_{\boldsymbol{\Xi}^{(k)}}[\|\bar{\theta}^{(k+1)} - \theta^\star\|^2] \leq \phi_{11}^{(k)}\mathbb{E}_{\boldsymbol{\Xi}^{(k-1)}}[\|\bar{\theta}^{(k)} - \theta^\star\|^2] + \phi_{12}^{(k)}\mathbb{E}_{\boldsymbol{\Xi}^{(k-1)}}[\|\boldsymbol{\Theta}^{(k)} - \mathbf{1}_m\bar{\theta}^{(k)}\|^2] + \psi_1^{(k)},$$

*where $\phi_{11}^{(k)} = 1 - \mu\alpha^{(k)}(1 + \mu\alpha^{(k)} - (\mu\alpha^{(k)})^2) + \frac{2\alpha^{(k)}}{\mu}(1 + \mu\alpha^{(k)})(1 - d_{\min}^{(k)})\beta^2$,*
*$\phi_{12}^{(k)} = (1 + \mu\alpha^{(k)})\frac{\alpha^{(k)}d_{\max}^{(k)}\beta^2}{m\mu}$, and $\psi_1^{(k)} = \frac{2\alpha^{(k)}}{\mu}(1 + \mu\alpha^{(k)})(1 - d_{\min}^{(k)})\delta^2 + \frac{(\alpha^{(k)})^2 d_{\max}^{(k)}\sigma^2}{m}$.*

In Lemma 4.7, the upper bound on the expected error at iteration $k + 1$ is expressed in terms of the scaled expected error $\phi_{11}^{(k)}\mathbb{E}_{\boldsymbol{\Xi}^{(k-1)}}[\|\bar{\theta}^{(k)} - \theta^\star\|^2]$, the scaled consensus error $\phi_{12}^{(k)}\mathbb{E}_{\boldsymbol{\Xi}^{(k-1)}}[\|\boldsymbol{\Theta}^{(k)} - \mathbf{1}_m\bar{\theta}^{(k)}\|^2]$ (which will be analyzed in Lemma 4.8), and the scalar $\psi_1^{(k)}$, all at iteration $k$. We can see that this bound captures the impact of sporadicity in local SGDs at devices, through $d_{\min}^{(k)}$ and $d_{\max}^{(k)}$. It recovers the bound for DGD when $d_{\min}^{(k)} = 1$, i.e., making $v_i^{(k)} = 1$ for all $i \in \mathcal{M}$ (e.g., see Lemma 5-b of Zehtabi et al. (2022)).

**Lemma 4.8 (Consensus error)** *(See Appendix F.2 for the proof.) Let Assumptions 4.1, 4.3 and 4.4 hold. For each iteration $k \geq 0$, we have the following bound on the expected consensus error:*

$$\mathbb{E}_{\boldsymbol{\Xi}^{(k)}}[\|\boldsymbol{\Theta}^{(k+1)} - \mathbf{1}_m\bar{\theta}^{(k+1)}\|^2] \leq \phi_{21}^{(k)}\mathbb{E}_{\boldsymbol{\Xi}^{(k-1)}}[\|\bar{\theta}^{(k)} - \theta^\star\|^2] + \phi_{22}^{(k)}\mathbb{E}_{\boldsymbol{\Xi}^{(k-1)}}[\|\boldsymbol{\Theta}^{(k)} - \mathbf{1}_m\bar{\theta}^{(k)}\|^2] + \psi_2^{(k)},$$

---

[1] Our notation of $\mathbb{E}_{\boldsymbol{\Xi}^{(k)}}[\circ]$ denotes the full expectation operator with respect to the random variables in matrix $\boldsymbol{\Xi}^{(k)}$, i.e., $\mathbb{E}_{\boldsymbol{\Xi}^{(k-1)}}[\mathbb{E}_{\xi^{(k)}}[\circ \mid \boldsymbol{\Xi}^{(k-1)}]] = \mathbb{E}_{\xi^{(0)}}[\mathbb{E}_{\xi^{(1)}}[\cdots\mathbb{E}_{\xi^{(k-1)}}[\mathbb{E}_{\xi^{(k)}}[\circ \mid \boldsymbol{\Xi}^{(k-1)}]\mid\boldsymbol{\Xi}^{(k-2)}]\cdots\mid\xi^{(0)}]]$.

[2] Note that $v_i^{(k)}$ and $\hat{v}_{ij}^{(k)}$ are thus Bernoulli random variables for each $k$. However, the expectation changes over time, i.e., varying $d_i^{(k)}$ and $b_{ij}^{(k)}$, making $v_i^{(k)}$ and $\hat{v}_{ij}^{(k)}$ dynamic Bernoulli variables.

*where* $\phi_{21}^{(k)} = 3\frac{1+\tilde{\rho}^{(k)}}{1-\tilde{\rho}^{(k)}}md_{\max}^{(k)}(\alpha^{(k)})^2(\zeta^2+2\beta^2(1-d_{\min}^{(k)}))$, $\phi_{22}^{(k)} = \frac{1+\tilde{\rho}^{(k)}}{2}+3\frac{1+\tilde{\rho}^{(k)}}{1-\tilde{\rho}^{(k)}}d_{\max}^{(k)}(\alpha^{(k)})^2(\zeta^2+2\beta^2)$, $\psi_2^{(k)} = m(\alpha^{(k)})^2 d_{\max}^{(k)}(3\frac{1+\tilde{\rho}^{(k)}}{1-\tilde{\rho}^{(k)}}\delta^2 + \sigma^2)$, *and* $\tilde{\rho}^{(k)}$ *is defined in Definition 4.5.*

Lemma 4.8 captures the impact of sporadicity in model exchanges, through $\tilde{\rho}^{(k)}$ (which is smaller when the mixing matrix is better connected), and in SGDs. It also shows how gradient diversity $(\zeta, \delta)$ makes the bound larger. It reduces to a bound for DGD when (a) $\zeta = 0$ and (b) $d_{\min}^{(k)} = 1$ (i.e., $d_i^{(k)} = 1$ for all $i$), in turn forcing $\phi_{21}^{(k)} = 0$ (e.g., see Lemma 5-c in Zehtabi et al. (2022)).

## 4.3 SUFFICIENT CONDITION FOR CONVERGENCE

We observe that the the average model error and consensus error from Lemmas 4.7 and 4.8 are coupled. We next characterize their joint evolution and provide a condition for `DSpodFL` convergence.

**Definition 4.9 (Error vector)** *We denote the error vector at iteration $k$ as $\nu^{(k)}$, defined as the concatenation of the average model error and the consensus error:*

$$\nu^{(k)} = \left[\mathbb{E}_{\mathbf{\Xi}^{(k-1)}}\left[\left\|\bar{\theta}^{(k)} - \theta^\star\right\|^2\right], \ \mathbb{E}_{\mathbf{\Xi}^{(k-1)}}\left[\left\|\mathbf{\Theta}^{(k)} - \mathbf{1}_m\bar{\theta}^{(k)}\right\|^2\right]\right]^T. \tag{6}$$

Using this definition, it follows that

$$\nu^{(k+1)} \leq \mathbf{\Phi}^{(k)}\nu^{(k)} + \mathbf{\Psi}^{(k)}, \tag{7}$$

with $\mathbf{\Phi}^{(k)} = [\phi_{ij}^{(k)}]_{1\leq i,j\leq 2}$ and $\mathbf{\Psi}^{(k)} = [\psi_1^{(k)}, \ \psi_2^{(k)}]^T$. Recursively expanding the inequalities in Eq. 7 gives us an explicit relationship between the expected model error and consensus error at each iteration, and their initial values, which will be useful in Theorem 4.11:

$$\nu^{(k+1)} \leq \mathbf{\Phi}^{(k:0)}\nu^{(0)} + \sum_{r=1}^{k}\mathbf{\Phi}^{(k:r)}\mathbf{\Psi}^{(r-1)} + \mathbf{\Psi}^{(k)}, \tag{8}$$

where we have defined $\mathbf{\Phi}^{(k:s)} = \mathbf{\Phi}^{(k)}\mathbf{\Phi}^{(k-1)}\cdots\mathbf{\Phi}^{(s)}$ for $k > s$, and $\mathbf{\Phi}^{(k:k)} = \mathbf{\Phi}^{(k)}$. Note that $\nu^{(0)} = [\|\bar{\theta}^{(0)} - \theta^\star\|^2, \|\mathbf{\Theta}^{(0)} - \mathbf{1}_m\bar{\theta}^{(0)}\|^2]^T$.

From Eqs. 7 and 8, we can apply linear system theory to identify a sufficient condition for convergence of `DSpodFL`: that the spectral radius of matrix $\mathbf{\Phi}^{(k)}$ is less than one, i.e., $\rho(\mathbf{\Phi}^{(k)}) < 1$. In the following proposition, we show this can be enforced through appropriate choice of learning rate.

**Proposition 4.10 (Spectral radius)** *(See Appendix F.3 for the proof.) Let Assumptions 4.1, 4.3 and 4.4 hold. If the learning rate satisfies the following condition for all $k \geq 0$:*

$$\alpha^{(k)} < \min\left\{\frac{1}{\mu}, \frac{1}{2\sqrt{3d_{\max}^{(k)}}}\frac{1-\tilde{\rho}^{(k)}}{\sqrt{1+\tilde{\rho}^{(k)}}}\frac{1}{\sqrt{\zeta^2+2\beta^2}}, \left(\frac{\mu}{12\left(\zeta^2+2\beta^2\left(1-d_{\min}^{(k)}\right)\right)}\right)^{1/3}\left(\frac{1-\tilde{\rho}^{(k)}}{2d_{\max}^{(k)}\beta}\right)^{2/3}\right\},$$

*then we have $\rho(\mathbf{\Phi}^{(k)}) < 1$ for all $k \geq 0$, in which $\rho(\cdot)$ denotes the spectral radius of a given matrix, and $\mathbf{\Phi}^{(k)}$ is given in Eq. 7. The exact value of $\rho(\mathbf{\Phi}^{(k)})$ is given in Appendix F.3.*

Proposition 4.10 implies that $\lim_{k\to\infty}\mathbf{\Phi}^{(k:0)} = 0$ in Eq. 8, which means the consensus and average model errors will converge. The exact convergence rate depends on the choice of learning rate $\alpha^{(k)}$.

## 4.4 MAIN THEOREM AND DISCUSSIONS FOR CONVEX CASE

We characterize the convergence bound of `DSpodFL` for the convex case in the following theorem.

**Theorem 4.11 (Strongly-convex convergence result)** *(See Appendix F.4 for the proof.) Let Assumptions 4.1, 4.3 and 4.4 hold, and suppose a constant step size $\alpha^{(k)} = \alpha > 0$ satisfying the conditions outlined in Proposition 4.10 is employed. Let $\tilde{\rho} = \max_{0\leq k\leq K}\tilde{\rho}^{(k)}$ be the maximum expected spectral radius of the mixing probabilities from Definition 4.5 and $d_{\min} = \min_{0\leq k\leq K, i\in\mathcal{M}} d_i^{(k)}$ be the minimum of the SGD probabilities. Then, we can rewrite Eq. 8 as*

$$\nu^{(K+1)} \leq \rho(\mathbf{\Phi})^{K+1}\nu^{(0)} + \frac{1}{1-\rho(\mathbf{\Phi})}\mathbf{\Psi}, \tag{9}$$

*in which* $\mathbf{\Psi} = \mathbf{\Psi}^{(k)}$ *and* $\mathbf{\Phi} = \mathbf{\Phi}^{(k)}$ *from Eq. 7 for all* $k$ *(given the bounds* $\tilde{\rho}^{(k)} \leq \tilde{\rho}$, $d_{\max}^{(k)} \leq 1$ *and* $d_{\min}^{(k)} \geq d_{\min}$*), and* $\nu^{(k)}$ *is the error vector. This means for large enough* $K$, *we have*

$$\lim_{K \to \infty} \nu^{(K+1)} \leq \frac{1}{2A} \begin{bmatrix} \frac{2}{\mu}\left(1 + \mu\alpha\right)\left(1 - d_{\min}\right)\delta^2 + \frac{\alpha\sigma^2}{m} \\ m\alpha\left(3\frac{1+\tilde{\rho}}{1-\tilde{\rho}}\delta^2 + \sigma^2\right) \end{bmatrix}, \tag{10}$$

*where* $A = \frac{\beta^2}{\mu}(\Gamma_2^\star - 1)(1 - \frac{1}{\Gamma_0})$ *with* $\Gamma_0, \Gamma_2^\star > 1$ *being constant scalars defined in Appendix F.3. Note that Proposition 4.10 ensures* $\rho(\mathbf{\Phi}) < 1$.

**Discussion on convergence.** The bound in Eq. 9 indicates that by using a constant step size, `DSpodFL` obtains a geometric convergence rate (i.e., $\mathcal{O}(\rho(\mathbf{\Phi})^K)$) as in other DFL methods (Mishchenko et al., 2022; Nedic & Ozdaglar, 2009), Eq. 10 characterizes the asymptotic optimality gap as $K \to \infty$. We observe that this bound holds for any choice of time-varying SGD and aggregation probabilities, $d_i^{(k)}$ and $b_{ij}^{(k)}$ through $\tilde{\rho}^{(k)}$ (see Appendix P.1), respectively. Additionally, the optimality gap is reduced by (i) choosing a smaller learning rate $\alpha$ and (ii) having a system with a larger minimum SGD probability $d_{\min}$. As discussed after Lemma 4.7, in the conventional DFL setting where SGDs occur at every iteration, we have $d_{\min} = 1$. In this setting, the optimality gap in Eq. 10 would be proportional to $\alpha$, showing that `DSpodFL` recovers well-known results of DGD-like algorithms (Maranjyan et al., 2022) in this special case of no sporadicity in local updates.

**Diminishing learning rate.** When diminishing step size $\alpha^{(k)}$ is used, `DSpodFL` achieves zero optimality gap with a sub-linear convergence rate $\mathcal{O}(\ln K/\sqrt{K})$, matching the rate achieved by existing DGD-based methods (Nedić & Olshevsky, 2014). See Appendix I for proofs and discussion.

**Effects of sporadicity terms.** Both $\rho(\mathbf{\Phi})$ (from Proposition 4.10) and the first term in the optimality gap vector given in Eq. 10 can be made smaller by choosing a larger $d_{min} = \min_{0 \leq k \leq K, i \in \mathcal{M}} d_i^{(k)}$, which will result in a faster convergence rate and a lower average model error. Also, the communication probabilities $b_{ij}^{(k)}$ affect the bounds through the spectral radius parameter $\tilde{\rho}$ (exact relationship given in Appendix E.4). To elaborate, increasing the frequency of communications will lower the spectral radius term $\tilde{\rho}$, leading to faster convergence and a smaller gap for the consensus error. However, this is not always desirable since choosing $d_i$ and $b_{ij}$ solely based on the convergence rate can result in longer iteration lengths, due to resource-limited clients. As we will see in Sec. 5, choosing $d_i$ and $b_{ij}$ considering resource availability leads to speedup in achieving performance targets.

## 4.5 Analysis for Non-Convex Case

Our discussion so far has revolved around strongly convex loss functions. For the non-convex case, we follow a similar roadmap for our analysis as in Sec. 4.2-4.4. The detailed supporting lemmas and propositions can be found in Appendix G. Ultimately, we arrive at the following theorem:

**Theorem 4.12 (Non-convex convergence result)** *(See Appendix H.7 for the proof.) Let Assumptions 4.2, 4.3 and 4.4 hold, and suppose a constant learning rate* $\alpha^{(k)} = \alpha$ *with* $\alpha > 0$ *satisfying the constraints given in Proposition G.6 is employed. Let* $\tilde{\rho} = \max_{0 \leq k \leq K} \tilde{\rho}^{(k)}$ *for the spectral radius and* $d_{\min} = \min_{0 \leq k \leq K, i \in \mathcal{M}} d_i^{(k)}$ *for the local SGDs. Then,*

$$\frac{\sum_{r=0}^K \mathbb{E}_{\mathbf{\Xi}^{(r-1)}}[\|\nabla F(\bar{\theta}^{(r)})\|^2]}{K+1} \leq \frac{1}{w_1}\left(\frac{F(\bar{\theta}^{(0)}) - F^\star}{\alpha(K+1)} + \frac{\|\mathbf{\Theta}^{(0)} - \mathbf{1}_m\bar{\theta}^{(0)}\|^2 w_2}{K+1} + \alpha^2 w_2 w_3 + (1 - d_{\min})w_4 + \alpha w_5\right),$$

*in which* $F^\star = \min_{\theta \in \mathbb{R}^n} F(\theta)$ *is the globally optimal loss, and* $w_1 = \frac{1}{2}(1 - \frac{1}{\Gamma_0})(1 - \frac{1}{\Gamma_2})(1 - \frac{1}{\Gamma_4^2})$, $w_2 = \frac{\beta^2 d_{\max}(1+\Gamma_3)}{m(1-\tilde{\rho})\left(1 - \frac{1}{\Gamma_1}\right)}$, $w_3 = md_{\max}(16\frac{1+\tilde{\rho}}{1-\tilde{\rho}}\delta^2 + \sigma^2)$, $w_4 = (1+\Gamma_3)\delta^2$ *and* $w_5 = \frac{\beta d_{\max}\sigma^2}{2m}$. *Also, the conditions on the constant scalars* $\Gamma_i$ *are* $\Gamma_0, \Gamma_1, \Gamma_2, \Gamma_4 > 1$ *and* $\Gamma_3 > 0$. *Letting* $K \to \infty$, *we obtain*

$$\lim_{K \to \infty} \frac{\sum_{r=0}^K \mathbb{E}_{\mathbf{\Xi}^{(r-1)}}[\|\nabla F(\bar{\theta}^{(r)})\|^2]}{K+1} \leq \frac{\alpha^2 w_2 w_3 + (1 - d_{\min})w_4 + \alpha w_5}{w_1}.$$

**Discussion.** Theorem 4.12 shows that many takeaways from Theorem 4.11 extend to the non-convex case. Specifically, Theorem 4.12 generalizes results of existing DFL works by capturing the effect of sporadicity with non-convex losses. As in Theorem 4.11, by setting $d_{min} = \min_{0 \leq k \leq K, i \in \mathcal{M}} d_i^{(k)} = 1$, our work recovers well-known convergence guarantees in DGD when there is no sporadicity in SGDs Koloskova et al. (2020) (see Appendix. P.4 for further discussion). Additionally, the stationarity gap, which is bounded regardless of the $d_i$ values, can be reduced by choosing a smaller

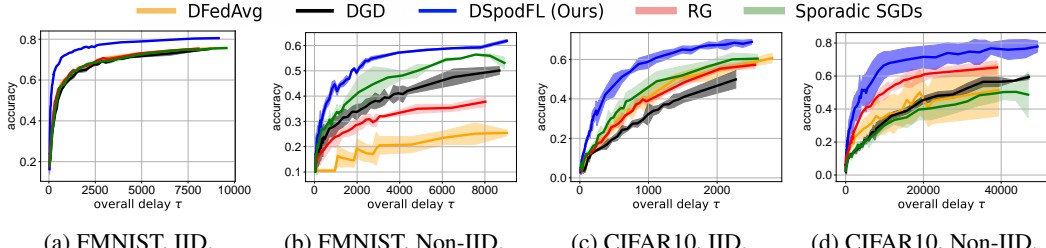

Figure 2: Accuracy vs. latency plots. `DSpodFL` achieves the target accuracy much faster with less delay, emphasizing the benefit of sporadicity in DFL for SGD iterations and model aggregations simultaneously.

learning rate $\alpha$ and/or a larger minimum SGD probability $d_{\min}$. One key difference from Theorem 4.11 is that Theorem 4.12 provides sub-linear convergence (i.e., $\mathcal{O}(1/K)$) of average gradient norms as opposed to geometric convergence of models' last iterates across clients. We also provide analysis for the diminishing learning rate case in Appendix K.

## 5 NUMERICAL EVALUATION

**Models and datasets.** To evaluate our methodology, we consider image classification tasks using the Fashion-MNIST (FMNIST) (Xiao et al., 2017) and CIFAR10 (Krizhevsky et al., 2009) datasets. We use FMNIST to train the Support Vector Machine (SVM) model (Cortes & Vapnik, 1995), while CIFAR10 is adopted for training VGG11 (Simonyan & Zisserman, 2015).

**Settings.** By default, we consider $m = 10$ clients connected via a random geometric graph (RGG) with radius $0.4$ (Penrose, 2003). We adopt a constant learning rate $\alpha = 0.01$, and use a batch size of $16$. The SGD probability $d_i$ for each client and exchange probability $b_{ij}$ for each link are randomly chosen according to either the Beta distribution $\text{Beta}(\alpha, \beta)$, uniform distribution $U[0, 1]$ or Bimodal Truncated Gaussian distribution $0.5(\tilde{N}_{[0,1]}(\mu_1, \sigma_1^2) + \tilde{N}_{[0,1]}(\mu_2, \sigma_2^2))$, and are held constant over iterations $k$. Choosing $\alpha = \beta < 1$ for the Beta distribution results in an inverted bell-shaped distribution, which corresponds to scenarios where the clients and communications links exhibit significant heterogeneity. For FMNIST, we use $d_i, b_{ij} \sim \text{Beta}(0.5, 0.5)$, and for CIFAR10, we use $d_i, b_{ij} \sim \text{Beta}(0.8, 0.8)$. We consider two different data distribution scenarios: (i) IID, where each client receives samples from all $10$ classes in the dataset, and (ii) non-IID, where each client receives samples from just $1$ class for the FMNIST dataset, and from $3$ classes for CIFAR10. Unless stated otherwise, our experiments are done under the non-IID setup. We measure the test accuracy of each scheme achieved over the average total delay incurred up to iteration $k$. Specifically, $\tau_{total}^{(k)} = \tau_{trans}^{(k)} + \tau_{proc}^{(k)}$, in which $\tau_{trans}^{(k)} = [\sum_{i=1}^{m} (1/|\mathcal{N}_i|) \sum_j \hat{v}_{ij}^{(k)}/b_{ij}]/[\sum_{i=1}^{m} (1/|\mathcal{N}_i|) \sum_j 1/b_{ij}]$ and $\tau_{proc}^{(k)} = [\sum_{i=1}^{m} v_i^{(k)}/d_i]/[\sum_{i=1}^{m} 1/d_i]$ are the per-client transmission delays incurred across links and processing delays incurred across clients in iteration $k$, respectively (see Appendix P.3 for further discussion). For a fair comparison, we determine the aggregation frequency $D$ for the DFedAvg algorithm (depicted in Fig. 1) based on these $d_i$, i.e., $D = \lceil(1/m)\sum_{i=1}^{m} 1/d_i\rceil$. We conduct the experiments based on a cluster of three NVIDIA A100 GPUs with 40GB memory. We run the experiments multiple times in each setup, and present the mean and 1-sigma standard deviation.

**Baselines.** We compare `DSpodFL` with four baselines that are tailored to decentralized settings: (a) Distributed Gradient Descent (DGD), where SGDs and local aggregations occur at every iteration (Nedic & Ozdaglar, 2009); (b) the Randomized Gossip (RG) algorithm (Koloskova et al., 2020), which is equivalent to Sporadic Aggregations (with constant SGDs); (c) Sporadic SGDs (with constant aggregations); and (d) Decentralized Federated Averaging (DFedAvg) (Sun et al., 2022). Note that all these baselines can be viewed as special cases of `DSpodFL` as elaborated in Fig. 1.

**Accuracy vs. delay comparisons.** Fig. 2 compares the test accuracies of different schemes in terms of overall delay. By allowing for sporadicity in both SGDs and aggregations, `DSpodFL` outperforms all baselines for both data distributions and models/datasets. In the IID setups of Figs. 2a and 2c, the performances of the baselines are reasonably similar. Meanwhile, our `DSpodFL` is able to significantly outperform those algorithms by an accuracy margin of $10 - 20\%$ in the initial stages of training. The gain of `DSpodFL` becomes more significant for non-IID data distributions as shown in Figs. 2b and 2d. Inter-client communications become more crucial in non-IID setups, as each client has access only to a small portion of the distribution of the whole dataset. Depending on the baseline and dataset, `DSpodFL` is able to achieve $10 - 40\%$ improvement in accuracy for a particular delay.

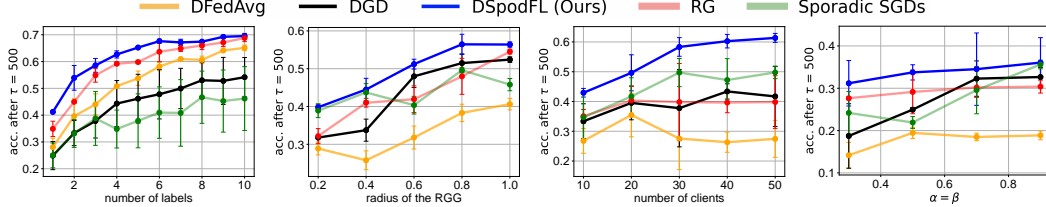

(a) Varying number of labels per client.

(b) Varying the radius of random geometric graph.

(c) Varying the number of clients $m$ in the network.

(d) Varying $\alpha = \beta$ in the $\mathrm{Beta}(\alpha, \beta)$ distribution.

Figure 3: Effects of system parameters on FMNIST. In Figs. 3a, 3b and 3c, client and link capabilities $d_i$ and $b_{ij}$ are sampled from a uniform distribution $\mathcal{U}(0, 1]$. The overall results confirm the advantage of `DSpodFL`.

**Effects of system parameters.** In Fig. 3, we study the accuracy reached by a certain training delay as system parameters are varied. In contrast to Fig. 2 where client/link capabilities were sampled from Beta distribution, for completeness, Figs. 3a-3c are sampled from uniform distribution. In Fig. 3a, we see how increasing number of labels possessed by each client (moving from non-IID to IID) improves the achieved accuracy of all methods. Further, `DSpodFL` outperforms the baselines for each data distribution. In Fig. 3b, we see that increasing the radius of the underlying RGG (which controls the density of connections) improves the achievable accuracy for all baselines. Again, `DSpodFL` performs the best for all choices of radii, confirming the benefit of integrating the notion of sporadicity in both communications and computations. Fig. 3c depicts the impact of the number of clients in the system. `DSpodFL` obtains the largest improvement as the size of the network increases, whereas the baselines are more likely to suffer from resource bottlenecks if weak nodes are added. Finally, in Fig. 3d, we analyze the effects of parameters $\alpha = \beta$ in the Beta distribution, which control the communication/computation heterogeneity across clients. Note that increasing these parameters to 1 brings the distribution closer to uniform. We see that `DSpodFL` is robust to the underlying resource distribution, and the gap between our approach and other baselines become more significant when levels of heterogeneity in client/link resources are higher, i.e., lower $\alpha = \beta$.

**Generalization of results to various settings.** In Fig. 4, we provide experimental results where some of the default system parameters that were used for Figs. 2 and 3 are changed. In Fig. 4a, a total of 50 clients are considered for the decentralized system. The results illustrate that the improvements become more pronounced with $m = 50$, confirming the trend seen with increasing $m$ in Fig. 3a. In Fig. 4b, a Bimodal Truncated Gaussian distribution is used to generate probabilities $d_i$ and $b_{ij}$, which for small values of variance when the means of the two modes are far from each other, gives heterogeneous clients. We see that when vary-

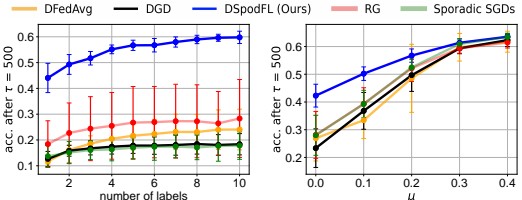

(a) Varying number of labels per client in a network of $m = 50$ clients.

(b) Varying $\mu$ in $d_i, b_{ij} \sim 0.5(\hat{N}_{[0,1]}(\mu, 0.01) + \hat{N}_{[0,1]}(1 - \mu, 0.01))$.

Figure 4: Scalability to larger clients and robustness against distributions on FMNIST.

ing $\mu$, a wider margin of improvement (15%) with lower $\mu$ where $\mu_1 = \mu$ and $\mu_2 = 1 - \mu$ is achieved. This confirms that `DSpodFL` is most advantageous relative to the baselines under higher levels of heterogeneity, similar to how the improvements under the inverted bell-shaped Beta distribution were more pronounced than those under the uniform distribution in Fig. 3d.

**Additional results.** In Appendix O, we report experimental results when time-varying SGD and aggregation probabilities are used, as well as under other configurations of resource heterogeneity.

# 6 CONCLUSION AND LIMITATIONS

We proposed `DSpodFL`, a DFL algorithmic framework that generalizes the notion of sporadicity to fully decentralized scenarios. By considering (i) sporadic gradient computations and (ii) sporadic client-to-client communications simultaneously, our approach tackles the challenges in heterogeneous and time-varying resource settings and subsumes well-known decentralized optimization algorithms. We analyzed the convergence behavior of `DSpodFL`, and showed how our results recover existing DGD-like algorithms under special cases of sporadicity. Through experiments, we demonstrated the advantage of `DSpodFL` compared to DFL baselines. Future work could consider further validating `DSpodFL` on larger datasets and more expansive set of tasks beyond image classification.

ACKNOWLEDGMENTS

This work was supported in part by the National Science Foundation (NSF) under grants CNS-2146171 and CPS-2313109, the Office of Naval Research (ONR) under grant N00014-22-1-2305, and the Air Force Office of Scientific Research (AFOSR) under grant FA9550-24-1-0083.

REPRODUCIBILITY STATEMENT

We use open-source datasets detailed in Section 5. Complete code for training and testing is available in the supplementary material. Pseudo-code of our algorithm is given in Appendix B. The assumptions for our theoretical results are outlined in Sec. 4.1, and the proofs of our Lemmas, Propositions and Theorems (Secs. 4.2-4.5) are given in Appendices D-H.

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

## A  NOTATION

Arguments for functions are denoted with parentheses, e.g., $f(x)$ implies $x$ is an argument for function $f$. The iteration index for a parameter is indicated via superscripts, e.g., $h^{(k)}$ is the value of the parameter $h$ at iteration $k$. client indices are given via subscripts, e.g., $h_i^{(k)}$ refers to parameter belonging to client $i$. We write a graph $\mathcal{G}$ with a set of nodes (clients) $\mathcal{V}$ and a set of edges (links) $\mathcal{E}$ as $\mathcal{G} = (\mathcal{V}, \mathcal{E})$.

We denote vectors with lowercase boldface, e.g., $\mathbf{x}$, and matrices with uppercase boldface, e.g., $\mathbf{X}$. All vectors $\mathbf{x} \in \mathbb{R}^{d \times 1}$ are column vectors, except in certain cases where average vectors $\bar{\mathbf{x}} \in \mathbb{R}^{1 \times d}$ and optimal vectors $\mathbf{w}^\star \in \mathbb{R}^{1 \times d}$ are row vectors. $\langle \mathbf{x}, \mathbf{x}' \rangle$ and $\langle \mathbf{X}, \mathbf{X}' \rangle$ denote the inner product of two vectors $\mathbf{x}, \mathbf{x}'$ of equal dimensions and the Frobenius inner product of two matrices $\mathbf{X}, \mathbf{X}'$ of equal dimensions, respectively. Moreover, $\|\mathbf{x}\|$ and $\|\mathbf{X}\|$ denote the 2-norm of the vector $\mathbf{x}$, and the Frobenius norm of the matrix $\mathbf{X}$, respectively. The spectral norm of the matrix $\mathbf{X}$ is written as $\rho(\mathbf{X})$.

## B  ALGORITHM PSEUDOCODE

---

**Algorithm 1** Decentralized Sporadic Federated Learning (`DSpodFL`)

---

1: **Input:** $K$, $\{\mathcal{G}^{(k)} = (\mathcal{M}, \mathcal{E}^{(k)})\}_{0 \leq k \leq K}$, $\{v_i^{(k)}\}_{i \in \mathcal{M}, 0 \leq k \leq K}$, $\{\hat{v}_{ij}^{(k)}\}_{(i,j) \in \mathcal{E}^{(k)}, 0 \leq k \leq K}$, $\{\alpha^{(k)}\}_{0 \leq k \leq K}$

2: **Output:** $\{\theta_i^{(K+1)}\}_{i \in \mathcal{M}}$

3: $k \leftarrow 0$, Initialize $\theta^{(0)}$, $\{\theta_i^{(0)} \leftarrow \theta^{(0)}\}_{i \in \mathcal{M}}$, $\{v_i^{(0)} \leftarrow 0\}_{i \in \mathcal{M}}$, $\{\hat{v}_{ij}^{(0)} \leftarrow 0\}_{(i,j) \in \mathcal{E}^{(0)}}$

4: **while** $k \leq K$ **do**

5:     **for all** $i \in \mathcal{M}$ **do**

6:         $g_i^{(k)} \leftarrow 0$, $\mathrm{aggr}_i^{(k)} \leftarrow 0$

7:         **if** $v_i^{(k)} = 1$ **then**

8:             sample mini-batch $\xi_i^{(k)} \in \mathcal{D}_i$

9:             $g_i^{(k)} \leftarrow \nabla F_i(\theta_i^{(k)}; \xi_i^{(k)})$

10:         **end if**

11:         **for all** $j \in \mathcal{E}_i^{(k)}$ **do**

12:             **if** $\hat{v}_{ij}^{(k)} = 1$ **then**

13:                 $r_{ij} \leftarrow 1/(1 + \max\{|\mathcal{N}_i|, |\mathcal{N}_j|\})$

14:                 $\mathrm{aggr}_i^{(k)} \leftarrow \mathrm{aggr}_i^{(k)} + r_{ij}\left(\theta_j^{(k)} - \theta_i^{(k)}\right)$

15:             **end if**

16:         **end for**

17:     **end for**

18:     **for all** $i \in \mathcal{M}$ **do**

19:         $\theta_i^{(k+1)} \leftarrow \theta_i^{(k)} + \mathrm{aggr}_i^{(k)} - \alpha^{(k)}\mathbf{g}_i^{(k)}$

20:     **end for**

21:     $k \leftarrow k + 1$

22: **end while**

---

## C  ASSUMPTION STATEMENTS

In this section, we state the mathematical inequalities that follow from the assumptions we made in Sec. 4.1, which are used in our subsequent Lemmas and Propositions.

- **Assumption 4.1**:

  (a): $\beta_i$-smoothness: $\|\nabla F_i(\theta) - \nabla F_i(\theta')\| \leq \beta_i \|\theta - \theta'\| \leq \beta \|\theta - \theta'\|$,

  (b): $\mu_i$-strong convexity: $\langle \nabla F_i(\theta) - \nabla F_i(\theta'), \theta - \theta' \rangle \geq \mu_i \|\theta - \theta'\|^2 \geq \mu \|\theta - \theta'\|^2$,

  (c): $\delta_i, \zeta_i$-gradient diversity: $\|\nabla F(\theta) - \nabla F_i(\theta)\| \leq \delta_i + \zeta_i \|\theta - \theta^\star\| \leq \delta + \zeta \|\theta - \theta^\star\|$,

for all $(\theta', \theta) \in \mathbb{R}^n \times \mathbb{R}^n$ and all $i \in \mathcal{M}$, where $\mu = \min_{i \in \mathcal{M}} \mu_i$, $\beta = \max_{i \in \mathcal{M}} \beta_i$, $\delta = \max_{i \in \mathcal{M}} \delta_i$ and $\zeta = \max_{i \in \mathcal{M}} \zeta_i$.

Note that these measures are related to each other via the inequalities $\mu \leq \mu_i \leq \beta_i \leq \beta$, $0 \leq \zeta_i < \beta_i + \beta$ and $0 \leq \zeta \leq 2\beta$ (see Appendix E.1). We will also find the relationship $F(\theta) \leq F(\theta') + \langle \nabla F(\theta'), \theta - \theta' \rangle + \frac{\beta}{2}\|\theta' - \theta\|^2$ useful in our treatment of smoothness.

- **Assumption 4.2**:

  Note that it is possible to make this assumption even for convex models instead of Assumption 4.1-(c), since smoothness implies that $\|\nabla F(\theta)\| \leq \beta\|\theta - \theta^\star\|$. However, we choose to use Assumption 4.1-(c) instead of Assumption 4.2-(b) for convex models, since it is a less strict assumption.

- **Assumption 4.3**:

  (a): Zero mean and bounded variance of stochastic gradient noise: $\mathbb{E}_{\epsilon_i^{(k)}}[\epsilon_i^{(k)}] = 0$,

  $\mathbb{E}[\|\epsilon_i^{(k)}\|_2^2] \leq \sigma_i^2 \leq \sigma^2$,

  where $\sigma^2 = \max_{i \in \mathcal{M}} \sigma_i^2$, for all $i \in \mathcal{M}$ and all $k \geq 0$.

  (b): Let us define SGD noise vectors $\epsilon_i^{(k)}$ and $\epsilon_j^{(k)}$, indicator variables $v_i^{(k)}$ and $v_j^{(k)}$, and $\hat{v}_{i,j}^{(k)}$ and $\hat{v}_{l,q}^{(k)}$. Then for all $i \neq j$, and $(i,j) \neq (l,q)$, we have:

  $\mathbb{E}_{\xi^{(k)}}[\langle \epsilon_i^{(k)}, \epsilon_j^{(k)} \rangle] = \left\langle \mathbb{E}_{\epsilon_i^{(k)}}[\epsilon_i^{(k)}], \mathbb{E}_{\epsilon_j^{(k)}}[\epsilon_j^{(k)}] \right\rangle$, $\mathbb{E}_{\xi^{(k)}}[v_i^{(k)} v_j^{(k)}] = \mathbb{E}_{v_i^{(k)}}[v_i^{(k)}]\mathbb{E}_{v_j^{(k)}}[v_j^{(k)}]$,

  $\mathbb{E}_{\xi^{(k)}}[\hat{v}_{i,j}^{(k)} \hat{v}_{l,q}^{(k)}] = \mathbb{E}_{\hat{v}_{i,j}^{(k)}}[\hat{v}_{i,j}^{(k)}]\mathbb{E}_{\hat{v}_{l,q}^{(k)}}[\hat{v}_{l,q}^{(k)}]$, $\mathbb{E}_{\xi^{(k)}}[\epsilon_i^{(k)} v_i^{(k)}] = \mathbb{E}_{\epsilon_i^{(k)}}[\epsilon_i^{(k)}]\mathbb{E}_{v_i^{(k)}}[v_i^{(k)}]$.

- **Assumption 4.4**:

  This assumption implies that if $\mathbf{P}^{(k)} = [p_{ij}^{(k)}]_{1 \leq i,j \leq m}$ and $\mathbf{R} = [r_{ij}]_{1 \leq i,j \leq m}$ as defined in Eqs. 4 and 2 are the doubly-stochastic mixing matrices assigned to $\mathcal{G}^{(k)}$ and $\mathcal{G}$, respectively, we have

  $\left\|\mathbf{P}^{(k)}\boldsymbol{\Theta}^{(k)} - \mathbf{1}_m\bar{\theta}^{(k)}\right\|^2 \leq \left\|\boldsymbol{\Theta}^{(k)} - \mathbf{1}_m\bar{\theta}^{(k)}\right\|^2$,

  $\left\|\mathbf{R}\boldsymbol{\Theta}^{(k)} - \mathbf{1}_m\bar{\theta}^{(k)}\right\|^2 \leq \rho_r^2 \cdot \left\|\boldsymbol{\Theta}^{(k)} - \mathbf{1}_m\bar{\theta}^{(k)}\right\|^2$,

  with $0 < \rho_r < 1$, where $\rho_r$ denotes the spectral radius of the matrix $\mathbf{R} - \frac{1}{m}\mathbf{1}_m\mathbf{1}_m^T$.

## C.1 GRADIENT DIVERSITY ASSUMPTIONS FOR CONVEX AND NON-CONVEX MODELS

In this section, we will discuss why we change the gradient diversity assumption when we treat non-convex models, i.e., move from Assumption 4.1-(c) to Assumption 4.2-(b).

We note that if we are dealing with strongly-convex models, i.e., we make Assumption 4.1-(b) for $\mu$-strongly convex models, the inequality in Assumption 4.2-(b) for data heterogeneity is a much stronger assumption than Assumption 4.1-(c). To show this, we first note that making the $\beta$-smoothness assumption (made in both Assumptions 4.1-(a) and 4.2-(a)) implies that

$$\|\nabla F(\theta)\| \leq \beta\|\theta - \theta^\star\|,$$

where $\theta^\star$ is a point where $\nabla F(\theta^\star) = 0$, i.e., a globally optimal point for convex models and a stationary point for non-convex models (note that we present and prove this result in Lemma D.1-(b)). Then, since we have $\|\nabla F_i(\theta)\| \leq \delta_i + \zeta_i\|\nabla F(\theta)\|$ according to Assumption 4.2-(b), we can conclude that

$$\|\nabla F(\theta) - \nabla F_i(\theta)\| \leq \|\nabla F(\theta)\| + \|\nabla F_i(\theta)\| \leq \delta_i + (1 + \zeta_i)\|\nabla F(\theta)\| \leq \delta_i + (1 + \zeta_i)\beta\|\theta - \theta^\star\|.$$

We observe that defining $\delta_i' = \delta_i$ and $\zeta_i' = (1 + \zeta_i)\beta > 0$, Assumption 4.1-(c) is implied with parameters $\delta_i'$ and $\zeta_i'$. The converse is not necessarily true though, i.e., Assumption 4.1-(c) does not imply Assumption 4.2-(b).

## D INTERMEDIARY LEMMAS

**Lemma D.1 (Gradient bounds)** *(See Appendix E.1 for the proof.) Let Assumption 4.1 hold. We have*

(a) *The global loss function $F(\theta)$ is $\beta$-smooth and $\mu$-strongly convex, i.e.,*

$$\|\nabla F(\theta) - \nabla F(\theta')\| \leq \beta \|\theta - \theta'\|, \qquad \langle \nabla F(\theta) - \nabla F(\theta'), \theta - \theta' \rangle \geq \mu \|\theta - \theta'\|^2.$$

(b) *The gradients of the global and local loss functions, and the gradient of the local loss function at the optimal point are bounded as*

$$\|\nabla F(\theta)\| \leq \beta \|\theta - \theta^\star\|, \quad \|\nabla F_i(\theta)\|^2 \leq 2\left(\beta_i^2 \|\theta - \theta^\star\|^2 + \delta_i^2\right), \quad \|\nabla F_i(\theta^\star)\| \leq \delta_i.$$

Part-(a) of Lemma D.1 outlines the smoothness and convexity behaviour of the global loss function based on the measures of local loss functions, and part (b) provides upper bounds on the gradients. Note how these show that we are not making the bounded gradients assumption for all $\theta \in \mathbb{R}^n$, but only bounded local gradients at the globally optimal point $\theta^\star$.

Next, we provide upper bounds on the expected Frobenius norms of the following quantities related to SGD noises.

**Lemma D.2 (Expected value of SGD noise average and deviation)** *(See Appendix E.2 for the proof.) Let Assumption 4.3 hold. For every iteration $k \geq 0$, the average SGD noise and their deviation from this average can be bounded as*

$$\mathbb{E}_{\xi^{(k)}}\left[\left\|\overline{\epsilon v}^{(k)}\right\|^2\right] \leq d_{\max}^{(k)} \sigma^2 / m, \qquad \mathbb{E}_{\xi^{(k)}}\left[\left\|\mathbf{V}^{(k)}\mathbf{E}^{(k)} - \mathbf{1}_m \overline{\epsilon v}^{(k)}\right\|^2\right] \leq m d_{\max}^{(k)} \sigma^2,$$

*in which $\overline{\epsilon v}^{(k)} = \frac{1}{m}\sum_{i=1}^m \epsilon_i^{(k)} v_i^{(k)}$.*

Note that by setting $d_{\max}^{(k)} = 1$ in D.2, we get back the well-known estimation bounds for these quantities (e.g., see Lemma 2 in Pu & Nedić (2021)).

Next, we find an upper bound on the expected deviation of the gradients from their average (similar to the second quantity in Lemma D.2).

**Lemma D.3 (Gradient deviation bound)** *(See Appendix E.3 for the proof.) Let Assumption 4.1 hold. For each iteration $k \geq 0$, we have the following bound on the expected error of gradients from their average*

$$\mathbb{E}_{\mathbf{\Xi}^{(k)}}\left[\left\|\mathbf{V}^{(k)}\nabla^{(k)} - \mathbf{1}_m \overline{\nabla v}^{(k)}\right\|^2\right] \leq 6 d_{\max}^{(k)}\left[m\delta^2 + \left(\zeta^2 + 2\beta^2\right)\mathbb{E}_{\mathbf{\Xi}^{(k-1)}}\left[\left\|\mathbf{\Theta}^{(k)} - \mathbf{1}_m \bar{\theta}^{(k)}\right\|^2\right]\right.$$

$$\left. + m\left(\zeta^2 + 2\beta^2\left(1 - d_{\min}^{(k)}\right)\right)\mathbb{E}_{\mathbf{\Xi}^{(k-1)}}\left[\left\|\bar{\theta}^{(k)} - \theta^\star\right\|^2\right]\right].$$

*in which $\nabla^{(k)}$ is a matrix whose rows are comprised of the gradient vectors $\nabla F_i(\theta_i^{(k)})$, and $\overline{\nabla v}^{(k)} = \frac{1}{m}\sum_{i=1}^m \nabla F_i(\theta_i^{(k)}) v_i^{(k)}$.*

Finally, we analyze the behaviour of the random mixing matrix $\mathbf{P}^{(k)}$ defined in Eqs. 3 and 4.

**Lemma D.4 (Expected mixing matrix)** *(See Appendix E.4 for the proof.) Let Assumption 4.4 hold. For each iteration $k \geq 0$, we have*

(a) *The expected mixing matrix, denoted as $\bar{\mathbf{R}}^{(k)}$, is irreducible and doubly-stochastic:*

$$\mathbb{E}_{\hat{\mathbf{V}}^{(k)}}\left[\mathbf{P}^{(k)}\right] \triangleq \bar{\mathbf{R}}^{(k)} = \left[\bar{r}_{ij}^{(k)}\right]_{1 \leq i,j \leq m}, \qquad \bar{r}_{ij}^{(k)} = \begin{cases} b_{ij}^{(k)} r_{ij} & i \neq j \\ 1 - \sum_{j=1}^m b_{ij}^{(k)} r_{ij} & i = j \end{cases}.$$

(b) $\mathbb{E}_{\hat{\mathbf{V}}^{(k)}}\left[\left(\mathbf{P}^{(k)}\right)^2\right] = \left(\bar{\mathbf{R}}^{(k)}\right)^2 + \mathbf{R}_0^{(k)} \triangleq \tilde{\mathbf{R}}^{(k)},$

*where $\mathbf{R}_0^{(k)}$ is a matrix whose rows and columns sum to zero. Thus, $\tilde{\mathbf{R}}^{(k)}$ will be irreducible and doubly-stochastic.*

*(c)* $\mathbb{E}_{\Xi^{(k)}}\left[\left\|\mathbf{P}^{(k)}\Theta^{(k)} - \mathbf{1}_m\bar{\theta}^{(k)}\right\|^2\right] \le \tilde{\rho}^{(k)}\mathbb{E}_{\Xi^{(k-1)}}\left[\left\|\Theta^{(k)} - \mathbf{1}_m\bar{\theta}^{(k)}\right\|^2\right],$

in which $\tilde{\rho}^{(k)}$ is the spectral radius of the matrix $\tilde{\mathbf{R}}^{(k)} - \frac{1}{m}\mathbf{1}_m\mathbf{1}_m^T$.

## E   PROOFS OF INTERMEDIARY LEMMAS

### E.1   PROOF OF LEMMA D.1

(a) First, we use Eq. 1, triangle inequality and the smoothness property given in Assumption 4.1-(a) to get

$$\|\nabla F(\theta) - \nabla F(\theta')\| = \left\|\frac{1}{m}\sum_{j=1}^m (\nabla F_j(\theta) - \nabla F_j(\theta'))\right\| \le \frac{1}{m}\sum_{j=1}^m \|\nabla F_j(\theta) - \nabla F_j(\theta')\|$$

$$\le \frac{1}{m}\sum_{j=1}^m \beta_j \|\theta - \theta'\| = \bar{\beta}\|\theta - \theta'\| \le \beta\|\theta - \theta'\|.$$

Next, using Eq. 1 and the strong convexity property of Assumption 4.1-(b), we have

$$\langle \nabla F(\theta) - \nabla F(\theta'), \theta - \theta'\rangle = \frac{1}{m}\sum_{j=1}^m \langle \nabla F_j(\theta) - \nabla F_j(\theta'), \theta - \theta'\rangle \ge \frac{1}{m}\sum_{j=1}^m \mu_j\|\theta - \theta'\|^2$$

$$= \bar{\mu}\|\theta - \theta'\|^2 \ge \mu\|\theta - \theta'\|^2.$$

(b) Since $\nabla F(\theta^\star) = 0$ by definition, we can use the results of part (a) of this lemma to show that

$$\|\nabla F(\theta)\| \le \beta\|\theta - \theta^\star\|.$$

Once again noting that $\nabla F(\theta^\star) = 0$, we next use the gradient diversity bound outlined in Assumption 4.1-(c) to get

$$\|\nabla F_i(\theta^\star)\| \le \delta_i. \tag{11}$$

Finally, using Eq. 11 and Assumption 4.1-(a), we write

$$\|\nabla F_i(\theta)\|^2 \le 2\left(\|\nabla F_i(\theta) - \nabla F_i(\theta^\star)\|^2 + \|\nabla F_i(\theta^\star)\|^2\right) \le 2\left(\beta_i^2\|\theta - \theta^\star\|^2 + \delta_i^2\right),$$

finishing the proof.

To explain the statement written after Assumption 4.1 on how these measures relate to each other, we first have

$$\mu \le \mu_i \le \beta_i \le \beta,$$

in which $\mu_i \le \beta_i$ is a well-known fact (see Bottou et al. (2018) as a reference), and $\mu \le \mu_i$ and $\beta_i \le \beta$ follow from the definitions given in Assumption 4.1. Moreover, if we upper-bound the gradient diversity term $\|\nabla F(\theta) - \nabla F_i(\theta)\|$ without using Assumption 4.1-(c), we will have

$$\begin{aligned}
\|\nabla F(\theta) - \nabla F_i(\theta)\| &\le \|\nabla F(\theta) - \nabla F_i(\theta^\star) + \nabla F_i(\theta^\star) - \nabla F_i(\theta)\| \\
&\le \|\nabla F(\theta)\| + \|\nabla F_i(\theta^\star)\| + \|\nabla F_i(\theta) - \nabla F_i(\theta^\star)\| \\
&\le \delta_i + \beta_i\|\theta - \theta^\star\| + \beta\|\theta - \theta^\star\| \le \delta_i + (\beta_i + \beta)\|\theta - \theta^\star\|,
\end{aligned} \tag{12}$$

in which we used the triangle inequality, Assumption 4.1-(a) and the results of Lemma D.1-(b). Now comparing Eq. 12 with the assumption made in 4.1-(c) for the same expression, we conclude that

$$\zeta_i \le \beta_i + \beta, \qquad \zeta \le 2\beta.$$

### E.2   PROOF OF LEMMA D.2

We start by finding an upper bound for the average SGD noise, by expanding the terms using their definitions and employing the properties given in Assumption 4.3 and Definition 4.6.

$$
\mathbb{E}_{\xi^{(k)}}\left[\left\|\overline{\epsilon v}^{(k)}\right\|^2\right] = \mathbb{E}_{\xi^{(k)}}\left[\left\|\frac{1}{m}\sum_{i=1}^m \epsilon_i^{(k)} v_i^{(k)}\right\|^2\right] = \mathbb{E}_{\xi^{(k)}}\left[\frac{1}{m^2}\sum_{i=1}^m\sum_{j=1}^m\left\langle \epsilon_i^{(k)} v_i^{(k)}, \epsilon_j^{(k)} v_j^{(k)}\right\rangle\right]
$$

$$
= \frac{1}{m^2}\sum_{i=1}^m \mathbb{E}_{\xi_i^{(k)}}\left[\left\|\epsilon_i^{(k)} v_i^{(k)}\right\|^2\right] + \frac{1}{m^2}\sum_{i=1}^m\sum_{\substack{j=1 \\ j\neq i}}^m\left\langle \mathbb{E}_{\xi_i^{(k)}}\left[\epsilon_i^{(k)} v_i^{(k)}\right], \mathbb{E}_{\xi_j^{(k)}}\left[\epsilon_j^{(k)} v_j^{(k)}\right]\right\rangle
$$

$$
= \frac{1}{m^2}\sum_{i=1}^m \mathbb{E}_{\xi_i^{(k)}}\left[\left\|\epsilon_i^{(k)}\right\|^2 v_i^{(k)}\right]
$$

$$
+ \frac{1}{m^2}\sum_{i=1}^m\sum_{\substack{j=1 \\ j\neq i}}^m\left\langle \mathbb{E}_{\epsilon_i^{(k)}}\left[\epsilon_i^{(k)}\right]\mathbb{E}_{v_i^{(k)}}\left[v_i^{(k)}\right], \mathbb{E}_{\epsilon_j^{(k)}}\left[\epsilon_j^{(k)}\right]\mathbb{E}_{v_j^{(k)}}\left[v_j^{(k)}\right]\right\rangle
$$

$$
= \frac{1}{m^2}\sum_{i=1}^m \mathbb{E}_{\epsilon_i^{(k)}}\left[\left\|\epsilon_i^{(k)}\right\|^2\right]\mathbb{E}_{v_i^{(k)}}\left[v_i^{(k)}\right] = \frac{1}{m^2}\sum_{i=1}^m d_i^{(k)}\sigma_i^2 \leq \frac{d_{\max}^{(k)}\sigma^2}{m}.
$$

Next, we found an upper bound for deviance of the error matrix from its average, using a similar approach as above. Noting that $\|\cdot\|$ is the Frobenius norm for matrices, Assumption 4.3 and Definition 4.6 implies that

$$
\mathbb{E}_{\xi^{(k)}}\left[\left\|\mathbf{V}^{(k)}\mathbf{E}^{(k)} - \mathbf{1}_m\overline{\epsilon v}^{(k)}\right\|^2\right]
$$

$$
= \mathbb{E}_{\xi^{(k)}}\left[\left\|\mathbf{V}^{(k)}\mathbf{E}^{(k)}\right\|^2 - 2\left\langle \mathbf{V}^{(k)}\mathbf{E}^{(k)}, \mathbf{1}_m\overline{\epsilon v}^{(k)}\right\rangle + \left\|\mathbf{1}_m\overline{\epsilon v}^{(k)}\right\|^2\right]
$$

$$
= \mathbb{E}_{\xi^{(k)}}\left[\sum_{i=1}^m\left\|\epsilon_i^{(k)} v_i^{(k)}\right\|^2\right] - 2\mathbb{E}_{\xi^{(k)}}\left[\sum_{i=1}^m\left\langle \epsilon_i^{(k)} v_i^{(k)}, \overline{\epsilon v}^{(k)}\right\rangle\right] + \mathbb{E}_{\xi^{(k)}}\left[m\left\|\overline{\epsilon v}^{(k)}\right\|^2\right]
$$

$$
= \sum_{i=1}^m \mathbb{E}_{\xi_i^{(k)}}\left[\left\|\epsilon_i^{(k)}\right\|^2 v_i^{(k)}\right] - \frac{2}{m}\sum_{i=1}^m \mathbb{E}_{\xi_i^{(k)}}\left[\left\|\epsilon_i^{(k)}\right\|^2 v_i^{(k)}\right]
$$

$$
- \frac{2}{m}\sum_{i=1}^m\left\langle \mathbb{E}_{\xi_i^{(k)}}\left[\epsilon_i^{(k)} v_i^{(k)}\right], \sum_{\substack{j=1 \\ j\neq i}}^m \mathbb{E}_{\xi_j^{(k)}}\left[\epsilon_j^{(k)} v_j^{(k)}\right]\right\rangle + m\mathbb{E}_{\xi^{(k)}}\left[\left\|\overline{\epsilon v}^{(k)}\right\|^2\right]
$$

$$
= \left(1 - \frac{2}{m} + \frac{1}{m}\right)\sum_{i=1}^m \mathbb{E}_{\epsilon_i^{(k)}}\left[\left\|\epsilon_i^{(k)}\right\|^2\right]\mathbb{E}_{v_i^{(k)}}\left[v_i^{(k)}\right]
$$

$$
- \frac{2}{m}\sum_{i=1}^m\sum_{\substack{j=1 \\ j\neq i}}^m\left\langle \mathbb{E}_{\epsilon_i^{(k)}}\left[\epsilon_i^{(k)}\right]\mathbb{E}_{v_i^{(k)}}\left[v_i^{(k)}\right], \mathbb{E}_{\epsilon_j^{(k)}}\left[\epsilon_j^{(k)}\right]\mathbb{E}_{v_j^{(k)}}\left[v_j^{(k)}\right]\right\rangle
$$

$$
= \left(1 - \frac{1}{m}\right)\sum_{i=1}^m d_i^{(k)}\sigma_i^2 \leq (m-1)d_{\max}^{(k)}\sigma^2 \leq md_{\max}^{(k)}\sigma^2.
$$

### E.3   PROOF OF LEMMA D.3

Noting that

$$
\left\|\mathbf{V}^{(k)}\nabla^{(k)} - \mathbf{1}_m\overline{\nabla v}^{(k)}\right\|^2 \leq 2\left\|\mathbf{V}^{(k)}\nabla^{(k)} - \mathbf{V}^{(k)}\nabla\mathbf{F}^{(k)}\right\|^2 + 2\left\|\mathbf{V}^{(k)}\nabla\mathbf{F}^{(k)} - \mathbf{1}_m\overline{\nabla v}^{(k)}\right\|^2,
$$

according to Young's inequality, we first find an upper bound for each of the two terms above separately. For the first term, noting that $\|\cdot\|$ is the Frobenius norm for matrices, we have

$$\left\|\mathbf{V}^{(k)}\nabla^{(k)} - \mathbf{V}^{(k)}\nabla\mathbf{F}^{(k)}\right\|^2 = \sum_{i=1}^{m}\left\|\nabla F_i\left(\theta_i^{(k)}\right)v_i^{(k)} - \nabla F\left(\theta_i^{(k)}\right)v_i^{(k)}\right\|^2$$

$$= \sum_{i=1}^{m}\left\|\nabla F_i\left(\theta_i^{(k)}\right) - \nabla F\left(\theta_i^{(k)}\right)\right\|^2 v_i^{(k)} \leq \sum_{i=1}^{m}\left(\delta_i + \zeta_i\left\|\theta_i^{(k)} - \theta^\star\right\|\right)^2 v_i^{(k)}$$

$$\leq \sum_{i=1}^{m}\left(\delta_i + \zeta_i\left\|\theta_i^{(k)} - \bar{\theta}^{(k)}\right\| + \zeta_i\left\|\bar{\theta}^{(k)} - \theta^\star\right\|\right)^2 v_i^{(k)}$$

$$\leq 3\sum_{i=1}^{m}\left(\delta_i^2 + \zeta_i^2\left\|\theta_i^{(k)} - \bar{\theta}^{(k)}\right\|^2 + \zeta_i^2\left\|\bar{\theta}^{(k)} - \theta^\star\right\|^2\right)v_i^{(k)},$$

where Assumption 4.1-(c), triangle inequality and Young's inequality were used for the inequalities, respectively. For the second term, again noting that $\|\cdot\|$ is the Frobenius norm for matrices, using Eq. 1 and the triangle inequality, we can write

$$\left\|\mathbf{V}^{(k)}\nabla\mathbf{F}^{(k)} - \mathbf{1}_m\overline{\nabla v}^{(k)}\right\|^2 = \sum_{i=1}^{m}\left\|\nabla F\left(\theta_i^{(k)}\right)v_i^{(k)} - \overline{\nabla v}^{(k)}\right\|^2$$

$$= \sum_{i=1}^{m}\left\|\frac{1}{m}\sum_{j=1}^{m}\left(\nabla F_j\left(\theta_i^{(k)}\right)v_i^{(k)} - \nabla F_j\left(\theta_j^{(k)}\right)v_j^{(k)}\right)\right\|^2$$

$$\leq \sum_{i=1}^{m}\frac{1}{m}\sum_{j=1}^{m}\left\|\nabla F_j\left(\theta_i^{(k)}\right)v_i^{(k)} - \nabla F_j\left(\theta_j^{(k)}\right)v_j^{(k)}\right\|^2,$$

Continuing from there and using Young's inequality and Assumption 4.1-(a), we have

$$= \frac{1}{m}\sum_{i=1}^{m}\sum_{j=1}^{m}\left\|\nabla F_j\left(\theta_i^{(k)}\right)v_i^{(k)} - \nabla F_j\left(\bar{\theta}^{(k)}\right)v_i^{(k)} + \nabla F_j\left(\bar{\theta}^{(k)}\right)v_i^{(k)} - \nabla F_j(\theta^\star)v_i^{(k)}\right.$$

$$\left.+ \nabla F_j(\theta^\star)v_i^{(k)} - \nabla F_j\left(\bar{\theta}^{(k)}\right)v_j^{(k)} + \nabla F_j\left(\bar{\theta}^{(k)}\right)v_j^{(k)} - \nabla F_j\left(\theta_j^{(k)}\right)v_j^{(k)}\right\|^2$$

$$= \frac{3}{m}\sum_{i=1}^{m}\sum_{j=1}^{m}\left(\left\|\nabla F_j\left(\theta_i^{(k)}\right) - \nabla F_j\left(\bar{\theta}^{(k)}\right)\right\|^2 v_i^{(k)} + \left\|\nabla F_j\left(\bar{\theta}^{(k)}\right) - \nabla F_j(\theta^\star)\right\|^2\left(v_i^{(k)} - v_j^{(k)}\right)^2\right.$$

$$\left.+ \left\|\nabla F_j\left(\bar{\theta}^{(k)}\right) - \nabla F_j\left(\theta_j^{(k)}\right)\right\|^2 v_j^{(k)}\right)$$

$$\leq \frac{3}{m}\sum_{i=1}^{m}\sum_{j=1}^{m}\left(\beta_j^2\left\|\theta_i^{(k)} - \bar{\theta}^{(k)}\right\|^2 v_i^{(k)} + \beta_j^2\left\|\bar{\theta}^{(k)} - \theta^\star\right\|^2\left(v_i^{(k)} + v_j^{(k)} - 2v_i^{(k)}v_j^{(k)}\right)\right.$$

$$\left.+ \beta_j^2\left\|\bar{\theta}^{(k)} - \theta_j^{(k)}\right\|^2 v_j^{(k)}\right)$$

$$= \frac{3}{m}\sum_{j=1}^{m}\beta_j^2\sum_{i=1}^{m}\left(\left\|\theta_i^{(k)} - \bar{\theta}^{(k)}\right\|^2 v_i^{(k)} + \left\|\bar{\theta}^{(k)} - \theta^\star\right\|^2\left(v_i^{(k)} + v_j^{(k)} - 2v_i^{(k)}v_j^{(k)}\right)\right.$$

$$\left.+ \left\|\bar{\theta}^{(k)} - \theta_j^{(k)}\right\|^2 v_j^{(k)}\right)$$

$$\leq \frac{3\bar{\beta}^2}{m}\left[m\sum_{i=1}^{m}\left\|\theta_i^{(k)} - \bar{\theta}^{(k)}\right\|^2 v_i^{(k)} + m\sum_{j=1}^{m}\left\|\bar{\theta}^{(k)} - \theta_j^{(k)}\right\|^2 v_j^{(k)}\right.$$

$$\left.+ \left\|\bar{\theta}^{(k)} - \theta^\star\right\|^2\left(m\sum_{i=1}^{m}v_i^{(k)} + m\sum_{j=1}^{m}v_j^{(k)} - 2\sum_{i=1}^{m}\sum_{j=1}^{m}v_i^{(k)}v_j^{(k)}\right)\right]$$

$$\leq 6\beta^2 \sum_{i=1}^{m} \left\|\theta_i^{(k)} - \bar{\theta}^{(k)}\right\|^2 v_i^{(k)} + 6\beta^2 \left\|\bar{\theta}^{(k)} - \theta^\star\right\|^2 \left(\sum_{i=1}^{m} v_i^{(k)} - \frac{1}{m}\left(\sum_{i=1}^{m} v_i^{(k)}\right)^2\right)$$

$$\leq 6\beta^2 \sum_{i=1}^{m} \left[\left\|\theta_i^{(k)} - \bar{\theta}^{(k)}\right\|^2 + \left\|\bar{\theta}^{(k)} - \theta^\star\right\|^2 \left(1 - \frac{1}{m}\sum_{j=1}^{m} v_j^{(k)}\right)\right] v_i^{(k)},$$

where for the last three inequalities we used the properties of the binary indicator random variables $v_i^{(k)} \in \{0, 1\}$. Now, by combining the two components together we get

$$\left\|\mathbf{V}^{(k)}\nabla^{(k)} - \mathbf{1}_m\overline{\nabla v}^{(k)}\right\|^2 = 2\left\|\mathbf{V}^{(k)}\nabla^{(k)} - \mathbf{V}^{(k)}\nabla\mathbf{F}^{(k)}\right\|^2 + 2\left\|\mathbf{V}^{(k)}\nabla\mathbf{F}^{(k)} - \mathbf{1}_m\overline{\nabla v}^{(k)}\right\|^2$$

$$\leq 6\sum_{i=1}^{m}\left[\delta_i^2 + (\zeta_i^2 + 2\beta^2)\left\|\theta_i^{(k)} - \bar{\theta}^{(k)}\right\|^2\right.$$

$$\left. + \left(\zeta_i^2 + 2\beta^2\left(1 - \frac{1}{m}\sum_{j=1}^{m} v_j^{(k)}\right)\right)\left\|\bar{\theta}^{(k)} - \theta^\star\right\|^2\right]v_i^{(k)}.$$

Finally, we have to take the expected value of the above inequality to conclude the proof. Towards this, we multiply $v_i^{(k)}$ inside the parentheses and use Definition 4.6 to get

$$\mathbb{E}_{\boldsymbol{\Xi}^{(k)}}\left[\left\|\mathbf{V}^{(k)}\nabla^{(k)} - \mathbf{1}_m\overline{\nabla v}^{(k)}\right\|^2\right]$$

$$\leq 6\sum_{i=1}^{m}\left[\left(\delta_i^2 + (\zeta_i^2 + 2\beta^2)\mathbb{E}_{\boldsymbol{\Xi}^{(k-1)}}\left[\left\|\theta_i^{(k)} - \bar{\theta}^{(k)}\right\|^2\right]\right)\mathbb{E}_{v_i^{(k)}}\left[v_i^{(k)}\right] + \left(\zeta_i^2\mathbb{E}_{v_i^{(k)}}\left[v_i^{(k)}\right]\right.\right.$$

$$\left.+2\beta^2\left(\mathbb{E}_{v_i^{(k)}}\left[v_i^{(k)}\right] - \frac{1}{m}\mathbb{E}_{v_i^{(k)}}\left[v_i^{(k)}\right] - \frac{1}{m}\mathbb{E}_{v_i^{(k)}}\left[v_i^{(k)}\right]\sum_{\substack{j=1\\j\neq i}}^{m}\mathbb{E}_{v_j^{(k)}}\left[v_j^{(k)}\right]\right)\right)$$

$$\mathbb{E}_{\boldsymbol{\Xi}^{(k-1)}}\left[\left\|\bar{\theta}^{(k)} - \theta^\star\right\|^2\right]\right]$$

$$= 6\sum_{i=1}^{m}\left[\left(\delta_i^2 + (\zeta_i^2 + 2\beta^2)\mathbb{E}_{\boldsymbol{\Xi}^{(k-1)}}\left[\left\|\theta_i^{(k)} - \bar{\theta}^{(k)}\right\|^2\right]\right)d_i^{(k)} + \left(\zeta_i^2 d_i^{(k)}\right.\right.$$

$$\left.+2\beta^2\left(d_i^{(k)} - \frac{1}{m}d_i^{(k)} - \frac{1}{m}d_i^{(k)}\sum_{\substack{j=1\\j\neq i}}^{m}d_j^{(k)}\right)\right)\mathbb{E}_{\boldsymbol{\Xi}^{(k-1)}}\left[\left\|\bar{\theta}^{(k)} - \theta^\star\right\|^2\right]\right]$$

$$= 6\sum_{i=1}^{m}d_i^{(k)}\left[\delta_i^2 + (\zeta_i^2 + 2\beta^2)\mathbb{E}_{\boldsymbol{\Xi}^{(k-1)}}\left[\left\|\theta_i^{(k)} - \bar{\theta}^{(k)}\right\|^2\right]\right.$$

$$\left.+\left(\zeta_i^2 + 2\beta^2\left(1 - \frac{1}{m}\left[1 + \sum_{\substack{j=1\\j\neq i}}^{m}d_j^{(k)}\right]\right)\right)\mathbb{E}_{\boldsymbol{\Xi}^{(k-1)}}\left[\left\|\bar{\theta}^{(k)} - \theta^\star\right\|^2\right]\right]$$

$$\leq 6d_{\max}^{(k)}\left[m\delta^2 + (\zeta^2 + 2\beta^2)\mathbb{E}_{\boldsymbol{\Xi}^{(k-1)}}\left[\left\|\boldsymbol{\Theta}^{(k)} - \mathbf{1}_m\bar{\theta}^{(k)}\right\|^2\right]\right.$$

$$\left.+m\left(\zeta^2 + 2\beta^2\left(1 - \frac{1}{m}\left[1 + (m-1)d_{\min}^{(k)}\right]\right)\right)\mathbb{E}_{\boldsymbol{\Xi}^{(k-1)}}\left[\left\|\bar{\theta}^{(k)} - \theta^\star\right\|^2\right]\right]$$

$$= 6d_{\max}^{(k)} \left[ m\delta^2 + \left( \zeta^2 + 2\beta^2 \right) \mathbb{E}_{\Xi^{(k-1)}} \left[ \left\| \Theta^{(k)} - \mathbf{1}_m \bar{\theta}^{(k)} \right\|^2 \right] \right.$$

$$\left. + m \left( \zeta^2 + 2\beta^2 \left( 1 - \frac{1}{m} \right) \left( 1 - d_{\min}^{(k)} \right) \right) \mathbb{E}_{\Xi^{(k-1)}} \left[ \left\| \bar{\theta}^{(k)} - \theta^\star \right\|^2 \right] \right]$$

$$\leq 6d_{\max}^{(k)} \left[ m\delta^2 + \left( \zeta^2 + 2\beta^2 \right) \mathbb{E}_{\Xi^{(k-1)}} \left[ \left\| \Theta^{(k)} - \mathbf{1}_m \bar{\theta}^{(k)} \right\|^2 \right] \right.$$

$$\left. + m \left( \zeta^2 + 2\beta^2 \left( 1 - d_{\min}^{(k)} \right) \right) \mathbb{E}_{\Xi^{(k-1)}} \left[ \left\| \bar{\theta}^{(k)} - \theta^\star \right\|^2 \right] \right],$$

where the last four lines are algebraic manipulations to simplify the bound.

### E.4 PROOF OF LEMMA D.4

(a) We take the expected value of the matrix $\mathbf{P}^{(k)}$ by looking at its individual elements. Using Eq. 4 and Definition 4.6, we have

$$\mathbb{E}_{\hat{\mathbf{V}}^{(k)}} \left[ \mathbf{P}^{(k)} \right] = \left[ \mathbb{E}_{\hat{v}_{ij}^{(k)}} \left[ p_{ij}^{(k)} \right] \right]_{1 \leq i,j \leq m} = \begin{cases} \left[ r_{ij} \mathbb{E}_{\hat{v}_{ij}^{(k)}} \left[ \hat{v}_{ij}^{(k)} \right] \right]_{1 \leq i,j \leq m} & i \neq j \\ \left[ 1 - \sum_{j=1}^m r_{ij} \mathbb{E}_{\hat{v}_{ij}^{(k)}} \left[ \hat{v}_{ij}^{(k)} \right] \right]_{1 \leq i \leq m} & i = j \end{cases}$$

$$= \begin{cases} \left[ b_{ij}^{(k)} r_{ij} \right]_{1 \leq i,j \leq m} & i \neq j \\ \left[ 1 - \sum_{j=1}^m b_{ij}^{(k)} r_{ij} \right]_{1 \leq i \leq m} & i = j \end{cases} = \bar{\mathbf{R}}^{(k)}$$

(b) Similar to the proof of the previous part, we take the expected value of $(\mathbf{P}^{(k)})^T \mathbf{P}^{(k)} = \left( \mathbf{P}^{(k)} \right)^2$ by looking at its individual elements. Again, using Eq. 4 and Definition 4.6, Assumption 4.3 implies that

if $i \neq j$ :

$$\mathbb{E}_{\hat{\mathbf{V}}^{(k)}} \left[ \sum_{l=1}^m p_{il}^{(k)} p_{lj}^{(k)} \right]$$

$$= \mathbb{E}_{\hat{\mathbf{V}}^{(k)}} \left[ \sum_{\substack{l=1 \\ l \neq i,j}}^m r_{il} r_{lj} \hat{v}_{il}^{(k)} \hat{v}_{lj}^{(k)} + \left( 1 - \sum_{q=1}^m r_{iq} \hat{v}_{iq}^{(k)} \right) r_{ij} \hat{v}_{ij}^{(k)} + \left( 1 - \sum_{q=1}^m r_{jq} \hat{v}_{jq}^{(k)} \right) r_{ij} \hat{v}_{ij}^{(k)} \right]$$

$$= \mathbb{E}_{\hat{\mathbf{V}}^{(k)}} \left[ \sum_{l=1}^m r_{il} r_{lj} \hat{v}_{il}^{(k)} \hat{v}_{lj}^{(k)} + \left( 2 - \sum_{q=1}^m \left( r_{iq} \hat{v}_{iq}^{(k)} + r_{jq} \hat{v}_{jq}^{(k)} \right) \right) r_{ij} \hat{v}_{ij}^{(k)} \right]$$

$$= \mathbb{E}_{\hat{\mathbf{V}}^{(k)}} \left[ \sum_{l=1}^m r_{il} r_{lj} \hat{v}_{il}^{(k)} \hat{v}_{lj}^{(k)} + \left( 2 - \sum_{\substack{q=1 \\ q \neq i,j}}^m \left( r_{iq} \hat{v}_{iq}^{(k)} + r_{jq} \hat{v}_{jq}^{(k)} \right) \right) r_{ij} \hat{v}_{ij}^{(k)} \right.$$

$$\left. - \left( r_{ij} \hat{v}_{ij}^{(k)} + r_{ji} \hat{v}_{ji}^{(k)} \right) r_{ij} \hat{v}_{ij}^{(k)} \right]$$

$$= \sum_{l=1}^m r_{il} r_{lj} \mathbb{E}_{\hat{v}_{il}^{(k)}} \left[ \hat{v}_{il}^{(k)} \right] \mathbb{E}_{\hat{v}_{lj}^{(k)}} \left[ \hat{v}_{lj}^{(k)} \right]$$

$$+ \left( 2 - \sum_{\substack{q=1 \\ q \neq i,j}}^m \left( r_{iq} \mathbb{E}_{\hat{v}_{iq}^{(k)}} \left[ \hat{v}_{iq}^{(k)} \right] + r_{jq} \mathbb{E}_{\hat{v}_{jq}^{(k)}} \left[ \hat{v}_{jq}^{(k)} \right] \right) \right) r_{ij} \mathbb{E}_{\hat{v}_{ij}^{(k)}} \left[ \hat{v}_{ij}^{(k)} \right] - 2 r_{ij}^2 \mathbb{E}_{\hat{v}_{ij}^{(k)}} \left[ \hat{v}_{ij}^{(k)} \right]$$

$$= \sum_{l=1}^{m} b_{il}^{(k)} r_{il} b_{lj}^{(k)} r_{lj} + \left( 2 - \sum_{\substack{q=1 \\ q \neq i,j}}^{m} \left( b_{iq}^{(k)} r_{iq} + b_{jq}^{(k)} r_{jq} \right) \right) b_{ij}^{(k)} r_{ij} - 2 b_{ij}^{(k)} r_{ij}^2$$

On the other hand,

if $i = j$ :

$$\mathbb{E}_{\hat{\mathbf{V}}^{(k)}} \left[ \sum_{l=1}^{m} p_{il}^{(k)} p_{lj}^{(k)} \right] = \mathbb{E}_{\hat{\mathbf{V}}^{(k)}} \left[ \sum_{\substack{l=1 \\ l \neq i}}^{m} r_{il} r_{li} \hat{v}_{il}^{(k)} \hat{v}_{li}^{(k)} + \left( 1 - \sum_{q=1}^{m} r_{iq} \hat{v}_{iq}^{(k)} \right)^2 \right]$$

$$= \mathbb{E}_{\hat{\mathbf{V}}^{(k)}} \left[ \sum_{l=1}^{m} r_{il}^2 \hat{v}_{il}^{(k)} + 1 - 2 \sum_{q=1}^{m} r_{iq} \hat{v}_{iq}^{(k)} + \sum_{q=1}^{m} \sum_{t=1}^{m} r_{iq} r_{it} \hat{v}_{iq}^{(k)} \hat{v}_{it}^{(k)} \right]$$

$$= \mathbb{E}_{\hat{\mathbf{V}}^{(k)}} \left[ \sum_{l=1}^{m} r_{il}^2 \hat{v}_{il}^{(k)} + 1 - 2 \sum_{q=1}^{m} r_{iq} \hat{v}_{iq}^{(k)} + \sum_{q=1}^{m} \sum_{\substack{t=1 \\ t \neq q}}^{m} r_{iq} r_{it} \hat{v}_{iq}^{(k)} \hat{v}_{it}^{(k)} + \sum_{q=1}^{m} r_{iq}^2 \hat{v}_{iq}^{(k)} \right]$$

$$= 2 \sum_{l=1}^{m} r_{il}^2 \mathbb{E}_{\hat{v}_{il}^{(k)}} \left[ \hat{v}_{il}^{(k)} \right] + 1 - 2 \sum_{q=1}^{m} r_{iq} \mathbb{E}_{\hat{v}_{iq}^{(k)}} \left[ \hat{v}_{iq}^{(k)} \right] + \sum_{q=1}^{m} \sum_{\substack{t=1 \\ t \neq q}}^{m} r_{iq} r_{it} \mathbb{E}_{\hat{v}_{iq}^{(k)}} \left[ \hat{v}_{iq}^{(k)} \right] \mathbb{E}_{\hat{v}_{it}^{(k)}} \left[ \hat{v}_{it}^{(k)} \right]$$

$$= 2 \sum_{l=1}^{m} b_{il}^{(k)} r_{il}^2 + 1 - 2 \sum_{q=1}^{m} b_{iq}^{(k)} r_{iq} + \sum_{q=1}^{m} \sum_{\substack{t=1 \\ t \neq q}}^{m} b_{iq}^{(k)} r_{iq} b_{it}^{(k)} r_{it}$$

Finally, comparing the above expression with the elements of $\left( \bar{\mathbf{R}}^{(k)} \right)^2$, we get

$$\mathbb{E}_{\hat{\mathbf{V}}^{(k)}} \left[ \left( \mathbf{P}^{(k)} \right)^2 \right] = \left[ \mathbb{E}_{\hat{\mathbf{V}}^{(k)}} \left[ \sum_{l=1}^{m} p_{il}^{(k)} p_{lj}^{(k)} \right] \right]_{1 \leq i,j \leq m}$$

$$= \left( \bar{\mathbf{R}}^{(k)} \right)^2 + \begin{cases} \left[ -2 b_{ij}^{(k)} \left( 1 - b_{ij}^{(k)} \right) r_{ij}^2 \right]_{1 \leq i,j \leq m} & i \neq j \\ \left[ \sum_{l=1}^{m} 2 b_{il}^{(k)} \left( 1 - b_{il}^{(k)} \right) r_{il}^2 \right]_{1 \leq i \leq m} & i = j \end{cases} = \left( \bar{\mathbf{R}}^{(k)} \right)^2 + \mathbf{R}_0^{(k)} \triangleq \tilde{\mathbf{R}}^{(k)},$$

in which $\mathbf{R}_0^{(k)}$ is a matrix whose rows and columns sum to zero.

(c) In order to prove this inequality, we expand the left-hand side Frobenius norm by its columns and use the results of part (b) of this lemma. We have

$$\mathbb{E}_{\boldsymbol{\Xi}^{(k)}} \left[ \left\| \mathbf{P}^{(k)} \boldsymbol{\Theta}^{(k)} - \mathbf{1}_m \bar{\theta}^{(k)} \right\|^2 \right] = \mathbb{E}_{\boldsymbol{\Xi}^{(k)}} \left[ \sum_{j=1}^{m} \left\| \mathbf{P}^{(k)} \theta_j^{(k)} - \bar{\theta}_j^{(k)} \mathbf{1}_m \right\|^2 \right]$$

$$= \mathbb{E}_{\boldsymbol{\Xi}^{(k)}} \left[ \sum_{j=1}^{m} \left\| \mathbf{P}^{(k)} \theta_j^{(k)} - \bar{\theta}_j^{(k)} \mathbf{P}^{(k)} \mathbf{1}_m \right\|^2 \right] = \mathbb{E}_{\boldsymbol{\Xi}^{(k)}} \left[ \sum_{j=1}^{m} \left\| \mathbf{P}^{(k)} \left( \theta_j^{(k)} - \bar{\theta}_j^{(k)} \mathbf{1}_m \right) \right\|^2 \right]$$

$$= \mathbb{E}_{\boldsymbol{\Xi}^{(k)}} \left[ \sum_{j=1}^{m} \left( \theta_j^{(k)} - \bar{\theta}_j^{(k)} \mathbf{1}_m \right)^T \left( \mathbf{P}^{(k)} \right)^T \mathbf{P}^{(k)} \left( \theta_j^{(k)} - \bar{\theta}_j^{(k)} \mathbf{1}_m \right) \right]$$

$$= \mathbb{E}_{\boldsymbol{\Xi}^{(k-1)}} \left[ \sum_{j=1}^{m} \left( \theta_j^{(k)} - \bar{\theta}_j^{(k)} \mathbf{1}_m \right)^T \mathbb{E}_{\hat{\mathbf{V}}^{(k)}} \left[ \mathbf{P}^{(k)} \mathbf{P}^{(k)} \right] \left( \theta_j^{(k)} - \bar{\theta}_j^{(k)} \mathbf{1}_m \right) \right]$$

$$= \mathbb{E}_{\boldsymbol{\Xi}^{(k-1)}} \left[ \sum_{j=1}^{m} \left( \theta_j^{(k)} - \bar{\theta}_j^{(k)} \mathbf{1}_m \right)^T \tilde{\mathbf{R}}^{(k)} \left( \theta_j^{(k)} - \bar{\theta}_j^{(k)} \mathbf{1}_m \right) \right]$$

$$\leq \mathbb{E}_{\mathbf{\Xi}^{(k-1)}}\left[\sum_{j=1}^{m}\tilde{\rho}^{(k)}\left\|\theta_j^{(k)}-\bar{\theta}_j^{(k)}\mathbf{1}_m\right\|^2\right]\leq\tilde{\rho}^{(k)}\mathbb{E}_{\mathbf{\Xi}^{(k-1)}}\left[\left\|\mathbf{\Theta}^{(k)}-\mathbf{1}_m\bar{\theta}^{(k)}\right\|^2\right].$$

Note how we used the double stochasticity of $\mathbf{P}^{(k)}$ in the second line.

# F PROOFS OF MAIN RESULTS

## F.1 PROOF OF LEMMA 4.7

Using Lemma D.1-(b) on $\bar{\theta}^{(k)}$, the average model parameters at iteration $k$, we get

$$\left\|\nabla F_i\left(\bar{\theta}^{(k)}\right)\right\|^2\leq 2\left(\beta_i^2\left\|\bar{\theta}^{(k)}-\theta^\star\right\|^2+\delta_i^2\right). \tag{13}$$

Now, if $0<\alpha^{(k)}\leq\frac{2}{\mu+\beta}$, we can write

$$\left\|\bar{\theta}^{(k+1)}-\theta^\star\right\|^2=\left\|\bar{\theta}^{(k)}-\alpha^{(k)}\overline{\mathbf{g}v}^{(k)}-\theta^\star\right\|^2$$

$$=\left\|\bar{\theta}^{(k)}-\alpha^{(k)}\overline{\nabla v}^{(k)}-\theta^\star\right\|^2-2\left\langle\bar{\theta}^{(k)}-\alpha^{(k)}\overline{\nabla v}^{(k)}-\theta^\star,\alpha^{(k)}\overline{\epsilon v}^{(k)}\right\rangle+\left(\alpha^{(k)}\right)^2\left\|\overline{\epsilon v}^{(k)}\right\|^2$$

$$\leq\left(1+\mu\alpha^{(k)}\right)\left\|\bar{\theta}^{(k)}-\alpha^{(k)}\nabla F\left(\bar{\theta}^{(k)}\right)-\theta^\star\right\|^2$$

$$+\left(1+\frac{1}{\mu\alpha^{(k)}}\right)\frac{\left(\alpha^{(k)}\right)^2}{m}\sum_{i=1}^{m}\left\|\nabla F_i\left(\bar{\theta}^{(k)}\right)-\nabla F_i\left(\theta_i^{(k)}\right)v_i^{(k)}\right\|^2$$

$$-2\left\langle\bar{\theta}^{(k)}-\alpha^{(k)}\overline{\nabla v}^{(k)}-\theta^\star,\alpha^{(k)}\overline{\epsilon v}^{(k)}\right\rangle+\left(\alpha^{(k)}\right)^2\left\|\overline{\epsilon v}^{(k)}\right\|^2$$

$$\leq\left(1+\mu\alpha^{(k)}\right)\left(1-\mu\alpha^{(k)}\right)^2\left\|\bar{\theta}^{(k)}-\theta^\star\right\|^2+\frac{\alpha^{(k)}}{m\mu}\left(1+\mu\alpha^{(k)}\right)\sum_{\substack{i=1\\v_i^{(k)}=1}}^{m}\beta_i^2\left\|\bar{\theta}^{(k)}-\theta_i^{(k)}\right\|^2$$

$$+\frac{2\alpha^{(k)}}{m\mu}\left(1+\mu\alpha^{(k)}\right)\sum_{\substack{i=1\\v_i^{(k)}=0}}^{m}\left(\beta_i^2\left\|\bar{\theta}^{(k)}-\theta^\star\right\|^2+\delta_i^2\right)$$

$$-2\left\langle\bar{\theta}^{(k)}-\alpha^{(k)}\overline{\nabla v}^{(k)}-\theta^\star,\alpha^{(k)}\overline{\epsilon v}^{(k)}\right\rangle+\left(\alpha^{(k)}\right)^2\left\|\overline{\epsilon v}^{(k)}\right\|^2$$

$$=\left[\left(1+\mu\alpha^{(k)}\right)\left(1-\mu\alpha^{(k)}\right)^2+\frac{2\alpha^{(k)}}{m\mu}\left(1+\mu\alpha^{(k)}\right)\sum_{i=1}^{m}\beta_i^2\left(1-v_i^{(k)}\right)\right]\left\|\bar{\theta}^{(k)}-\theta^\star\right\|^2$$

$$+\frac{\alpha^{(k)}}{m\mu}\left(1+\mu\alpha^{(k)}\right)\sum_{i=1}^{m}\beta_i^2\left\|\theta_i^{(k)}-\bar{\theta}^{(k)}\right\|^2v_i^{(k)}+\frac{2\alpha^{(k)}}{m\mu}\left(1+\mu\alpha^{(k)}\right)\sum_{i=1}^{m}\delta_i^2\left(1-v_i^{(k)}\right)$$

$$-2\left\langle\bar{\theta}^{(k)}-\alpha^{(k)}\overline{\nabla v}^{(k)}-\theta^\star,\alpha^{(k)}\overline{\epsilon v}^{(k)}\right\rangle+\left(\alpha^{(k)}\right)^2\left\|\overline{\epsilon v}^{(k)}\right\|^2,$$

in which the relationship in first four lines follow from (i) Eq. 5, (ii) $\mathbf{g}_i^{(k)}=\nabla_i^{(k)}+\epsilon_i^{(k)}$ for all $i\in\mathcal{M}$, (iii) Young's inequality, (iv) Lemma 10 in Qu & Li (2017), Assumption 4.1-(a) and Eq. 13. Next, we take the expected value of the above inequality and use Definition 4.6, Assumption 4.3 and Lemma D.2 to get

$$\mathbb{E}_{\mathbf{\Xi}^{(k)}}\left[\left\|\bar{\theta}^{(k+1)}-\theta^\star\right\|^2\right]\leq\mathbb{E}_{\mathbf{v}^{(k)}}\left[\left(1+\mu\alpha^{(k)}\right)\left(1-\mu\alpha^{(k)}\right)^2\right.$$

$$\left.+\frac{2\alpha^{(k)}}{m\mu}\left(1+\mu\alpha^{(k)}\right)\sum_{i=1}^{m}\beta_i^2\left(1-v_i^{(k)}\right)\right]\mathbb{E}_{\mathbf{\Xi}^{(k-1)}}\left[\left\|\bar{\theta}^{(k)}-\theta^\star\right\|^2\right]$$

$$+ \frac{\alpha^{(k)}}{m\mu}\left(1 + \mu\alpha^{(k)}\right)\sum_{i=1}^{m}\beta_i^2 \mathbb{E}_{\boldsymbol{\Xi}^{(k-1)}}\left[\left\|\theta_i^{(k)} - \bar{\theta}^{(k)}\right\|^2\right]\mathbb{E}_{v_i^{(k)}}\left[v_i^{(k)}\right]$$

$$+ \frac{2\alpha^{(k)}}{m\mu}\left(1 + \mu\alpha^{(k)}\right)\sum_{i=1}^{m}\delta_i^2 \mathbb{E}_{v_i^{(k)}}\left[1 - v_i^{(k)}\right]$$

$$- 2\mathbb{E}_{\boldsymbol{\Xi}^{(k-1)}\cup\mathbf{V}^{(k)}}\left[\left\langle \bar{\theta}^{(k)} - \alpha^{(k)}\overline{\nabla v}^{(k)} - \theta^\star, \alpha^{(k)}\mathbb{E}_{\mathbf{E}^{(k)}}\left[\overline{\epsilon v}^{(k)}\right]\right\rangle\right] + \left(\alpha^{(k)}\right)^2 \mathbb{E}_{\xi^{(k)}}\left[\left\|\overline{\epsilon v}^{(k)}\right\|^2\right]$$

$$\leq \left[\left(1 + \mu\alpha^{(k)}\right)\left(1 - \mu\alpha^{(k)}\right)^2 + \frac{2\alpha^{(k)}}{m\mu}\left(1 + \mu\alpha^{(k)}\right)\sum_{i=1}^{m}\beta^2\left(1 - d_i^{(k)}\right)\right]$$

$$\mathbb{E}_{\boldsymbol{\Xi}^{(k-1)}}\left[\left\|\bar{\theta}^{(k)} - \theta^\star\right\|^2\right]$$

$$+ \frac{\alpha^{(k)}}{m\mu}\left(1 + \mu\alpha^{(k)}\right)\sum_{i=1}^{m}d_i^{(k)}\beta_i^2 \mathbb{E}_{\boldsymbol{\Xi}^{(k-1)}}\left[\left\|\theta_i^{(k)} - \bar{\theta}^{(k)}\right\|^2\right]$$

$$+ \frac{2\alpha^{(k)}}{m\mu}\left(1 + \mu\alpha^{(k)}\right)\sum_{i=1}^{m}\delta_i^2\left(1 - d_i^{(k)}\right) + \frac{\left(\alpha^{(k)}\right)^2 d_{\max}^{(k)}\sigma^2}{m}$$

$$- 2\mathbb{E}_{\boldsymbol{\Xi}^{(k-1)}\cup\mathbf{V}^{(k)}}\left[\left\langle \bar{\theta}^{(k)} - \alpha^{(k)}\overline{\nabla v}^{(k)} - \theta^\star, \alpha^{(k)}\mathbb{E}_{\mathbf{E}^{(k)}}\left[\overline{\epsilon v}^{(k)}\right]\right\rangle\right] + \left(\alpha^{(k)}\right)^2 \mathbb{E}_{\xi^{(k)}}\left[\left\|\overline{\epsilon v}^{(k)}\right\|^2\right]$$

$$= \left[1 - \mu\alpha^{(k)}\left(1 + \mu\alpha^{(k)} - \left(\mu\alpha^{(k)}\right)^2\right) + \frac{2\alpha^{(k)}}{\mu}\left(1 + \mu\alpha^{(k)}\right)\left(1 - d_{\min}^{(k)}\right)\beta^2\right]$$

$$\mathbb{E}_{\boldsymbol{\Xi}^{(k-1)}}\left[\left\|\bar{\theta}^{(k)} - \theta^\star\right\|^2\right]$$

$$+ \left(1 + \mu\alpha^{(k)}\right)\frac{\alpha^{(k)}d_{\max}^{(k)}\beta^2}{m\mu}\mathbb{E}_{\boldsymbol{\Xi}^{(k-1)}}\left[\left\|\boldsymbol{\Theta}^{(k)} - \mathbf{1}_m\bar{\theta}^{(k)}\right\|^2\right]$$

$$+ \frac{2\alpha^{(k)}}{\mu}\left(1 + \mu\alpha^{(k)}\right)\left(1 - d_{\min}^{(k)}\right)\delta^2 + \frac{\left(\alpha^{(k)}\right)^2 d_{\max}^{(k)}\sigma^2}{m}.$$

## F.2 PROOF OF LEMMA 4.8

Using Eqs. 3 and 5, we first expand the left-hand side norm, and then use Young's inequality to get

$$\left\|\boldsymbol{\Theta}^{(k+1)} - \mathbf{1}_m\bar{\theta}^{(k+1)}\right\|^2 = \left\|\mathbf{P}^{(k)}\boldsymbol{\Theta}^{(k)} - \mathbf{1}_m\bar{\theta}^{(k)} - \alpha^{(k)}\left(\mathbf{V}^{(k)}\mathbf{G}^{(k)} - \mathbf{1}_m\overline{\mathbf{g}v}^{(k)}\right)\right\|^2$$

$$\leq \left\|\mathbf{P}^{(k)}\boldsymbol{\Theta}^{(k)} - \mathbf{1}_m\bar{\theta}^{(k)} - \alpha^{(k)}\left(\mathbf{V}^{(k)}\nabla^{(k)} - \mathbf{1}_m\overline{\nabla v}^{(k)}\right)\right\|^2$$

$$- 2\alpha^{(k)}\left\langle \mathbf{P}^{(k)}\boldsymbol{\Theta}^{(k)} - \mathbf{1}_m\bar{\theta}^{(k)} - \alpha^{(k)}\left(\mathbf{V}^{(k)}\nabla^{(k)} - \mathbf{1}_m\overline{\nabla v}^{(k)}\right), \mathbf{V}^{(k)}\mathbf{E}^{(k)} - \mathbf{1}_m\overline{\epsilon v}^{(k)}\right\rangle$$

$$+ \left(\alpha^{(k)}\right)^2\left\|\mathbf{V}^{(k)}\mathbf{E}^{(k)} - \mathbf{1}_m\overline{\epsilon v}^{(k)}\right\|^2$$

$$\leq \left(1 + \frac{1 - \tilde{\rho}^{(k)}}{2\tilde{\rho}^{(k)}}\right)\left\|\mathbf{P}^{(k)}\boldsymbol{\Theta}^{(k)} - \mathbf{1}_m\bar{\theta}^{(k)}\right\|^2 \tag{14}$$

$$+ \left(1 + \frac{2\tilde{\rho}^{(k)}}{1 - \tilde{\rho}^{(k)}}\right)\left(\alpha^{(k)}\right)^2\left\|\mathbf{V}^{(k)}\nabla^{(k)} - \mathbf{1}_m\overline{\nabla v}^{(k)}\right\|^2$$

$$- 2\alpha^{(k)}\left\langle \mathbf{P}^{(k)}\boldsymbol{\Theta}^{(k)} - \mathbf{1}_m\bar{\theta}^{(k)} - \alpha^{(k)}\left(\mathbf{V}^{(k)}\nabla^{(k)} - \mathbf{1}_m\overline{\nabla v}^{(k)}\right),\right.$$

$$\left.\left(\mathbf{I}_m - \frac{1}{m}\mathbf{1}_m\mathbf{1}_m^T\right)\mathbf{V}^{(k)}\mathbf{E}^{(k)}\right\rangle$$

$$+ \left(\alpha^{(k)}\right)^2\left\|\mathbf{V}^{(k)}\mathbf{E}^{(k)} - \mathbf{1}_m\overline{\epsilon v}^{(k)}\right\|^2.$$

Next, we take the expected value of the above inequality and use Lemmas D.2, D.3 and D.4-(c) to get

$$
\mathbb{E}_{\boldsymbol{\Xi}^{(k)}}\left[\left\|\boldsymbol{\Theta}^{(k+1)} - \mathbf{1}_m\bar{\theta}^{(k+1)}\right\|^2\right] \leq \frac{1+\tilde{\rho}^{(k)}}{2\tilde{\rho}^{(k)}}\tilde{\rho}^{(k)}\mathbb{E}_{\boldsymbol{\Xi}^{(k-1)}}\left[\left\|\boldsymbol{\Theta}^{(k)} - \mathbf{1}_m\bar{\theta}^{(k)}\right\|^2\right]
$$

$$
+3\frac{1+\tilde{\rho}^{(k)}}{1-\tilde{\rho}^{(k)}}d_{\max}^{(k)}\left(\alpha^{(k)}\right)^2\left(m\delta^2 + \left(\zeta^2 + 2\beta^2\right)\mathbb{E}_{\boldsymbol{\Xi}^{(k-1)}}\left[\left\|\boldsymbol{\Theta}^{(k)} - \mathbf{1}_m\bar{\theta}^{(k)}\right\|^2\right]\right.
$$

$$
\left. + m\left(\zeta^2 + 2\beta^2\left(1 - d_{\min}^{(k)}\right)\right)\mathbb{E}_{\boldsymbol{\Xi}^{(k-1)}}\left[\left\|\bar{\theta}^{(k)} - \theta^\star\right\|^2\right]\right)
$$

$$
-2\alpha^{(k)}\mathbb{E}_{\boldsymbol{\Xi}^{(k)}\setminus\mathbf{E}^{(k)}}\left[\left\langle \mathbf{P}^{(k)}\boldsymbol{\Theta}^{(k)} - \mathbf{1}_m\bar{\theta}^{(k)} - \alpha^{(k)}\left(\mathbf{V}^{(k)}\nabla^{(k)} - \mathbf{1}_m\overline{\nabla v}^{(k)}\right),\right.\right.
$$

$$
\left.\left.\left(\mathbf{I}_m - \frac{1}{m}\mathbf{1}_m\mathbf{1}_m^T\right)\mathbf{V}^{(k)}\mathbb{E}_{\mathbf{E}^{(k)}}\left[\mathbf{E}^{(k)}\right]\right\rangle\right]
$$

$$
+ m\left(\alpha^{(k)}\right)^2 d_{\max}^{(k)}\sigma^2
$$

$$
\leq \left[\frac{1+\tilde{\rho}^{(k)}}{2} + 3\frac{1+\tilde{\rho}^{(k)}}{1-\tilde{\rho}^{(k)}}d_{\max}^{(k)}\left(\alpha^{(k)}\right)^2\left(\zeta^2 + 2\beta^2\right)\right]\mathbb{E}_{\boldsymbol{\Xi}^{(k-1)}}\left[\left\|\boldsymbol{\Theta}^{(k)} - \mathbf{1}_m\bar{\theta}^{(k)}\right\|^2\right]
$$

$$
+ 3\frac{1+\tilde{\rho}^{(k)}}{1-\tilde{\rho}^{(k)}}md_{\max}^{(k)}\left(\alpha^{(k)}\right)^2\left(\zeta^2 + 2\beta^2\left(1 - d_{\min}^{(k)}\right)\right)\mathbb{E}_{\boldsymbol{\Xi}^{(k-1)}}\left[\left\|\bar{\theta}^{(k)} - \theta^\star\right\|^2\right]
$$

$$
+ m\left(\alpha^{(k)}\right)^2 d_{\max}^{(k)}\left(3\frac{1+\tilde{\rho}^{(k)}}{1-\tilde{\rho}^{(k)}}\delta^2 + \sigma^2\right).
$$

### F.3 PROOF OF PROPOSITION 4.10

**Step 1: Setting up the proof.** We want to find the conditions under which we will have $\rho\left(\boldsymbol{\Phi}^{(k)}\right) < 1$. As we have $\boldsymbol{\Phi}^{(k)} = [\phi_{ij}]_{1\leq i,j\leq 2}$ and $\rho\left(\boldsymbol{\Phi}^{(k)}\right) = \max\left\{\left|\lambda_1^{(k)}\right|, \left|\lambda_2^{(k)}\right|\right\}$ where $\lambda_i^{(k)}$ are the eigenvalues of the matrix $\boldsymbol{\Phi}^{(k)}$ for $i = 1, 2$, we need to show that $\max\left\{\left|\lambda_1^{(k)}\right|, \left|\lambda_2^{(k)}\right|\right\} < 1$. Therefore, we first write the eigenvalue equation of the matrix as

$$
\left(\lambda - \phi_{11}^{(k)}\right)\left(\lambda - \phi_{22}^{(k)}\right) - \phi_{12}^{(k)}\phi_{21}^{(k)} = 0 \quad \Rightarrow \quad \lambda^2 - \left(\phi_{11}^{(k)} + \phi_{22}^{(k)}\right)\lambda + \phi_{11}^{(k)}\phi_{22}^{(k)} - \phi_{12}^{(k)}\phi_{21}^{(k)} = 0.
$$

Since this is a quadratic equation in the form of $a\lambda^2 + b\lambda + c = 0$, we know that if $b < 0$, and $a, c > 0$ and the determinant is positive, we will have $\max\left\{\left|\lambda_1^{(k)}\right|, \left|\lambda_2^{(k)}\right|\right\} = \frac{-b+\sqrt{b^2-4ac}}{2a}$. Therefore, we solve for $\frac{-b+\sqrt{b^2-4ac}}{2a} < 1$ as follows

$$
\sqrt{b^2 - 4ac} < b + 2a \quad \Rightarrow \quad 4a\left(b + c\right) + 4a^2 > 0 \quad \Rightarrow \quad a + b + c > 0.
$$

Now, rewriting the above inequality in terms of the actual coefficients, we get

$$
1 - \phi_{11}^{(k)} - \phi_{22}^{(k)} + \phi_{11}^{(k)}\phi_{22}^{(k)} - \phi_{12}^{(k)}\phi_{21}^{(k)} > 0 \quad \Rightarrow \quad \left(1 - \phi_{11}^{(k)}\right)\left(1 - \phi_{22}^{(k)}\right) > \phi_{12}^{(k)}\phi_{21}^{(k)}. \tag{15}
$$

Furthermore, note that $a = 1 > 0$, $b = -\left(\phi_{11}^{(k)} + \phi_{22}^{(k)}\right) < 0$ and $b^2 - 4ac = \left(\phi_{11}^{(k)} - \phi_{22}^{(k)}\right)^2 + 4\phi_{12}^{(k)}\phi_{21}^{(k)} > 0$ hold by definition, so we only need to check for

$$
c = \phi_{11}^{(k)}\phi_{22}^{(k)} - \phi_{12}^{(k)}\phi_{21}^{(k)} > 0. \tag{16}
$$

Eqs. 15 and 16 lay out the necessary conditions in order to get $\rho\left(\boldsymbol{\Phi}^{(k)}\right) < 1$.

**Step 2: Simplifying the conditions.** Starting off with the more important of the two, we first solve for Eq. 15. In order to simplify this inequality, we choose to have (i) $0 < \phi_{11}^{(k)} \leq 1$ and (ii)

$0 < \phi_{22}^{(k)} \leq 1$ for the main diagonal entries. For $\phi_{11}^{(k)}$ as defined in Lemma 4.7, we have

$$\phi_{11}^{(k)} \leq 1 \quad \Rightarrow \quad \frac{1 + \mu\alpha^{(k)} - \left(\mu\alpha^{(k)}\right)^2}{1 + \mu\alpha^{(k)}} \geq \frac{2\beta^2}{\mu^2}\left(1 - d_{\min}^{(k)}\right). \tag{17}$$

To better characterize the condition on $\alpha^{(k)}$ based on the above inequality, we put the following constraints on $d_{\min}^{(k)}$ and $\alpha^{(k)}$ to get

$$\text{Constraints 1: } d_{\min}^{(k)} \geq \frac{1}{\Gamma_0^{(k)}}, \qquad \alpha^{(k)} \leq \frac{\Gamma_1^{(k)}}{\mu}$$

$$\Rightarrow \quad \mu\alpha^{(k)} \geq \frac{2\beta^2}{\mu^2}\left(1 - \frac{1}{\Gamma_0^{(k)}}\right)\left(1 + \Gamma_1^{(k)}\right) + \left(\Gamma_1^{(k)}\right)^2 - 1,$$

where $\Gamma_0^{(k)} \geq 1$ and $\Gamma_1^{(k)} > 0$ are scalars. The above condition requires the learning rate $\alpha^{(k)}$ to be lower-bounded, which is something we want to avoid. Thus, if the right-hand side of the inequality is non-positive, this condition only requires us to choose a non-negative value for the learning rate, which is sensible. So, we have

$$\frac{2\beta^2}{\mu^2}\left(1 - \frac{1}{\Gamma_0^{(k)}}\right)\left(1 + \Gamma_1^{(k)}\right) + \left(\Gamma_1^{(k)}\right)^2 - 1 \leq 0$$

$$\Rightarrow \quad \left(\Gamma_1^{(k)}\right)^2 + \left(\frac{2\beta^2}{\mu^2}\left(1 - \frac{1}{\Gamma_0^{(k)}}\right)\right)\Gamma_1^{(k)} + \left(\frac{2\beta^2}{\mu^2}\left(1 - \frac{1}{\Gamma_0^{(k)}}\right) - 1\right) \leq 0$$

$$\Rightarrow \quad \left|\Gamma_1^{(k)} - \frac{\beta^2}{\mu^2}\left(1 - \frac{1}{\Gamma_0^{(k)}}\right)\right| \leq \left|\frac{\beta^2}{\mu^2}\left(1 - \frac{1}{\Gamma_0^{(k)}}\right) - 1\right|$$

$$\Rightarrow \quad \begin{cases} \frac{1}{\Gamma_0^{(k)}} \leq 1 - \frac{\mu^2}{\beta^2} : & 1 \leq \Gamma_1^{(k)} \leq \frac{2\beta^2}{\mu^2}\left(1 - \frac{1}{\Gamma_0^{(k)}}\right) - 1 \\ \frac{1}{\Gamma_0^{(k)}} \geq 1 - \frac{\mu^2}{\beta^2} : & \frac{2\beta^2}{\mu^2}\left(1 - \frac{1}{\Gamma_0^{(k)}}\right) - 1 \leq \Gamma_1^{(k)} \leq 1 \end{cases}$$

$$\Rightarrow \quad \min\left\{1, \frac{2\beta^2}{\mu^2}\left(1 - \frac{1}{\Gamma_0^{(k)}}\right) - 1\right\} \leq \Gamma_1^{(k)} \leq \max\left\{1, \frac{2\beta^2}{\mu^2}\left(1 - \frac{1}{\Gamma_0^{(k)}}\right) - 1\right\}.$$

We observe that we found a lower and upper bound for the choice of $\Gamma_1^{(k)}$. Next, in order to simplify $\phi_{11}^{(k)}$ as defined in Lemma 4.7, we can use Eq. 17 to write

$$\text{Constraint 2: } \frac{1 + \mu\alpha^{(k)} - \left(\mu\alpha^{(k)}\right)^2}{1 + \mu\alpha^{(k)}} \geq \frac{2\beta^2}{\mu^2}\left(1 - \frac{1}{\Gamma_0^{(k)}}\right)\Gamma_2^{(k)},$$

in which $\Gamma_2^{(k)} \geq 1$ ensures that the constraint is satisfied, since we solved for Eq. 17 and found the conditions on $\alpha^{(k)}$, $\Gamma_0^{(k)}$ and $\Gamma_1^{(k)}$ to do so. Hence, for the bounds defined in Lemma 4.7, we get

$$\phi_{11}^{(k)} \leq 1 - \frac{2}{\mu}\left(1 + \Gamma_1^{(k)}\right)\left(\Gamma_2^{(k)} - 1\right)\left(1 - \frac{1}{\Gamma_0^{(k)}}\right)\beta^2\alpha^{(k)}, \qquad \phi_{12}^{(k)} \leq \frac{\left(1 + \Gamma_1^{(k)}\right)d_{\max}^{(k)}\beta^2}{m\mu}\alpha^{(k)},$$

$$\psi_1^{(k)} \leq \frac{2\alpha^{(k)}}{\mu}\left(1 + \Gamma_1^{(k)}\right)\left(1 - d_{\min}^{(k)}\right)\delta^2 + \frac{\left(\alpha^{(k)}\right)^2 d_{\max}^{(k)}\sigma^2}{m},$$

and there are no changes to the upper bounds of $\phi_{21}^{(k)}$, $\phi_{22}^{(k)}$ and $\psi_2^{(k)}$, which were defined in Lemma 4.8. Note that matrix $\mathbf{\Phi}^{(k)}$ and vector $\mathbf{\Psi}^{(k)}$ in Eq. 7 were used as upper bounds, therefore we can always replace their values with new upper bounds for them. Furthermore, note that in $\psi_1^{(k)}$, the term $d_{\min}^{(k)}$ was intentionally not interchanged with its lower bound. Consequently, with this new value for $\phi_{11}^{(k)}$, we continue as

$$\phi_{11}^{(k)} > 0 \quad \Rightarrow \quad \alpha^{(k)} < \frac{\mu}{2\left(1 + \Gamma_1^{(k)}\right)\left(\Gamma_2^{(k)} - 1\right)\left(1 - \frac{1}{\Gamma_0^{(k)}}\right)\beta^2}.$$

Finally, we check the next conditions on $\phi_{22}^{(k)}$ defined in Lemma 4.8, i.e., $0 < \phi_{22}^{(k)} \leq 1$. Note that for $\phi_{11}^{(k)}$, $\phi_{12}^{(k)}$ and $\phi_{21}^{(k)}$ the lower bound is 0, but for $\phi_{22}^{(k)}$ it is $\frac{1+\tilde{\rho}^{(k)}}{2}$. Therefore, the lower-bound condition of $\phi_{22}^{(k)} > 0$ is already met. For the upper-bound condition $\phi_{22}^{(k)} \leq 1$, noting that we have $\frac{3+\tilde{\rho}^{(k)}}{4} < 1$, we can write $\phi_{22}^{(k)} \leq \frac{3+\tilde{\rho}^{(k)}}{4}$ to enforce this constraint. We have

$$
\frac{1+\tilde{\rho}^{(k)}}{2} \leq \phi_{22}^{(k)} \leq \frac{3+\tilde{\rho}^{(k)}}{4} \quad \Rightarrow \quad 0 \leq \alpha^{(k)} \leq \frac{1}{2\sqrt{3d_{\max}^{(k)}}} \frac{1-\tilde{\rho}^{(k)}}{\sqrt{1+\tilde{\rho}^{(k)}}} \frac{1}{\sqrt{\zeta^2 + 2\beta^2}}
$$

**Step 3: Determining the constraints.** Now that we have made sure that (i) $0 < \phi_{11}^{(k)} \leq 1$ and (ii) $0 < \phi_{22}^{(k)} \leq 1$ in the previous step, we can continue to solve Eq. 15. For the left-hand side of the inequality, we have

$$
\left(1 - \phi_{11}^{(k)}\right)\left(1 - \phi_{22}^{(k)}\right) = \left[\frac{2}{\mu}\left(1 + \Gamma_1^{(k)}\right)\left(\Gamma_2^{(k)} - 1\right)\left(1 - \frac{1}{\Gamma_0^{(k)}}\right)\beta^2\alpha^{(k)}\right]\left(1 - \phi_{22}^{(k)}\right)
$$
$$
\geq \left[\frac{2}{\mu}\left(1 + \Gamma_1^{(k)}\right)\left(\Gamma_2^{(k)} - 1\right)\left(1 - \frac{1}{\Gamma_0^{(k)}}\right)\beta^2\alpha^{(k)}\right]\frac{1-\tilde{\rho}^{(k)}}{4}.
$$

Now, putting this back to Eq. 15, we get

$$
\left[\frac{2}{\mu}\left(1 + \Gamma_1^{(k)}\right)\left(\Gamma_2^{(k)} - 1\right)\left(1 - \frac{1}{\Gamma_0^{(k)}}\right)\beta^2\alpha^{(k)}\right]\frac{1-\tilde{\rho}^{(k)}}{4} > \phi_{12}^{(k)}\phi_{21}^{(k)}
$$

$$
\Rightarrow \quad \left[\frac{\left(1 + \Gamma_1^{(k)}\right)d_{\max}^{(k)}\beta^2}{m\mu}\alpha^{(k)}\right]\left[3md_{\max}^{(k)}\frac{1+\tilde{\rho}^{(k)}}{1-\tilde{\rho}^{(k)}}\left(\zeta^2 + 2\beta^2\left(1 - d_{\min}^{(k)}\right)\right)\left(\alpha^{(k)}\right)^2\right]
$$
$$
< \left[\frac{2}{\mu}\left(1 + \Gamma_1^{(k)}\right)\left(\Gamma_2^{(k)} - 1\right)\left(1 - \frac{1}{\Gamma_0^{(k)}}\right)\beta^2\alpha^{(k)}\right]\frac{1-\tilde{\rho}^{(k)}}{4}
$$

$$
\Rightarrow \quad \alpha^{(k)} < \frac{\sqrt{\Gamma_2^{(k)} - 1}\sqrt{1 - \frac{1}{\Gamma_0^{(k)}}}\left(1 - \tilde{\rho}^{(k)}\right)}{\sqrt{6}d_{\max}^{(k)}\sqrt{1+\tilde{\rho}^{(k)}}\sqrt{\zeta^2 + 2\beta^2\left(1 - d_{\min}^{(k)}\right)}}.
$$

Finally, we solve for Eq. 16. Noting that by solving Eq. 15 we made sure that $1 - \phi_{11}^{(k)} - \phi_{22}^{(k)} + \phi_{11}^{(k)}\phi_{22}^{(k)} - \phi_{12}^{(k)}\phi_{21}^{(k)} > 0$, we can write

$$
c > 0 \quad \Rightarrow \quad \phi_{11}^{(k)}\phi_{22}^{(k)} - \phi_{12}^{(k)}\phi_{21}^{(k)} > 0 \quad \Rightarrow \quad \phi_{11}^{(k)} + \phi_{22}^{(k)} - 1 > 0
$$
$$
\Rightarrow \quad 1 - \frac{2}{\mu}\left(1 + \Gamma_1^{(k)}\right)\left(\Gamma_2^{(k)} - 1\right)\left(1 - \frac{1}{\Gamma_0^{(k)}}\right)\beta^2\alpha^{(k)} + \frac{1+\tilde{\rho}^{(k)}}{2} - 1 > 0
$$
$$
\Rightarrow \quad \alpha^{(k)} < \frac{\mu\left(1 + \tilde{\rho}^{(k)}\right)}{4\left(1 + \Gamma_1^{(k)}\right)\left(\Gamma_2^{(k)} - 1\right)\left(1 - \frac{1}{\Gamma_0^{(k)}}\right)\beta^2}.
$$

**Step 4: Putting all the constraints together.** Reviewing all the constraints on $\alpha^{(k)}$ from the beginning of this appendix, we can collect all of the constraints together and simplify them as

$$\alpha^{(k)} < \min\left\{ \frac{\Gamma_1^{(k)}}{\mu}, \frac{1}{2\sqrt{3d_{\max}^{(k)}}}\frac{1-\tilde{\rho}^{(k)}}{\sqrt{1+\tilde{\rho}^{(k)}}}\frac{1}{\sqrt{\zeta^2+2\beta^2}}, \frac{\mu}{2\left(1+\Gamma_1^{(k)}\right)\left(\Gamma_2^{(k)}-1\right)\left(1-\frac{1}{\Gamma_0^{(k)}}\right)\beta^2}, \right.$$

$$\frac{\mu\left(1+\tilde{\rho}^{(k)}\right)}{4\left(1+\Gamma_1^{(k)}\right)\left(\Gamma_2^{(k)}-1\right)\left(1-\frac{1}{\Gamma_0^{(k)}}\right)\beta^2},$$

$$\left. \frac{\sqrt{\Gamma_2^{(k)}-1}\sqrt{1-\frac{1}{\Gamma_0^{(k)}}}\left(1-\tilde{\rho}^{(k)}\right)}{\sqrt{6}d_{\max}^{(k)}\sqrt{1+\tilde{\rho}^{(k)}}\sqrt{\zeta^2+2\beta^2\left(1-d_{\min}^{(k)}\right)}} \right\}$$

$$= \min\left\{ \frac{\Gamma_1^{(k)}}{\mu}, \frac{1}{2\sqrt{3d_{\max}^{(k)}}}\frac{1-\tilde{\rho}^{(k)}}{\sqrt{1+\tilde{\rho}^{(k)}}}\frac{1}{\sqrt{\zeta^2+2\beta^2}}, \frac{\mu\left(1+\tilde{\rho}^{(k)}\right)}{4\left(1+\Gamma_1^{(k)}\right)\left(\Gamma_2^{(k)}-1\right)\left(1-\frac{1}{\Gamma_0^{(k)}}\right)\beta^2}, \right.$$

$$\left. \frac{\sqrt{\Gamma_2^{(k)}-1}\sqrt{1-\frac{1}{\Gamma_0^{(k)}}}\left(1-\tilde{\rho}^{(k)}\right)}{\sqrt{6}d_{\max}^{(k)}\sqrt{1+\tilde{\rho}^{(k)}}\sqrt{\zeta^2+2\beta^2\left(1-d_{\min}^{(k)}\right)}} \right\},$$

while satisfying

$$\Gamma_0^{(k)} \geq 1, \qquad \Gamma_2^{(k)} > 1,$$

$$\max\left\{0, \min\left\{1, \frac{2\beta^2}{\mu^2}\left(1-\frac{1}{\Gamma_0^{(k)}}\right)-1\right\}\right\} \leq \Gamma_1^{(k)} \leq \max\left\{1, \frac{2\beta^2}{\mu^2}\left(1-\frac{1}{\Gamma_0^{(k)}}\right)-1\right\}. \tag{18}$$

Note that one of the terms in the above minimization function was trivially removed since $\frac{1+\tilde{\rho}^{(k)}}{2} < 1$. In order to simply the condition on $\alpha^{(k)}$ further, we take the minimum of these terms with respect to each variable separately to get

$$\alpha^{(k)} < \min\left\{ \min_{\Gamma_1^{(k)}}\left\{ \frac{\Gamma_1^{(k)}}{\mu}, \min_{\Gamma_2^{(k)},\Gamma_0^{(k)}}\left\{ \frac{\mu\left(1+\tilde{\rho}^{(k)}\right)}{4\left(1+\Gamma_1^{(k)}\right)\left(\Gamma_2^{(k)}-1\right)\left(1-\frac{1}{\Gamma_0^{(k)}}\right)\beta^2}, \right. \right. \right.$$

$$\left. \left. \left. \frac{\sqrt{\Gamma_2^{(k)}-1}\sqrt{1-\frac{1}{\Gamma_0^{(k)}}}\left(1-\tilde{\rho}^{(k)}\right)}{\sqrt{6}d_{\max}^{(k)}\sqrt{1+\tilde{\rho}^{(k)}}\sqrt{\zeta^2+2\beta^2\left(1-d_{\min}^{(k)}\right)}} \right\} \right\}, \right. \tag{19}$$

$$\left. \frac{1}{2\sqrt{3d_{\max}^{(k)}}}\frac{1-\tilde{\rho}^{(k)}}{\sqrt{1+\tilde{\rho}^{(k)}}}\frac{1}{\sqrt{\zeta^2+2\beta^2}} \right\}.$$

Solving for the inner minimization in Eq. 19 first using $\Gamma_2^{(k)}$ by defining $c_1^{(k)} = \frac{\sqrt{1-\frac{1}{\Gamma_0^{(k)}}}\left(1-\tilde{\rho}^{(k)}\right)}{\sqrt{6}d_{\max}^{(k)}\sqrt{1+\tilde{\rho}^{(k)}}\sqrt{\zeta^2+2\beta^2\left(1-d_{\min}^{(k)}\right)}}$ and $c_2^{(k)} = \frac{4\left(1+\Gamma_1^{(k)}\right)\left(1-\frac{1}{\Gamma_0^{(k)}}\right)\beta^2}{\mu\left(1+\tilde{\rho}^{(k)}\right)}$, we have

$$\begin{cases} c_1^{(k)}\sqrt{\Gamma_2^{(k)}-1} \leq \frac{1}{c_2^{(k)}\left(\Gamma_2^{(k)}-1\right)}; & 1 < \Gamma_2^{(k)} \leq \Gamma_2^{\star(k)} \\ \frac{1}{c_2^{(k)}\left(\Gamma_2^{(k)}-1\right)} \leq c_1^{(k)}\sqrt{\Gamma_2^{(k)}-1}; & \Gamma_2^{(k)} \geq \Gamma_2^{\star(k)} \end{cases}, \tag{20}$$

in which $\Gamma_2^{(k)} > 1$ is due to Eq. 18. We can see that in Eq. 20, one of the expressions is increasing with respect to $\Gamma_2^{(k)}$, and the other one is decreasing. Thus, we find the optimal value for it $\Gamma_2^{\star(k)}$ as

$$\sqrt{\Gamma_2^{\star(k)} - 1}^3 = \frac{1}{c_1^{(k)} c_2^{(k)}} \qquad \Rightarrow \qquad \Gamma_2^{\star(k)} = \frac{1}{\left(c_1^{(k)} c_2^{(k)}\right)^{2/3}} + 1$$

$$\Rightarrow \Gamma_2^{\star(k)} = \left( \frac{\sqrt{6} d_{\max}^{(k)} \sqrt{1 + \tilde{\rho}^{(k)}} \sqrt{\zeta^2 + 2\beta^2 \left(1 - d_{\min}^{(k)}\right)}}{\sqrt{1 - \frac{1}{\Gamma_0^{(k)}}} \left(1 - \tilde{\rho}^{(k)}\right)} \frac{\mu \left(1 + \tilde{\rho}^{(k)}\right)}{4 \left(1 + \Gamma_1^{(k)}\right) \left(1 - \frac{1}{\Gamma_0^{(k)}}\right) \beta^2} \right)^{2/3} + 1$$

$$= \frac{(1 + \tilde{\rho}^{(k)}) \sqrt[3]{3} \sqrt[3]{\zeta^2 + 2\beta^2 \left(1 - d_{\min}^{(k)}\right)}}{2 \left(1 - \frac{1}{\Gamma_0^{(k)}}\right)} \left( \frac{d_{\max}^{(k)} \mu}{(1 - \tilde{\rho}^{(k)}) \left(1 + \Gamma_1^{(k)}\right) \beta^2} \right)^{2/3} + 1$$

Choosing $\Gamma_2^{(k)} = \Gamma_2^{(k)\star}$, we get

$$\min_{\Gamma_2^{(k)}} \left\{ \frac{\mu \left(1 + \tilde{\rho}^{(k)}\right)}{4 \left(1 + \Gamma_1^{(k)}\right) \left(\Gamma_2^{(k)} - 1\right) \left(1 - d_{\min}^{(k)}\right) \beta^2}, \frac{\sqrt{\Gamma_2^{(k)} - 1} \sqrt{1 - d_{\min}^{(k)}} \left(1 - \tilde{\rho}^{(k)}\right)}{\sqrt{6} d_{\max}^{(k)} \sqrt{1 + \tilde{\rho}^{(k)}} \sqrt{\zeta^2 + 2\beta^2 \left(1 - d_{\min}^{(k)}\right)}} \right\}$$

$$= \frac{\mu \left(1 + \tilde{\rho}^{(k)}\right) \left(c_1^{(k)} c_2^{(k)}\right)^{2/3}}{4 \left(1 + \Gamma_1^{(k)}\right) \left(1 - d_{\min}^{(k)}\right) \beta^2} = \left( \frac{\left(c_1^{(k)}\right)^2}{c_2^{(k)}} \right)^{1/3}$$

$$= \left( \frac{\mu}{6(1 + \Gamma_1^{(k)}) \left(\zeta^2 + 2\beta^2 \left(1 - d_{\min}^{(k)}\right)\right)} \right)^{1/3} \left( \frac{1 - \tilde{\rho}^{(k)}}{2 d_{\max}^{(k)} \beta} \right)^{2/3}.$$

Note that by making this minimization over $\Gamma_2^{(k)}$, the dependency on $\Gamma_0^{(k)}$ was removed as well. Moving on to the second minimization in Eq. 19 using $\Gamma_1^{(k)}$, we note that finding the optimal value $\Gamma_1^{\star(k)}$ would be analytically cumbersome due to the conditions that need to be satisfied for it; First, $\Gamma_1^{(k)} > 0$, and second, $\min\left\{1, \frac{2\beta^2}{\mu^2} \left(1 - d_{\min}^{(k)}\right) - 1\right\} \leq \Gamma_1^{(k)} \leq \max\left\{1, \frac{2\beta^2}{\mu^2} \left(1 - d_{\min}^{(k)}\right) - 1\right\}$. Thus, in order to get a more intuitive upper bound for $\alpha^{(k)}$, we settle for a possible suboptimal value for it. If we choose $\Gamma_1^{(k)} = 1$ which is the only point satisfying the conditions in Eq. 18 and it also does not rely on the value of $d_{\min}^{(k)}$, we get

$$\alpha^{(k)} < \min\left\{ \frac{1}{\mu}, \left( \frac{\mu}{12 \left(\zeta^2 + 2\beta^2 \left(1 - d_{\min}^{(k)}\right)\right)} \right)^{1/3} \left( \frac{1 - \tilde{\rho}^{(k)}}{2 d_{\max}^{(k)} \beta} \right)^{2/3}, \right.$$

$$\left. \frac{1}{2\sqrt{3 d_{\max}^{(k)}}} \frac{1 - \tilde{\rho}^{(k)}}{\sqrt{1 + \tilde{\rho}^{(k)}}} \frac{1}{\sqrt{\zeta^2 + 2\beta^2}} \right\}.$$

**Step 5: Obtaining $\rho(\mathbf{\Phi}^{(k)})$.** We established $\rho(\mathbf{\Phi}^{(k)}) < 1$ in the previous steps. The last step is to determine what $\rho(\mathbf{\Phi}^{(k)})$ is. We have

$$\rho\left(\mathbf{\Phi}^{(k)}\right) = \frac{-b + \sqrt{b^2 - 4ac}}{2a} = \frac{\phi_{11}^{(k)} + \phi_{22}^{(k)} + \sqrt{\left(\phi_{11}^{(k)} + \phi_{22}^{(k)}\right)^2 - 4\left(\phi_{11}^{(k)}\phi_{22}^{(k)} - \phi_{12}^{(k)}\phi_{21}^{(k)}\right)}}{2}$$

$$= \frac{\phi_{11}^{(k)} + \phi_{22}^{(k)} + \sqrt{\left(\phi_{11}^{(k)} - \phi_{22}^{(k)}\right)^2 + 4\phi_{12}^{(k)}\phi_{21}^{(k)}}}{2}$$

$$= \frac{1}{2}\Bigg[1 - \frac{2}{\mu}\left(1 + \Gamma_1^{(k)}\right)\left(\Gamma_2^{(k)} - 1\right)\left(1 - \frac{1}{\Gamma_0^{(k)}}\right)\beta^2\alpha^{(k)} + \frac{1 + \tilde{\rho}^{(k)}}{2}$$

$$+ 3\frac{1 + \tilde{\rho}^{(k)}}{1 - \tilde{\rho}^{(k)}}d_{\max}^{(k)}\left(\alpha^{(k)}\right)^2\left(\zeta^2 + 2\beta^2\right)\Bigg]$$

$$+ \frac{1}{2}\Bigg[\left(1 - \frac{2}{\mu}\left(1 + \Gamma_1^{(k)}\right)\left(\Gamma_2^{(k)} - 1\right)\left(1 - \frac{1}{\Gamma_0^{(k)}}\right)\beta^2\alpha^{(k)}\right.$$

$$\left. - \frac{1 + \tilde{\rho}^{(k)}}{2} - 3\frac{1 + \tilde{\rho}^{(k)}}{1 - \tilde{\rho}^{(k)}}d_{\max}^{(k)}\left(\alpha^{(k)}\right)^2\left(\zeta^2 + 2\beta^2\right)\right)^2$$

$$+ 4\frac{\left(1 + \Gamma_1^{(k)}\right)d_{\max}^{(k)}\beta^2}{m\mu}\alpha^{(k)}3\frac{1 + \tilde{\rho}^{(k)}}{1 - \tilde{\rho}^{(k)}}md_{\max}^{(k)}\left(\alpha^{(k)}\right)^2\left(\zeta^2 + 2\beta^2\left(1 - d_{\min}^{(k)}\right)\right)\Bigg]^{1/2}$$

$$= \frac{3 + \tilde{\rho}^{(k)}}{4} - \frac{1}{\mu}\left(1 + \Gamma_1^{(k)}\right)\left(\Gamma_2^{(k)} - 1\right)\left(1 - \frac{1}{\Gamma_0^{(k)}}\right)\beta^2\alpha^{(k)}$$

$$+ \frac{3}{2}\frac{1 + \tilde{\rho}^{(k)}}{1 - \tilde{\rho}^{(k)}}d_{\max}^{(k)}\left(\zeta^2 + 2\beta^2\right)\left(\alpha^{(k)}\right)^2$$

$$+ \frac{1}{2}\Bigg[\left(\frac{1 - \tilde{\rho}^{(k)}}{2} - \frac{2}{\mu}\left(1 + \Gamma_1^{(k)}\right)\left(\Gamma_2^{(k)} - 1\right)\left(1 - \frac{1}{\Gamma_0^{(k)}}\right)\beta^2\alpha^{(k)}\right.$$

$$\left. - 3\frac{1 + \tilde{\rho}^{(k)}}{1 - \tilde{\rho}^{(k)}}d_{\max}^{(k)}\left(\zeta^2 + 2\beta^2\right)\left(\alpha^{(k)}\right)^2\right)^2$$

$$+ 12\frac{\left(1 + \Gamma_1^{(k)}\right)\beta^2}{\mu}\frac{1 + \tilde{\rho}^{(k)}}{1 - \tilde{\rho}^{(k)}}\left(d_{\max}^{(k)}\right)^2\left(\zeta^2 + 2\beta^2\left(1 - d_{\min}^{(k)}\right)\right)\left(\alpha^{(k)}\right)^3\Bigg]^{1/2}.$$

Therefore, $\rho(\mathbf{\Phi}^{(k)})$ follows as $\rho(\mathbf{\Phi}^{(k)}) = \frac{3 + \tilde{\rho}^{(k)}}{4} - A^{(k)}\alpha^{(k)} + B^{(k)}(\alpha^{(k)})^2 + \frac{1}{2}\sqrt{\left(\frac{1 - \tilde{\rho}^{(k)}}{2} - 2(A^{(k)}\alpha^{(k)} + B^{(k)}(\alpha^{(k)})^2)\right)^2 + C^{(k)}(\alpha^{(k)})^3}$, where $A^{(k)} = \frac{1}{\mu}(\Gamma_2^{\star(k)} - 1)(1 - \frac{1}{\Gamma_0^{(k)}})\beta^2$, $B^{(k)} = \frac{3}{2}\frac{1 + \tilde{\rho}^{(k)}}{1 - \tilde{\rho}^{(k)}}d_{\max}^{(k)}(\zeta^2 + 2\beta^2)$ and $C^{(k)} = 24\frac{\beta^2}{\mu}\frac{1 + \tilde{\rho}^{(k)}}{1 - \tilde{\rho}^{(k)}}(d_{\max}^{(k)})^2(\zeta^2 + 2\beta^2(1 - d_{\min}^{(k)}))$. The value for the constant $\Gamma_2^{\star(k)} > 1$ was given in step 4.

### F.4 PROOF OF THEOREM 4.11

Note that by the properties of spectral radius, we have that $\mathbf{\Phi}\| \cdot \| \leq \rho(\mathbf{\Phi})\| \cdot \|$. Now, using Eq. 8, we can write

$$\begin{bmatrix} \mathbb{E}_{\mathbf{\Xi}^{(K)}}\left[\|\bar{\theta}^{(K+1)} - \theta^\star\|^2\right] \\ \mathbb{E}_{\mathbf{\Xi}^{(K)}}\left[\|\mathbf{\Theta}^{(K+1)} - \mathbf{1}_m\bar{\theta}^{(K+1)}\|^2\right] \end{bmatrix} \leq \rho(\mathbf{\Phi})^{K+1}\begin{bmatrix} \|\bar{\theta}^{(0)} - \theta^\star\|^2 \\ \|\mathbf{\Theta}^{(0)} - \mathbf{1}_m\bar{\theta}^{(0)}\|^2 \end{bmatrix} + \sum_{r=1}^{K}\rho(\mathbf{\Phi})^{K-r+1}\mathbf{\Psi} + \mathbf{\Psi}.$$

We emphasize that the time index $k$ in $\mathbf{\Phi}^{(k)}$ and $\mathbf{\Psi}^{(k)}$ was dropped, since we are using a constant learning rate, and substituting the bounds for $d_{\max}^{(k)} \leq 1$ and $d_{\min}^{(k)} \geq d_{\min}$ and $\tilde{\rho}^{(k)} \leq \tilde{\rho}$. This

results in the constant matrix $\mathbf{\Phi}^{(k)} = \mathbf{\Phi}$ and the constant vector $\mathbf{\Psi}^{(k)} = \mathbf{\Psi}$. Focusing on the term $\sum_{r=1}^{K} \rho(\mathbf{\Phi})^{K-r+1}\mathbf{\Psi} + \mathbf{\Psi}$, we get

$$\sum_{r=1}^{K} \rho(\mathbf{\Phi})^{K-r+1}\mathbf{\Psi} + \mathbf{\Psi} = \sum_{r=1}^{K+1} \rho(\mathbf{\Phi})^{K-r+1}\mathbf{\Psi} = \left(\sum_{u=0}^{K} \rho(\mathbf{\Phi})^u\right)\mathbf{\Psi} \leq \left(\sum_{u=0}^{\infty} \rho(\mathbf{\Phi})^u\right)\mathbf{\Psi}$$

$$= \frac{1}{1 - \rho(\mathbf{\Phi})}\mathbf{\Psi} = \frac{1}{|1 - \rho(\mathbf{\Phi})|}\mathbf{\Psi}.$$

Putting the above inequalities together concludes the proof of Eq. 9. Finally, noting that $\rho(\mathbf{\Phi}) < 1$ following 4.10, We can let $K \to \infty$ to get Eq. 10. Note that $1 - \rho(\mathbf{\Phi}) \geq 2A\alpha$, where $A = \frac{\beta^2}{\mu}(\Gamma_2^\star - 1)(1 - \frac{1}{\Gamma_0})$ with $\Gamma_0, \Gamma_2^\star > 1$ being constant scalars defined in Appendix F.3.

We also note that Eqs. 9 and 10 are derived for the consensus error and the average model error themselves, i.e., their last iterates. As summarized in Table 1, this is an improvement over existing works with sporadic aggregations (Koloskova et al., 2020; Lian et al., 2017; Sundhar Ram et al., 2010) where only the Cesaro sums (i.e., the running averages of the iterates) of these error terms are bounded.

## G  NON-CONVEX ANALYSIS

In this appendix, we analyze the convergence of our methodology when non-convex loss functions are utilized. Our approach will be entirely different than the one done in Sec. 4, as we will be using the non-convexity assumption (Assumption 4.2) instead of strong convexity (Assumption 4.1).

We will still characterize the expected consensus error as $\mathbb{E}_{\mathbf{\Xi}^{(k)}}[\|\mathbf{\Theta}^{(k+1)} - \mathbf{1}_m\bar{\theta}^{(k+1)}\|^2]$, but contrary to what was done in Sec. 4, instead of the distance of the average model from the optimal solution, we will analyze the norm of the average model gradients $\mathbb{E}_{\mathbf{\Xi}^{(k)}}[\|\nabla F(\bar{\theta}^{(k+1)})\|^2]$. As an alternative to Lemma 4.7, we first provide an upper bound on the average model performance at each iteration for the non-convex case, i.e., $\mathbb{E}_{\mathbf{\Xi}^{(k)}}[F(\bar{\theta}^{(k+1)})]$, in Lemma G.3. Then, as an alternative to Lemma 4.8, we also calculate an upper bound on the consensus error for non-convex models, i.e., $\mathbb{E}_{\mathbf{\Xi}^{(k)}}[\|\mathbf{\Theta}^{(k+1)} - \mathbf{1}_m\bar{\theta}^{(k+1)}\|^2]$, in Lemma G.4.

We first need two preliminary Lemmas, each of which will be useful in the proof of Lemmas G.3 and G.4, respectively.

**Lemma G.1 (Gradient bounds for non-convex models)** *(See Appendix H.1) Let Assumptions 4.1-(a), 4.1-(c) and M.1 hold. The following upper bounds related to the gradient of the global loss function can be obtained in terms of the gradient norms $\mathbb{E}_{\mathbf{\Xi}^{(k-1)}}[\|\nabla F(\bar{\theta}^{(k)})\|^2]$ and the consensus error $\mathbb{E}_{\mathbf{\Xi}^{(k-1)}}[\|\mathbf{\Theta}^{(k)} - \mathbf{1}_m\bar{\theta}^{(k)}\|^2]$.*

*(a)* $\mathbb{E}_{\mathbf{\Xi}^{(k)}}[\|\nabla F(\bar{\theta}^{(k)}) - \overline{\nabla v}^{(k)}\|^2] \leq 2\zeta^2(1 - d_{\min}^{(k)})\mathbb{E}_{\mathbf{\Xi}^{(k-1)}}[\|\nabla F(\bar{\theta}^{(k)})\|^2] + \frac{\beta^2 d_{\max}^{(k)}}{m}\mathbb{E}_{\mathbf{\Xi}^{(k-1)}}[\|\mathbf{\Theta}^{(k)} - \mathbf{1}_m\bar{\theta}^{(k)}\|^2] + 2(1 - d_{\min}^{(k)})\delta^2.$

*(b)* $-\mathbb{E}_{\mathbf{\Xi}^{(k)}}[\langle\nabla F(\bar{\theta}^{(k)}), \overline{\nabla v}^{(k)}\rangle] \leq -(\frac{1}{2} - \zeta^2(1 - d_{\min}^{(k)}))\mathbb{E}_{\mathbf{\Xi}^{(k-1)}}[\|\nabla F(\bar{\theta}^{(k)})\|^2] + \frac{\beta^2 d_{\max}^{(k)}}{2m}\mathbb{E}_{\mathbf{\Xi}^{(k-1)}}[\|\mathbf{\Theta}^{(k)} - \mathbf{1}_m\bar{\theta}^{(k)}\|^2] + (1 - d_{\min}^{(k)})\delta^2.$

*(c)* $\frac{1}{2}\mathbb{E}_{\mathbf{\Xi}^{(k)}}[\|\overline{\nabla v}^{(k)}\|^2] \leq [1 + 2\zeta^2(1 - d_{\min}^{(k)})]\mathbb{E}_{\mathbf{\Xi}^{(k-1)}}[\|\nabla F(\bar{\theta}^{(k)})\|^2] + \frac{\beta^2 d_{\max}^{(k)}}{m}\mathbb{E}_{\mathbf{\Xi}^{(k-1)}}[\|\mathbf{\Theta}^{(k)} - \mathbf{1}_m\bar{\theta}^{(k)}\|^2] + 2(1 - d_{\min}^{(k)})\delta^2.$

Next, we find an upper bound on the expected deviation of the gradients from their average for non-convex models, similar to Lemma D.3 which was derived for strongly convex models.

**Lemma G.2 (Gradient deviation bound for non-convex models)** *(See Appendix H.2 for the proof.) Let Assumption 4.2 hold. For each iteration $k \geq 0$, we have the following bound on the*

*expected error of gradients from their average*

$$\mathbb{E}_{\boldsymbol{\Xi}^{(k)}}\left[\left\|\mathbf{V}^{(k)}\nabla^{(k)} - \mathbf{1}_m\overline{\nabla v}^{(k)}\right\|^2\right] \leq 8d_{\max}^{(k)}\left[\beta^2\mathbb{E}_{\boldsymbol{\Xi}^{(k-1)}}\left[\left\|\boldsymbol{\Theta}^{(k)} - \mathbf{1}_m\bar{\theta}^{(k)}\right\|^2\right]\right.$$

$$\left. + 2m\zeta^2\mathbb{E}_{\boldsymbol{\Xi}^{(k-1)}}\left[\left\|\nabla F\left(\bar{\theta}^{(k)}\right)\right\|^2\right] + 2m\delta^2\right].$$

*in which $\nabla^{(k)}$ is a matrix whose rows are comprised of the gradient vectors $\nabla F_i(\theta_i^{(k)})$, and $\overline{\nabla v}^{(k)} = \frac{1}{m}\sum_{i=1}^m \nabla F_i(\theta_i^{(k)})v_i^{(k)}$.*

Now, we can continue with our key lemmas for the non-convex case. Similar to the convex case, we first derive counterparts for average model error and consensus error, i.e., Lemmas 4.7 and 4.8 for convex models, respectively. we first provide an upper bound on the average model performance $\mathbb{E}_{\boldsymbol{\Xi}^{(k)}}[F(\bar{\theta}^{(k+1)})]$ (Lemma G.3), and also upper bound the consensus error $\mathbb{E}_{\boldsymbol{\Xi}^{(k)}}[\|\boldsymbol{\Theta}^{(k+1)} - \mathbf{1}_m\bar{\theta}^{(k+1)}\|^2]$ (Lemma G.4), at each $k$.

**Lemma G.3 (Average model performance for non-convex models)** *(See Appendix H.3 for the proof.) Let Assumptions 4.2 and 4.3 hold. For each iteration $k \geq 0$, we have the following bound on the expected average model performance:*

$$\mathbb{E}_{\boldsymbol{\Xi}^{(k)}}[F(\bar{\theta}^{(k+1)})] \leq \mathbb{E}_{\boldsymbol{\Xi}^{(k-1)}}[F(\bar{\theta}^{(k)})] - \phi_{11}^{(k)}\mathbb{E}_{\boldsymbol{\Xi}^{(k-1)}}[\|\nabla F(\bar{\theta}^{(k)})\|^2] + \phi_{12}^{(k)}\mathbb{E}_{\boldsymbol{\Xi}^{(k-1)}}[\|\boldsymbol{\Theta}^{(k)} - \mathbf{1}_m\bar{\theta}^{(k)}\|^2] + \psi_1^{(k)},$$

*where $\phi_{11}^{(k)} = \alpha^{(k)}[\frac{1}{2} - \zeta^2(1 - d_{\min}^{(k)}) - \beta(1 + 2\zeta^2(1 - d_{\min}^{(k)}))\alpha^{(k)}]$, $\phi_{12}^{(k)} = \frac{\beta^2 d_{\max}^{(k)}}{2m}\alpha^{(k)}(1 + 2\beta\alpha^{(k)})$ and $\psi_1^{(k)} = \alpha^{(k)}[(1 - d_{\min}^{(k)})(1 + 2\beta\alpha^{(k)})\delta^2 + \frac{\beta}{2}\alpha^{(k)}\frac{d_{\max}\sigma^2}{m}]$.*

In Lemma G.3, the upper bound on the expected error at iteration $k + 1$ is expressed in terms of the expected performance $\mathbb{E}_{\boldsymbol{\Xi}^{(k-1)}}[F(\bar{\theta}^{(k)})]$, the scaled gradient norms $\phi_{11}^{(k)}\mathbb{E}_{\boldsymbol{\Xi}^{(k-1)}}[\|\nabla F(\bar{\theta}^{(k)})\|^2]$, the scaled consensus error $\phi_{12}^{(k)}\mathbb{E}_{\boldsymbol{\Xi}^{(k-1)}}[\|\boldsymbol{\Theta}^{(k)} - \mathbf{1}_m\bar{\theta}^{(k)}\|^2]$ (which will be presented in Lemma G.4), and a scalar $\psi_1^{(k)}$, all at iteration $k$. We next bound the consensus error in the following lemma.

**Lemma G.4 (Consensus error for non-convex models)** *(See Appendix H.4 for the proof.) Let Assumptions 4.2, 4.3 and 4.4 hold. For each iteration $k \geq 0$, we have the following bound on the expected consensus error:*

$$\mathbb{E}_{\boldsymbol{\Xi}^{(k)}}[\|\boldsymbol{\Theta}^{(k+1)} - \mathbf{1}_m\bar{\theta}^{(k+1)}\|^2] \leq \phi_{21}^{(k)}\mathbb{E}_{\boldsymbol{\Xi}^{(k-1)}}[\|\nabla F(\bar{\theta}^{(k)})\|^2] + \phi_{22}^{(k)}\mathbb{E}_{\boldsymbol{\Xi}^{(k-1)}}[\|\boldsymbol{\Theta}^{(k)} - \mathbf{1}_m\bar{\theta}^{(k)}\|^2] + \psi_2^{(k)},$$

*where $\phi_{21}^{(k)} = 16\frac{1+\tilde{\rho}^{(k)}}{1-\tilde{\rho}^{(k)}}md_{\max}^{(k)}(\alpha^{(k)})^2\zeta^2$, $\phi_{22}^{(k)} = \frac{1+\tilde{\rho}^{(k)}}{2} + 8\frac{1+\tilde{\rho}^{(k)}}{1-\tilde{\rho}^{(k)}}d_{\max}^{(k)}(\alpha^{(k)})^2\beta^2$ and $\psi_2^{(k)} = m(\alpha^{(k)})^2d_{\max}^{(k)}(16\frac{1+\tilde{\rho}^{(k)}}{1-\tilde{\rho}^{(k)}}\delta^2 + \sigma^2)$, where $\tilde{\rho}^{(k)}$ is defined in Definition 4.5 and Lemma D.4-(c).*

From this point forward, our analysis method will differ from the one done in Sec. 4. Instead of forming an error vector like Eq. 6 and analyzing their joint behavior, we first expand the consensus error recursively to get the next lemma.

**Lemma G.5 (Explicit consensus error for non-convex models)** *(See Appendix H.5 for the proof.) Let Assumptions 4.2 and 4.4 hold. If the learning rate $\alpha^{(k)}$ satisfies $\alpha^{(k)} \leq \frac{1}{\Gamma_1^{(k)}}\frac{1-\tilde{\rho}^{(k)}}{4\beta\sqrt{d_{\max}^{(k)}(1+\tilde{\rho}^{(k)})}}$, with $\Gamma_1^{(k)} > 1$,*

*then for each iteration $k \geq 0$, we have the following bound on the expected consensus error:*

$$\mathbb{E}_{\boldsymbol{\Xi}^{(k)}}[\|\boldsymbol{\Theta}^{(k+1)} - \mathbf{1}_m\bar{\theta}^{(k+1)}\|^2] \leq \phi_{22}^{(k:0)}\|\boldsymbol{\Theta}^{(0)} - \mathbf{1}_m\bar{\theta}^{(0)}\|^2 + \sum_{r=0}^k \phi_{22}^{(k:r+1)}\phi_{21}^{(r)}\mathbb{E}_{\boldsymbol{\Xi}^{(r-1)}}[\|\nabla F(\bar{\theta}^{(r)})\|^2] + \sum_{r=0}^k \phi_{22}^{(k:r+1)}\psi_2^{(r)},$$

*where $\phi_{22}^{(k:r)} = \prod_{s=r}^k \phi_{22}^{(s)}$, and the values of $\phi_{21}^{(k)}$, $\phi_{22}^{(k)}$ and $\psi_2^{(k)}$ are given in Lemma G.4.*

Finally, we use the upper bound derived in Lemma G.5 in Lemma G.3 to derive the following proposition.

**Proposition G.6 (Explicit average model performance for non-convex models)** *(See Appendix H.6 for the proof.) Let Assumptions 4.2 and 4.3 hold. If a non-increasing learning rate is used, i.e., $\alpha^{(k+1)} \leq \alpha^{(k)}$, which also satisfies the following condition*

$$
\alpha^{(k)} < \min \left\{ \frac{\Gamma_3^{(k)}}{2\beta}, \frac{1}{\Gamma_1^{(k)}} \frac{1 - \tilde{\rho}^{(k)}}{4\beta \sqrt{d_{\max}^{(k)}(1 + \tilde{\rho}^{(k)})}}, \frac{1}{\Gamma_2^{(k)}} \frac{\frac{1}{2} - \zeta^2 \left(1 - d_{\min}^{(k)}\right)}{\beta \left(1 + 2\zeta^2 \left(1 - d_{\min}^{(k)}\right)\right)}, \right.
$$

$$
\left. \frac{\sqrt{\left(1 - \frac{1}{(\Gamma_1^{(k)})^2}\right)\left(1 - \frac{1}{\Gamma_2^{(k)}}\right)}}{4\Gamma_4^{(k)}\sqrt{1 + \Gamma_3^{(k)}}} \frac{\sqrt{\frac{1}{2} - \zeta^2 \left(1 - d_{\min}^{(k)}\right)}}{d_{\max}^{(k)} \zeta \beta} \frac{1 - \tilde{\rho}^{(k)}}{\sqrt{1 + \tilde{\rho}^{(k)}}} \right\},
$$

*then for each iteration $k \geq 0$ we have the following bound on the expected average model performance:*

$$
\mathbb{E}_{\boldsymbol{\Xi}^{(k)}}[F(\bar{\theta}^{(k+1)})] \leq F(\bar{\theta}^{(0)}) - \sum_{r=0}^{k} \alpha^{(r)} w_1^{(r)} \mathbb{E}_{\boldsymbol{\Xi}^{(r-1)}}[\|\nabla F(\bar{\theta}^{(r)})\|^2] +
$$
$$
\alpha^{(0)} w_2^{(0)} \|\boldsymbol{\Theta}^{(0)} - \mathbf{1}_m \bar{\theta}^{(0)}\|^2 + \sum_{r=0}^{k-1} \alpha^{(r)} w_2^{(r)} \psi_2^{(r)} + \sum_{r=0}^{k} \psi_1^{(r)},
$$

*in which $w_1^{(k)} = (\frac{1}{2} - \zeta^2(1 - d_{\min}^{(k)}))(1 - \frac{1}{\Gamma_2^{(k)}})(1 - \frac{1}{(\Gamma_4^{(k)})^2})$, $w_2^{(k)} = \frac{\beta^2 d_{\max}^{(k)}(1 + \Gamma_3^{(k)})}{m(1 - \tilde{\rho}^{(k)})(1 - \frac{1}{(\Gamma_1^{(k)})^2})}$, and the values of $\psi_1^{(k)}$ and $\psi_2^{(k)}$ were given in Lemmas G.3 and G.4.*

# H  PROOFS FOR NON-CONVEX ANALYSIS

## H.1  PROOF OF LEMMA G.1

(a) For this deviation term, we use Eq. 1 and triangle inequality to write

$$
\left\|\nabla F(\bar{\theta}^{(k)}) - \overline{\nabla v}^{(k)}\right\|^2 = \left\|\frac{1}{m}\sum_{i=1}^{m}\left(\nabla F_i(\bar{\theta}^{(k)}) - \nabla F_i(\theta_i^{(k)})v_i^{(k)}\right)\right\|^2
$$

$$
\leq \frac{1}{m}\sum_{i=1}^{m}\left\|\nabla F_i(\bar{\theta}^{(k)}) - \nabla F_i(\theta_i^{(k)})v_i^{(k)}\right\|^2
$$

$$
= \frac{1}{m}\sum_{\substack{i=1 \\ v_i^{(k)}=1}}^{m}\left\|\nabla F_i(\bar{\theta}^{(k)}) - \nabla F_i(\theta_i^{(k)})\right\|^2 + \frac{1}{m}\sum_{\substack{i=1 \\ v_i^{(k)}=0}}^{m}\left\|\nabla F_i(\bar{\theta}^{(k)})\right\|^2
$$

$$
\leq \frac{1}{m}\sum_{\substack{i=1 \\ v_i^{(k)}=1}}^{m}\beta_i^2\left\|\bar{\theta}^{(k)} - \theta_i^{(k)}\right\|^2 + \frac{2}{m}\sum_{\substack{i=1 \\ v_i^{(k)}=0}}^{m}\left(\delta_i^2 + \zeta_i^2\left\|\nabla F(\bar{\theta}^{(k)})\right\|^2\right)
$$

$$
= \frac{1}{m}\sum_{i=1}^{m}\beta_i^2\left\|\bar{\theta}^{(k)} - \theta_i^{(k)}\right\|^2 v_i^{(k)} + \frac{2}{m}\sum_{i=1}^{m}\left(\delta_i^2 + \zeta_i^2\left\|\nabla F(\bar{\theta}^{(k)})\right\|^2\right)\left(1 - v_i^{(k)}\right),
$$

where in the line second to last, smoothness and gradient diversity (Assumptions 4.2-(a) and 4.2-(b)) were used. Taking the expected value of the above inequality concludes the proof.

(b) Second, for this inner product term, we have

$$
-\left\langle \nabla F(\bar{\theta}^{(k)}), \overline{\nabla v}^{(k)}\right\rangle = -\left\langle \nabla F(\bar{\theta}^{(k)}), \overline{\nabla v}^{(k)} - \nabla F(\bar{\theta}^{(k)}) + \nabla F(\bar{\theta}^{(k)})\right\rangle
$$

$$
= -\left\|\nabla F(\bar{\theta}^{(k)})\right\|^2 + \left\langle \nabla F(\bar{\theta}^{(k)}), \nabla F(\bar{\theta}^{(k)}) - \overline{\nabla v}^{(k)}\right\rangle
$$

$$
\leq \frac{-1}{2}\left\|\nabla F(\bar{\theta}^{(k)})\right\|^2 + \frac{1}{2}\left\|\nabla F(\bar{\theta}^{(k)}) - \overline{\nabla v}^{(k)}\right\|^2.
$$

Now, taking the expected value of this inequality and using part (a) of this lemma, we get

$$
\begin{aligned}
-\mathbb{E}_{\boldsymbol{\Xi}^{(k)}}\left[\left\langle \nabla F(\bar{\theta}^{(k)}), \overline{\nabla v}^{(k)}\right\rangle\right] \leq &-\frac{1}{2}\mathbb{E}_{\boldsymbol{\Xi}^{(k-1)}}\left[\left\|\nabla F(\bar{\theta}^{(k)})\right\|^2\right] \\
&+\frac{\beta^2 d_{\max}^{(k)}}{2m}\mathbb{E}_{\boldsymbol{\Xi}^{(k-1)}}\left[\left\|\boldsymbol{\Theta}^{(k)}-\mathbf{1}_m\bar{\theta}^{(k)}\right\|^2\right] \\
&+\zeta^2\left(1-d_{\min}^{(k)}\right)\mathbb{E}_{\boldsymbol{\Xi}^{(k-1)}}\left[\left\|\nabla F(\bar{\theta}^{(k)})\right\|^2\right] \\
&+\left(1-d_{\min}^{(k)}\right)\delta^2 \\
\leq &-\left(\frac{1}{2}-\zeta^2\left(1-d_{\min}^{(k)}\right)\right)\mathbb{E}_{\boldsymbol{\Xi}^{(k-1)}}\left[\left\|\nabla F(\bar{\theta}^{(k)})\right\|^2\right] \\
&+\frac{\beta^2 d_{\max}^{(k)}}{2m}\mathbb{E}_{\boldsymbol{\Xi}^{(k-1)}}\left[\left\|\boldsymbol{\Theta}^{(k)}-\mathbf{1}_m\bar{\theta}^{(k)}\right\|^2\right]+\left(1-d_{\min}^{(k)}\right)\delta^2.
\end{aligned}
$$

(c) Finally, for the norm term, we have

$$
\frac{1}{2}\left\|\overline{\nabla v}^{(k)}\right\|^2 = \frac{1}{2}\left\|\overline{\nabla v}^{(k)}-\nabla F(\bar{\theta}^{(k)})+\nabla F(\bar{\theta}^{(k)})\right\|^2 \leq \left\|\nabla F(\bar{\theta}^{(k)})\right\|^2+\left\|\nabla F(\bar{\theta}^{(k)})-\overline{\nabla v}^{(k)}\right\|^2.
$$

Taking the expected value of this inequality and utilizing part (a) of this lemma

$$
\begin{aligned}
\frac{1}{2}\mathbb{E}_{\boldsymbol{\Xi}^{(k)}}\left[\left\|\overline{\nabla v}^{(k)}\right\|^2\right] \leq &\,\mathbb{E}_{\boldsymbol{\Xi}^{(k-1)}}\left[\left\|\nabla F(\bar{\theta}^{(k)})\right\|^2\right]+\frac{\beta^2 d_{\max}^{(k)}}{m}\mathbb{E}_{\boldsymbol{\Xi}^{(k-1)}}\left[\left\|\boldsymbol{\Theta}^{(k)}-\mathbf{1}_m\bar{\theta}^{(k)}\right\|^2\right] \\
&+2\zeta^2\left(1-d_{\min}^{(k)}\right)\mathbb{E}_{\boldsymbol{\Xi}^{(k-1)}}\left[\left\|\nabla F(\bar{\theta}^{(k)})\right\|^2\right]+2\left(1-d_{\min}^{(k)}\right)\delta^2 \\
\leq &\left[1+2\zeta^2\left(1-d_{\min}^{(k)}\right)\right]\mathbb{E}_{\boldsymbol{\Xi}^{(k-1)}}\left[\left\|\nabla F(\bar{\theta}^{(k)})\right\|^2\right]+\frac{\beta^2 d_{\max}^{(k)}}{m}\mathbb{E}_{\boldsymbol{\Xi}^{(k-1)}}\left[\left\|\boldsymbol{\Theta}^{(k)}-\mathbf{1}_m\bar{\theta}^{(k)}\right\|^2\right] \\
&+2\left(1-d_{\min}^{(k)}\right)\delta^2.
\end{aligned}
$$

## H.2 PROOF OF LEMMA G.2

We have using the definition of Frobenius norms on matrices and triangle inequality that

$$
\begin{aligned}
\left\|\mathbf{V}^{(k)}\nabla^{(k)}-\mathbf{1}_m\overline{\nabla v}^{(k)}\right\|^2 &= \sum_{i=1}^{m}\left\|\nabla F_i(\theta_i^{(k)})v_i^{(k)}-\overline{\nabla v}^{(k)}\right\|^2 \\
&= \sum_{i=1}^{m}\left\|\frac{1}{m}\sum_{j=1}^{m}\left(\nabla F_i(\theta_i^{(k)})v_i^{(k)}-\nabla F_j(\theta_j^{(k)})v_j^{(k)}\right)\right\|^2 \\
&\leq \frac{1}{m}\sum_{i=1}^{m}\sum_{j=1}^{m}\left\|\nabla F_i(\theta_i^{(k)})v_i^{(k)}-\nabla F_j(\theta_j^{(k)})v_j^{(k)}\right\|^2 \\
&\leq \frac{1}{m}\sum_{i=1}^{m}\sum_{j=1}^{m}\left\|\nabla F_i(\theta_i^{(k)})v_i^{(k)}-\nabla F_i(\bar{\theta}^{(k)})v_i^{(k)}+\nabla F_i(\bar{\theta}^{(k)})v_i^{(k)}-\nabla F_j(\bar{\theta}^{(k)})v_j^{(k)}\right. \\
&\qquad\qquad\qquad\qquad\left.+\nabla F_j(\bar{\theta}^{(k)})v_j^{(k)}-\nabla F_j(\theta_j^{(k)})v_j^{(k)}\right\|^2 \\
&\leq \frac{4}{m}\sum_{i=1}^{m}\sum_{j=1}^{m}\left\|\nabla F_i(\theta_i^{(k)})-\nabla F_i(\bar{\theta}^{(k)})\right\|^2 v_i^{(k)}+\left\|\nabla F_i(\bar{\theta}^{(k)})\right\|^2 v_i^{(k)}+\left\|\nabla F_j(\bar{\theta}^{(k)})\right\|^2 v_j^{(k)} \\
&\qquad\qquad\qquad\qquad +\left\|\nabla F_j(\bar{\theta}^{(k)})-\nabla F_j(\theta_j^{(k)})\right\|^2 v_j^{(k)}
\end{aligned}
$$

$$\leq \frac{4}{m} \sum_{i=1}^{m} \sum_{j=1}^{m} \beta_i^2 \left\| \theta_i^{(k)} - \bar{\theta}^{(k)} \right\|^2 v_i^{(k)} + 2 \left( \delta_i^2 + \zeta_i^2 \left\| \nabla F\left(\bar{\theta}^{(k)}\right) \right\|^2 \right) v_i^{(k)}$$

$$+ 2 \left( \delta_j^2 + \zeta_j^2 \left\| \nabla F\left(\bar{\theta}^{(k)}\right) \right\|^2 \right) v_j^{(k)} + \beta_j^2 \left\| \bar{\theta}^{(k)} - \theta_j^{(k)} \right\|^2 v_j^{(k)}$$

$$= 4 \left[ \sum_{i=1}^{m} \beta_i^2 \left\| \theta_i^{(k)} - \bar{\theta}^{(k)} \right\|^2 v_i^{(k)} + 2 \sum_{i=1}^{m} \left( \delta_i^2 + \zeta_i^2 \left\| \nabla F\left(\bar{\theta}^{(k)}\right) \right\|^2 \right) v_i^{(k)} \right.$$

$$\left. + 2 \sum_{j=1}^{m} \left( \delta_j^2 + \zeta_j^2 \left\| \nabla F\left(\bar{\theta}^{(k)}\right) \right\|^2 \right) v_j^{(k)} + \sum_{j=1}^{m} \beta_j^2 \left\| \bar{\theta}^{(k)} - \theta_j^{(k)} \right\|^2 v_j^{(k)} \right]$$

$$= 8 \sum_{i=1}^{m} \left[ \beta_i^2 \left\| \theta_i^{(k)} - \bar{\theta}^{(k)} \right\|^2 + 2 \left( \delta_i^2 + \zeta_i^2 \left\| \nabla F\left(\bar{\theta}^{(k)}\right) \right\|^2 \right) \right] v_i^{(k)}$$

$$\leq 8\beta^2 \sum_{i=1}^{m} \left\| \theta_i^{(k)} - \bar{\theta}^{(k)} \right\|^2 v_i^{(k)} + 16 \left( \zeta^2 \left\| \nabla F\left(\bar{\theta}^{(k)}\right) \right\|^2 + \delta^2 \right) \sum_{i=1}^{m} v_i^{(k)},$$

where in the sixth line above, smoothness and gradient diversity assumptions of Assumption 4.2 were used. We next take the expected value of the expression above and use uncorrelatedness or random variables of Assumption 4.3-(b) and Definition 4.6 to get

$$\mathbb{E}_{\boldsymbol{\Xi}^{(k)}} \left[ \left\| \mathbf{V}^{(k)} \nabla^{(k)} - \mathbf{1}_m \overline{\nabla v}^{(k)} \right\|^2 \right]$$

$$\leq 8\beta^2 \sum_{i=1}^{m} \mathbb{E}_{\boldsymbol{\Xi}^{(k-1)}} \left[ \left\| \theta_i^{(k)} - \bar{\theta}^{(k)} \right\|^2 \right] \mathbb{E}_{v_i^{(k)}} \left[ v_i^{(k)} \right]$$

$$+ 16 \left( \zeta^2 \mathbb{E}_{\boldsymbol{\Xi}^{(k-1)}} \left[ \left\| \nabla F\left(\bar{\theta}^{(k)}\right) \right\|^2 \right] + \delta^2 \right) \sum_{i=1}^{m} \mathbb{E}_{v_i^{(k)}} \left[ v_i^{(k)} \right]$$

$$\leq 8\beta^2 \sum_{i=1}^{m} \mathbb{E}_{\boldsymbol{\Xi}^{(k-1)}} \left[ \left\| \theta_i^{(k)} - \bar{\theta}^{(k)} \right\|^2 \right] d_i^{(k)} + 16 \left( \zeta^2 \mathbb{E}_{\boldsymbol{\Xi}^{(k-1)}} \left[ \left\| \nabla F\left(\bar{\theta}^{(k)}\right) \right\|^2 \right] + \delta^2 \right) \sum_{i=1}^{m} d_i^{(k)}$$

$$\leq 8\beta^2 d_{\max}^{(k)} \mathbb{E}_{\boldsymbol{\Xi}^{(k-1)}} \left[ \left\| \boldsymbol{\Theta}^{(k)} - \mathbf{1}_m \bar{\theta}^{(k)} \right\|^2 \right] + 16 m d_{\max}^{(k)} \left( \zeta^2 \mathbb{E}_{\boldsymbol{\Xi}^{(k-1)}} \left[ \left\| \nabla F\left(\bar{\theta}^{(k)}\right) \right\|^2 \right] + \delta^2 \right)$$

$$= 8 d_{\max}^{(k)} \left[ \beta^2 \mathbb{E}_{\boldsymbol{\Xi}^{(k-1)}} \left[ \left\| \boldsymbol{\Theta}^{(k)} - \mathbf{1}_m \bar{\theta}^{(k)} \right\|^2 \right] + 2m \zeta^2 \mathbb{E}_{\boldsymbol{\Xi}^{(k-1)}} \left[ \left\| \nabla F\left(\bar{\theta}^{(k)}\right) \right\|^2 \right] + 2m \delta^2 \right].$$

### H.3    PROOF OF LEMMA G.3

We have

$$F(\bar{\theta}^{(k+1)}) \leq F(\bar{\theta}^{(k)}) + \left\langle \nabla F(\bar{\theta}^{(k)}), \bar{\theta}^{(k+1)} - \bar{\theta}^{(k)} \right\rangle + \frac{\beta}{2} \left\| \bar{\theta}^{(k)} - \bar{\theta}^{(k+1)} \right\|^2$$

$$= F(\bar{\theta}^{(k)}) + \left\langle \nabla F(\bar{\theta}^{(k)}), -\alpha^{(k)} \overline{gv}^{(k)} \right\rangle + \frac{\beta}{2} \left\| \alpha^{(k)} \overline{gv}^{(k)} \right\|^2$$

$$= F(\bar{\theta}^{(k)}) - \alpha^{(k)} \left\langle \nabla F(\bar{\theta}^{(k)}), \overline{\nabla v}^{(k)} \right\rangle - \alpha^{(k)} \left\langle \nabla F(\bar{\theta}^{(k)}), \overline{\epsilon v}^{(k)} \right\rangle$$

$$+ \frac{\beta}{2} \left( \alpha^{(k)} \right)^2 \left\| \overline{\nabla v}^{(k)} \right\|^2 + \frac{\beta}{2} \left( \alpha^{(k)} \right)^2 \left\| \overline{\epsilon v}^{(k)} \right\|^2 + \beta \left( \alpha^{(k)} \right)^2 \left\langle \overline{\nabla v}^{(k)}, \overline{\epsilon v}^{(k)} \right\rangle,$$

in which the relationship in each of the three lines follow from (i) Smoothness (Assumption 4.2-(a)), (ii) Eq. 5, (iii) $\mathbf{g}_i^{(k)} = \nabla_i^{(k)} + \epsilon_i^{(k)}$ for all $i \in \mathcal{M}$. Next, we take the expected value of the above

inequality and use Assumptions 4.3, and Lemmas D.2 and G.1 to get

$$
\mathbb{E}_{\boldsymbol{\Xi}^{(k)}}\left[F(\bar{\theta}^{(k+1)})\right] \leq \mathbb{E}_{\boldsymbol{\Xi}^{(k-1)}}\left[F(\bar{\theta}^{(k)})\right] - \alpha^{(k)}\left(\frac{1}{2} - \zeta^2\left(1 - d_{\min}^{(k)}\right)\right)\mathbb{E}_{\boldsymbol{\Xi}^{(k-1)}}\left[\left\|\nabla F(\bar{\theta}^{(k)})\right\|^2\right]
$$
$$
+ \frac{\beta^2 d_{\max}^{(k)}}{2m}\alpha^{(k)}\mathbb{E}_{\boldsymbol{\Xi}^{(k-1)}}\left[\left\|\boldsymbol{\Theta}^{(k)} - \mathbf{1}_m\bar{\theta}^{(k)}\right\|^2\right] + \alpha^{(k)}\left(1 - d_{\min}^{(k)}\right)\delta^2
$$
$$
+ \beta\left[1 + 2\zeta^2\left(1 - d_{\min}^{(k)}\right)\right]\left(\alpha^{(k)}\right)^2\mathbb{E}_{\boldsymbol{\Xi}^{(k-1)}}\left[\left\|\nabla F(\bar{\theta}^{(k)})\right\|^2\right]
$$
$$
+ \frac{\beta^3 d_{\max}^{(k)}}{m}\left(\alpha^{(k)}\right)^2\mathbb{E}_{\boldsymbol{\Xi}^{(k-1)}}\left[\left\|\boldsymbol{\Theta}^{(k)} - \mathbf{1}_m\bar{\theta}^{(k)}\right\|^2\right]
$$
$$
+ 2\beta\left(1 - d_{\min}^{(k)}\right)\delta^2\left(\alpha^{(k)}\right)^2 + \frac{\beta}{2}\left(\alpha^{(k)}\right)^2 d_{\max}^{(k)}\frac{\sigma^2}{m}
$$
$$
\leq \mathbb{E}_{\boldsymbol{\Xi}^{(k-1)}}\left[F(\bar{\theta}^{(k)})\right]
$$
$$
- \alpha^{(k)}\left[\frac{1}{2} - \zeta^2\left(1 - d_{\min}^{(k)}\right) - \beta\left(1 + 2\zeta^2\left(1 - d_{\min}^{(k)}\right)\right)\alpha^{(k)}\right]\mathbb{E}_{\boldsymbol{\Xi}^{(k-1)}}\left[\left\|\nabla F(\bar{\theta}^{(k)})\right\|^2\right]
$$
$$
+ \frac{\beta^2 d_{\max}^{(k)}}{2m}\alpha^{(k)}\left(1 + 2\beta\alpha^{(k)}\right)\mathbb{E}_{\boldsymbol{\Xi}^{(k-1)}}\left[\left\|\boldsymbol{\Theta}^{(k)} - \mathbf{1}_m\bar{\theta}^{(k)}\right\|^2\right]
$$
$$
+ \alpha^{(k)}\left[\left(1 - d_{\min}^{(k)}\right)\left(1 + 2\beta\alpha^{(k)}\right)\delta^2 + \frac{\beta}{2}\alpha^{(k)}\frac{d_{\max}\sigma^2}{m}\right].
$$

## H.4 PROOF OF LEMMA G.4

Note that we proved a similar bound for the case of convex models in Lemma 4.8. Thus, we start from Eq. 14 derived in the proof of that lemma in Appendix 4.8. We take the expected value of the inequality in Eq. 14 and use Lemmas D.2, G.2 and D.4-(c) to get

$$
\mathbb{E}_{\boldsymbol{\Xi}^{(k)}}\left[\left\|\boldsymbol{\Theta}^{(k+1)} - \mathbf{1}_m\bar{\theta}^{(k+1)}\right\|^2\right] \leq \frac{1 + \tilde{\rho}^{(k)}}{2\tilde{\rho}^{(k)}}\tilde{\rho}^{(k)}\mathbb{E}_{\boldsymbol{\Xi}^{(k-1)}}\left[\left\|\boldsymbol{\Theta}^{(k)} - \mathbf{1}_m\bar{\theta}^{(k)}\right\|^2\right]
$$
$$
+ 8\frac{1 + \tilde{\rho}^{(k)}}{1 - \tilde{\rho}^{(k)}}d_{\max}^{(k)}\left(\alpha^{(k)}\right)^2\left[\beta^2\mathbb{E}_{\boldsymbol{\Xi}^{(k-1)}}\left[\left\|\boldsymbol{\Theta}^{(k)} - \mathbf{1}_m\bar{\theta}^{(k)}\right\|^2\right]\right.
$$
$$
\left. + 2m\zeta^2\mathbb{E}_{\boldsymbol{\Xi}^{(k-1)}}\left[\left\|\nabla F\left(\bar{\theta}^{(k)}\right)\right\|^2\right] + 2m\delta^2\right]
$$
$$
- 2\alpha^{(k)}\mathbb{E}_{\boldsymbol{\Xi}^{(k)}\backslash\mathbf{E}^{(k)}}\left[\left\langle \mathbf{P}^{(k)}\boldsymbol{\Theta}^{(k)} - \mathbf{1}_m\bar{\theta}^{(k)} - \alpha^{(k)}\left(\mathbf{V}^{(k)}\nabla^{(k)} - \mathbf{1}_m\overline{\nabla v}^{(k)}\right),\right.\right.
$$
$$
\left.\left.\left(\mathbf{I}_m - \frac{1}{m}\mathbf{1}_m\mathbf{1}_m^T\right)\mathbf{V}^{(k)}\mathbb{E}_{\mathbf{E}^{(k)}}\left[\mathbf{E}^{(k)}\right]\right\rangle\right]
$$
$$
+ m\left(\alpha^{(k)}\right)^2 d_{\max}^{(k)}\sigma^2
$$
$$
\leq \left[\frac{1 + \tilde{\rho}^{(k)}}{2} + 8\frac{1 + \tilde{\rho}^{(k)}}{1 - \tilde{\rho}^{(k)}}d_{\max}^{(k)}\left(\alpha^{(k)}\right)^2\beta^2\right]\mathbb{E}_{\boldsymbol{\Xi}^{(k-1)}}\left[\left\|\boldsymbol{\Theta}^{(k)} - \mathbf{1}_m\bar{\theta}^{(k)}\right\|^2\right]
$$
$$
+ 16\frac{1 + \tilde{\rho}^{(k)}}{1 - \tilde{\rho}^{(k)}}md_{\max}^{(k)}\left(\alpha^{(k)}\right)^2\zeta^2\mathbb{E}_{\boldsymbol{\Xi}^{(k-1)}}\left[\left\|\nabla F\left(\bar{\theta}^{(k)}\right)\right\|^2\right]
$$
$$
+ m\left(\alpha^{(k)}\right)^2 d_{\max}^{(k)}\left(16\frac{1 + \tilde{\rho}^{(k)}}{1 - \tilde{\rho}^{(k)}}\delta^2 + \sigma^2\right).
$$

## H.5    PROOF OF LEMMA G.5

We expand the bound derived in Lemma G.4 to get

$$\mathbb{E}_{\Xi^{(k)}}\left[\left\|\Theta^{(k+1)} - \mathbf{1}_m\bar{\theta}^{(k+1)}\right\|^2\right] \leq$$

$$\left(\prod_{r=0}^{k}\phi_{22}^{(r)}\right)\left\|\Theta^{(0)} - \mathbf{1}_m\bar{\theta}^{(0)}\right\|^2 + \sum_{r=0}^{k}\left(\prod_{s=r+1}^{k}\phi_{22}^{(s)}\right)\left[\phi_{21}^{(r)}\mathbb{E}_{\Xi^{(r-1)}}\left[\left\|\nabla F\left(\bar{\theta}^{(r)}\right)\right\|^2\right] + \psi_2^{(r)}\right].$$

We need to make sure that the consensus error diminishes over the iterations. Towards this goal, we ensure that $\phi_{22}^{(k)} < 1$ for all $k \geq 0$. Using the definition of $\phi_{22}^{(k)}$ from Lemma G.4, we have

$$\frac{1 + \tilde{\rho}^{(k)}}{2} + 8\frac{1 + \tilde{\rho}^{(k)}}{1 - \tilde{\rho}^{(k)}}d_{\max}^{(k)}(\alpha^{(k)})^2\beta^2 < 1 \qquad \Rightarrow \qquad \alpha^{(k)} < \frac{1 - \tilde{\rho}^{(k)}}{4\beta\sqrt{d_{\max}^{(k)}(1 + \tilde{\rho}^{(k)})}}.$$

We will find it useful later in the proof of Proposition G.6 to define the following equivalent constraint

$$\alpha^{(k)} \leq \frac{1}{\Gamma_1^{(k)}}\frac{1 - \tilde{\rho}^{(k)}}{4\beta\sqrt{d_{\max}^{(k)}(1 + \tilde{\rho}^{(k)})}}, \qquad \Gamma_1^{(k)} > 1.$$

This results in $\phi_{22}^{(k)} \leq \frac{1 + \tilde{\rho}^{(k)}}{2} + \frac{1}{(\Gamma_1^{(k)})^2}\frac{1 - \tilde{\rho}^{(k)}}{2}$.

## H.6    PROOF OF PROPOSITION G.6

**Step 1: Recursively expanding consensus error.** We substitute the upper bound derived in Lemma G.5 in Lemma G.3 to get

$$\mathbb{E}_{\Xi^{(k)}}\left[F(\bar{\theta}^{(k+1)})\right] \leq \mathbb{E}_{\Xi^{(k-1)}}\left[F(\bar{\theta}^{(k)})\right] - \phi_{11}^{(k)}\mathbb{E}_{\Xi^{(k-1)}}\left[\left\|\nabla F(\bar{\theta}^{(k)})\right\|^2\right]$$

$$+ \phi_{12}^{(k)}\left\{\phi_{22}^{(k-1:0)}\left\|\Theta^{(0)} - \mathbf{1}_m\bar{\theta}^{(0)}\right\|^2\right.$$

$$\left. + \sum_{r=0}^{k-1}\phi_{22}^{(k-1:r+1)}\left[\phi_{21}^{(r)}\mathbb{E}_{\Xi^{(r-1)}}\left[\left\|\nabla F\left(\bar{\theta}^{(r)}\right)\right\|^2\right] + \psi_2^{(r)}\right]\right\} + \psi_1^{(k)}$$

$$\leq \mathbb{E}_{\Xi^{(k-1)}}\left[F(\bar{\theta}^{(k)})\right] - \phi_{11}^{(k)}\mathbb{E}_{\Xi^{(k-1)}}\left[\left\|\nabla F(\bar{\theta}^{(k)})\right\|^2\right] + \phi_{12}^{(k)}\phi_{22}^{(k-1:0)}\left\|\Theta^{(0)} - \mathbf{1}_m\bar{\theta}^{(0)}\right\|^2$$

$$+ \phi_{12}^{(k)}\sum_{r=0}^{k-1}\phi_{22}^{(k-1:r+1)}\phi_{21}^{(r)}\mathbb{E}_{\Xi^{(r-1)}}\left[\left\|\nabla F\left(\bar{\theta}^{(r)}\right)\right\|^2\right] + \phi_{12}^{(k)}\sum_{r=0}^{k-1}\phi_{22}^{(k-1:r+1)}\psi_2^{(r)} + \psi_1^{(k)}.$$

Now, summing both sides of the inequality from $k = 0$ to $k = K$, we get

$$\mathbb{E}_{\Xi^{(K)}}\left[F(\bar{\theta}^{(K+1)})\right] \leq F(\bar{\theta}^{(0)}) - \sum_{k=0}^{K}\phi_{11}^{(k)}\mathbb{E}_{\Xi^{(k-1)}}\left[\left\|\nabla F(\bar{\theta}^{(k)})\right\|^2\right]$$

$$+ \sum_{k=0}^{K}\phi_{12}^{(k)}\phi_{22}^{(k-1:0)}\left\|\Theta^{(0)} - \mathbf{1}_m\bar{\theta}^{(0)}\right\|^2$$

$$+ \sum_{k=0}^{K}\phi_{12}^{(k)}\sum_{r=0}^{k-1}\phi_{22}^{(k-1:r+1)}\phi_{21}^{(r)}\mathbb{E}_{\Xi^{(r-1)}}\left[\left\|\nabla F\left(\bar{\theta}^{(r)}\right)\right\|^2\right]$$

$$+ \sum_{k=0}^{K}\phi_{12}^{(k)}\sum_{r=0}^{k-1}\phi_{22}^{(k-1:r+1)}\psi_2^{(r)} + \sum_{k=0}^{K}\psi_1^{(k)}$$

$$
\begin{aligned}
= {}& F(\bar{\theta}^{(0)}) - \sum_{k=0}^{K} \phi_{11}^{(k)} \mathbb{E}_{\boldsymbol{\Xi}^{(k-1)}}\left[\left\|\nabla F(\bar{\theta}^{(k)})\right\|^2\right] + \left\|\boldsymbol{\Theta}^{(0)} - \mathbf{1}_m \bar{\theta}^{(0)}\right\|^2 \sum_{k=0}^{K} \phi_{12}^{(k)} \phi_{22}^{(k-1:0)} \\
& + \sum_{r=0}^{K-1} \phi_{21}^{(r)} \mathbb{E}_{\boldsymbol{\Xi}^{(r-1)}}\left[\left\|\nabla F\left(\bar{\theta}^{(r)}\right)\right\|^2\right] \sum_{k=r+1}^{K} \phi_{12}^{(k)} \phi_{22}^{(k-1:r+1)} \\
& + \sum_{r=0}^{K-1} \psi_2^{(r)} \sum_{k=r+1}^{K} \phi_{12}^{(k)} \phi_{22}^{(k-1:r+1)} + \sum_{k=0}^{K} \psi_1^{(k)} \\
= {}& F(\bar{\theta}^{(0)}) - \sum_{k=0}^{K-1}\left[\phi_{11}^{(k)} - \phi_{21}^{(k)} \sum_{r=k+1}^{K} \phi_{12}^{(r)} \phi_{22}^{(r-1:k+1)}\right] \mathbb{E}_{\boldsymbol{\Xi}^{(k-1)}}\left[\left\|\nabla F(\bar{\theta}^{(k)})\right\|^2\right] \\
& - \phi_{11}^{(K)} \mathbb{E}_{\boldsymbol{\Xi}^{(K-1)}}\left[\left\|\nabla F(\bar{\theta}^{(K)})\right\|^2\right] + \left\|\boldsymbol{\Theta}^{(0)} - \mathbf{1}_m \bar{\theta}^{(0)}\right\|^2 \sum_{k=0}^{K} \phi_{12}^{(k)} \phi_{22}^{(k-1:0)} \\
& + \sum_{k=0}^{K-1} \psi_2^{(k)} \sum_{r=k+1}^{K} \phi_{12}^{(r)} \phi_{22}^{(r-1:k+1)} + \sum_{k=0}^{K} \psi_1^{(k)}
\end{aligned}
$$

**Step 2: Simplifying using non-increasing learning rate.** Using the definitions of $\phi_{12}^{(k)}$ and $\phi_{22}^{(k)}$ from Lemmas G.3 and G.4, a non-increasing learning rate means that the upper bounds for $\phi_{12}^{(k)}$ and $\phi_{22}^{(k)}$ is non-increasing as well, knowing that $d_i^{(k)} \le 1$ and $\tilde{\rho}^{(k)} < 1$. Thus, we have

$$
\begin{aligned}
\mathbb{E}_{\boldsymbol{\Xi}^{(K)}}\left[F(\bar{\theta}^{(K+1)})\right] \le {}& F(\bar{\theta}^{(0)}) \\
& - \sum_{k=0}^{K-1}\left[\phi_{11}^{(k)} - \phi_{21}^{(k)} \phi_{12}^{(k+1)} \sum_{r=k+1}^{K}\left(\phi_{22}^{(k+1)}\right)^{r-1-k}\right] \mathbb{E}_{\boldsymbol{\Xi}^{(k-1)}}\left[\left\|\nabla F(\bar{\theta}^{(k)})\right\|^2\right] \\
& - \phi_{11}^{(K)} \mathbb{E}_{\boldsymbol{\Xi}^{(K-1)}}\left[\left\|\nabla F(\bar{\theta}^{(K)})\right\|^2\right] + \left\|\boldsymbol{\Theta}^{(0)} - \mathbf{1}_m \bar{\theta}^{(0)}\right\|^2 \phi_{12}^{(0)} \sum_{k=0}^{K}\left(\phi_{22}^{(0)}\right)^k \\
& + \sum_{k=0}^{K-1} \psi_2^{(k)} \phi_{12}^{(k+1)} \sum_{r=k+1}^{K}\left(\phi_{22}^{(k+1)}\right)^{r-1-k} + \sum_{k=0}^{K} \psi_1^{(k)} \\
\le {}& F(\bar{\theta}^{(0)}) - \sum_{k=0}^{K-1}\left[\phi_{11}^{(k)} - \phi_{21}^{(k)} \phi_{12}^{(k+1)} \sum_{u=0}^{K-1-k}\left(\phi_{22}^{(k+1)}\right)^{u}\right] \mathbb{E}_{\boldsymbol{\Xi}^{(k-1)}}\left[\left\|\nabla F(\bar{\theta}^{(k)})\right\|^2\right] \\
& - \phi_{11}^{(K)} \mathbb{E}_{\boldsymbol{\Xi}^{(K-1)}}\left[\left\|\nabla F(\bar{\theta}^{(K)})\right\|^2\right] + \left\|\boldsymbol{\Theta}^{(0)} - \mathbf{1}_m \bar{\theta}^{(0)}\right\|^2 \phi_{12}^{(0)} \sum_{k=0}^{\infty}\left(\phi_{22}^{(0)}\right)^k \\
& + \sum_{k=0}^{K-1} \psi_2^{(k)} \phi_{12}^{(k+1)} \sum_{u=0}^{K-1-k}\left(\phi_{22}^{(k+1)}\right)^{u} + \sum_{k=0}^{K} \psi_1^{(k)} \\
\le {}& F(\bar{\theta}^{(0)}) - \sum_{k=0}^{K-1}\left[\phi_{11}^{(k)} - \phi_{21}^{(k)} \phi_{12}^{(k+1)} \sum_{u=0}^{\infty}\left(\phi_{22}^{(k+1)}\right)^{u}\right] \mathbb{E}_{\boldsymbol{\Xi}^{(k-1)}}\left[\left\|\nabla F(\bar{\theta}^{(k)})\right\|^2\right] \\
& - \phi_{11}^{(K)} \mathbb{E}_{\boldsymbol{\Xi}^{(K-1)}}\left[\left\|\nabla F(\bar{\theta}^{(K)})\right\|^2\right] + \left\|\boldsymbol{\Theta}^{(0)} - \mathbf{1}_m \bar{\theta}^{(0)}\right\|^2 \phi_{12}^{(0)} \sum_{k=0}^{\infty}\left(\phi_{22}^{(0)}\right)^k \\
& + \sum_{k=0}^{K-1} \psi_2^{(k)} \phi_{12}^{(k+1)} \sum_{u=0}^{\infty}\left(\phi_{22}^{(k+1)}\right)^{u} + \sum_{k=0}^{K} \psi_1^{(k)}
\end{aligned}
$$

$$\leq F(\bar{\theta}^{(0)}) - \sum_{k=0}^{K-1} \left[ \phi_{11}^{(k)} - \frac{\phi_{21}^{(k)} \phi_{12}^{(k+1)}}{1 - \phi_{22}^{(k+1)}} \right] \mathbb{E}_{\Xi^{(k-1)}} \left[ \left\| \nabla F(\bar{\theta}^{(k)}) \right\|^2 \right]$$

$$- \phi_{11}^{(K)} \mathbb{E}_{\Xi^{(K-1)}} \left[ \left\| \nabla F(\bar{\theta}^{(K)}) \right\|^2 \right] + \frac{\left\| \Theta^{(0)} - \mathbf{1}_m \bar{\theta}^{(0)} \right\|^2 \phi_{12}^{(0)}}{1 - \phi_{22}^{(0)}} + \sum_{k=0}^{K-1} \frac{\psi_2^{(k)} \phi_{12}^{(k+1)}}{1 - \phi_{22}^{(k+1)}}$$

$$+ \sum_{k=0}^{K} \psi_1^{(k)}$$

$$\leq F(\bar{\theta}^{(0)}) - \sum_{k=0}^{K-1} \left[ \phi_{11}^{(k)} - \frac{\phi_{21}^{(k)} \phi_{12}^{(k)}}{1 - \phi_{22}^{(k)}} \right] \mathbb{E}_{\Xi^{(k-1)}} \left[ \left\| \nabla F(\bar{\theta}^{(k)}) \right\|^2 \right]$$

$$- \phi_{11}^{(K)} \mathbb{E}_{\Xi^{(K-1)}} \left[ \left\| \nabla F(\bar{\theta}^{(K)}) \right\|^2 \right] + \frac{\left\| \Theta^{(0)} - \mathbf{1}_m \bar{\theta}^{(0)} \right\|^2 \phi_{12}^{(0)}}{1 - \phi_{22}^{(0)}} + \sum_{k=0}^{K-1} \frac{\psi_2^{(k)} \phi_{12}^{(k)}}{1 - \phi_{22}^{(k)}}$$

$$+ \sum_{k=0}^{K} \psi_1^{(k)}$$

**Step 3:** We need to ensure that the gradient descents result in lowering the average model loss. Thus, we need to make sure that $\phi_{11}^{(k)} > 0$ and $\phi_{11}^{(k)} - \frac{\phi_{21}^{(k)} \phi_{12}^{(k)}}{1 - \phi_{22}^{(k)}} > 0$. Using the definitions of $\phi_{11}^{(k)}$ from Lemmas G.3, we first solve for $\phi_{11}^{(k)} > 0$ to get

$$\alpha^{(k)} \left[ \frac{1}{2} - \zeta^2 \left( 1 - d_{\min}^{(k)} \right) - \beta \left( 1 + 2\zeta^2 \left( 1 - d_{\min}^{(k)} \right) \right) \alpha^{(k)} \right] > 0$$

$$\Rightarrow \qquad 0 < \alpha^{(k)} < \frac{\frac{1}{2} - \zeta^2 \left( 1 - d_{\min}^{(k)} \right)}{\beta \left( 1 + 2\zeta^2 \left( 1 - d_{\min}^{(k)} \right) \right)}.$$

Note that the above inequality implicitly assumes the following constraint as well

$$\frac{1}{2} - \zeta^2 \left( 1 - d_{\min}^{(k)} \right) > 0 \qquad \Rightarrow \qquad d_{\min}^{(k)} > 1 - \frac{1}{2\zeta^2},$$

which alternatively is equal to

$$d_{\min}^{(k)} \geq 1 - \frac{1}{\Gamma_0^{(k)}} \frac{1}{2\zeta^2}, \qquad \Gamma_0^{(k)} > 1.$$

Next, to solve for $\phi_{11}^{(k)} - \frac{\phi_{21}^{(k)} \phi_{12}^{(k)}}{1 - \phi_{22}^{(k)}} > 0$, we need two extra constraints on $\alpha^{(k)}$. The first one follows as

$$\alpha^{(k)} \leq \frac{1}{\Gamma_2^{(k)}} \frac{1 - \frac{1}{\Gamma_0^{(k)}}}{2\beta \left( 1 + \frac{1}{\Gamma_0^{(k)}} \right)}, \qquad \Gamma_2^{(k)} > 1,$$

which results in $\phi_{11}^{(k)} \geq \frac{\alpha^{(k)}}{2} \left( 1 - \frac{1}{\Gamma_0^{(k)}} \right) \left( 1 - \frac{1}{\Gamma_2^{(k)}} \right)$. The second constraint on $\alpha^{(k)}$ would be

$$\alpha^{(k)} \leq \frac{\Gamma_3^{(k)}}{2\beta}, \qquad \Gamma_3^{(k)} > 0,$$

which implies $\phi_{12}^{(k)} \leq \frac{\beta^2 d_{\max}^{(k)}}{2m} (1 + \Gamma_3^{(k)}) \alpha^{(k)}$. Using the previous two results and the upper bound for $\phi_{22}^{(k)}$ derived in Lemma G.5, we can continue as

$$\frac{\alpha^{(k)}}{2} \left( 1 - \frac{1}{\Gamma_0^{(k)}} \right) \left( 1 - \frac{1}{\Gamma_2^{(k)}} \right) \frac{1 - \tilde{\rho}^{(k)}}{2} \left( 1 - \frac{1}{\left( \Gamma_1^{(k)} \right)^2} \right) > \phi_{21}^{(k)} \phi_{12}^{(k)}$$

$$\Rightarrow \quad \frac{\alpha^{(k)}}{2}\left(1 - \frac{1}{\Gamma_0^{(k)}}\right)\left(1 - \frac{1}{\Gamma_2^{(k)}}\right)\frac{1 - \tilde{\rho}^{(k)}}{2}\left(1 - \frac{1}{\left(\Gamma_1^{(k)}\right)^2}\right)$$

$$> \left(16\frac{1 + \tilde{\rho}^{(k)}}{1 - \tilde{\rho}^{(k)}}md_{\max}^{(k)}\left(\alpha^{(k)}\right)^2\zeta^2\right)\left(\frac{\beta^2 d_{\max}^{(k)}}{2m}\alpha^{(k)}\left(1 + \Gamma_3^{(k)}\right)\right)$$

$$\Rightarrow \quad \alpha^{(k)} < \frac{\sqrt{\left(1 - \frac{1}{\left(\Gamma_1^{(k)}\right)^2}\right)\left(1 - \frac{1}{\Gamma_2^{(k)}}\right)}}{4\sqrt{2}\sqrt{1 + \Gamma_3^{(k)}}}\frac{1 - \tilde{\rho}^{(k)}}{\sqrt{1 + \tilde{\rho}^{(k)}}}\frac{\sqrt{1 - \frac{1}{\Gamma_0^{(k)}}}}{\zeta\beta d_{\max}^{(k)}}$$

Finally, setting the following constraint

$$\alpha^{(k)} \le \frac{1}{\Gamma_4^{(k)}}\frac{\sqrt{\left(1 - \frac{1}{\left(\Gamma_1^{(k)}\right)^2}\right)\left(1 - \frac{1}{\Gamma_2^{(k)}}\right)}}{4\sqrt{2}\sqrt{1 + \Gamma_3^{(k)}}}\frac{1 - \tilde{\rho}^{(k)}}{\sqrt{1 + \tilde{\rho}^{(k)}}}\frac{\sqrt{1 - \frac{1}{\Gamma_0^{(k)}}}}{\zeta\beta d_{\max}^{(k)}}, \qquad \Gamma_4^{(k)} > 1,$$

we obtain

$$\phi_{11}^{(k)} - \frac{\phi_{21}^{(k)}\phi_{12}^{(k)}}{1 - \phi_{22}^{(k)}} \ge \frac{\alpha^{(k)}}{2}\left(1 - \frac{1}{\Gamma_0^{(k)}}\right)\left(1 - \frac{1}{\Gamma_2^{(k)}}\right)\left(1 - \frac{1}{\left(\Gamma_4^{(k)}\right)^2}\right).$$

## H.7 Proof of Theorem 4.12

We are given that a constant learning rate $\alpha^{(k)} = \alpha$ is being used, and we have that $\tilde{\rho}^{(k)} \le \tilde{\rho}$ with $\tilde{\rho} = \max_{0 \le k \le K}\tilde{\rho}^{(k)}$ for the spectral radius and $d_i^{(k)} \ge d_{\min}$ with $d_{\min} = \min_{0 \le k \le K, i \in \mathcal{M}} d_i^{(k)}$ for SGD probabilities. Thus, we can simplify the results of Proposition G.6 to get

$$\mathbb{E}_{\mathbf{\Xi}^{(K)}}\left[F(\bar{\theta}^{(K+1)})\right] \le F\left(\bar{\theta}^{(0)}\right) - w_1\alpha\sum_{r=0}^{K}\mathbb{E}_{\mathbf{\Xi}^{(r-1)}}\left[\left\|\nabla F(\bar{\theta}^{(r)})\right\|^2\right]$$

$$+ \alpha w_2\left\|\mathbf{\Theta}^{(0)} - \mathbf{1}_m\bar{\theta}^{(0)}\right\|^2 + \alpha w_2\psi_2 K + \psi_1(K+1).$$

Therefore, if the learning rate satisfies

$$\alpha < \min\left\{\frac{\Gamma_3}{2\beta}, \frac{1}{\Gamma_1}\frac{1 - \tilde{\rho}}{4\beta\sqrt{1 + \tilde{\rho}}}, \frac{1}{2\Gamma_2}\frac{1 - \frac{1}{\Gamma_0}}{\beta\left(1 + \frac{1}{\Gamma_0}\right)},\right.$$

$$\left.\frac{\sqrt{\left(1 - \frac{1}{\Gamma_1}\right)\left(1 - \frac{1}{\Gamma_2}\right)}}{4\sqrt{2}\Gamma_4\sqrt{1 + \Gamma_3}}\frac{\sqrt{1 - \frac{1}{\Gamma_0}}}{\zeta\beta}\frac{1 - \tilde{\rho}}{\sqrt{1 + \tilde{\rho}}}\right\},$$

then we have

$$\frac{1}{K+1}\sum_{r=0}^{K}\mathbb{E}_{\mathbf{\Xi}^{(r-1)}}\left[\left\|\nabla F(\bar{\theta}^{(r)})\right\|^2\right] \le \frac{1}{\alpha w_1(K+1)}\left[F\left(\bar{\theta}^{(0)}\right) - F^\star + \left\|\mathbf{\Theta}^{(0)} - \mathbf{1}_m\bar{\theta}^{(0)}\right\|^2\alpha w_2\right.$$

$$\left. + \alpha w_2\psi_2 K + \psi_1(K+1)\right].$$

$$\le \frac{1}{w_1}\left[\frac{F\left(\bar{\theta}^{(0)}\right) - F^\star}{\alpha(k+1)} + \frac{\left\|\mathbf{\Theta}^{(0)} - \mathbf{1}_m\bar{\theta}^{(0)}\right\|^2 w_2}{k+1} + \alpha^2 w_2 w_3 + (1 - d_{\min})w_4 + \alpha w_5\right],$$

where $w_1 = \frac{1}{2}(1 - \frac{1}{\Gamma_0})(1 - \frac{1}{\Gamma_2})(1 - \frac{1}{\Gamma_4})$, $w_2 = \frac{\beta^2 d_{\max}(1 + \Gamma_3)}{m(1 - \tilde{\rho})\left(1 - \frac{1}{\Gamma_1}\right)}$, $w_3 = md_{\max}(16\frac{1 + \tilde{\rho}}{1 - \tilde{\rho}}\delta^2 + \sigma^2)$,

$w_4 = (1 + \Gamma_3)\delta^2$ and $w_5 = \frac{\beta d_{\max}\sigma^2}{2m}$. Also, the conditions on the constant scalars $\Gamma_i$ with $0 \le i \le 4$ are $\Gamma_0, \Gamma_1, \Gamma_2, \Gamma_4 > 1$ and $\Gamma_3 > 0$.

# I   DIMINISHING LEARNING RATE POLICY FOR CONVEX MODELS

In this appendix, we do the convergence analysis of our methodology under a diminishing learning rate policy, i.e., when $\alpha^{(k+1)} < \alpha^{(k)}$ for all $k \geq 0$. We will show that convergence to the globally optimal point is possible if the frequency of SGDs, i.e., $d_i^{(k)}$, is increasing over time. Thus, a few preliminary lemmas are first required, to re-derive the counterpart of Proposition 4.10 for the increasing $d_i^{(k)}$ strategy.

**Proposition I.1 (Spectral radius with diminishing learning rate)** *(See Appendix J.1 for the proof.) Let Assumptions 4.1, 4.3 and 4.4 hold. If the SGD probabilities are chosen as $d_i^{(k)} = 1 - \eta_i^{(k)} \alpha^{(k)}$ with $0 \leq \eta_i^{(k)} \leq \frac{1}{\alpha^{(k)}}$ and $\eta_{\min}^{(k)} = \min_{i \in \mathcal{M}} \eta_i^{(k)}$, and the learning rate satisfies the following condition for all $k \geq 0$*

$$\alpha^{(k)} < \min \left\{ \frac{\Gamma_1^{(k)}}{\mu}, \frac{1}{2\sqrt{3}} \frac{1 - \tilde{\rho}^{(k)}}{\sqrt{1 + \tilde{\rho}^{(k)}}} \frac{1}{\sqrt{\zeta^2 + 2\beta^2}}, \right.$$
$$\left. \left( \frac{\mu}{6 \left( \zeta^2 + 2\eta_{\min}^{(k)} \Gamma_1^{(k)} \frac{\beta^2}{\mu} \right) \left( 1 + \Gamma_1^{(k)} \right)} \right)^{1/3} \left( \frac{1 - \tilde{\rho}^{(k)}}{2\beta} \right)^{2/3} \right\}.$$

*then we have $\rho\left( \mathbf{\Phi}^{(k)} \right) < 1$ for all $k \geq 0$, in which $\rho(\cdot)$ denotes the spectral radius of a given matrix, and $\mathbf{\Phi}^{(k)}$ is given in the linear system of inequalities of Eq. 7. $\rho(\mathbf{\Phi}^{(k)})$ is given by*

$\rho(\mathbf{\Phi}^{(k)}) = 1 - h(\alpha^{(k)})$, *where* $h(\alpha^{(k)}) = \frac{1 - \tilde{\rho}^{(k)}}{4} + A^{(k)} \alpha^{(k)} - B^{(k)} (\alpha^{(k)})^2 - \frac{1}{2} \sqrt{\left( \frac{1 - \tilde{\rho}^{(k)}}{2} - 2(A^{(k)} \alpha^{(k)} + B^{(k)} (\alpha^{(k)})^2) \right)^2 + C^{(k)} (\alpha^{(k)})^3}$, *and* $A^{(k)} = \frac{\eta_{\min}^{(k)} \Gamma_1^{(k)}}{\mu} (1 + \Gamma_1^{(k)}) (\Gamma_2^{\star(k)} - 1)\beta^2$, $B^{(k)} = \frac{3}{2} \frac{1 + \tilde{\rho}^{(k)}}{1 - \tilde{\rho}^{(k)}} (\zeta^2 + 2\beta^2)$, *and* $C^{(k)} = 12 \frac{(1 + \Gamma_1^{(k)})\beta^2}{\mu} \frac{1 + \tilde{\rho}^{(k)}}{1 - \tilde{\rho}^{(k)}} (\zeta^2 + 2\eta_{\min}^{(k)} \Gamma_1^{(k)} \frac{\beta^2}{\mu})$. *The value for $\Gamma_2^{\star(k)}$ is given in Appendix J.1, and $0 < \Gamma_1^{(k)} < 1/(1 + 2\beta^2 \eta_{\min}^{(k)}/\mu^2)$ is an arbitrary scalar value.*

Proposition I.1 implies that $\lim_{k \to \infty} \mathbf{\Phi}^{(k:0)} = 0$ in Eq. 8. However, note that this is only the asymptotic behaviour of $\mathbf{\Phi}^{(k:0)}$, and the exact convergence rate will depend on the choice of the learning rate $\alpha^{(k)}$. Furthermore, noting that the first expression in Eq. 8 asymptotically approaches zero, Proposition I.1 also implies that the optimality gap is determined by the terms $\sum_{r=1}^{k} \mathbf{\Phi}^{(k:r)} \mathbf{\Psi}^{(r-1)} + \mathbf{\Psi}^{(k)}$, and it can be made zero if the learning rate $\alpha^{(k)}$ satisfies certain conditions, which we will discuss in Theorem I.5.

Proposition I.1 outlines the necessary constraint on the learning rate $\alpha^{(k)}$ at each iteration $k \geq 0$. We next provide a corollary to Proposition I.1, in which we show that under certain conditions, the above-mentioned constraint needs to be satisfied only on the initial value of the learning rate, i.e, $\alpha^{(0)}$.

**Corollary I.2 (Constraint on diminishing learning rate initialization for convex models)** *(Corollary to Proposition I.1) If the learning rate $\alpha^{(k)}$ is non-increasing, i.e., $\alpha^{(k+1)} \leq \alpha^{(k)}$, the SGD probabilities are determined as $d_i^{(k)} = 1 - \eta_i \alpha^{(k)}$ with $0 \leq \eta_i \leq \frac{1}{\alpha^{(0)}}$ and $\eta_{\min} = \min_{i \in \mathcal{M}} \eta_i$, for all $k \geq 0$, and we have constant aggregations probabilities $b_{ij}^{(k)} = b_{ij}$, then the constraints in Proposition I.1 simplify to*

$$\alpha^{(0)} < \min \left\{ \frac{\Gamma_1}{\mu}, \frac{1}{2\sqrt{3}} \frac{1 - \tilde{\rho}}{\sqrt{1 + \tilde{\rho}}} \frac{1}{\sqrt{\zeta^2 + 2\beta^2}}, \left( \frac{\mu}{6 \left( \zeta^2 + 2\eta_{\min} \Gamma_1 \frac{\beta^2}{\mu} \right) \left( 1 + \Gamma_1 \right)} \right)^{1/3} \left( \frac{1 - \tilde{\rho}}{2\beta} \right)^{2/3} \right\}.$$

*This also results in time-invariant scalars $A = A^{(k)}$, $B = B^{(k)}$ and $C = C^{(k)}$, where these quantities were defined in Proposition I.1.*

In the above Corollary, we obtained the constraints on the initial value of the learning rate, i.e., $\alpha^{(0)}$, that lead to the spectral radius of $\mathbf{\Phi}^{(k)}$ being less than 1, i.e., $\rho(\mathbf{\Phi}^{(k)}) < 1$. Note that since

the learning rate is diminishing, i.e., $\lim_{k\to\infty} \alpha^{(k)} = 0$, it follows that $d_i^{(0)} = 1 - \eta_i \alpha^{(0)}$ and $\lim_{k\to\infty} d_i^{(k)} = 1$, which means that all clients will basically do SGDs at every iteration for large enough values of $k$.

Two final building blocks are necessary for the proof of Theorem I.5. We present these in the following lemmas.

**Lemma I.3 (Bound for product in summation form)** *(See Lemma 1 in Zehtabi et al. (2022) for the proof.) Let $\{\zeta_r\}_{r=0}^{\infty}$ be a scalar sequence where $0 < \zeta_r \leq 1$, $\forall r \geq 0$. For any $p \geq 1$, we have*

$$\prod_{r=s}^{k} (1 - \zeta_r)^p \leq \frac{1}{p \sum_{r=s}^{k} \zeta_r}.$$

Next, we outline another crucial lemma for our analysis.

**Lemma I.4 (Bounds for learning-rate-based quantities)** *(See Appendix J.2 for the proof.) Let a diminishing learning rate $\alpha^{(k)} = \alpha^{(0)}/\sqrt{1 + k/\gamma}$ be used, which satisfies the properties*

$$\alpha^{(k+1)} < \alpha^{(k)}, \qquad \sum_{k=0}^{\infty} \alpha^{(k)} = \infty, \qquad \sum_{k=0}^{\infty} \left(\alpha^{(k)}\right)^2 < \infty^3. \tag{21}$$

*Under the setup of Proposition I.1 for the SGD probabilities, i.e., $d_i^{(k)}$, if the aggregation probabilities are constant, i.e., $b_{ij}^{(k)} = b_{ij}$ for all $i \in \mathcal{M}$ and $(i, j) \in \mathcal{E}^{(k)}$, then the following bounds hold*

*(a)* $\sum_{q=r}^{k} \alpha^{(q)} \geq 2\alpha^{(0)} \left(\sqrt{1 + \frac{k}{\gamma}} - \sqrt{1 + \frac{r}{\gamma}}\right),$

*(b)* $h(\alpha^{(k)}) \geq 2A\alpha^{(k)},$

*(c)* $\sum_{r=1}^{k} \frac{(\alpha^{(r-1)})^2}{\sum_{q=r}^{k} h(\alpha^{(q)})}$

$\leq \frac{\alpha^{(0)}}{4A} \left[\frac{1}{\sqrt{1 + \frac{k}{\gamma}} - \sqrt{1 + \frac{1}{\gamma}}} + \frac{2(\gamma+1)}{\sqrt{1 + \frac{k}{\gamma}}} \left(\ln \sqrt{1 + \frac{k}{\gamma}} + \ln \frac{1}{\sqrt{1 + \frac{1}{\gamma+k-1}} - 1}\right) + \frac{2\sqrt{1 + \frac{k}{\gamma}}}{1 + \frac{k-1}{\gamma}}\right].$

*where $h(\alpha^{(k)})$ and the constant $A^{(k)}$ were defined in Proposition I.1.*

Using Corollary I.2 and Lemmas I.3 and I.4, our main theorem follows.

**Theorem I.5 (Strongly-convex convergence result with a diminishing learning rate)** *(See Appendix J.3 for the proof.) Let Assumptions 4.1, 4.3 and 4.4 hold. If a diminishing learning rate policy $\alpha^{(k)} = \alpha^{(0)}/\sqrt{1 + k/\gamma}$ with $\gamma > 0$ satisfying the conditions outlined in Corollary I.2 is employed, and the SGD probabilities are set to $d_i^{(k)} = 1 - ((1 - d_i^{(0)})/\alpha^{(0)})\alpha^{(k)}$ for all $i \in \mathcal{M}$, while aggregation probabilities are set to constant values, i.e., $b_{ij}^{(k)} = b_{ij}$ for all $i \in \mathcal{M}$ and $(i, j) \in \mathcal{E}^{(k)}$, then we can rewrite Eq. 8 as*

$$\nu^{(K+1)} \leq \mathcal{O}\left(\frac{1}{\sqrt{K}}\right) \nu^{(0)}$$

$$+ \left(\alpha^{(0)}\right)^2 \left(3\mathcal{O}\left(\frac{1}{\sqrt{K}}\right) + \mathcal{O}\left(\frac{\ln K}{\sqrt{K}}\right) + \mathcal{O}\left(\frac{1}{k}\right)\right) \begin{bmatrix} \frac{2(1 + \mu\alpha^{(0)})(1 - d_{\max}^{(0)})}{\mu\alpha^{(0)}}\delta^2 + \frac{\sigma^2}{m} \\ m\left(3\frac{1+\tilde{\rho}}{1-\tilde{\rho}}\delta^2 + \sigma^2\right) \end{bmatrix}. \tag{22}$$

*Letting $K \to \infty$, we get*

$$\lim_{K\to\infty} \nu^{(K+1)} = 0. \tag{23}$$

---

[3]Note that the last condition implies $\lim_{k\to\infty} \alpha^{(k)} = 0$

The bound in Eq. 22 of Theorem I.5 indicates that by using a diminishing learning rate policy of $\alpha^{(k)} = \alpha^{(0)}/\sqrt{1 + k/\gamma}$, DSpodFL achieves a sub-linear convergence rate of $\mathcal{O}(\ln K/\sqrt{K})$, and Eq. 23 shows that asymptotic zero optimality gap as $k \to \infty$ can be achieved.

However, it is worth noting that choosing the SGD probabilities based on the learning rate, i.e., $d_i^{(k)} = 1 - \alpha^{(k)}/\alpha^{(0)}$ for all $i \in \mathcal{M}$ and $k \geq 0$, is only of theoretical value in this paper. This is because our motivation of introducing the notion of SGD probabilities was to capture computational capabilities of heterogeneous clients in real-world settings, therefore, it is an independent uncontrollable parameter and cannot be chosen based on the learning rate.

Finally, note that setting $d_i^{(k)} = 1 - \alpha^{(k)}/\alpha^{(0)}$ is equivalent to having all clients in the decentralized system to conduct SGD at each iteration as $k \to \infty$. This result is akin to Wang & Nedić (2023), in which an increasing similarity between the learning rates of clients is needed for convergence, despite them being initially uncoordinated.

## J PROOFS FOR DIMINISHING LEARNING RATE POLICY FOR CONVEX MODELS

### J.1 PROOF OF PROPOSITION I.1

Let the SGD probabilities $d_i^{(k)}$ be chosen as the following for all $i \in \mathcal{M}$:

$$d_i^{(k)} = 1 - \eta_i^{(k)}\alpha^{(k)}, \qquad 0 \leq \eta_i^{(k)} \leq \frac{1}{\alpha^{(k)}}, \qquad \eta_{\max}^{(k)} = \max_{i \in \mathcal{M}} \eta_i^{(k)}, \qquad \eta_{\min}^{(k)} = \min_{i \in \mathcal{M}} \eta_i^{(k)},$$

where $\alpha^{(k)}$ is the learning rate with a diminishing policy, i.e., $\lim_{k \to \infty} \alpha^{(k)} = 0$. Based on this relationship that we put between the SGD probabilities and the learning rate, we first rewrite the bounds for matrices $\mathbf{\Phi}^{(k)}$ and $\mathbf{\Psi}^{(k)}$ which were given in Lemmas 4.7 and 4.8. We have

$$\phi_{11}^{(k)} = 1 - \mu\alpha^{(k)}\left(1 + \mu\alpha^{(k)} - \left(\mu\alpha^{(k)}\right)^2\right) + \frac{2\eta_{\min}^{(k)}\left(\alpha^{(k)}\right)^2}{\mu}\left(1 + \mu\alpha^{(k)}\right)\beta^2,$$

$$\phi_{12}^{(k)} = \left(1 + \mu\alpha^{(k)}\right)\left(1 - \eta_{\max}^{(k)}\alpha^{(k)}\right)\frac{\alpha^{(k)}\beta^2}{m\mu},$$

$$\phi_{21}^{(k)} = 3\frac{1 + \tilde{\rho}^{(k)}}{1 - \tilde{\rho}^{(k)}}m\left(1 - \eta_{\max}^{(k)}\alpha^{(k)}\right)\left(\alpha^{(k)}\right)^2\left(\zeta^2 + 2\beta^2\eta_{\min}^{(k)}\alpha^{(k)}\right)$$

$$\phi_{22}^{(k)} = \frac{1 + \tilde{\rho}^{(k)}}{2} + 3\frac{1 + \tilde{\rho}^{(k)}}{1 - \tilde{\rho}^{(k)}}\left(1 - \eta_{\max}^{(k)}\alpha^{(k)}\right)\left(\alpha^{(k)}\right)^2\left(\zeta^2 + 2\beta^2\right), \tag{24}$$

$$\psi_1^{(k)} = \left(\alpha^{(k)}\right)^2\left[\frac{2\eta_{\min}^{(k)}}{\mu}\left(1 + \mu\alpha^{(k)}\right)\delta^2 + \left(1 - \eta_{\max}^{(k)}\alpha^{(k)}\right)\frac{\sigma^2}{m}\right],$$

$$\psi_2^{(k)} = m\left(1 - \eta_{\max}^{(k)}\alpha^{(k)}\right)\left(\alpha^{(k)}\right)^2\left(3\frac{1 + \tilde{\rho}^{(k)}}{1 - \tilde{\rho}^{(k)}}\delta^2 + \sigma^2\right).$$

The important difference with the terms in Eq. 24 and the corresponding ones outlined in Lemmas 4.7 and 4.8 is the fact that we get a $\left(\alpha^{(k)}\right)^2$ factor for $\psi_1^{(k)}$ and $\psi_2^{(k)}$. This factor will help us show in Theorem I.5 that zero optimality gap can be reached, which follows mainly from Eq. 21.

Next, we do an analysis similar to the proof of Proposition 4.10, which was given in Appendix F.3.

**Step 1: Setting up the proof.** We skip repeating the explanations for this step, as they are exactly the same as step 1 in Appendix F.3.

**Step 2: Simplifying the conditions.** Recall that we have to ensure (i) $0 < \phi_{11}^{(k)} \leq 1$ and (ii) $0 < \phi_{22}^{(k)} \leq 1$. For $\phi_{11}^{(k)}$ as defined in Eq. 24, we have

$$\phi_{11}^{(k)} \leq 1 \qquad \Rightarrow \qquad \frac{1 + \mu\alpha^{(k)} - \left(\mu\alpha^{(k)}\right)^2}{\left(1 + \mu\alpha^{(k)}\right)\alpha^{(k)}} \geq \frac{2\beta^2\eta_{\min}^{(k)}}{\mu^2}. \tag{25}$$

We then put the following constraint on $\alpha^{(k)}$ to get a tighter lower bound for Eq. 25. We have

$$\text{Constraint 1: } \alpha^{(k)} \leq \frac{\Gamma_1^{(k)}}{\mu}, \quad \Rightarrow \quad \mu\alpha^{(k)} > \frac{2\beta^2\eta_{\min}^{(k)}}{\mu^2}\Gamma_1^{(k)}\left(1 + \Gamma_1^{(k)}\right) + \left(\Gamma_1^{(k)}\right)^2 - 1,$$

where $\Gamma_1^{(k)} > 0$ is a scalar. In order to avoid a positive lower-bound on the learning rate $\alpha^{(k)}$, we find the conditions under which the right-hand side of the above inequality is negative. We have

$$\frac{2\beta^2\eta_{\min}^{(k)}}{\mu^2}\Gamma_1^{(k)}\left(1 + \Gamma_1^{(k)}\right) + \left(\Gamma_1^{(k)}\right)^2 - 1 < 0$$

$$\Rightarrow \quad \left(1 + \frac{2\beta^2\eta_{\min}^{(k)}}{\mu^2}\right)\left(\Gamma_1^{(k)}\right)^2 + \frac{2\beta^2\eta_{\min}^{(k)}}{\mu^2}\Gamma_1^{(k)} - 1 < 0,$$

$$\Rightarrow \quad \left(\left(1 + \frac{2\beta^2\eta_{\min}^{(k)}}{\mu^2}\right)\Gamma_1^{(k)} - 1\right)\left(\Gamma_1^{(k)} + 1\right) < 0 \quad \Rightarrow \quad -1 < \Gamma_1^{(k)} < \frac{1}{1 + \frac{2\beta^2\eta_{\min}^{(k)}}{\mu^2}}.$$

Next, in order to simplify $\phi_{11}^{(k)}$ further, we add another constraint using Eq. 25 to parameterize the lower bound in Eq. 25. We have

$$\frac{1 + \mu\alpha^{(k)} - \left(\mu\alpha^{(k)}\right)^2}{1 + \mu\alpha^{(k)}} \geq \frac{2\beta^2\eta_{\min}^{(k)}}{\mu^2}\Gamma_1^{(k)},$$

$$\text{Constraint 2: } \frac{1 + \mu\alpha^{(k)} - \left(\mu\alpha^{(k)}\right)^2}{1 + \mu\alpha^{(k)}} > \frac{2\beta^2\eta_{\min}^{(k)}}{\mu^2}\Gamma_1^{(k)}\Gamma_2^{(k)}, \quad \Gamma_2^{(k)} > 1,$$

in which $\Gamma_2^{(k)} > 1$ makes sure that the constraint in Eq. 25 is satisfied. Hence, we can update the entries of matrices $\mathbf{\Phi}^{(k)}$ and $\mathbf{\Psi}^{(k)}$ as

$$\phi_{11}^{(k)} \leq 1 - \frac{2\eta_{\min}^{(k)}\Gamma_1^{(k)}}{\mu}\left(1 + \Gamma_1^{(k)}\right)\left(\Gamma_2^{(k)} - 1\right)\beta^2\alpha^{(k)},$$

$$\phi_{12}^{(k)} \leq \frac{\left(1 + \Gamma_1^{(k)}\right)\beta^2}{m\mu}\alpha^{(k)}, \quad \phi_{21}^{(k)} \leq 3\frac{1 + \tilde{\rho}^{(k)}}{1 - \tilde{\rho}^{(k)}}m\left(\alpha^{(k)}\right)^2\left(\zeta^2 + 2\eta_{\min}^{(k)}\Gamma_1^{(k)}\frac{\beta^2}{\mu}\right)$$

$$\phi_{22}^{(k)} \leq \frac{1 + \tilde{\rho}^{(k)}}{2} + 3\frac{1 + \tilde{\rho}^{(k)}}{1 - \tilde{\rho}^{(k)}}\left(\alpha^{(k)}\right)^2\left(\zeta^2 + 2\beta^2\right),$$

$$\psi_1^{(k)} \leq \left(\alpha^{(k)}\right)^2\left[\frac{2\eta_{\min}^{(k)}}{\mu}\left(1 + \Gamma_1^{(k)}\right)\delta^2 + \frac{\sigma^2}{m}\right], \quad \psi_2^{(k)} \leq m\left(\alpha^{(k)}\right)^2\left(3\frac{1 + \tilde{\rho}^{(k)}}{1 - \tilde{\rho}^{(k)}}\delta^2 + \sigma^2\right).$$

Note that matrix $\mathbf{\Phi}^{(k)}$ and vector $\mathbf{\Psi}^{(k)}$ in Eq. 7 were used as upper bounds, therefore we can always replace their values with new upper bounds for them. Consequently, with this new value for $\phi_{11}^{(k)}$, we continue as

$$\phi_{11}^{(k)} > 0 \quad \Rightarrow \quad \alpha^{(k)} < \frac{\mu}{2\eta_{\min}^{(k)}\Gamma_1^{(k)}\left(1 + \Gamma_1^{(k)}\right)\left(\Gamma_2^{(k)} - 1\right)\beta^2}.$$

Finally, we check the next condition $0 < \phi_{22}^{(k)} \leq 1$. Noting that we have $\frac{3 + \tilde{\rho}^{(k)}}{4} < 1$, we can enforce $\phi_{22}^{(k)} \leq 1$ by setting $\phi_{22}^{(k)} \leq \frac{3 + \tilde{\rho}^{(k)}}{4}$. We have

$$\frac{1 + \tilde{\rho}^{(k)}}{2} \leq \phi_{22}^{(k)} \leq \frac{3 + \tilde{\rho}^{(k)}}{4} \quad \Rightarrow \quad 0 \leq \alpha^{(k)} \leq \frac{1}{2\sqrt{3}}\frac{1 - \tilde{\rho}^{(k)}}{\sqrt{1 + \tilde{\rho}^{(k)}}}\frac{1}{\sqrt{\zeta^2 + 2\beta^2}}.$$

**Step 3: Determining the constraints.** Having made sure that (i) $0 < \phi_{11}^{(k)} \leq 1$ and (ii) $0 < \phi_{22}^{(k)} \leq 1$ in the previous step, we can continue to solve Eq. 15. For the left-hand side of the inequality, we have

$$\left(1 - \phi_{11}^{(k)}\right)\left(1 - \phi_{22}^{(k)}\right) = \left[\frac{2\eta_{\min}^{(k)}\Gamma_1^{(k)}}{\mu}\left(1 + \Gamma_1^{(k)}\right)\left(\Gamma_2^{(k)} - 1\right)\beta^2\alpha^{(k)}\right]\left(1 - \phi_{22}^{(k)}\right)$$

$$\geq \left[\frac{2\eta_{\min}^{(k)}\Gamma_1^{(k)}}{\mu}\left(1 + \Gamma_1^{(k)}\right)\left(\Gamma_2^{(k)} - 1\right)\beta^2\alpha^{(k)}\right]\frac{1 - \tilde{\rho}^{(k)}}{4}$$

Now, putting this back to Eq. 15, we get

$$\left[\frac{2\eta_{\min}^{(k)}\Gamma_1^{(k)}}{\mu}\left(1+\Gamma_1^{(k)}\right)\left(\Gamma_2^{(k)}-1\right)\beta^2\alpha^{(k)}\right]\frac{1-\tilde{\rho}^{(k)}}{4} > \phi_{12}^{(k)}\phi_{21}^{(k)}$$

$$\Rightarrow \quad \left[\frac{\left(1+\Gamma_1^{(k)}\right)\beta^2}{m\mu}\alpha^{(k)}\right]\left[3m\frac{1+\tilde{\rho}^{(k)}}{1-\tilde{\rho}^{(k)}}\left(\zeta^2+2\eta_{\min}^{(k)}\Gamma_1^{(k)}\frac{\beta^2}{\mu}\right)\left(\alpha^{(k)}\right)^2\right]$$

$$< \left[\frac{2\eta_{\min}^{(k)}\Gamma_1^{(k)}}{\mu}\left(1+\Gamma_1^{(k)}\right)\left(\Gamma_2^{(k)}-1\right)\beta^2\alpha^{(k)}\right]\frac{1-\tilde{\rho}^{(k)}}{4}$$

$$\Rightarrow \quad \alpha^{(k)} < \sqrt{\frac{\eta_{\min}^{(k)}\Gamma_1^{(k)}\left(\Gamma_2^{(k)}-1\right)\left(1-\tilde{\rho}^{(k)}\right)^2}{6\left(1+\tilde{\rho}^{(k)}\right)\left(\zeta^2+2\eta_{\min}^{(k)}\Gamma_1^{(k)}\frac{\beta^2}{\mu}\right)}}.$$

Finally, we solve for Eq. 16, i.e., $c = \phi_{11}^{(k)}\phi_{22}^{(k)} - \phi_{12}^{(k)}\phi_{21}^{(k)} > 0$. Noting that by solving Eq. 15 we made sure that $1 - \phi_{11}^{(k)} - \phi_{22}^{(k)} + \phi_{11}^{(k)}\phi_{22}^{(k)} - \phi_{12}^{(k)}\phi_{21}^{(k)} > 0$, we can write

$$c > 0 \quad \Rightarrow \quad \phi_{11}^{(k)} + \phi_{22}^{(k)} - 1 > 0$$

$$\Rightarrow \quad 1 - \frac{2\eta_{\min}^{(k)}\Gamma_1^{(k)}}{\mu}\left(1+\Gamma_1^{(k)}\right)\left(\Gamma_2^{(k)}-1\right)\beta^2\alpha^{(k)} + \frac{1+\tilde{\rho}^{(k)}}{2} - 1 > 0$$

$$\Rightarrow \quad \alpha^{(k)} < \frac{\mu\left(1+\tilde{\rho}^{(k)}\right)}{4\eta_{\min}^{(k)}\Gamma_1^{(k)}\left(1+\Gamma_1^{(k)}\right)\left(\Gamma_2^{(k)}-1\right)\beta^2},$$

in which we have used the value of $\phi_{11}^{(k)}$ itself, but the lower bound of $\phi_{22}^{(k)}$.

**Step 4: Putting all the constraints together.** Reviewing all the constraints on $\alpha^{(k)}$ from the beginning of this appendix, we can collect all of the constraints together and simplify them as

$$\alpha^{(k)} < \min\left\{\frac{\Gamma_1^{(k)}}{\mu}, \frac{1}{2\sqrt{3}}\frac{1-\tilde{\rho}^{(k)}}{\sqrt{1+\tilde{\rho}^{(k)}}}\frac{1}{\sqrt{\zeta^2+2\beta^2}}, \frac{\mu}{2\eta_{\min}^{(k)}\Gamma_1^{(k)}\left(1+\Gamma_1^{(k)}\right)\left(\Gamma_2^{(k)}-1\right)\beta^2},\right.$$

$$\left.\frac{\mu\left(1+\tilde{\rho}^{(k)}\right)}{4\eta_{\min}^{(k)}\Gamma_1^{(k)}\left(1+\Gamma_1^{(k)}\right)\left(\Gamma_2^{(k)}-1\right)\beta^2}, \sqrt{\frac{\eta_{\min}^{(k)}\Gamma_1^{(k)}\left(\Gamma_2^{(k)}-1\right)\left(1-\tilde{\rho}^{(k)}\right)^2}{6\left(1+\tilde{\rho}^{(k)}\right)\left(\zeta^2+2\eta_{\min}^{(k)}\Gamma_1^{(k)}\frac{\beta^2}{\mu}\right)}}\right\}$$

$$= \min\left\{\frac{\Gamma_1^{(k)}}{\mu}, \frac{1}{2\sqrt{3}}\frac{1-\tilde{\rho}^{(k)}}{\sqrt{1+\tilde{\rho}^{(k)}}}\frac{1}{\sqrt{\zeta^2+2\beta^2}}, \frac{\mu\left(1+\tilde{\rho}^{(k)}\right)}{4\eta_{\min}^{(k)}\Gamma_1^{(k)}\left(1+\Gamma_1^{(k)}\right)\left(\Gamma_2^{(k)}-1\right)\beta^2},\right.$$

$$\left.\sqrt{\frac{\eta_{\min}^{(k)}\Gamma_1^{(k)}\left(\Gamma_2^{(k)}-1\right)\left(1-\tilde{\rho}^{(k)}\right)^2}{6\left(1+\tilde{\rho}^{(k)}\right)\left(\zeta^2+2\eta_{\min}^{(k)}\Gamma_1^{(k)}\frac{\beta^2}{\mu}\right)}}\right\},$$

$$(26)$$

while satisfying

$$\max\{-1,0\} = 0 < \Gamma_1^{(k)} < \min\left\{1, \frac{1}{1+\frac{2\beta^2\eta_{\min}^{(k)}}{\mu^2}}\right\} = \frac{1}{1+\frac{2\beta^2\eta_{\min}^{(k)}}{\mu^2}},$$

$$\Gamma_2^{(k)} > 1, \qquad 0 \le \eta_{\min}^{(k)} \le \frac{1}{\alpha^{(k)}}. \tag{27}$$

Note that one of the terms in Eq. 26 was trivially removed since $\frac{1+\tilde{\rho}^{(k)}}{2} < 1$. Consequently, we obtain

$$
\alpha^{(k)} < \min_{\Gamma_1^{(k)}} \left\{ \frac{\Gamma_1^{(k)}}{\mu}, \frac{1}{2\sqrt{3}} \frac{1-\tilde{\rho}^{(k)}}{\sqrt{1+\tilde{\rho}^{(k)}}} \frac{1}{\sqrt{\zeta^2 + 2\beta^2}}, \min_{\Gamma_2^{(k)}, \eta_{\min}^{(k)}} \left\{ \frac{\mu\left(1+\tilde{\rho}^{(k)}\right)}{4\eta_{\min}^{(k)}\Gamma_1^{(k)}\left(1+\Gamma_1^{(k)}\right)\left(\Gamma_2^{(k)}-1\right)\beta^2}, \right. \right.
$$
$$
\left. \left. \sqrt{\frac{\eta_{\min}^{(k)}\Gamma_1^{(k)}\left(\Gamma_2^{(k)}-1\right)\left(1-\tilde{\rho}^{(k)}\right)^2}{6\left(1+\tilde{\rho}^{(k)}\right)\left(\zeta^2 + 2\eta_{\min}^{(k)}\Gamma_1^{(k)}\frac{\beta^2}{\mu}\right)}} \right\} \right\},
$$

(28)

First, we focus on minimizing the inner expression in Eq. 28 using $\Gamma_2^{(k)}$ by defining $c_1^{(k)} = \sqrt{\frac{\eta_{\min}^{(k)}\Gamma_1^{(k)}\left(1-\tilde{\rho}^{(k)}\right)^2}{6\left(1+\tilde{\rho}^{(k)}\right)\left(\zeta^2 + 2\eta_{\min}^{(k)}\Gamma_1^{(k)}\frac{\beta^2}{\mu}\right)}}$ and $c_2^{(k)} = \frac{4\eta_{\min}^{(k)}\Gamma_1^{(k)}\left(1+\Gamma_1^{(k)}\right)\beta^2}{\mu\left(1+\tilde{\rho}^{(k)}\right)}$. We can see that one of the above

expressions is increasing with respect to $\Gamma_2^{(k)}$, and the other one is decreasing. Thus, we have

$$
\begin{cases} c_1^{(k)}\sqrt{\Gamma_2^{(k)}-1} \leq \frac{1}{c_2^{(k)}\left(\Gamma_2^{(k)}-1\right)}; & 1 < \Gamma_2^{(k)} \leq \Gamma_2^{\star(k)} \\ \frac{1}{c_2^{(k)}\left(\Gamma_2^{(k)}-1\right)} \leq c_1^{(k)}\sqrt{\Gamma_2^{(k)}-1}; & \Gamma_2^{(k)} \geq \Gamma_2^{\star(k)}. \end{cases}
$$

in which $\Gamma_2^{(k)} > 1$ is due to Eq. 27. Hence, we find the optimal value for it, i.e., $\Gamma_2^{\star(k)}$, as

$$
\sqrt{\Gamma_2^{\star(k)}-1}^3 = \frac{1}{c_1^{(k)}c_2^{(k)}} \qquad \Rightarrow \qquad \Gamma_2^{\star(k)} = \frac{1}{\left(c_1^{(k)}c_2^{(k)}\right)^{2/3}} + 1
$$

$$
\Rightarrow \Gamma_2^{\star(k)} = \left( \sqrt{\frac{6\left(1+\tilde{\rho}^{(k)}\right)\left(\zeta^2 + 2\eta_{\min}^{(k)}\Gamma_1^{(k)}\frac{\beta^2}{\mu}\right)}{\eta_{\min}^{(k)}\Gamma_1^{(k)}\left(1-\tilde{\rho}^{(k)}\right)^2}} \frac{\mu\left(1+\tilde{\rho}^{(k)}\right)}{4\eta_{\min}^{(k)}\Gamma_1^{(k)}\left(1+\Gamma_1^{(k)}\right)\beta^2} \right)^{2/3} + 1
$$

$$
= \frac{1+\tilde{\rho}^{(k)}}{2\eta_{\min}^{(k)}\Gamma_1^{(k)}\beta}\left(\frac{3\left(\zeta^2 + 2\eta_{\min}^{(k)}\Gamma_1^{(k)}\frac{\beta^2}{\mu}\right)}{\beta}\right)^{1/3}\left(\frac{\mu}{\left(1-\tilde{\rho}^{(k)}\right)\left(1+\Gamma_1^{(k)}\right)}\right)^{2/3} + 1.
$$

We choose $\Gamma_2^{(k)} = \Gamma_2^{\star(k)}$ (see the explanation given in related step of Appendix F.3) to get

$$
\min_{\Gamma_2^{(k)}} \left\{ \frac{\mu\left(1+\tilde{\rho}^{(k)}\right)}{4\eta_{\min}^{(k)}\Gamma_1^{(k)}\left(1+\Gamma_1^{(k)}\right)\left(\Gamma_2^{(k)}-1\right)\beta^2}, \sqrt{\frac{\eta_{\min}^{(k)}\Gamma_1^{(k)}\left(\Gamma_2^{(k)}-1\right)\left(1-\tilde{\rho}^{(k)}\right)^2}{6\left(1+\tilde{\rho}^{(k)}\right)\left(\zeta^2 + 2\eta_{\min}^{(k)}\Gamma_1^{(k)}\frac{\beta^2}{\mu}\right)}} \right\}
$$

$$
\geq \frac{\mu\left(1+\tilde{\rho}^{(k)}\right)\left(c_1^{(k)}c_2^{(k)}\right)^{2/3}}{4\eta_{\min}^{(k)}\Gamma_1^{(k)}\left(1+\Gamma_1^{(k)}\right)\beta^2} = \left(\frac{\left(c_1^{(k)}\right)^2}{c_2^{(k)}}\right)^{1/3}
$$

$$
= \left(\frac{\mu}{6\left(\zeta^2 + 2\eta_{\min}^{(k)}\Gamma_1^{(k)}\frac{\beta^2}{\mu}\right)\left(1+\Gamma_1^{(k)}\right)}\right)^{1/3}\left(\frac{1-\tilde{\rho}^{(k)}}{2\beta}\right)^{2/3}.
$$

Note that in the process of minimizing Eq. 28 over $\Gamma_2^{(k)}$, two out of the three dependencies on $\eta_{\min}^{(k)}$, and two out of four dependencies on $\Gamma_1^{(k)}$ were removed. Hence, we get

$$
\alpha^{(k)} < \min \left\{ \frac{\Gamma_1^{(k)}}{\mu}, \frac{1}{2\sqrt{3}} \frac{1-\tilde{\rho}^{(k)}}{\sqrt{1+\tilde{\rho}^{(k)}}} \frac{1}{\sqrt{\zeta^2 + 2\beta^2}}, \right.
$$
$$
\left. \left(\frac{\mu}{6\left(\zeta^2 + 2\eta_{\min}^{(k)}\Gamma_1^{(k)}\frac{\beta^2}{\mu}\right)\left(1+\Gamma_1^{(k)}\right)}\right)^{1/3}\left(\frac{1-\tilde{\rho}^{(k)}}{2\beta}\right)^{2/3} \right\}.
$$

Finally, we make a remark that we do not minimize over $\eta_{\min}^{(k)}$ here, as we take it as a given deterministic value based on the choice of $d_i^{(k)} = 1 - \eta_i^{(k)}\alpha^{(k)}$.

**Step 5: Obtaining $\rho(\mathbf{\Phi}^{(k)})$.** We established $\rho(\mathbf{\Phi}^{(k)}) < 1$ in the previous steps. The last step is to determine what $\rho(\mathbf{\Phi}^{(k)})$ is. We have

$$
\rho\left(\mathbf{\Phi}^{(k)}\right) = \frac{-b + \sqrt{b^2 - 4ac}}{2a} = \frac{\phi_{11}^{(k)} + \phi_{22}^{(k)} + \sqrt{\left(\phi_{11}^{(k)} + \phi_{22}^{(k)}\right)^2 - 4\left(\phi_{11}^{(k)}\phi_{22}^{(k)} - \phi_{12}^{(k)}\phi_{21}^{(k)}\right)}}{2}
$$

$$
= \frac{\phi_{11}^{(k)} + \phi_{22}^{(k)} + \sqrt{\left(\phi_{11}^{(k)} - \phi_{22}^{(k)}\right)^2 + 4\phi_{12}^{(k)}\phi_{21}^{(k)}}}{2}
$$

$$
= \frac{1 - \frac{2\eta_{\min}^{(k)}\Gamma_1^{(k)}}{\mu}\left(1 + \Gamma_1^{(k)}\right)\left(\Gamma_2^{(k)} - 1\right)\beta^2\alpha^{(k)} + \frac{1+\tilde{\rho}^{(k)}}{2} + 3\frac{1+\tilde{\rho}^{(k)}}{1-\tilde{\rho}^{(k)}}\left(\alpha^{(k)}\right)^2\left(\zeta^2 + 2\beta^2\right)}{2}
$$

$$
+ \frac{1}{2}\Bigg[\left(1 - \frac{2\eta_{\min}^{(k)}\Gamma_1^{(k)}}{\mu}\left(1 + \Gamma_1^{(k)}\right)\left(\Gamma_2^{(k)} - 1\right)\beta^2\alpha^{(k)} - \frac{1+\tilde{\rho}^{(k)}}{2}\right.
$$

$$
\left. - 3\frac{1+\tilde{\rho}^{(k)}}{1-\tilde{\rho}^{(k)}}\left(\alpha^{(k)}\right)^2\left(\zeta^2 + 2\beta^2\right)\right)^2
$$

$$
+ 4\frac{\left(1 + \Gamma_1^{(k)}\right)\beta^2}{m\mu}\alpha^{(k)}3\frac{1+\tilde{\rho}^{(k)}}{1-\tilde{\rho}^{(k)}}m\left(\alpha^{(k)}\right)^2\left(\zeta^2 + 2\eta_{\min}^{(k)}\Gamma_1^{(k)}\frac{\beta^2}{\mu}\right)\Bigg]^{1/2}
$$

$$
= \frac{3 + \tilde{\rho}^{(k)}}{4} - \frac{\eta_{\min}^{(k)}\Gamma_1^{(k)}}{\mu}\left(1 + \Gamma_1^{(k)}\right)\left(\Gamma_2^{(k)} - 1\right)\beta^2\alpha^{(k)} + \frac{3}{2}\frac{1+\tilde{\rho}^{(k)}}{1-\tilde{\rho}^{(k)}}\left(\alpha^{(k)}\right)^2\left(\zeta^2 + 2\beta^2\right)
$$

$$
+ \frac{1}{2}\Bigg[\left(\frac{1-\tilde{\rho}^{(k)}}{2} - \frac{2\eta_{\min}^{(k)}\Gamma_1^{(k)}}{\mu}\left(1 + \Gamma_1^{(k)}\right)\left(\Gamma_2^{(k)} - 1\right)\beta^2\alpha^{(k)}\right.
$$

$$
\left. - 3\frac{1+\tilde{\rho}^{(k)}}{1-\tilde{\rho}^{(k)}}\left(\alpha^{(k)}\right)^2\left(\zeta^2 + 2\beta^2\right)\right)^2
$$

$$
+ 12\frac{\left(1 + \Gamma_1^{(k)}\right)\beta^2}{\mu}\frac{1+\tilde{\rho}^{(k)}}{1-\tilde{\rho}^{(k)}}\left(\alpha^{(k)}\right)^3\left(\zeta^2 + 2\eta_{\min}^{(k)}\Gamma_1^{(k)}\frac{\beta^2}{\mu}\right)\Bigg]^{1/2}
$$

$$
= \frac{3 + \tilde{\rho}^{(k)}}{4} - A^{(k)}\alpha^{(k)} + B^{(k)}\left(\alpha^{(k)}\right)^2
$$

$$
+ \frac{1}{2}\sqrt{\left(\frac{1-\tilde{\rho}^{(k)}}{2} - 2\left(A^{(k)}\alpha^{(k)} + B^{(k)}\left(\alpha^{(k)}\right)^2\right)\right)^2 + C^{(k)}\left(\alpha^{(k)}\right)^3}.
$$

## J.2 PROOF OF LEMMA I.4

(a) Since $\alpha^{(k)} = \frac{\alpha^{(0)}}{\sqrt{1+\frac{k}{\gamma}}}$, we have

$$
\sum_{q=r}^{k} \alpha^{(q)} = \sum_{q=r}^{k} \frac{\alpha^{(0)}}{\sqrt{1+\frac{q}{\gamma}}} \geq \int_r^k \frac{\alpha^{(0)}\,dx}{\sqrt{1+\frac{x}{\gamma}}} \geq 2\alpha^{(0)}\left(\sqrt{1+\frac{k}{\gamma}} - \sqrt{1+\frac{r}{\gamma}}\right).
$$

(b) Based on the equation $\rho(\mathbf{\Phi}^{(k)}) = 1 - h(\alpha^{(k)})$ given in Proposition I.1, $h(\alpha^{(k)})$ was given as

$$
h(\alpha^{(k)}) = \frac{1-\tilde{\rho}}{4} + A\alpha^{(k)} - B\left(\alpha^{(k)}\right)^2
$$

$$
- \frac{1}{2}\sqrt{\left(\frac{1-\tilde{\rho}}{2} - 2\left(A\alpha^{(k)} + B\left(\alpha^{(k)}\right)^2\right)\right)^2 + C\left(\alpha^{(k)}\right)^3}.
$$

Further note that since we established $0 \leq \rho(\mathbf{\Phi}^{(k)}) < 1$ in Proposition I.1, it would mean $0 < h(\alpha^{(k)}) \leq 1$. Using triangle inequality, we have

$$
\begin{aligned}
h(\alpha^{(k)}) &\geq \frac{1 - \tilde{\rho}}{4} + A\alpha^{(k)} - B\left(\alpha^{(k)}\right)^2 - \left(\frac{1 - \tilde{\rho}}{4} - \left(A\alpha^{(k)} + B\left(\alpha^{(k)}\right)^2\right)\right) - \frac{\sqrt{C}}{2}\left(\alpha^{(k)}\right)^{3/2} \\
&\geq 2A\alpha^{(k)}.
\end{aligned}
$$

(c)

$$
\begin{aligned}
\sum_{r=1}^{k-1} \frac{\left(\alpha^{(r-1)}\right)^2}{\sum_{q=r}^{k} h(\alpha^{(q)})} &\leq \sum_{r=1}^{k} \frac{\left(\alpha^{(r-1)}\right)^2}{\sum_{q=r}^{k} 2A^{(k)}\alpha^{(q)}} = \frac{1}{2A^{(k)}} \sum_{r=1}^{k} \frac{\left(\frac{\alpha^{(0)}}{\sqrt{1+\frac{r-1}{\gamma}}}\right)^2}{\sum_{q=r}^{k} \alpha^{(q)}} \\
&\leq \frac{1}{2A^{(k)}} \sum_{r=1}^{k} \frac{\frac{(\alpha^{(0)})^2}{1+\frac{r-1}{\gamma}}}{2\alpha^{(0)}\left(\sqrt{1+\frac{k}{\gamma}} - \sqrt{1+\frac{r}{\gamma}}\right)} \\
&= \frac{\alpha^{(0)}}{4A^{(k)}} \sum_{r=1}^{k-1} \frac{1}{\left(1+\frac{r-1}{\gamma}\right)\left(\sqrt{1+\frac{k}{\gamma}} - \sqrt{1+\frac{r}{\gamma}}\right)} \\
&\leq \frac{\alpha^{(0)}}{4A^{(k)}} \sum_{r=1}^{k-1} \frac{1+\frac{1}{\gamma}}{\left(1+\frac{r}{\gamma}\right)\left(\sqrt{1+\frac{k}{\gamma}} - \sqrt{1+\frac{r}{\gamma}}\right)} \\
&\leq \frac{\alpha^{(0)}}{4A^{(k)}} \left[ \frac{1}{\sqrt{1+\frac{k}{\gamma}} - \sqrt{1+\frac{1}{\gamma}}} + \int_1^{k-1} \frac{\left(1+\frac{1}{\gamma}\right) dx}{\left(1+\frac{x}{\gamma}\right)\left(\sqrt{1+\frac{k}{\gamma}} - \sqrt{1+\frac{x}{\gamma}}\right)} \right].
\end{aligned}
$$

Focusing only on the integral and defining $u(x) = \sqrt{1+\frac{x}{\gamma}}$, we have

$$
\begin{aligned}
\int_1^{k-1} \frac{\left(1+\frac{1}{\gamma}\right) dx}{\left(1+\frac{x}{\gamma}\right)\left(\sqrt{1+\frac{k}{\gamma}} - \sqrt{1+\frac{x}{\gamma}}\right)} &= \int_1^{k-1} \frac{2\gamma\left(1+\frac{1}{\gamma}\right) du(x)}{u(x)(u(k) - u(x))} \\
&= \int_1^{k-1} \frac{2(\gamma+1)}{u(k)} \left(\frac{1}{u(x)} + \frac{1}{u(k) - u(x)}\right) du(x) \\
&= \frac{2(\gamma+1)}{u(k)} \left(\ln \frac{u(k-1)}{u(1)} + \ln \frac{u(k) - u(1)}{u(k) - u(k-1)}\right) \\
&= \frac{2(\gamma+1)}{u(k)} \ln \frac{u(k-1)u(k)}{u(k) - u(k-1)} = \frac{2(\gamma+1)}{u(k)} \left(\ln u(k) + \ln \frac{1}{\frac{u(k)}{u(k-1)} - 1}\right)
\end{aligned}
$$

$$
= \frac{2(\gamma+1)}{\sqrt{1+\frac{k}{\gamma}}} \left(\ln \sqrt{1+\frac{k}{\gamma}} + \ln \frac{1}{\sqrt{1+\frac{1}{\gamma+k-1}} - 1}\right).
$$

Putting everything back together concludes the proof.

### J.3 PROOF OF THEOREM I.5

First, using the fact that $\mathbf{\Phi}^{(k)}\|\cdot\| \leq \rho(\mathbf{\Phi^{(k)}})\|\cdot\|$ for each iteration $k \geq 0$, we can rewrite Eq. 8 to get

$$
\begin{bmatrix} \mathbb{E}_{\mathbf{\Xi}^{(k)}}\left[\left\|\bar{\theta}^{(k+1)} - \theta^\star\right\|^2\right] \\ \mathbb{E}_{\mathbf{\Xi}^{(k)}}\left[\left\|\mathbf{\Theta}^{(k+1)} - \mathbf{1}_m\bar{\theta}^{(k+1)}\right\|^2\right] \end{bmatrix} \leq \left(\prod_{q=0}^{k}\rho\left(\mathbf{\Phi}^{(q)}\right)\right)\begin{bmatrix} \left\|\bar{\theta}^{(0)} - \theta^\star\right\|^2 \\ \left\|\mathbf{\Theta}^{(0)} - \mathbf{1}_m\bar{\theta}^{(0)}\right\|^2 \end{bmatrix}
$$
$$
+ \sum_{r=1}^{k}\left(\prod_{q=r}^{k}\rho\left(\mathbf{\Phi}^{(q)}\right)\right)\left(\alpha^{(r-1)}\right)^2\begin{bmatrix} \frac{2(1-d_{\max}^{(0)})}{\mu\alpha^{(0)}}\left(1 + \mu\alpha^{(0)}\right)\delta^2 + \frac{\sigma^2}{m} \\ m\left(3\frac{1+\tilde{\rho}}{1-\tilde{\rho}}\delta^2 + \sigma^2\right) \end{bmatrix} + \mathbf{\Psi}^{(k)},
$$

$$(29)$$

where the $\mathbf{\Psi}^{(k)}$ matrix was written using Eq. 24.

Next, in order to obtain Eq. 22 when $k \to \infty$, we need to simplify each of the three terms in Eq. 29. The easiest one to show is the last term, i.e., $\mathbf{\Psi}^{(k)}$. Based on Eq. 24, both of its entries $\psi_1^{(k)}$ and $\psi_2^{(k)}$ have a factor $\left(\alpha^{(k)}\right)^2$ multiplied by a value that can be upper-bounded by a constant. Thus, we have

$$
\mathbf{\Psi}^{(k)} \leq \frac{\left(\alpha^{(0)}\right)^2}{1 + \frac{k}{\gamma}}\begin{bmatrix} \frac{2\left(1+\mu\alpha^{(0)}\right)\left(1-d_{\max}^{(0)}\right)}{\mu\alpha^{(0)}}\delta^2 + \frac{\sigma^2}{m} \\ m\left(3\frac{1+\tilde{\rho}}{1-\tilde{\rho}}\delta^2 + \sigma^2\right) \end{bmatrix}.
$$

Regarding the first and the second term, i.e., $\prod_{q=0}^{k}\rho\left(\mathbf{\Phi}^{(q)}\right)$ and $\sum_{r=1}^{k}\left(\prod_{q=r}^{k}\rho\left(\mathbf{\Phi}^{(q)}\right)\right)\left(\alpha^{(r-1)}\right)^2$, respectively, we have

$$
\prod_{q=0}^{k}\rho\left(\mathbf{\Phi}^{(q)}\right) = \prod_{q=0}^{k}\left(1 - h(\alpha^{(q)})\right) \leq \frac{1}{\sum_{q=0}^{k}h(\alpha^{(q)})},
$$

$$
\sum_{r=1}^{k}\left(\prod_{q=r}^{k}\rho\left(\mathbf{\Phi}^{(q)}\right)\right)\left(\alpha^{(r-1)}\right)^2 = \sum_{r=1}^{k}\left(\prod_{q=r}^{k}\left(1 - h(\alpha^{(q)})\right)\right)\left(\alpha^{(r-1)}\right)^2 \leq \sum_{r=1}^{k}\frac{\left(\alpha^{(r-1)}\right)^2}{\sum_{q=r}^{k}h(\alpha^{(q)})},
$$

in both of which Lemma I.3 was used, since $0 \leq h(\alpha^{(k)}) < 1$. Next, we employ Lemma I.4 on the above expressions. The proof easily follows. Note that $\ln\frac{1}{\sqrt{1+\frac{1}{\gamma+k-1}}-1} \leq \mathcal{O}(\ln k)$.

## K DIMINISHING LEARNING RATE POLICY FOR NON-CONVEX MODELS

In this appendix, we do the convergence analysis of our methodology under a diminishing learning rate policy, i.e., when $\alpha^{(k+1)} < \alpha^{(k)}$ for all $k \geq 0$. We will show that convergence to a stationary point with zero optimality error is possible if the frequency of SGDs, i.e., $d_i^{(k)}$, is increasing over time.

First, we rewrite the coefficients $\phi_{ij}^{(k)}$ and $\psi_i^{(k)}$ with $1 \leq i, j \leq 2$, using the fact that the SGD probability is chosen as $d_i^{(k)} = 1 - \eta_i^{(k)}\alpha^{(k)}$.

**Corollary K.1 (Average model performance and consensus error simplifications)** *(Corollary to Lemmas G.3 and G.4) Let Assumptions 4.2, 4.3 and 4.4 hold. If the SGD probabilities are chosen as $d_i^{(k)} = 1 - \eta_i^{(k)}\alpha^{(k)}$ with $0 \leq \eta_i^{(k)} \leq \frac{1}{\alpha^{(k)}}$ and $\eta_{\min}^{(k)} = \min_{i \in \mathcal{M}}\eta_i^{(k)}$, then the quantities in Lemma G.3 simplify to*

$\phi_{11}^{(k)} = \alpha^{(k)}[\frac{1}{2} - (\zeta^2\eta_{\min}^{(k)} + \beta(1 + 2\zeta^2\eta_{\min}^{(k)}))\alpha^{(k)}]$, $\phi_{12}^{(k)} = \frac{\beta^2}{2m}\alpha^{(k)}(1 + 2\beta\alpha^{(k)})$ *and* $\psi_1^{(k)} = [\eta_{\min}^{(k)}(1 + 2\beta\alpha^{(k)})\delta^2 + \frac{\beta\sigma^2}{2m}](\alpha^{(k)})^2$.

*Similarly, the quantities in Lemma G.4 also simplify to*

$\phi_{21}^{(k)} = 16\frac{1+\tilde{\rho}^{(k)}}{1-\tilde{\rho}^{(k)}}m\zeta^2(\alpha^{(k)})^2$, $\phi_{22}^{(k)} = \frac{1+\tilde{\rho}^{(k)}}{2} + 8\frac{1+\tilde{\rho}^{(k)}}{1-\tilde{\rho}^{(k)}}\beta^2(\alpha^{(k)})^2$ *and* $\psi_2^{(k)} = m(16\frac{1+\tilde{\rho}^{(k)}}{1-\tilde{\rho}^{(k)}}\delta^2 + \sigma^2)(\alpha^{(k)})^2$.

Next, using the simplified results derived in Corollary K.1 we further simplify the results of Proposition G.6 to arrive at the following Proposition.

**Proposition K.2 (Stationary point for non-convex models with a diminishing learning rate)**
*(See Appendix L.1 for the proof.) Let Assumptions 4.2 and 4.3 hold. If the SGD probabilities are chosen as $d_i^{(k)} = 1 - \eta_i^{(k)} \alpha^{(k)}$ and a non-increasing learning rate is used, i.e., $\alpha^{(k+1)} \leq \alpha^{(k)}$, then the constraints on the learning rate given in Proposition G.6 are simplified to*

$$\alpha^{(k)} < \min \left\{ \frac{\Gamma_3^{(k)}}{2\beta}, \frac{1}{\Gamma_1^{(k)}} \frac{1 - \tilde{\rho}^{(k)}}{4\beta\sqrt{1 + \tilde{\rho}^{(k)}}}, \frac{1}{\Gamma_2^{(k)}} \frac{\frac{1}{2}}{\zeta^2 \eta_{\min}^{(k)} + \beta \left(1 + 2\zeta^2 \eta_{\min}^{(k)}\right)}, \right.$$

$$\left. \frac{\sqrt{\left(1 - \frac{1}{(\Gamma_1^{(k)})^2}\right)\left(1 - \frac{1}{\Gamma_2^{(k)}}\right)}}{4\sqrt{2}\Gamma_4^{(k)}\sqrt{1 + \Gamma_3^{(k)}}} \frac{1}{\zeta\beta} \frac{1 - \tilde{\rho}^{(k)}}{\sqrt{1 + \tilde{\rho}^{(k)}}} \right\},$$

*where $\eta_{\min}^{(k)} = \min_{i \in \mathcal{M}} \eta_i^{(k)}$. Consequently, for each iteration $k \geq 0$, we get the following bound on the expected average model performance:*

$$\mathbb{E}_{\mathbf{\Xi}^{(k)}}[F(\bar{\theta}^{(k+1)})] \leq F(\bar{\theta}^{(0)}) - \sum_{r=0}^{k} \alpha^{(r)} w_1^{(r)} \mathbb{E}_{\mathbf{\Xi}^{(r-1)}}[\|\nabla F(\bar{\theta}^{(r)})\|^2] + \alpha^{(0)} w_2^{(0)} \|\mathbf{\Theta}^{(0)} - \mathbf{1}_m \bar{\theta}^{(0)}\|^2 + \sum_{r=0}^{k-1} \alpha^{(r)} w_2^{(r)} \psi_2^{(r)} + \sum_{r=0}^{k} \psi_1^{(r)},$$

*in which $w_1^{(k)} = \frac{1}{2}(1 - \frac{1}{\Gamma_2^{(k)}})(1 - \frac{1}{(\Gamma_4^{(k)})^2})$, $w_2^{(k)} = \frac{\beta^2 (1 + \Gamma_3^{(k)})}{m(1 - \tilde{\rho}^{(k)})(1 - \frac{1}{(\Gamma_1^{(k)})^2})}$, and the values of $\psi_1^{(k)}$ and $\psi_2^{(k)}$ were given in Corollary K.1.*

Note that the conditions laid out for the learning rate in Proposition K.2 have to be satisfied for all iterations $k \geq 0$. As one last step, we outline the sufficient conditions under which we can derive a single constraint on the initial value of the learning rate.

**Corollary K.3 (Constraint on diminishing learning rate initialization for non-convex models)**
*(Corollary to Proposition K.2) If the learning rate $\alpha^{(k)}$ is non-increasing, i.e., $\alpha^{(k+1)} \leq \alpha^{(k)}$, the SGD probabilities are determined as $d_i^{(k)} = 1 - \eta_i \alpha^{(k)}$ with $0 \leq \eta_i \leq \frac{1}{\alpha^{(0)}}$ and $\eta_{\min} = \min_{i \in \mathcal{M}} \eta_i$, for all $k \geq 0$, and we have constant aggregations probabilities $b_{ij}^{(k)} = b_{ij}$, then the constraints in Proposition K.2 simplify to*

$$\alpha^{(0)} < \min \left\{ \frac{\Gamma_3}{2\beta}, \frac{1}{\Gamma_1} \frac{1 - \tilde{\rho}}{4\beta\sqrt{1 + \tilde{\rho}}}, \frac{1}{\Gamma_2} \frac{\frac{1}{2}}{\zeta^2 \eta_{\min} + \beta(1 + 2\zeta^2 \eta_{\min})}, \frac{\sqrt{\left(1 - \frac{1}{\Gamma_1^2}\right)\left(1 - \frac{1}{\Gamma_2}\right)}}{4\sqrt{2}\Gamma_4 \sqrt{1 + \Gamma_3}} \frac{1}{\zeta\beta} \frac{1 - \tilde{\rho}}{\sqrt{1 + \tilde{\rho}}} \right\},$$

*This also results in time-invariant scalars $A = A^{(k)}$, $B = B^{(k)}$ and $C = C^{(k)}$, where these quantities were defined in Proposition K.2.*

With Corollary K.3, we have most of the ingredients required to prove the main Theorem. We will just need two extra upper bounds on summations involving the learning rate, as presented in the next Lemma.

**Lemma K.4 (Bounds for powers of learning rate sums)** *(See Appendix L.2 for the proof.) Let a diminishing learning rate $\alpha^{(k)} = \alpha^{(0)}/\sqrt{1 + k/\gamma}$ be used, which satisfies the properties in Eq. 21. Under the setup of Proposition K.2 for the SGD probabilities, i.e., $d_i^{(k)}$, if the aggregation probabilities are constant, i.e., $b_{ij}^{(k)} = b_{ij}$ for all $i \in \mathcal{M}$ and $(i, j) \in \mathcal{E}^{(k)}$, then the following bounds hold*

*(a) $\sum_{q=r}^{k} \left(\alpha^{(q)}\right)^2 \leq \left(\alpha^{(0)}\right)^2 (1 + \gamma \ln(1 + k/\gamma))$,*

*(b) $\sum_{q=r}^{k} \left(\alpha^{(q)}\right)^3 \leq \left(\alpha^{(0)}\right)^3 (1 + 2\gamma)$.*

**Theorem K.5 (Non-convex convergence result with a diminishing learning rate)** *(See Appendix L.3 for the proof.) Let Assumptions 4.2, 4.3 and 4.4 hold. If a diminishing learning rate policy $\alpha^{(k)} = \alpha^{(0)}/\sqrt{1 + k/\gamma}$ with $\gamma > 0$ satisfying the conditions outlined in Corollary K.3 is employed, and the SGD probabilities are set to $d_i^{(k)} = 1 - ((1 - d_i^{(0)})/\alpha^{(0)})\alpha^{(k)}$ for all $i \in \mathcal{M}$, while aggregation probabilities are set to constant values, i.e., $b_{ij}^{(k)} = b_{ij}$ for all $i \in \mathcal{M}$ and $(i, j) \in \mathcal{E}^{(k)}$, then we have*

$$\frac{1}{\sum_{r=0}^{K} \alpha^{(r)}} \sum_{r=0}^{K} \alpha^{(r)} \mathbb{E}_{\mathbf{\Xi}^{(r-1)}} \left[ \left\| \nabla F(\bar{\theta}^{(r)}) \right\|^2 \right] \leq \frac{1}{2w_1} \left[ \frac{F(\bar{\theta}^{(0)}) - F^\star}{\alpha^{(0)}} \mathcal{O}\left( \frac{1}{\sqrt{K}} \right) \right.$$

$$+ w_2 \left\| \mathbf{\Theta}^{(0)} - \mathbf{1}_m \bar{\theta}^{(0)} \right\|^2 \mathcal{O}\left( \frac{1}{\sqrt{K}} \right) + w_2 w_3 \left( \alpha^{(0)} \right)^2 (1 + 2\gamma) \mathcal{O}\left( \frac{1}{\sqrt{K}} \right) \quad (30)$$

$$\left. + w_4 \alpha^{(0)} \left( 1 + \gamma \mathcal{O}\left( \frac{\ln K}{\sqrt{K}} \right) \right) \right].$$

*in which $F^\star = \min_{\theta \in \mathbb{R}^n} F(\theta)$, and the values of scalars $w_i$ with $1 \leq i \leq 5$ are given in Appendix H.7. On letting $K \to \infty$, we obtain*

$$\lim_{K \to \infty} \frac{1}{\sum_{r=0}^{K} \alpha^{(r)}} \sum_{r=0}^{K} \alpha^{(r)} \mathbb{E}_{\mathbf{\Xi}^{(r-1)}} \left[ \left\| \nabla F(\bar{\theta}^{(r)}) \right\|^2 \right] = 0. \quad (31)$$

Eq. 30 in Theorem K.5 illustrates the convergence rate that can be achieved when a diminishing learning rate policy is used to train non-convex models. We recover the well-known $\mathcal{O}(\ln K/\sqrt{K})$ rate for DGD methods here. Furthermore, letting $K \to \infty$ in Eq. 30, we observe in Eq. 31 that the stationarity gap becomes zero.

## L PROOFS FOR DIMINISHING LEARNING RATE POLICY FOR NON-CONVEX MODELS

### L.1 PROOF OF PROPOSITION K.2

Considering the fact that the learning rate is diminishing, we can simplify the learning rate constraints in Proposition G.6. Using the updated coefficients derived in Corollary K.1, we rewrite the constraints from Appendix G.5 and Step 3 in Appendix H.6 to get

(i)

$$\phi_{22}^{(k)} < 1 \qquad \Rightarrow \qquad \alpha^{(k)} < \frac{1 - \tilde{\rho}^{(k)}}{4\sqrt{1 + \tilde{\rho}^{(k)}}\beta}$$

$$\Rightarrow \qquad \alpha^{(k)} \leq \frac{1}{\Gamma_1^{(k)}} \frac{1 - \tilde{\rho}^{(k)}}{4\sqrt{1 + \tilde{\rho}^{(k)}}\beta}, \qquad \Gamma_1^{(k)} > 1$$

$$\Rightarrow \qquad 1 - \phi_{22}^{(k)} \geq \frac{1 - \tilde{\rho}^{(k)}}{2} \left( 1 - \frac{1}{\left(\Gamma_1^{(k)}\right)^2} \right).$$

(ii)

$$\phi_{11}^{(k)} > 0 \qquad \Rightarrow \qquad 0 < \alpha^{(k)} < \frac{\frac{1}{2}}{\zeta^2 \eta_{\min}^{(k)} + \beta(1 + 2\zeta^2 \eta_{\min}^{(k)})}$$

$$\Rightarrow \qquad \alpha^{(k)} \leq \frac{1}{\Gamma_2^{(k)}} \frac{\frac{1}{2}}{\zeta^2 \eta_{\min}^{(k)} + \beta(1 + 2\zeta^2 \eta_{\min}^{(k)})} \qquad \Rightarrow \qquad \Gamma_2^{(k)} > 1$$

$$\Rightarrow \qquad \phi_{11}^{(k)} \geq \frac{1}{2} \left( 1 - \frac{1}{\Gamma_2^{(k)}} \right) \alpha^{(k)}.$$

(iii)

$$\alpha^{(k)} \le \frac{\Gamma_3^{(k)}}{2\beta}, \qquad \Gamma_3^{(k)} > 0$$

$$\Rightarrow \quad \phi_{12}^{(k)} \le \frac{\beta^2}{2m}(1+\Gamma_3^{(k)})\alpha^{(k)}, \qquad \psi_1^{(k)} \le \left[\eta_{\min}^{(k)}(1+\Gamma_3^{(k)})\delta^2 + \frac{\beta\sigma^2}{2m}\right]\left(\alpha^{(k)}\right)^2.$$

(iv)

$$\phi_{11}^{(k)} - \frac{\phi_{21}^{(k)}\phi_{12}^{(k)}}{1-\phi_{22}^{(k)}} > 0$$

$$\Rightarrow \quad \frac{1}{2}\left(1-\frac{1}{\Gamma_2^{(k)}}\right)\alpha^{(k)} - \frac{16(1+\tilde{\rho}^{(k)})\zeta^2\left(\alpha^{(k)}\right)^3\beta^2(1+\Gamma_3^{(k)})}{(1-\tilde{\rho}^{(k)})^2\left(1-\frac{1}{(\Gamma_1^{(k)})^2}\right)} > 0$$

$$\Rightarrow \quad \alpha^{(k)} < \frac{\sqrt{\left(1-\frac{1}{\Gamma_2^{(k)}}\right)\left(1-\frac{1}{(\Gamma_1^{(k)})^2}\right)}(1-\tilde{\rho}^{(k)})}{4\sqrt{2}\sqrt{1+\tilde{\rho}^{(k)}}\zeta\beta\sqrt{1+\Gamma_3^{(k)}}}$$

$$\Rightarrow \quad \alpha^{(k)} \le \frac{1}{\Gamma_4^{(k)}}\frac{\sqrt{\left(1-\frac{1}{\Gamma_2^{(k)}}\right)\left(1-\frac{1}{(\Gamma_1^{(k)})^2}\right)}(1-\tilde{\rho}^{(k)})}{4\sqrt{2}\sqrt{1+\tilde{\rho}^{(k)}}\zeta\beta\sqrt{1+\Gamma_3^{(k)}}}, \qquad \Gamma_4^{(k)} > 1$$

$$\Rightarrow \quad \phi_{11}^{(k)} - \frac{\phi_{21}^{(k)}\phi_{12}^{(k)}}{1-\phi_{22}^{(k)}} \ge \frac{1}{2}\left(1-\frac{1}{\Gamma_2^{(k)}}\right)\left(1-\frac{1}{(\Gamma_4^{(k)})^2}\right)\alpha^{(k)}.$$

Putting together all of these constraints concludes the proof.

### L.2 PROOF OF LEMMA K.4

(a)

$$\sum_{r=0}^{k}\left(\alpha^{(r)}\right)^2 \le \left(\alpha^{(0)}\right)^2 + \int_0^k \left(\alpha^{(x)}\right)^2 dx = \left(\alpha^{(0)}\right)^2 + \int_0^k \frac{\left(\alpha^{(0)}\right)^2}{1+x/\gamma}dx$$

$$= \left(\alpha^{(0)}\right)^2 + \left(\alpha^{(0)}\right)^2\gamma\ln\left(1+k/\gamma\right) = \left(\alpha^{(0)}\right)^2\left(1+\gamma\ln\left(1+k/\gamma\right)\right).$$

(b)

$$\sum_{r=0}^{k}\left(\alpha^{(r)}\right)^3 \le \left(\alpha^{(0)}\right)^3 + \int_0^k \left(\alpha^{(x)}\right)^3 dx = \left(\alpha^{(0)}\right)^3 + \int_0^k \frac{\left(\alpha^{(0)}\right)^3}{(1+x/\gamma)^{3/2}}dx$$

$$= \left(\alpha^{(0)}\right)^3 + 2\left(\alpha^{(0)}\right)^3\gamma\left(1-\frac{1}{\sqrt{1+k/\gamma}}\right) \le \left(\alpha^{(0)}\right)^3\left(1+2\gamma\right).$$

### L.3 PROOF OF THEOREM K.5

For each iteration $k \ge 0$, (i) the learning rate being diminishing, (ii) the fact that $d_i^{(k)} = 1 - ((1-d_i^{(0)})/\alpha^{(0)})\alpha^{(k)}$ and (iii) $b_{ij}^{(k)} = b_{ij}$ which results in $\tilde{\rho}^{(k)} = \tilde{\rho}$, results in Proposition K.2 implying

$$\mathbb{E}_{\Xi^{(k)}}\left[F(\bar{\theta}^{(k+1)})\right] \le F(\bar{\theta}^{(0)}) - w_1\sum_{r=0}^{k}\alpha^{(r)}\mathbb{E}_{\Xi^{(r-1)}}\left[\left\|\nabla F(\bar{\theta}^{(r)})\right\|^2\right] + \alpha^{(0)}w_2\left\|\Theta^{(0)} - \mathbf{1}_m\bar{\theta}^{(0)}\right\|^2$$

$$+ w_2\sum_{r=0}^{k-1}\alpha^{(r)}\psi_2^{(r)} + \sum_{r=0}^{k}\psi_1^{(r)}.$$

Substituting the values of $\psi_1^{(k)}$ and $\psi_2^{(k)}$ from Corollary K.1, then rearranging the inequality and dividing it by $\sum_{r=0}^{k} \alpha^{(r)}$, we get

$$\frac{1}{\sum_{r=0}^{k} \alpha^{(r)}} \sum_{r=0}^{k} \alpha^{(r)} \mathbb{E}_{\boldsymbol{\Xi}^{(r-1)}} \left[ \left\| \nabla F(\bar{\theta}^{(r)}) \right\|^2 \right] \leq \frac{1}{w_1 \sum_{r=0}^{k} \alpha^{(r)}} \left[ F(\bar{\theta}^{(0)}) - F^{\star} \right.$$

$$\left. + \alpha^{(0)} w_2 \left\| \boldsymbol{\Theta}^{(0)} - \mathbf{1}_m \bar{\theta}^{(0)} \right\|^2 + w_2 w_3 \sum_{r=0}^{k-1} \left( \alpha^{(r)} \right)^3 + w_4 \sum_{r=0}^{k} \left( \alpha^{(r)} \right)^2 \right].$$

in which $w_1 = \frac{1}{2}(1 - \frac{1}{\Gamma_2})(1 - \frac{1}{(\Gamma_4)^2})$, $w_2 = \frac{\beta^2(1+\Gamma_3)}{m(1-\tilde{\rho})(1-\frac{1}{(\Gamma_1)^2})}$, $w_3 = m(16\frac{1+\tilde{\rho}}{1-\tilde{\rho}}\delta^2 + \sigma^2)$ and $w_4 = \frac{1-d_{\max}^{(0)}}{\alpha^{(0)}}(1 + \Gamma_3)\delta^2 + \frac{\beta\sigma^2}{2m}$.

Finally, using Lemmas I.4-(a) and K.4, we conclude that

$$\frac{1}{\sum_{r=0}^{k} \alpha^{(r)}} \sum_{r=0}^{k} \alpha^{(r)} \mathbb{E}_{\boldsymbol{\Xi}^{(r-1)}} \left[ \left\| \nabla F(\bar{\theta}^{(r)}) \right\|^2 \right] \leq \frac{1}{2w_1 \alpha^{(0)} \left( \sqrt{1+k/\gamma} - 1 \right)} \left[ F(\bar{\theta}^{(0)}) - F^{\star} \right.$$

$$\left. + \alpha^{(0)} w_2 \left\| \boldsymbol{\Theta}^{(0)} - \mathbf{1}_m \bar{\theta}^{(0)} \right\|^2 + w_2 w_3 \left( \alpha^{(0)} \right)^3 (1 + 2\gamma) + w_4 \left( \alpha^{(0)} \right)^2 (1 + \gamma \ln(1 + k/\gamma)) \right].$$

## M  NON-CONVEX ANALYSIS UNDER THE PL CONDITION

In this appendix, we make the convergence analysis of our developed framework when non-convex ML models satisfying the PL condition are employed. Our approach will be quite similar to Sec. 4, with the key difference that we will use the milder Polyak-Lojasiewicz (PL) condition (Xin et al., 2021) (Assumption M.1) instead of strong convexity (Assumption 4.1-(b)) to make this generalization.

**Assumption M.1 (PL inequality)** *The global loss function $F$ meets the PL condition $\|\nabla F(\theta)\|^2 \geq 2\mu\left(F(\theta) - F^{\star}\right)$ with some $\mu > 0$, where $F^{\star}$ is the optimal value of $F$.*

Under this assumption, we further know that $F$ satisfies the quadratic growth condition (QG-condition) $\|\theta - \theta^{\star}\|^2 \leq (2/\mu)(F(\theta) - F^{\star})$, where $\theta^{\star}$ is the nearest point to the optimal solution of the minimization problem under consideration. This will also be useful in our analysis.

We will still characterize the expected consensus error as $\mathbb{E}_{\boldsymbol{\Xi}^{(k)}}[\|\boldsymbol{\Theta}^{(k+1)} - \mathbf{1}_m \bar{\theta}^{(k+1)}\|^2]$, but contrary to what was done in Sec. 4, the distance of the average model from the optimal solution will be captured via $\mathbb{E}_{\boldsymbol{\Xi}^{(k)}}[F(\bar{\theta}^{(k+1)}) - F^{\star}]$. As an alternative to Lemma 4.7, we first provide an upper bound on the expected error in the average model at each iteration for the non-convex case, i.e., $\mathbb{E}_{\boldsymbol{\Xi}^{(k)}}[F(\bar{\theta}^{(k+1)}) - F^{\star}]$, in Lemma M.3. Then, as an alternative to Lemma 4.8, we also calculate an upper bound on the consensus error for non-convex models, i.e., $\mathbb{E}_{\boldsymbol{\Xi}^{(k)}}[\|\boldsymbol{\Theta}^{(k+1)} - \mathbf{1}_m \bar{\theta}^{(k+1)}\|^2]$, in Corollary M.4.

We first need a preliminary Lemma which will be useful in the proof of Lemma M.3.

**Lemma M.2 (Gradient bounds under the PL condition)** *(See Appendix N.1 for the proof.) Let Assumptions 4.1-(a), 4.1-(c) and M.1 hold. The following upper bounds related to the gradient of the global loss function can be obtained in terms of the optimality error $\mathbb{E}_{\boldsymbol{\Xi}^{(k-1)}}[F(\bar{\theta}^{(k)}) - F^{\star}]$ and the consensus error $\mathbb{E}_{\boldsymbol{\Xi}^{(k-1)}}[\|\boldsymbol{\Theta}^{(k)} - \mathbf{1}_m \bar{\theta}^{(k)}\|^2]$.*

(a) $\mathbb{E}_{\boldsymbol{\Xi}^{(k)}}[\|\nabla F(\bar{\theta}^{(k)}) - \overline{\nabla v}^{(k)}\|^2] \leq \frac{4\beta^2}{\mu}(1 - d_{\min}^{(k)})\mathbb{E}_{\boldsymbol{\Xi}^{(k-1)}}[F(\bar{\theta}^{(k)}) - F^{\star}] + \frac{\beta^2 d_{\max}^{(k)}}{m}\mathbb{E}_{\boldsymbol{\Xi}^{(k-1)}}[\|\boldsymbol{\Theta}^{(k)} - \mathbf{1}_m \bar{\theta}^{(k)}\|^2] + 2(1 - d_{\min}^{(k)})\delta^2.$

(b) $-\mathbb{E}_{\boldsymbol{\Xi}^{(k)}}[\langle \nabla F(\bar{\theta}^{(k)}), \overline{\nabla v}^{(k)} \rangle] \leq -\mu(1 - \frac{2\beta^2}{\mu^2}(1 - d_{\min}^{(k)}))\mathbb{E}_{\boldsymbol{\Xi}^{(k-1)}}[F(\bar{\theta}^{(k)}) - F^{\star}] + \frac{\beta^2 d_{\max}^{(k)}}{2m}\mathbb{E}_{\boldsymbol{\Xi}^{(k-1)}}[\|\boldsymbol{\Theta}^{(k)} - \mathbf{1}_m \bar{\theta}^{(k)}\|^2] + (1 - d_{\min}^{(k)})\delta^2.$

*(c)* $\frac{1}{2}\mathbb{E}_{\boldsymbol{\Xi}^{(k)}}[\|\overline{\nabla v}^{(k)}\|^2] \quad\leq\quad \frac{2\beta^2}{\mu}(3 \quad-\quad 2d_{\min}^{(k)})\mathbb{E}_{\boldsymbol{\Xi}^{(k-1)}}[F(\bar{\theta}^{(k)}) - F^\star] \quad+$
$\frac{\beta^2 d_{\max}^{(k)}}{m}\mathbb{E}_{\boldsymbol{\Xi}^{(k-1)}}[\|\boldsymbol{\Theta}^{(k)} - \mathbf{1}_m\bar{\theta}^{(k)}\|^2] + 2(1 - d_{\min}^{(k)})\delta^2.$

Now can continue with the key lemma and corollary.

**Lemma M.3 (Average error for non-convex models satisfying the PL condition)** *(See Appendix N.2 for the proof.) Let Assumptions 4.1-(a), 4.1-(c), 4.3 and M.1 hold. For each iteration $k \geq 0$, we have the following bound on the expected average model error*

$$\mathbb{E}_{\boldsymbol{\Xi}^{(k)}}[F(\bar{\theta}^{(k+1)}) - F^\star] \leq \phi_{11}^{(k)}\mathbb{E}_{\boldsymbol{\Xi}^{(k-1)}}[F(\bar{\theta}^{(k)}) - F^\star] + \phi_{12}^{(k)}\mathbb{E}_{\boldsymbol{\Xi}^{(k-1)}}[\|\boldsymbol{\Theta}^{(k)} - \mathbf{1}_m\bar{\theta}^{(k)}\|^2] + \psi_1^{(k)},$$

*where $\phi_{11}^{(k)} = 1 + \frac{2\beta^3}{\mu}(3 - 2d_{\min}^{(k)})(\alpha^{(k)})^2 - \mu\alpha^{(k)}(1 - \frac{2\beta^2}{\mu^2}(1 - d_{\min}^{(k)}))$, $\phi_{12}^{(k)} = \frac{\beta^2 d_{\max}^{(k)}}{2m}\alpha^{(k)}(1 + 2\beta\alpha^{(k)})$, and*
$\psi_1^{(k)} = \alpha^{(k)}(1 - d_{\min}^{(k)})\delta^2 + \frac{\beta}{2}(\alpha^{(k)})^2[4(1 - d_{\min}^{(k)})\delta^2 + \frac{d_{\max}\sigma^2}{m}].$

Similar to our discussion around Lemma 4.7, note again how the coefficients simplify when $d_{\min}^{(k)} = 1$, which is essentially equivalent to the conventional DFL setup where clients perform SGDs at every iteration, i.e., $v_i^{(k)} = 1$ for all $i \in \mathcal{M}$.

We next bound the consensus error at each iteration, i.e., $\mathbb{E}_{\boldsymbol{\Xi}^{(k)}}[\|\boldsymbol{\Theta}^{(k+1)} - \mathbf{1}_m\bar{\theta}^{(k+1)}\|^2]$, which measures the deviation of ML model parameters of clients from the average non-convex ML model.

**Corollary M.4 (Consensus error for non-convex models satisfying the PL condition)**
*(Corollary to Lemma 4.8) Let Assumptions 4.1-(a), 4.1-(c), 4.3, 4.4 and M.1 hold. For each iteration $k \geq 0$, we have the following bound on the expected consensus error*

$$\mathbb{E}_{\boldsymbol{\Xi}^{(k)}}[\|\boldsymbol{\Theta}^{(k+1)} - \mathbf{1}_m\bar{\theta}^{(k+1)}\|^2] \leq \phi_{21}^{(k)}\mathbb{E}_{\boldsymbol{\Xi}^{(k-1)}}[F(\bar{\theta}^{(k)}) - F^\star] + \phi_{22}^{(k)}\mathbb{E}_{\boldsymbol{\Xi}^{(k-1)}}[\|\boldsymbol{\Theta}^{(k)} - \mathbf{1}_m\bar{\theta}^{(k)}\|^2] + \psi_2^{(k)},$$

*where $\phi_{21}^{(k)} = \frac{6}{\mu}\frac{1+\tilde{\rho}^{(k)}}{1-\tilde{\rho}^{(k)}}md_{\max}^{(k)}(\alpha^{(k)})^2(\zeta^2 + 2\beta^2(1 - d_{\min}^{(k)}))$,*

$\phi_{22}^{(k)} = \frac{1+\tilde{\rho}^{(k)}}{2} + 3\frac{1+\tilde{\rho}^{(k)}}{1-\tilde{\rho}^{(k)}}d_{\max}^{(k)}(\alpha^{(k)})^2(\zeta^2 + 2\beta^2)$, *and $\psi_2^{(k)} = m(\alpha^{(k)})^2 d_{\max}^{(k)}(3\frac{1+\tilde{\rho}^{(k)}}{1-\tilde{\rho}^{(k)}}\delta^2 + \sigma^2)$.*

**Proof.** *Lemma 4.8 states that* $\mathbb{E}_{\boldsymbol{\Xi}^{(k)}}[\|\boldsymbol{\Theta}^{(k+1)} - \mathbf{1}_m\bar{\theta}^{(k+1)}\|^2] \leq \phi_{21}^{(k)}\mathbb{E}_{\boldsymbol{\Xi}^{(k-1)}}[\|\bar{\theta}^{(k)} - \theta^\star\|^2] + \phi_{22}^{(k)}\mathbb{E}_{\boldsymbol{\Xi}^{(k-1)}}[\|\boldsymbol{\Theta}^{(k)} - \mathbf{1}_m\bar{\theta}^{(k)}\|^2] + \psi_2^{(k)}$. *Now, using the PL condition (Assumption M.1), we know that* $\|\bar{\theta}^{(k)} - \theta^\star\|^2 \leq \frac{2}{\mu}(F(\bar{\theta}^{(k)}) - F^\star)$. *Hence, we get* $\phi_{21}^{(k)} \leftarrow \frac{2}{\mu}\phi_{21}^{(k)}$.

Corollary M.4 is almost the same as Lemma 4.8, with the only difference being in $\phi_{21}^{(k)}$. Note again that $\phi_{21}^{(k)} = 0$ in the conventional DFL setup, where (a) $\zeta = 0$ and (b) $d_{\min}^{(k)} = 1$, resulting $d_i^{(k)} = 1$ for all $i \in \mathcal{M}$.

Let us denote the error vector at iteration $k$ with $\nu_{nc}^{(k)}$, defined as

$$\nu_{nc}^{(k)} = \begin{bmatrix} \mathbb{E}_{\boldsymbol{\Xi}^{(k-1)}}\left[F(\bar{\theta}^{(k)}) - F^\star\right] \\ \mathbb{E}_{\boldsymbol{\Xi}^{(k-1)}}\left[\|\boldsymbol{\Theta}^{(k)} - \mathbf{1}_m\bar{\theta}^{(k)}\|^2\right] \end{bmatrix}. \tag{32}$$

With this definition, putting the results of Lemmas 4.7 and 4.8 together form the following linear system of inequalities:

Putting the results of Lemma M.3 and Corollary M.4 together form the following linear system of inequalities:

$$\nu_{nc}^{(k+1)} \leq \boldsymbol{\Phi}^{(k)}\nu_{nc}^{(k)} + \boldsymbol{\Psi}^{(k)}, \tag{33}$$

with $\boldsymbol{\Phi}^{(k)} = [\phi_{ij}^{(k)}]_{1 \leq i,j \leq 2}$ and $\boldsymbol{\Psi}^{(k)} = [\psi_1^{(k)} \quad \psi_2^{(k)}]^T$. Recursively expanding the inequalities in Eq. 33 gives us an explicit relationship between the expected model error and consensus error at each iteration and their initial values:

$$\nu_{nc}^{(k+1)} \leq \boldsymbol{\Phi}^{(k:0)}\nu_{nc}^{(k)} + \sum_{r=1}^{k} \boldsymbol{\Phi}^{(k:r)}\boldsymbol{\Psi}^{(r-1)} + \boldsymbol{\Psi}^{(k)}, \tag{34}$$

where we have defined $\mathbf{\Phi}^{(k:s)} = \mathbf{\Phi}^{(k)}\mathbf{\Phi}^{(k-1)}\cdots\mathbf{\Phi}^{(s)}$ for $k > s$, and $\mathbf{\Phi}^{(k:k)} = \mathbf{\Phi}^{(k)}$.

In order for us formalize the convergence bound of `DSpodFL`, we have to show that the spectral radius of matrix $\mathbf{\Phi}^{(k)}$ given in Eq. 33 is less than one, i.e., $\rho(\mathbf{\Phi}^{(k)}) < 1$. This part was outlined in Proposition M.5 in the main text.

**Proposition M.5 (Spectral radius under the PL condition)** *(See Appendix N.3 for the proof.) Let Assumptions 4.1-(a), 4.1-(c), 4.3, 4.4 and M.1 hold. If the learning rate satisfies the following condition for all $k \geq 0$*

$$\alpha^{(k)} < \min\left\{ \frac{1 - \frac{2\beta^2}{\mu^2}\left(1 - d_{\min}^{(k)}\right)}{10\left(3 - 2d_{\min}^{(k)}\right)} \frac{\mu^2}{\beta^3}, \frac{5\left(1 + \tilde{\rho}^{(k)}\right)}{8\mu\left(1 - \frac{2\beta^2}{\mu^2}\left(1 - d_{\min}^{(k)}\right)\right)}, \right.$$

$$\left. \frac{1}{6\sqrt{2d_{\max}^{(k)}}} \frac{\mu}{\beta^2} \frac{1 - \tilde{\rho}^{(k)}}{\sqrt{1 + \tilde{\rho}^{(k)}}} \sqrt{\frac{1 - \frac{2\beta^2}{\mu^2}\left(1 - d_{\min}^{(k)}\right)}{1 + \frac{2\beta^2}{\zeta^2}\left(1 - d_{\min}^{(k)}\right)}} \right\},$$

*then we have $\rho(\mathbf{\Phi}^{(k)}) < 1$ for all $k \geq 0$, in which $\rho(\cdot)$ denotes the spectral radius of a given matrix, and $\mathbf{\Phi}^{(k)}$ is the linear system of inequalities of governing the dynamics of optimality and consensus errors. $\rho(\mathbf{\Phi}^{(k)})$ follows as $\rho(\mathbf{\Phi}^{(k)}) = \frac{3 + \tilde{\rho}^{(k)}}{4} - A^{(k)}\alpha^{(k)} + B^{(k)}(\alpha^{(k)})^2 + \frac{1}{2}\sqrt{\left(\frac{1 - \tilde{\rho}^{(k)}}{2} - 2(A^{(k)}\alpha^{(k)} + B^{(k)}(\alpha^{(k)})^2)\right)^2 + C^{(k)}(\alpha^{(k)})^3}$, where $A^{(k)} = \frac{2\mu}{5}(1 - \frac{2\beta^2}{\mu^2}(1 - d_{\min}^{(k)}))$, $B^{(k)} = \frac{3}{2}\frac{1 + \tilde{\rho}^{(k)}}{1 - \tilde{\rho}^{(k)}}(\zeta^2 + 2\beta^2)$ and $C^{(k)} = \frac{72\beta^2}{5\mu}\frac{1 + \tilde{\rho}^{(k)}}{1 - \tilde{\rho}^{(k)}}(d_{\max}^{(k)})^2(\zeta^2 + 2\beta^2(1 - d_{\min}^{(k)}))$.*

Proposition M.5 enables us to guarantee convergence of `DSpodFL` when non-convex models are used. The argument follows along the lines of the things we discussed in Sec. 4.4. Proposition M.5 implies that $\lim_{k \to \infty} \mathbf{\Phi}^{(k:0)} = 0$ in Eq. 34. However, this is only the asymptotic behavior of $\mathbf{\Phi}^{(k:0)}$, and the exact convergence rate will depend on the choice of the learning rate $\alpha^{(k)}$. Furthermore, since the first expression in Eq. 34 asymptotically approaches zero, Proposition M.5 also implies that the non-negative optimality gap is determined by the terms $\sum_{r=1}^{k} \mathbf{\Phi}^{(k:r)}\mathbf{\Psi}^{(r-1)} + \mathbf{\Psi}^{(k)}$, and it can be either zero or a positive value depending on the choice of $\alpha^{(k)}$.

Proposition M.5 outlines the necessary constraint on the learning rate $\alpha^{(k)}$ at each iteration $k \geq 0$. We next provide a corollary to Proposition M.5, in which we show that under certain conditions, the above-mentioned constraint needs to be satisfied only on the initial value of the learning rate, i.e, $\alpha^{(0)}$.

**Corollary M.6 (Constraint on learning rate initialization under the PL condition)** *(Corollary to Proposition M.5) If the learning rate $\alpha^{(k)}$ is non-increasing and the probabilities of SGDs $d_i^{(k)}$ and aggregations $b_{ij}^{(k)}$ are constant, i.e., $\alpha^{(k+1)} \leq \alpha^{(k)}$, $d_i^{(k)} = d_i$, $b_{ij}^{(k)} = b_{ij}$, for all $k \geq 0$, and we have then the constraints in Proposition M.5 simplify to*

$$\alpha^{(0)} < \min\left\{ \frac{1 - \frac{2\beta^2}{\mu^2}\left(1 - d_{\min}\right)}{10\left(3 - 2d_{\min}\right)} \frac{\mu^2}{\beta^3}, \frac{5}{8\mu\left(1 - \frac{2\beta^2}{\mu^2}\left(1 - d_{\min}\right)\right)}, \right.$$

$$\left. \frac{1}{6\sqrt{2d_{\max}}} \frac{\zeta}{\beta^2} \frac{1 - \tilde{\rho}}{\sqrt{1 + \tilde{\rho}}} \sqrt{\frac{\mu^2 - 2\beta^2\left(1 - d_{\min}\right)}{\zeta^2 + 2\beta^2\left(1 - d_{\min}\right)}} \right\}.$$

**Theorem M.7 (Non-convex result under the PL condition)** *(Proof is similar to the proof of Theorem 4.11 given in Appendix F.4.) Let Assumptions 4.1-(a), 4.1-(c) and 4.3, 4.4 and M.1 hold. Let a constant learning rate $\alpha^{(k)} = \alpha$ with $\alpha > 0$ satisfying the conditions of Proposition M.5 be employed, and the probabilities of SGDs and aggregations be time-invariant, i.e., $d_i^{(k)} = d_i$ and $b_{ij}^{(k)} = b_{ij}$, for all $k \geq 0$. Then, the convergence rate is geometric, specifically $\rho(\mathbf{\Phi})^{K+1}$, with an optimality gap*

$$\lim_{K \to \infty} \nu_{nc}^{(k+1)} \le \frac{1}{A} \left[ \begin{matrix} (1 - d_{\min})\delta^2 + \frac{\alpha\beta}{2} \left( 4 \left( 1 - d_{\min} \right) \delta^2 + \frac{\sigma^2}{m} \right) \\ m\alpha \left( 3\frac{1+\tilde{\rho}}{1-\tilde{\rho}} \delta^2 + \sigma^2 \right) \end{matrix} \right], \qquad (35)$$

*for $\boldsymbol{\Phi}$ given in Lemma M.3 and Corollary M.4, and $\nu_{nc}^{(k)}$ defined in Eq. 32. Proposition M.5 ensures $\rho(\boldsymbol{\Phi}) < 1$.*

We see that the optimality gap in Eq. 35 can be decreased by choosing a smaller learning rate $\alpha$ and a larger minimum SGD probability $d_{\min}$, similar to the argument made for Theorem 4.11.

# N    PROOFS FOR NON-CONVEX ANALYSIS UNDER THE PL CONDITION

## N.1    PROOF OF LEMMA M.2

(a) For this deviation term, we have

$$\left\| \nabla F(\bar{\theta}^{(k)}) - \overline{\nabla v}^{(k)} \right\|^2 = \left\| \frac{1}{m} \sum_{i=1}^{m} \left( \nabla F_i(\bar{\theta}^{(k)}) - \nabla F_i(\theta_i^{(k)}) v_i^{(k)} \right) \right\|^2$$

$$\le \frac{1}{m} \sum_{i=1}^{m} \left\| \nabla F_i(\bar{\theta}^{(k)}) - \nabla F_i(\theta_i^{(k)}) v_i^{(k)} \right\|^2$$

$$= \frac{1}{m} \sum_{\substack{i=1 \\ v_i^{(k)}=1}}^{m} \left\| \nabla F_i(\bar{\theta}^{(k)}) - \nabla F_i(\theta_i^{(k)}) \right\|^2 + \frac{1}{m} \sum_{\substack{i=1 \\ v_i^{(k)}=0}}^{m} \left\| \nabla F_i(\bar{\theta}^{(k)}) \right\|^2$$

$$\le \frac{1}{m} \sum_{\substack{i=1 \\ v_i^{(k)}=1}}^{m} \beta_i^2 \left\| \bar{\theta}^{(k)} - \theta_i^{(k)} \right\|^2 + \frac{2}{m} \sum_{\substack{i=1 \\ v_i^{(k)}=0}}^{m} \left( \beta_i^2 \left\| \bar{\theta}^{(k)} - \theta^\star \right\|^2 + \delta_i^2 \right)$$

$$= \frac{1}{m} \sum_{i=1}^{m} \beta_i^2 \left\| \bar{\theta}^{(k)} - \theta_i^{(k)} \right\|^2 v_i^{(k)} + \frac{2}{m} \sum_{i=1}^{m} \left( \frac{2\beta_i^2}{\mu} \left( F(\bar{\theta}^{(k)}) - F^\star \right) + \delta_i^2 \right) \left( 1 - v_i^{(k)} \right),$$

where in the last two lines, (i) Smoothness (Assumption 4.1-(a)) and Lemma D.1-(b) and (ii) PL condition (Assumption M.1) was used, respectively. Taking the expected value of the above inequality concludes the proof.

(b) Second, for this inner product term, we have

$$- \left\langle \nabla F(\bar{\theta}^{(k)}), \overline{\nabla v}^{(k)} \right\rangle = - \left\langle \nabla F(\bar{\theta}^{(k)}), \overline{\nabla v}^{(k)} - \nabla F(\bar{\theta}^{(k)}) + \nabla F(\bar{\theta}^{(k)}) \right\rangle$$

$$= - \left\| \nabla F(\bar{\theta}^{(k)}) \right\|^2 + \left\langle \nabla F(\bar{\theta}^{(k)}), \nabla F(\bar{\theta}^{(k)}) - \overline{\nabla v}^{(k)} \right\rangle$$

$$\le \frac{-1}{2} \left\| \nabla F(\bar{\theta}^{(k)}) \right\|^2 + \frac{1}{2} \left\| \nabla F(\bar{\theta}^{(k)}) - \overline{\nabla v}^{(k)} \right\|^2.$$

Now, taking the expected value of this inequality and using part (a) of this lemma alongside the PL condition (Assumption M.1), we get

$$-\mathbb{E}_{\boldsymbol{\Xi}^{(k)}}\left[\left\langle\nabla F(\bar{\theta}^{(k)}),\overline{\nabla v}^{(k)}\right\rangle\right] \leq -\mu\mathbb{E}_{\boldsymbol{\Xi}^{(k-1)}}\left[F(\bar{\theta}^{(k)}) - F^{\star}\right]$$

$$+ \frac{\beta^2 d_{\max}^{(k)}}{2m}\mathbb{E}_{\boldsymbol{\Xi}^{(k-1)}}\left[\left\|\boldsymbol{\Theta}^{(k)} - \mathbf{1}_m\bar{\theta}^{(k)}\right\|^2\right]$$

$$+ \frac{2\beta^2}{\mu}\left(1 - d_{\min}^{(k)}\right)\mathbb{E}_{\boldsymbol{\Xi}^{(k-1)}}\left[F(\bar{\theta}^{(k)}) - F^{\star}\right]$$

$$+ \left(1 - d_{\min}^{(k)}\right)\delta^2$$

$$\leq -\mu\left(1 - \frac{2\beta^2}{\mu^2}\left(1 - d_{\min}^{(k)}\right)\right)\mathbb{E}_{\boldsymbol{\Xi}^{(k-1)}}\left[F(\bar{\theta}^{(k)}) - F^{\star}\right] + \frac{\beta^2 d_{\max}^{(k)}}{2m}\mathbb{E}_{\boldsymbol{\Xi}^{(k-1)}}\left[\left\|\boldsymbol{\Theta}^{(k)} - \mathbf{1}_m\bar{\theta}^{(k)}\right\|^2\right]$$

$$+ \left(1 - d_{\min}^{(k)}\right)\delta^2.$$

(c) Finally, for the norm term, we have

$$\frac{1}{2}\left\|\overline{\nabla v}^{(k)}\right\|^2 = \frac{1}{2}\left\|\overline{\nabla v}^{(k)} - \nabla F(\bar{\theta}^{(k)}) + \nabla F(\bar{\theta}^{(k)})\right\|^2 \leq \left\|\nabla F(\bar{\theta}^{(k)})\right\|^2 + \left\|\nabla F(\bar{\theta}^{(k)}) - \overline{\nabla v}^{(k)}\right\|^2.$$

Taking the expected value of this inequality and utilizing part (a) of this lemma alongside the PL condition (Assumption M.1)

$$\frac{1}{2}\mathbb{E}_{\boldsymbol{\Xi}^{(k)}}\left[\left\|\overline{\nabla v}^{(k)}\right\|^2\right] \leq \frac{2\beta^2}{\mu}\mathbb{E}_{\boldsymbol{\Xi}^{(k-1)}}\left[F(\bar{\theta}^{(k)}) - F^{\star}\right] + \frac{\beta^2 d_{\max}^{(k)}}{m}\mathbb{E}_{\boldsymbol{\Xi}^{(k-1)}}\left[\left\|\boldsymbol{\Theta}^{(k)} - \mathbf{1}_m\bar{\theta}^{(k)}\right\|^2\right]$$

$$+ \frac{4\beta^2}{\mu}\left(1 - d_{\min}^{(k)}\right)\mathbb{E}_{\boldsymbol{\Xi}^{(k-1)}}\left[F(\bar{\theta}^{(k)}) - F^{\star}\right] + 2\left(1 - d_{\min}^{(k)}\right)\delta^2$$

$$\leq \frac{2\beta^2}{\mu}\left(3 - 2d_{\min}^{(k)}\right)\mathbb{E}_{\boldsymbol{\Xi}^{(k-1)}}\left[F(\bar{\theta}^{(k)}) - F^{\star}\right] + \frac{\beta^2 d_{\max}^{(k)}}{m}\mathbb{E}_{\boldsymbol{\Xi}^{(k-1)}}\left[\left\|\boldsymbol{\Theta}^{(k)} - \mathbf{1}_m\bar{\theta}^{(k)}\right\|^2\right]$$

$$+ 2\left(1 - d_{\min}^{(k)}\right)\delta^2.$$

## N.2 PROOF OF LEMMA M.3

Using Lemma D.1-(b) on $\bar{\theta}^{(k)}$, the average model parameters at iteration $k$, and then employing Assumption M.1, we get

$$\left\|\nabla F(\bar{\theta}^{(k)})\right\|^2 \leq \beta^2\left\|\bar{\theta}^{(k)} - \theta^{\star}\right\|^2 \leq \frac{2\beta^2}{\mu}\left(F(\bar{\theta}^{(k)}) - F^{\star}\right).$$

Now, we can write

$$F(\bar{\theta}^{(k+1)}) - F^{\star} \leq F(\bar{\theta}^{(k)}) + \left\langle\nabla F(\bar{\theta}^{(k)}), \bar{\theta}^{(k+1)} - \bar{\theta}^{(k)}\right\rangle + \frac{\beta}{2}\left\|\bar{\theta}^{(k)} - \bar{\theta}^{(k+1)}\right\|^2 - F^{\star}$$

$$= F(\bar{\theta}^{(k)}) + \left\langle\nabla F(\bar{\theta}^{(k)}), -\alpha^{(k)}\overline{gv}^{(k)}\right\rangle + \frac{\beta}{2}\left\|\alpha^{(k)}\overline{gv}^{(k)}\right\|^2 - F^{\star}$$

$$= F(\bar{\theta}^{(k)}) - F^{\star} - \alpha^{(k)}\left\langle\nabla F(\bar{\theta}^{(k)}), \overline{\nabla v}^{(k)}\right\rangle - \alpha^{(k)}\left\langle\nabla F(\bar{\theta}^{(k)}), \overline{\epsilon v}^{(k)}\right\rangle$$

$$+ \frac{\beta}{2}\left(\alpha^{(k)}\right)^2\left\|\overline{\nabla v}^{(k)}\right\|^2 + \frac{\beta}{2}\left(\alpha^{(k)}\right)^2\left\|\overline{\epsilon v}^{(k)}\right\|^2$$

$$+ \beta\left(\alpha^{(k)}\right)^2\left\langle\overline{\nabla v}^{(k)}, \overline{\epsilon v}^{(k)}\right\rangle,$$

in which the relationship in each of the three lines follow from (i) Smoothness (Assumption 4.1-(a)), (ii) Eq. 5, (iii) $\mathbf{g}_i^{(k)} = \nabla_i^{(k)} + \epsilon_i^{(k)}$ for all $i \in \mathcal{M}$. Next, we take the expected value of the above inequality and use Assumption 4.3, and Lemmas D.2 and M.2 to get

$$\mathbb{E}_{\mathbf{\Xi}^{(k)}}\left[F(\bar{\theta}^{(k+1)}) - F^\star\right] \le \mathbb{E}_{\mathbf{\Xi}^{(k-1)}}\left[F(\bar{\theta}^{(k)}) - F^\star\right]$$

$$- \mu\alpha^{(k)}\left(1 - \frac{2\beta^2}{\mu^2}\left(1 - d_{\min}^{(k)}\right)\right)\mathbb{E}_{\mathbf{\Xi}^{(k-1)}}\left[F(\bar{\theta}^{(k)}) - F^\star\right]$$

$$+ \frac{\beta^2 d_{\max}^{(k)}}{2m}\alpha^{(k)}\mathbb{E}_{\mathbf{\Xi}^{(k-1)}}\left[\left\|\mathbf{\Theta}^{(k)} - \mathbf{1}_m\bar{\theta}^{(k)}\right\|^2\right] + \alpha^{(k)}\left(1 - d_{\min}^{(k)}\right)\delta^2$$

$$+ \frac{2\beta^3}{\mu}\left(3 - 2d_{\min}^{(k)}\right)\left(\alpha^{(k)}\right)^2\mathbb{E}_{\mathbf{\Xi}^{(k-1)}}\left[F(\bar{\theta}^{(k)}) - F^\star\right]$$

$$+ \frac{\beta^3 d_{\max}^{(k)}}{m}\left(\alpha^{(k)}\right)^2\mathbb{E}_{\mathbf{\Xi}^{(k-1)}}\left[\left\|\mathbf{\Theta}^{(k)} - \mathbf{1}_m\bar{\theta}^{(k)}\right\|^2\right] + 2\beta\left(1 - d_{\min}^{(k)}\right)\delta^2\left(\alpha^{(k)}\right)^2$$

$$+ \frac{\beta}{2}\left(\alpha^{(k)}\right)^2 d_{\max}^{(k)}\frac{\sigma^2}{m}$$

$$\le \left[1 + \frac{2\beta^3}{\mu}\left(3 - 2d_{\min}^{(k)}\right)\left(\alpha^{(k)}\right)^2 - \mu\alpha^{(k)}\left(1 - \frac{2\beta^2}{\mu^2}\left(1 - d_{\min}^{(k)}\right)\right)\right]\mathbb{E}_{\mathbf{\Xi}^{(k-1)}}\left[F(\bar{\theta}^{(k)}) - F^\star\right]$$

$$+ \frac{\beta^2 d_{\max}^{(k)}}{2m}\alpha^{(k)}\left(1 + 2\beta\alpha^{(k)}\right)\mathbb{E}_{\mathbf{\Xi}^{(k-1)}}\left[\left\|\mathbf{\Theta}^{(k)} - \mathbf{1}_m\bar{\theta}^{(k)}\right\|^2\right] + \alpha^{(k)}\left(1 - d_{\min}^{(k)}\right)\delta^2$$

$$+ \frac{\beta}{2}\left(\alpha^{(k)}\right)^2\left[4\left(1 - d_{\min}^{(k)}\right)\delta^2 + \frac{d_{\max}\sigma^2}{m}\right].$$

### N.3 PROOF OF PROPOSITION M.5

We will do an analysis similar to the proof of Propositions 4.10 and I.1, which were given in Appendices F.3 and J.1, respectively.

**Step 1: Setting up the proof.** We skip repeating the explanations for this step, as they are exactly the same as step 1 in Appendix F.3.

**Step 2: Simplifying the conditions.** Recall that we have to ensure (i) $0 < \phi_{11}^{(k)} \le 1$ and (ii) $0 < \phi_{22}^{(k)} \le 1$. For $\phi_{11}^{(k)}$ as defined in Lemma M.3, we have

$$\phi_{11}^{(k)} \le 1 \quad \Rightarrow \quad \alpha^{(k)} \le \frac{\mu^2\left(1 - \frac{2\beta^2}{\mu^2}\left(1 - d_{\min}^{(k)}\right)\right)}{2\beta^3\left(3 - 2d_{\min}^{(k)}\right)} \quad \Rightarrow \quad d_{\min}^{(k)} > 1 - \frac{\mu^2}{2\beta^2}.$$

We can see that we got a requirement for $d_{\min}^{(k)}$ here, and it should be lower bounded. Therefore, contrary to when strongly convex models were being used that $d_i^{(k)}$ could have had any value for all $i \in \mathcal{M}$ and $k \ge 0$, when using a non-convex model this is no longer the case, and $d_i^{(k)}$ have to be larger than a threshold $1 - \frac{\mu^2}{2\beta^2}$. To put this into better context, note that $1 - \frac{\mu^2}{2\beta^2} > \frac{1}{2}$, and thus at the best possible scenario we can allow $d_i^{(k)} > \frac{1}{2}$.

We then put the following constraint on $\alpha^{(k)}$ to get a more compact form for $\phi_{11}^{(k)}$, defined in Lemma M.3. We have

$$\text{Constraint 1: } \alpha^{(k)} \le \Gamma_1^{(k)}\frac{\mu^2\left(1 - \frac{2\beta^2}{\mu^2}\left(1 - d_{\min}^{(k)}\right)\right)}{2\beta^3\left(3 - 2d_{\min}^{(k)}\right)}, \qquad 0 < \Gamma_1^{(k)} \le 1,$$

Note that although the above constraint has to be satisfied for $\alpha^{(k)}$, we also obtain an upper bound for the condition for theoretical analysis purposes. We have

$$\alpha^{(k)} \le \Gamma_1^{(k)}\frac{\mu^2}{2\beta^3} \le \frac{\Gamma_1^{(k)}}{2\beta}.$$

Hence, we can update $\phi_{11}^{(k)}$ and $\phi_{12}^{(k)}$ as defined in Lemma M.3 as the follows

$$\phi_{11}^{(k)} \le 1 - \left(1 - \Gamma_1^{(k)}\right)\left(1 - \frac{2\beta^2}{\mu^2}\left(1 - d_{\min}^{(k)}\right)\right)\mu\alpha^{(k)}, \qquad \phi_{12}^{(k)} \le \frac{\beta^2 d_{\max}^{(k)}}{2m}\left(1 + \Gamma_1^{(k)}\right)\alpha^{(k)}.$$

Note that other entries of matrices $\mathbf{\Phi}^{(k)}$ and $\mathbf{\Psi}^{(k)}$ remain the same as initially given in Lemma M.3 and Corollary M.4. Moreover, since matrix $\mathbf{\Phi}^{(k)}$ and vector $\mathbf{\Psi}^{(k)}$ in Eq. 33 were used as upper bounds, therefore we can always replace their values with new upper bounds for them. Consequently, with this new value for $\phi_{11}^{(k)}$, we continue as

$$\phi_{11}^{(k)} > 0 \qquad \Rightarrow \qquad \alpha^{(k)} < \frac{1}{\mu\left(1 - \Gamma_1^{(k)}\right)\left(1 - \frac{2\beta^2}{\mu^2}\left(1 - d_{\min}^{(k)}\right)\right)}.$$

Finally, we check the next condition $0 < \phi_{22}^{(k)} \le 1$. Noting that we have $\frac{3 + \tilde{\rho}^{(k)}}{4} < 1$, we can enforce $\phi_{22}^{(k)} \le 1$ by setting $\phi_{22}^{(k)} \le \frac{3 + \tilde{\rho}^{(k)}}{4}$. We have

$$\frac{1 + \tilde{\rho}^{(k)}}{2} \le \phi_{22}^{(k)} \le \frac{3 + \tilde{\rho}^{(k)}}{4} \qquad \Rightarrow \qquad 0 \le \alpha^{(k)} \le \frac{1}{2\sqrt{3d_{\max}^{(k)}}}\frac{1 - \tilde{\rho}^{(k)}}{\sqrt{1 + \tilde{\rho}^{(k)}}}\frac{1}{\sqrt{\zeta^2 + 2\beta^2}}.$$

**Step 3: Determining the constraints.** Having made sure that (i) $0 < \phi_{11}^{(k)} \le 1$ and (ii) $0 < \phi_{22}^{(k)} \le 1$ in the previous step, we can continue to solve Eq. 15. For the left-hand side of the inequality, we have

$$\left(1 - \phi_{11}^{(k)}\right)\left(1 - \phi_{22}^{(k)}\right) = \left[\left(1 - \Gamma_1^{(k)}\right)\left(1 - \frac{2\beta^2}{\mu^2}\left(1 - d_{\min}^{(k)}\right)\right)\mu\alpha^{(k)}\right]\left(1 - \phi_{22}^{(k)}\right)$$

$$\ge \left[\left(1 - \Gamma_1^{(k)}\right)\left(1 - \frac{2\beta^2}{\mu^2}\left(1 - d_{\min}^{(k)}\right)\right)\mu\alpha^{(k)}\right]\frac{1 - \tilde{\rho}^{(k)}}{4}$$

Now, putting this back to Eq. 15, we get

$$\left[\left(1 - \Gamma_1^{(k)}\right)\left(1 - \frac{2\beta^2}{\mu^2}\left(1 - d_{\min}^{(k)}\right)\right)\mu\alpha^{(k)}\right]\frac{1 - \tilde{\rho}^{(k)}}{4} > \phi_{12}^{(k)}\phi_{21}^{(k)}$$

$$\Rightarrow \qquad \left[\frac{\beta^2 d_{\max}^{(k)}}{2m}\left(1 + \Gamma_1^{(k)}\right)\alpha^{(k)}\right]\left[\frac{6}{\mu}\frac{1 + \tilde{\rho}^{(k)}}{1 - \tilde{\rho}^{(k)}}md_{\max}^{(k)}\left(\alpha^{(k)}\right)^2\left(\zeta^2 + 2\beta^2\left(1 - d_{\min}^{(k)}\right)\right)\right]$$

$$< \left[\left(1 - \Gamma_1^{(k)}\right)\left(1 - \frac{2\beta^2}{\mu^2}\left(1 - d_{\min}^{(k)}\right)\right)\mu\alpha^{(k)}\right]\frac{1 - \tilde{\rho}^{(k)}}{4}$$

$$\Rightarrow \qquad \alpha^{(k)} < \frac{1}{2\sqrt{3}d_{\max}^{(k)}}\sqrt{\frac{1 - \Gamma_1^{(k)}}{1 + \Gamma_1^{(k)}}}\frac{1 - \tilde{\rho}^{(k)}}{\sqrt{1 + \tilde{\rho}^{(k)}}}\frac{\mu}{\beta\zeta}\sqrt{\frac{1 - \frac{2\beta^2}{\mu^2}\left(1 - d_{\min}^{(k)}\right)}{1 + \frac{2\beta^2}{\zeta^2}\left(1 - d_{\min}^{(k)}\right)}}.$$

Finally, we solve for Eq. 16, i.e., $c = \phi_{11}^{(k)}\phi_{22}^{(k)} - \phi_{12}^{(k)}\phi_{21}^{(k)} > 0$. Noting that by solving Eq. 15 we made sure that $1 - \phi_{11}^{(k)} - \phi_{22}^{(k)} + \phi_{11}^{(k)}\phi_{22}^{(k)} - \phi_{12}^{(k)}\phi_{21}^{(k)} > 0$, we can write

$$c > 0 \qquad \Rightarrow \qquad \phi_{11}^{(k)} + \phi_{22}^{(k)} - 1 > 0$$

$$\Rightarrow \qquad 1 - \left(1 - \Gamma_1^{(k)}\right)\left(1 - \frac{2\beta^2}{\mu^2}\left(1 - d_{\min}^{(k)}\right)\right)\mu\alpha^{(k)} + \frac{1 + \tilde{\rho}^{(k)}}{2} - 1 > 0$$

$$\Rightarrow \qquad \alpha^{(k)} < \frac{1 + \tilde{\rho}^{(k)}}{2\mu\left(1 - \Gamma_1^{(k)}\right)\left(1 - \frac{2\beta^2}{\mu^2}\left(1 - d_{\min}^{(k)}\right)\right)},$$

in which we have used the value of $\phi_{11}^{(k)}$ itself, but the lower bound of $\phi_{22}^{(k)}$.

**Step 4: Putting all the constraints together.** Reviewing all the constraints on $\alpha^{(k)}$ from the beginning of this appendix, we can collect all of the constraints together and simplify them as

$$
\alpha^{(k)} < \min \left\{ \Gamma_1^{(k)} \frac{\mu^2 \left(1 - \frac{2\beta^2}{\mu^2}\left(1 - d_{\min}^{(k)}\right)\right)}{2\beta^3 \left(3 - 2d_{\min}^{(k)}\right)}, \frac{1}{\mu \left(1 - \Gamma_1^{(k)}\right)\left(1 - \frac{2\beta^2}{\mu^2}\left(1 - d_{\min}^{(k)}\right)\right)}, \right.
$$
$$
\frac{1}{2\sqrt{3d_{\max}^{(k)}}} \frac{1 - \tilde{\rho}^{(k)}}{\sqrt{1 + \tilde{\rho}^{(k)}}} \frac{1}{\sqrt{\zeta^2 + 2\beta^2}},
$$
$$
\frac{1}{2\sqrt{3d_{\max}^{(k)}}} \sqrt{\frac{1 - \Gamma_1^{(k)}}{1 + \Gamma_1^{(k)}}} \frac{1 - \tilde{\rho}^{(k)}}{\sqrt{1 + \tilde{\rho}^{(k)}}} \frac{\mu}{\beta\zeta} \sqrt{\frac{1 - \frac{2\beta^2}{\mu^2}\left(1 - d_{\min}^{(k)}\right)}{1 + \frac{2\beta^2}{\zeta^2}\left(1 - d_{\min}^{(k)}\right)}},
$$
$$
\left. \frac{1 + \tilde{\rho}^{(k)}}{2\mu \left(1 - \Gamma_1^{(k)}\right)\left(1 - \frac{2\beta^2}{\mu^2}\left(1 - d_{\min}^{(k)}\right)\right)} \right\} \tag{36}
$$
$$
= \min \left\{ \Gamma_1^{(k)} \frac{\mu^2 \left(1 - \frac{2\beta^2}{\mu^2}\left(1 - d_{\min}^{(k)}\right)\right)}{2\beta^3 \left(3 - 2d_{\min}^{(k)}\right)}, \frac{1 + \tilde{\rho}^{(k)}}{2\mu \left(1 - \Gamma_1^{(k)}\right)\left(1 - \frac{2\beta^2}{\mu^2}\left(1 - d_{\min}^{(k)}\right)\right)}, \right.
$$
$$
\frac{1}{2\sqrt{3d_{\max}^{(k)}}} \frac{1 - \tilde{\rho}^{(k)}}{\sqrt{1 + \tilde{\rho}^{(k)}}} \frac{1}{\sqrt{\zeta^2 + 2\beta^2}},
$$
$$
\left. \frac{1}{2\sqrt{3d_{\max}^{(k)}}} \sqrt{\frac{1 - \Gamma_1^{(k)}}{1 + \Gamma_1^{(k)}}} \frac{1 - \tilde{\rho}^{(k)}}{\sqrt{1 + \tilde{\rho}^{(k)}}} \frac{\mu}{\beta\zeta} \sqrt{\frac{1 - \frac{2\beta^2}{\mu^2}\left(1 - d_{\min}^{(k)}\right)}{1 + \frac{2\beta^2}{\zeta^2}\left(1 - d_{\min}^{(k)}\right)}} \right\}
$$

while satisfying

$$
0 < \Gamma_1^{(k)} \leq 1.
$$

Note that one of the terms in Eq. 36 was trivially removed since $\frac{1 + \tilde{\rho}^{(k)}}{2} < 1$. Furthermore, for the last two terms, we have

(a) $\quad \dfrac{1}{2\sqrt{3d_{\max}^{(k)}}} \dfrac{1 - \tilde{\rho}^{(k)}}{\sqrt{1 + \tilde{\rho}^{(k)}}} \dfrac{1}{\sqrt{\zeta^2 + 2\beta^2}} > \dfrac{1}{2\sqrt{3d_{\max}^{(k)}}} \dfrac{1 - \tilde{\rho}^{(k)}}{\sqrt{1 + \tilde{\rho}^{(k)}}} \dfrac{1}{\sqrt{6}\beta},$

(b) $\quad \dfrac{1}{2\sqrt{3d_{\max}^{(k)}}} \sqrt{\dfrac{1 - \Gamma_1^{(k)}}{1 + \Gamma_1^{(k)}}} \dfrac{1 - \tilde{\rho}^{(k)}}{\sqrt{1 + \tilde{\rho}^{(k)}}} \dfrac{\mu}{\beta\zeta} \sqrt{\dfrac{1 - \frac{2\beta^2}{\mu^2}\left(1 - d_{\min}^{(k)}\right)}{1 + \frac{2\beta^2}{\zeta^2}\left(1 - d_{\min}^{(k)}\right)}}$

$$
< \dfrac{1}{2\sqrt{3d_{\max}^{(k)}}} \dfrac{1 - \tilde{\rho}^{(k)}}{\sqrt{1 + \tilde{\rho}^{(k)}}} \dfrac{1}{\zeta\sqrt{d_{\max}^{(k)}}} \sqrt{\dfrac{1 - \Gamma_1^{(k)}}{1 + \Gamma_1^{(k)}}}.
$$

We found a lower bound for $(a)$, and an upper bound for $(b)$. Since the constraint on $\alpha^{(k)}$ includes the minimum of these two terms, showing that the upper bound for $(b)$ is less than the lower bound for $(a)$, will constitute the fact that the $(b) \leq (a)$. We have

$$
\frac{1 - \Gamma_1^{(k)}}{1 + \Gamma_1^{(k)}} \leq \frac{\zeta^2 d_{\max}^{(k)}}{6\beta^2} \quad \Rightarrow \quad \Gamma_1^{(k)} \geq \frac{1 - \frac{\zeta^2 d_{\max}^{(k)}}{6\beta^2}}{1 + \frac{\zeta^2 d_{\max}^{(k)}}{6\beta^2}} = \Gamma_1^\star.
$$

Therefore, choosing $\Gamma_1^{(k)} = \Gamma_1^\star$ will give us tightest possible bounds. However, in order to get simpler expressions for the first two terms in Eq. 36 which would give us better intuition, we choose

the infimum of $\Gamma_1^{(k)}$ for them, i.e., $1/5$, to obtain

$$
\alpha^{(k)} < \min \left\{ \frac{\mu^2 \left(1 - \frac{2\beta^2}{\mu^2}\left(1 - d_{\min}^{(k)}\right)\right)}{10\beta^3 \left(3 - 2d_{\min}^{(k)}\right)}, \frac{5}{4} \frac{1 + \tilde{\rho}^{(k)}}{2\mu \left(1 - \frac{2\beta^2}{\mu^2}\left(1 - d_{\min}^{(k)}\right)\right)}, \right.
$$
$$
\left. \frac{1}{6\sqrt{2d_{\max}^{(k)}}} \frac{\mu}{\beta^2} \frac{1 - \tilde{\rho}^{(k)}}{\sqrt{1 + \tilde{\rho}^{(k)}}} \sqrt{\frac{1 - \frac{2\beta^2}{\mu^2}\left(1 - d_{\min}^{(k)}\right)}{1 + \frac{2\beta^2}{\zeta^2}\left(1 - d_{\min}^{(k)}\right)}} \right\}
$$

**Step 5: Obtaining $\rho(\mathbf{\Phi}^{(k)})$.** We established $\rho(\mathbf{\Phi}^{(k)}) < 1$ in the previous steps. The last step is to determine what $\rho(\mathbf{\Phi}^{(k)})$ is. We have

$$
\rho\left(\mathbf{\Phi}^{(k)}\right) = \frac{-b + \sqrt{b^2 - 4ac}}{2a} = \frac{\phi_{11}^{(k)} + \phi_{22}^{(k)} + \sqrt{\left(\phi_{11}^{(k)} + \phi_{22}^{(k)}\right)^2 - 4\left(\phi_{11}^{(k)}\phi_{22}^{(k)} - \phi_{12}^{(k)}\phi_{21}^{(k)}\right)}}{2}
$$

$$
= \frac{\phi_{11}^{(k)} + \phi_{22}^{(k)} + \sqrt{\left(\phi_{11}^{(k)} - \phi_{22}^{(k)}\right)^2 + 4\phi_{12}^{(k)}\phi_{21}^{(k)}}}{2}
$$

$$
= \frac{1 - \left(1 - \Gamma_1^{(k)}\right)\left(1 - \frac{2\beta^2}{\mu^2}\left(1 - d_{\min}^{(k)}\right)\right)\mu\alpha^{(k)} + \frac{1+\tilde{\rho}^{(k)}}{2} + 3\frac{1+\tilde{\rho}^{(k)}}{1-\tilde{\rho}^{(k)}}\left(\alpha^{(k)}\right)^2\left(\zeta^2 + 2\beta^2\right)}{2}
$$
$$
+ \frac{1}{2}\left[\left(1 - \left(1 - \Gamma_1^{(k)}\right)\left(1 - \frac{2\beta^2}{\mu^2}\left(1 - d_{\min}^{(k)}\right)\right)\mu\alpha^{(k)} - \frac{1+\tilde{\rho}^{(k)}}{2}\right.\right.
$$
$$
\left.\left. - 3\frac{1+\tilde{\rho}^{(k)}}{1-\tilde{\rho}^{(k)}}\left(\alpha^{(k)}\right)^2\left(\zeta^2 + 2\beta^2\right)\right)^2\right.
$$
$$
\left. + 4\frac{\beta^2 d_{\max}^{(k)}}{2m}\left(1 + \Gamma_1^{(k)}\right)\alpha^{(k)}\frac{6}{\mu}\frac{1+\tilde{\rho}^{(k)}}{1-\tilde{\rho}^{(k)}}md_{\max}^{(k)}\left(\alpha^{(k)}\right)^2\left(\zeta^2 + 2\beta^2\left(1 - d_{\min}^{(k)}\right)\right)\right]^{1/2}
$$

$$
= \frac{3 + \tilde{\rho}^{(k)}}{4} - \frac{1}{2}\left(1 - \Gamma_1^{(k)}\right)\left(1 - \frac{2\beta^2}{\mu^2}\left(1 - d_{\min}^{(k)}\right)\right)\mu\alpha^{(k)} + \frac{3}{2}\frac{1+\tilde{\rho}^{(k)}}{1-\tilde{\rho}^{(k)}}\left(\alpha^{(k)}\right)^2\left(\zeta^2 + 2\beta^2\right)
$$
$$
+ \frac{1}{2}\left[\left(\frac{1-\tilde{\rho}^{(k)}}{2} - \left(1 - \Gamma_1^{(k)}\right)\left(1 - \frac{2\beta^2}{\mu^2}\left(1 - d_{\min}^{(k)}\right)\right)\mu\alpha^{(k)}\right.\right.
$$
$$
\left.\left. - 3\frac{1+\tilde{\rho}^{(k)}}{1-\tilde{\rho}^{(k)}}\left(\alpha^{(k)}\right)^2\left(\zeta^2 + 2\beta^2\right)\right)^2\right.
$$
$$
\left. + \frac{12\beta^2}{\mu}\left(1 + \Gamma_1^{(k)}\right)\frac{1+\tilde{\rho}^{(k)}}{1-\tilde{\rho}^{(k)}}\left(d_{\max}^{(k)}\right)^2\left(\alpha^{(k)}\right)^3\left(\zeta^2 + 2\beta^2\left(1 - d_{\min}^{(k)}\right)\right)\right]^{1/2}
$$

$$
= \frac{3 + \tilde{\rho}^{(k)}}{4} - A^{(k)}\alpha^{(k)} + B^{(k)}\left(\alpha^{(k)}\right)^2
$$
$$
+ \frac{1}{2}\sqrt{\left(\frac{1-\tilde{\rho}^{(k)}}{2} - 2\left(A^{(k)}\alpha^{(k)} + B^{(k)}\left(\alpha^{(k)}\right)^2\right)\right)^2 + C^{(k)}\left(\alpha^{(k)}\right)^3}.
$$

## O  FURTHER EXPERIMENTS

### O.1  ACCURACY VS. DELAY WITH UNIFORM DISTRIBUTION

In the experiments we provided in Fig. 2, the SGD and aggregation probabilities were sampled from a Beta distribution, e.g., $d_i^{(k)}, b_{ij}^{(k)} \sim \text{Beta}(\alpha, \beta)$. In this section, we investigate sampling these probabilities from the uniform distribution, denoted as $\mathcal{U}(0, 1]$.

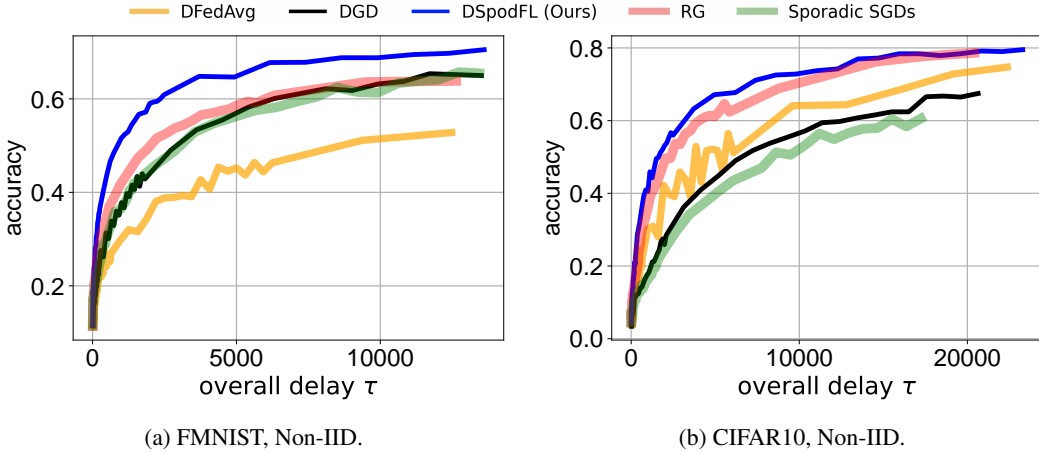

(a) FMNIST, Non-IID.    (b) CIFAR10, Non-IID.

Figure 5: Accuracy vs. latency plots obtained in different setups where the SGD and aggregation probabilities are sampled from the uniform distribution $\mathcal{U}(0, 1]$. `DSpodFL` achieves the target accuracy much faster with less delay, emphasizing the benefit of sporadicity in DFL for SGD iterations and model aggregations simultaneously.

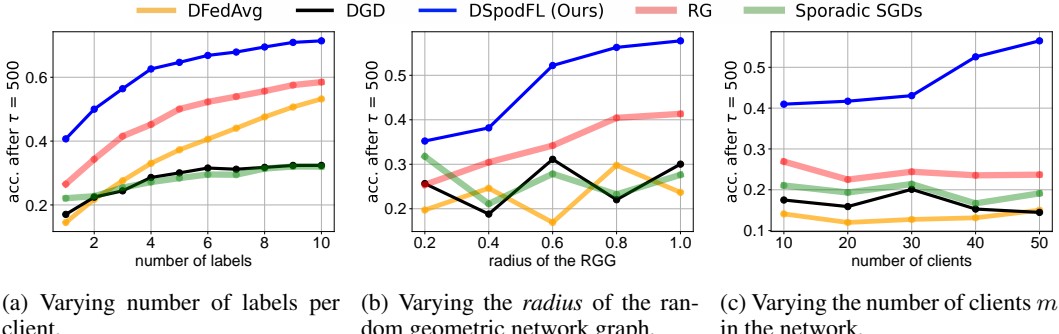

(a) Varying number of labels per client.
(b) Varying the *radius* of the random geometric network graph.
(c) Varying the number of clients $m$ in the network.

Figure 6: Effects of system parameters on FMNIST, where client and link capabilities $d_i$ and $b_{ij}$ are sampled from a Beta distribution $\mathrm{Beta}(0.5, 0.5)$. The overall results confirm the advantage of `DSpodFL` in various settings.

We have provided experimental results for only the non-IID cases in Fig. 5, under exactly the same setup outlined in Sec. 5. It can be observed that the findings discussed in Sec. 5 also hold here. In other words, our `DSpodFL` method outperforms the baselines in terms of accuracy per overall delay.

We will explain the intuitive reason of why `DSpodFL` is outperforming other baselines in both Figs. 2 and 5. Let $\mathcal{G} = (\mathcal{M}, \mathcal{E})$ be given a network graph, and assume there exits two paths between nodes $i$ and $j$. Let one of these paths have a communication cost $k$ times more than the other path, where $k \gg 1$. In our `DSpodFL` method, the path with lower cost will be utilized roughly $k$ times more than the other path, thus resulting in lower communication overhead while still preserving information flow between nodes $i$ and $j$. Meanwhile, other methods, especially DGD and DFedAvg methods, do not take this into account.

## O.2    EFFECTS OF SYSTEM PARAMETERS WITH BETA DISTRIBUTION

In the experiments we provided in Fig. 3, specifically Figs. 3a, 3b and 3c, the SGD and aggregation probabilities were sampled from a uniform distribution, e.g., $d_i^{(k)}, b_{ij}^{(k)} \sim \mathcal{U}(0, 1]$. In this section, we investigate sampling these probabilities from the Beta distribution, denoted as $\mathrm{Beta}(0.5, 0.5)$.

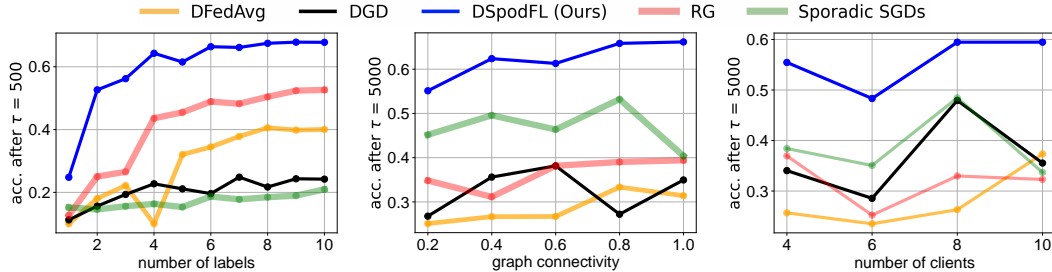

(a) Varying number of labels per client.

(b) Varying the *radius* of the random geometric network graph.

(c) Varying the number of clients $m$ in the network.

Figure 7: Effects of system parameters on CIFAR10. In all figures, client and link capabilities $d_i$ and $b_{ij}$ are sampled from a Beta distribution $\text{Beta}(0.8, 0.8)$. The overall results confirm the advantage of `DSpodFL` in various settings.

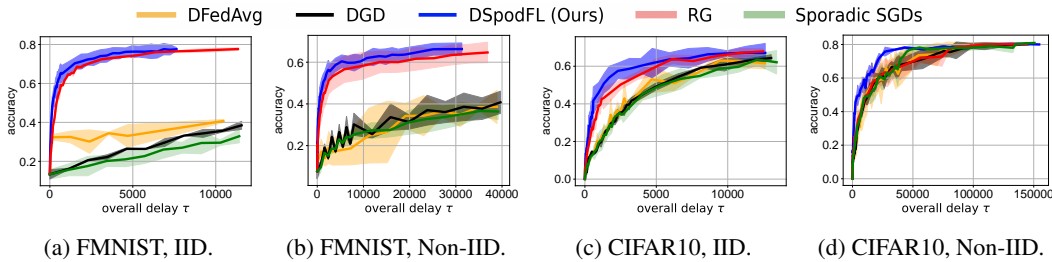

(a) FMNIST, IID.

(b) FMNIST, Non-IID.

(c) CIFAR10, IID.

(d) CIFAR10, Non-IID.

Figure 8: Accuracy vs. latency plots when SGD and aggregation probabilities are time-varying. `DSpodFL` achieves the target accuracy much faster with less delay, emphasizing the benefit of sporadicity in DFL for SGD iterations and model aggregations simultaneously.

We have provided experimental results for only the non-IID cases in Fig. 6, under exactly the same setup outlined in 5. It can be observed that the findings discussed in Sec. 5 also hold here. In other words, the performance gain of our `DSpodFL` method compared to the baselines is robust regardless of the variation in system parameters, i.e, (i) data heterogeneity level, (ii) level of graph connectivity level and (iii) number of clients in the system.

## O.3 EFFECTS OF SYSTEM PARAMETERS ON CIFAR10

In both Sections 5 and O.2, we analyzed the effects of system parameters on the FMNIST dataset. Here, we will provide similar experimental results for the CIFAR10 dataset. Note that while an SVM model was trained on the FMNIST dataset, for CIFAR10 we use the VGG11 model. We sample the SGD and aggregation probabilities from the Beta distribution, i.e., $d_i^{(k)} \sim \text{Beta}(0.8, 0.8)$ and $b_{ij}^{(k)} \sim \text{Beta}(0.8, 0.8)$, respectively.

The results in Fig. 7 are carried out in the non-IID regime as well, under the setup described in Sec. 5. Again, the findings discussed in Sec. 5 and O.2 can be validated here, showing that `DSpodFL` outperforms the state-of-the-art in various settings. This demonstrates that our results hold regardless of the dataset in question and the ML model being used, adding yet another dimension of robustness to our methodology.

## O.4 DYNAMIC SGD AND AGGREGATION PROBABILITIES

In the experiments done in Sec. 5, the SGD and aggregations probabilities where set to constant values for all clients, i.e., $d_i^{(k)} = d_i$ and $b_{ij}^{(k)} = b_{ij}$, for all $k \geq 0$. In this section, we conduct experiments by letting these probabilities to be time-varying. This setup corresponds to situations where the computation/communication resources of clients vary over time. To be specific, we change probabilities $d_i^{(k)}$ and $b_{ij}^{(k)}$ every 1000 iterations of model training. However, note that we still randomly

| Algorithm | IID | | | Non-IID | | |
|---|---|---|---|---|---|---|
| | Iter. 1000 | Iter. 2500 | Iter. 3500 | Iter. 5000 | Iter. 10000 | Iter. 15000 |
| DGD | 0.80 | 0.81 | 0.82 | 0.72 | 0.73 | 0.75 |
| DFedAvg | 0.79 | 0.81 | 0.81 | 0.39 | 0.41 | 0.41 |
| RG | 0.80 | 0.82 | 0.82 | 0.60 | 0.65 | 0.64 |
| Sporadic SGDs | 0.77 | 0.80 | 0.80 | 0.70 | 0.74 | 0.74 |
| **DSpodFL** | 0.77 | 0.80 | 0.80 | 0.63 | 0.67 | 0.70 |

Table 2: Accuracy vs. iteration results for experiments in Fig. 2 done for the FMNIST dataset.

| Algorithm | IID | | | Non-IID | | |
|---|---|---|---|---|---|---|
| | Iter. 1500 | Iter. 3000 | Iter. 4500 | Iter. 3000 | Iter. 6000 | Iter. 9500 |
| DGD | 0.73 | 0.73 | 0.74 | 0.77 | 0.80 | 0.81 |
| DFedAvg | 0.73 | 0.74 | 0.75 | 0.56 | 0.70 | 0.76 |
| RG | 0.72 | 0.74 | 0.74 | 0.73 | 0.80 | 0.80 |
| Sporadic SGDs | 0.74 | 0.74 | 0.75 | 0.71 | 0.76 | 0.79 |
| **DSpodFL** | 0.72 | 0.73 | 0.73 | 0.65 | 0.72 | 0.76 |

Table 3: Accuracy vs. iteration results for experiments in Fig. 2 done for the CIFAR10 dataset.

sample them from the same distribution, i.e., $\text{Beta}(0.5, 0.5)$ for FMNIST and $\text{Beta}(0.8, 0.8)$ for CIFAR10.

It can be observed that the findings discussed in Sec. 5 also hold here. In other words, our `DSpodFL` method outperforms the baselines in terms of accuracy per overall delay regardless of the time-variation in SGD and aggregation probabilities.

### O.5    ACCURACY VS. ITERATION

We have provided the accuracy vs. iteration results for the experiments done in Fig. 2. In Tables 2 and 3, we give the results for several sampled iterations.

We observe that given enough time, i.e., after sufficient epochs, the achievable accuracy for `DSpodFL` matches the achievable accuracy for the other baselines. The accuracy for `DSpodFL` being slightly lower than DGD at some iterations is due to the fact that gradient and consensus operations occur at every iteration for all devices in DGD, without taking resource availability into account. In `DSpodFL`, on the other hand, at each iteration some devices do not compute gradients and/or some of the links are not utilized for communications. Thus, it is not unexpected for the accuracy of `DSpodFL` to be lower than DGD across the iterations, since it is actually designed to achieve faster convergence in terms of the actual physical delay incurred as seen in Fig. 2. Regardless, the final achievable accuracy is the same for all baselines, as they all fit within the `DSpodFL` framework, and we theoretically prove in the paper that all of these algorithms will converge to the same global ML model.

### O.6    DECOMPOSING ACCURACY VS. DELAY RESULTS

In Tables 4 and 5, we have decomposed the results from Fig. 2 into their processing and transmission delay components. We have reported how long it takes for different algorithms to achieve a specific accuracy in terms of 1) processing delay, 2) transmission delay and 3) overall delay. We can see that to reach a certain accuracy, `DSpodFL` strikes the best balance between transmission and processing delays, leading to a better overall delay. Specifically, it obtains a similar processing delay to the Sporadic SGDs algorithm and transmission delay to RG, which are the best baselines in those respective categories. Note that the reported delays below are in units of time.

### O.7    RESULTS WITH A TRUNCATED GAUSSIAN DISTRIBUTION

In Fig. 9 we have further explored the performance of `DSpodFL` under a truncated Gaussian distribution $\hat{N}_{[0,1]}(\mu, \sigma^2)$ to generate probabilities $d_i$ and $b_{ij}$, which for small values of variance $\sigma^2$ gives

| Algorithm | IID, with accuracy = 0.75 | | | Non-IID, with accuracy = 0.40 | | |
|---|---|---|---|---|---|---|
| | Process. | Transm. | Overall | Process. | Transm. | Overall |
| DGD | 4605.26 | 1915.95 | 6521.22 | 2470.37 | 1027.76 | 3498.14 |
| DFedAvg | 5947.19 | 88.82 | 6036.01 | 221418.66 | 3070.60 | 224489.26 |
| RG | 6130.19 | 331.40 | 6461.58 | 9180.18 | 451.15 | 9631.18 |
| Sporadic SGDs | 591.84 | 7614.84 | 8206.68 | 127.38 | 1270.18 | 1397.56 |
| **DSpodFL** | 593.69 | 893.32 | **1487.00** | 257.68 | 430.96 | **688.65** |

Table 4: Decomposition of accuracy vs. delay results in Fig. 2 for the FMNIST dataset into their processing an transmission delay components.

| Algorithm | IID, with accuracy = 0.65 | | | Non-IID, with accuracy = 0.30 | | |
|---|---|---|---|---|---|---|
| | Process. | Transm. | Overall | Process. | Transm. | Overall |
| DGD | 3462.06 | 2571.64 | 6033.70 | 2250.12 | 8807.37 | 11057.49 |
| DFedAvg | 2784.81 | 482.01 | 3266.83 | 9627.35 | 2065.34 | 11692.70 |
| RG | 3462.06 | 614.68 | 4076.74 | 2250.12 | 161.15 | 2411.27 |
| Sporadic SGDs | 440.73 | 2888.25 | 3328.98 | 160.13 | 13081.37 | 13241.50 |
| **DSpodFL** | 628.97 | 990.24 | **1619.21** | 327.27 | 647.24 | **974.51** |

Table 5: Decomposition of accuracy vs. delay results in Fig. 2 for the CIFAR10 dataset into their processing an transmission delay components.

the setting of relatively homogeneous and static clients. Fig. 9a demonstrates accuracy vs. latency when using this distribution with a mean and standard deviation of 0.5, and Fig. 9b shows the result for different $\sigma^2$. We see a wider margin of improvement with a relatively larger $\sigma^2$. This confirms that `DSpodFL` is most advantageous relative to the baselines under extreme levels of heterogeneity and dynamics, similar to how the improvements under the inverted bell-shaped beta distribution are more pronounced than those under the uniform distribution as seen in Fig. 3d. We have also experimented with fixing the variance of the truncated Gaussian distribution and varying its mean in Fig. 9c.

## O.8 FURTHER GENERALIZATION AND SCALABILITY VERIFICATION

In Fig. 10, we present further experimental results under different settings than the setup given in the main text. In Fig. 10a, we analyze the effect of graph connectivity on the overall performance when a total of $m = 50$ clients are present in the network. We observe that the improvement gap between `DSpodFL` and other baselines becomes even more significant compared to Fig. 3b. In Figs. 10b and 10c, we isolate the effect of varying $b_{ij}$ and $d_i$ separately, in contrast to Fig. 3d where both the SGD probabilities $d_i$ (i.e., computation capabilities) and aggregation probabilities $b_{ij}$ (i.e., communication capabilities) were varied together. The results are show that `DSpodFL` is more robust to variations in $b_{ij}$ or $d_i$ compared to the baselines, respectively. In Figs. 10b (and 10c), while the heterogeneity in computational (communication) resources is preserved across the whole experiment, moving from $\alpha = \beta = 0.5$ to $\alpha = \beta = 1$ brings us from a heterogeneous regime to a homogeneous one in terms of communication (computational) resources. Thus, the Sporadic SGDs (Sporadic Aggregations) component of `DSpodFL`, i.e., green curve (red curve), becomes key to improvement over other baselines when communication (computational) resources are homogeneous.

## O.9 EFFECT OF LEARNING RATE

We vary the learning rate in the range $\alpha \in \{0.0001, 0.001, 0.01, 0.1\}$ in our experiments. In Fig. 11, we present the test accuracy results for our `DSpodFL` algorithm and other baselines, when they reach a total delay of $\tau = 500$ for FMNIST and $\tau = 5000$ for CIFAR10. For FMNIST, we observe that even the best performance of the baselines is lower than our `DSpodFL` algorithm with learning rates $\alpha = 0.01$ or $\alpha = 0.1$. For CIFAR10, we make a similar observation where the best performance of the baselines is lower than `DSpodFL` with learning rates $\alpha = 0.001$ and $\alpha = 0.01$.

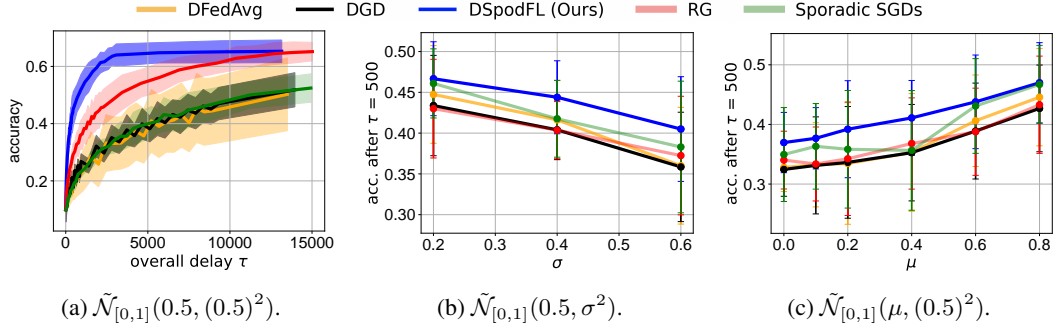

(a) $\tilde{\mathcal{N}}_{[0,1]}(0.5,(0.5)^2)$.     (b) $\tilde{\mathcal{N}}_{[0,1]}(0.5,\sigma^2)$.     (c) $\tilde{\mathcal{N}}_{[0,1]}(\mu,(0.5)^2)$.

Figure 9: We investigate the FMNIST, Non-IID setup from Figs. 2 and 3 using a Truncated Gaussian Distribution to sample SGD and aggregation probabilities $d_i$ and $b_{ij}$, respectively. We denote this distribution as $\tilde{\mathcal{N}}_{[0,1]}(\mu,\sigma^2)$, which only has values in the interval $[0,1]$.

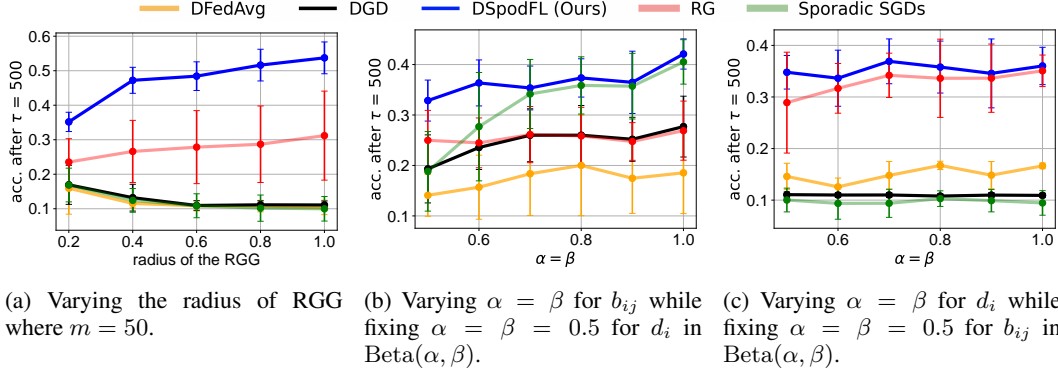

(a) Varying the radius of RGG where $m = 50$.    (b) Varying $\alpha = \beta$ for $b_{ij}$ while fixing $\alpha = \beta = 0.5$ for $d_i$ in $\mathrm{Beta}(\alpha,\beta)$.    (c) Varying $\alpha = \beta$ for $d_i$ while fixing $\alpha = \beta = 0.5$ for $b_{ij}$ in $\mathrm{Beta}(\alpha,\beta)$.

Figure 10: Further investigation of `DSpodFL`'s generalization to various setups.

## P    REMARKS

### P.1    EFFECT OF COMMUNICATION SPORADICITY ON ANALYTICAL BOUNDS

We will make a remark about this in two parts:

**1. Effect of $b_{ij}$ on the spectral radius $\tilde{\rho}^{(k)}$:** As mentioned in Sec. 4.4's "Discussion on convergence", the probabilities $b_{ij}$ affect the bounds through the spectral radius of the expected mixing matrix, i.e., $\tilde{\rho}^{(k)} = \rho(\tilde{\mathbf{R}}^{(k)} - (1/m)\mathbf{1}_m\mathbf{1}_m^T)$ from Definition 4.5. We can refer to the elements of matrix $\tilde{\mathbf{R}}^{(k)}$ in Appendix E.4-(b), given as

$$\tilde{\mathbf{R}}^{(k)} \triangleq \left(\bar{\mathbf{R}}^{(k)}\right)^2 + \begin{cases} \left[-2b_{ij}^{(k)}\left(1-b_{ij}^{(k)}\right)r_{ij}^2\right]_{1\leq i,j\leq m} & i \neq j \\ \left[\sum_{l=1}^m 2b_{il}^{(k)}\left(1-b_{il}^{(k)}\right)r_{il}^2\right]_{1\leq i\leq m} & i = j \end{cases}$$

where $\bar{\mathbf{R}}^{(k)}$ is defined in Lemma D.4-(a) as

$$\bar{\mathbf{R}}^{(k)} = \begin{cases} \left[b_{ij}^{(k)}r_{ij}\right]_{1\leq i,j\leq m} & i \neq j \\ \left[1 - \sum_{j=1}^m b_{ij}^{(k)}r_{ij}\right]_{1\leq i,j\leq m} & i = j \end{cases}$$

As we can see, the values of $b_{ij}^{(k)}$ directly affect $\tilde{\mathbf{R}}^{(k)}$, and hence the communication sporadicity parameters $b_{ij}^{(k)}$ affect all the convergence bounds and learning rate constraints through $\tilde{\rho}^{(k)}$, the spectral radius of $\tilde{\mathbf{R}}^{(k)} - (1/m)\mathbf{1}_m\mathbf{1}_m^T$.

From the definition of $\tilde{\mathbf{R}}^{(k)}$, we can draw some intuitions on the relation between $b_{ij}^{(k)}$ and $\tilde{\rho}^{(k)}$: If $b_{ij}^{(k)} \to 0$ for all $(i,j) \in \mathcal{E}$, we will have a diagonal $\tilde{\mathbf{R}}^{(k)}$, meaning that $\tilde{\rho}^{(k)} = 1$. On the other

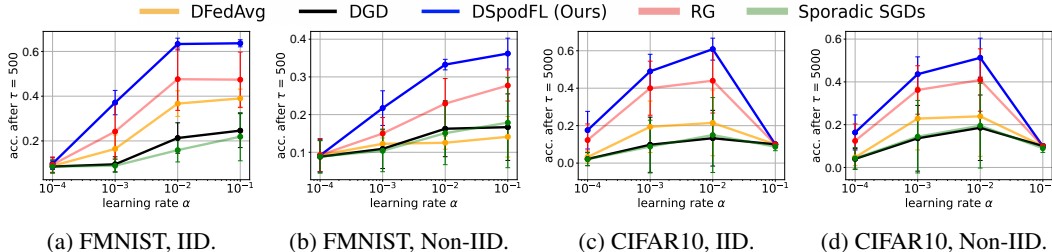

|  |  |  |  |
|---|---|---|---|
| (a) FMNIST, IID. | (b) FMNIST, Non-IID. | (c) CIFAR10, IID. | (d) CIFAR10, Non-IID. |

Figure 11: Effect of the learning rate $\alpha$.

extreme, if $b_{ij}^{(k)} \to 1$ for all $(i,j) \in \mathcal{E}$, we will have the main diagonal of $\tilde{\mathbf{R}}^{(k)} = \mathbf{R}^2$, meaning that $\tilde{\rho}^{(k)} = \rho_r^2$. This would be the lowest achievable spectral radius $\tilde{\rho}^{(k)}$, as it is utilizing all the communication links as frequently as possible. All other cases of $b_{ij}^{(k)}$ result in $\tilde{\rho}^{(k)}$ between these two extremes, with higher connectivity decreasing $\tilde{\rho}^{(k)}$.

**2. Effect of $\tilde{\rho}^{(k)}$ on the analytical bounds:** Based on the relationship between $b_{ij}^{(k)}$ and $\tilde{\rho}^{(k)}$, we can establish the connection between $b_{ij}^{(k)}$ and the convergence bound. We can see in Propositions 4.10 and G.6 (convex and non-convex cases) that if $\tilde{\rho} = 1$ (one possible scenario being that all $b_{ij}^{(k)} = 0$, as explained in the previous paragraph) the constraint on the learning rate becomes $\alpha^{(k)} < 0$. Intuitively, no learning rate can make DSpodFL converge if none of the communication links are ever utilized. On the other hand, having all $b_{ij}^{(k)} = 1$ results in the minimum achievable $\tilde{\rho}$, allowing a larger learning rate to be chosen based on the constraints given in Propositions 4.10 and G.6. In other words, as the connectivity of the graph induced by $b_{ij}^{(k)}$ increases, larger step sizes can be tolerated while still guaranteeing convergence in Theorems 4.11 and 4.12.

### P.2 PERFORMANCE SUPERIORITY OF `DSpodFL`

Our performance improvements over the baselines come from sporadic operations of both local SGDs and aggregations. In decentralized FL, resource availability is often heterogeneous and dynamic, as discussed in Sec. 1: there are (i) variations in computation capabilities at clients, causing bottlenecks in assuming consistent participation in SGD computations, and (ii) variations in link bandwidth, causing bottlenecks during model aggregation. Referring to Algorithm 1 in Appendix B, `DSpodFL` overcomes these limitations as follows:

- Client $i$ conducts an SGD in iteration $k$ only if $v_i^{(k)} = 1$ (line 7), which has a probability $d_i$ proportional to its processing availability. Consider that multiple clients may possess overlapping data distributions. In such scenarios, the system can benefit from more frequent SGD updates from the clients with the highest resource availability, as they provide information representative of multiple clients and finish iterations faster.

- Link $(i, j)$ is used for model sharing in iteration $k$ only if $v_{ij}^{(k)} = 1$ (line 12), which has a probability $b_{ij}$ proportional to its bandwidth availability. `DSpodFL` takes advantage of the fact that model information can propagate through the system rapidly over the fastest links in the graph. For example, if link $(i, j)$ has low bandwidth, our method will make client $i$'s relevant local update information reach client $j$ more rapidly through a series of other high bandwidth links.

Unlike `DSpodFL`, the baselines evaluated in Sec. 5 work under (a) fixed SGDs and/or (b) fixed aggregations. DGD assumes local SGDs and aggregations at every iteration, RG employs constant SGDs (but sporadic aggregations), Sporadic SGDs assumes constant aggregations, and DFedAvg is DGD with decreased communication frequencies.

## P.3 COMPARISON METRICS FOR EXPERIMENTS

As outlined in Sec. 5, the average total delay $\tau_{total}^{(k)}$ at iteration $k$ is defined as the sum of average processing delay $\tau_{proc}^{(k)}$ and average transmission delay $\tau_{trans}^{(k)}$ up to iteration $k$. At each iteration, depending on whether a client $i$ computed SGDs or not, i.e., $v_i^{(k)} \in \{0, 1\}$, there will be a processing delay incurred to finish the computation proportional to $1/d_i^{(k)}$. Similarly, depending on $\hat{v}_{ij}^{(k)} \in \{0, 1\}$, some of the links $(i, j)$ in the network graph will be utilized for communications, incurring a transmission delay proportional to $1/b_{ij}^{(k)}$ for that link. In order to obtain comparable result regardless of the size, connectivity and the topology of the underlying graph, we normalize (i.e., average) both processing and transmission delay to obtain $\tau_{proc}^{(k)}, \tau_{trans}^{(k)}$. Formally, these are defined as $\tau_{trans}^{(k)} = [\sum_{i=1}^{m} (1/|\mathcal{N}_i|) \sum_j \hat{v}_{ij}^{(k)}/b_{ij}]/[\sum_{i=1}^{m} (1/|\mathcal{N}_i|) \sum_j 1/b_{ij}]$ and $\tau_{proc}^{(k)} = [\sum_{i=1}^{m} v_i^{(k)}/d_i]/[\sum_{i=1}^{m} 1/d_i]$. Finally, we define $\tau_{total}^{(k)} = \tau_{trans}^{(k)} + \tau_{proc}^{(k)}$ as the average total delay as a metric to compare `DSpodFL` with the baselines.

Having explained the average total delay $\tau_{total}^{(k)} = \tau_{trans}^{(k)} + \tau_{proc}^{(k)}$ above, we can now see what happens to different baselines under our proposed heterogeneity framework, and how $d_i$ and $b_{ij}$ translate to the speed of a client and a link, respectively. In DGD, we have that computations and communications occur at every iteration, i.e., $v_i^{(k)} = \hat{v}_{ij}^{(k)} = 1$ for all $i \in \mathcal{M}$, $(i, j) \in \mathcal{E}$ and $k \geq 0$. Thus, we will have that $\tau_{trans}^{(k)} = [\sum_{i=1}^{m} (1/|\mathcal{N}_i|) \sum_j 1/b_{ij}]/[\sum_{i=1}^{m} (1/|\mathcal{N}_i|) \sum_j 1/b_{ij}] = 1$ and $\tau_{proc}^{(k)} = [\sum_{i=1}^{m} 1/d_i]/[\sum_{i=1}^{m} 1/d_i] = 1$, and thus each iteration of training will incur a total average delay of $\tau_{total}^{(k)} = 2$ for this baseline. For DFedAvg, we still have $\tau_{proc}^{(k)} = 1$ as SGDs occur at every iteration. But since aggregations occur every $D$ iterations, i.e., clients conduct $D$ consecutive iterations of local SGD steps before communications, we will have $\tau_{trans}^{(k)} = 0$ for $D$ iterations and then $\tau_{trans}^{(k)} = 1$ for the next iteration. This deterministic cycle will then continue for DFedAvg.

In RG and the Sporadic SGDs baselines, we only have one of these operations occurring at every iteration, i.e., computations and communications, respectively. This means that in RG, we have $v_i^{(k)} = 1$ is deterministic but $\hat{v}_{ij}^{(k)}$ is stochastic, and for Sporadic SGDs, we have deterministic $\hat{v}ij^{(k)} = 1$ but stochastic $v_i^{(k)}$. Thus, for RG we will have $\tau_{total}^{(k)} = \tau_{trans}^{(k)} + 1 < 2$ implied by $\tau_{proc}^{(k)} = 1$, and for Sporadic SGDs, we will have that $\tau_{total}^{(k)} = 1 + \tau_{proc}^{(k)} < 2$ implied by $\tau_{trans}^{(k)} = 1$.

However, note that in `DSpodFL`, both computation and communication operations are carried out in a stochastic way, and thus each iteration of training requires less delay incurred on the whole decentralized system to start the next round of training. Note however, that less computation and communication might come at the cost of losing performance at each iteration, but our motivation in `DSpodFL` was to prove that in fact if we evaluate these algorithms based on their total delay, it can outperform existing baselines. The intuition is that due to the data distribution of clients available in the network, their processing capabilities, the graph topology and the link bandwidth capabilities, it is not necessary to over utilize all of these resources at every iteration for fast convergence. In fact, a resource-aware approach like `DSpodFL`, takes a step at optimally utilizing the resources to achieve the same final solutions in a shorter amount of time.

## P.4 CONVERGENCE RATE COMPARISON WITH RELATED WORK

**Convergence rate in Big O notation.** First, we note that as outlined in Table 1 of Sec. 1, we provide convergence for last iterates of model parameters when dealing with strongly-convex models in our paper, in contrast to the majority of existing literature which only provide convergence for average iterates (Koloskova et al., 2020). Therefore when dealing with strongly-convex models, we can compare our theoretical results only for the DGD baseline because the DGD-like algorithms given in Mishchenko et al. (2022); Maranjyan et al. (2022) are among the few to show convergence for the last iterates of model parameters as well. According to Theorems 3.6, 5.5, 5.7 and D.1 in Mishchenko et al. (2022) and Theorems 3.5, 4.5 and B.2 in Maranjyan et al. (2022), the convergence rate of the algorithms ProxSkip, Decentralized Scaffnew, SplitSkip (Mishchenko et al., 2022), GradSkip, GradSkip+ and VR-GradSkip+ (Maranjyan et al., 2022) are all geometric, i.e., $\mathcal{O}(\rho^K)$

with $0 < \rho < 1$ (note that most of these algorithms are FL algorithms, and not decentralized FL algorithms). This rate agrees with the rate we provide in Eq. 9 of Theorem 4.11.

When dealing with non-convex models, we can compare our rate with all the baselines, i.e., Nedic & Ozdaglar (2009) for DGD, Koloskova et al. (2020) for RG and Sun et al. (2022) for DFedAvg. Since DGD is a special case of RG with no sporadicity in aggregations, i.e., $b_{ij} = 1$ for all $(i, j) \in \mathcal{E}$, we will compare our work with the more recent paper Koloskova et al. (2020). For DGD and RG, Lemma 17 of Koloskova et al. (2020) shows the convergence upper bound before tuning the constant learning rate $\alpha$, which is

$$\mathcal{O}\left(\frac{\mathbb{E}[F(\bar{\theta}^{(0)})] - F^\star}{\alpha(K+1)}\right) + \mathcal{O}(\alpha) + \mathcal{O}(\alpha^2).$$

For DFedAvg, Theorem 1 in Sun et al. (2022) with a zero momentum ($\theta = 0$) obtains the bound

$$\mathcal{O}\left(\frac{\mathbb{E}[F(\bar{\theta}^{(0)})] - F^\star}{\alpha(K+1)}\right) + \mathcal{O}(\alpha) + \mathcal{O}(\alpha^2) + \mathcal{O}(\alpha^3).$$

We observe that by setting $d_{min} = 1$ in Theorem 4.12 of our paper, these convergence rates are recovered. Recall that as discussed in Fig. 1 of our paper, all of these baseline algorithms fit within the general framework of DSpodFL with $d_i = 1$ for all clients $i \in \mathcal{M}$. That is why we can substitute $d_{min} = 1$ in Theorem 4.12 to compare our analytical results with the ones provided in Nedic & Ozdaglar (2009); Koloskova et al. (2020); Sun et al. (2022).

**Comparison of convergence bound for non-convex models with Koloskova et al. (2020).** Let us examine the bound derived in the proof of Lemma 16 in Koloskova et al. (2020) before tuning the learning rate, i.e.,

$$\frac{1}{2(T+1)}\sum_{t=0}^{T}\|\nabla f(\bar{x}^{(t)})\|_2^2 \le \frac{\mathbb{E}f(\bar{x}^{(0)}) - f^\star}{(T+1)\eta} + \frac{L\hat{\sigma}^2}{n}\eta + 64\frac{L^2[2\hat{\sigma}^2 + 2(\frac{6\tau}{p} + M)\hat{\zeta}^2]\tau}{p}\eta^2.$$

Translating these parameters to the setup of our paper, we have

$$T \to K, t \to r, f \to F, x \to \theta, \eta \to \alpha, L \to \beta, \hat{\sigma} \to \sigma, n \to m, \tau \to 1, p \to 1 - \tilde{\rho}, M \to 0.$$

As a consequence, the bound in Koloskova et al. (2020) using the notation and setup of our paper becomes

$$\frac{1}{2(K+1)}\sum_{r=0}^{K}\|\nabla F(\bar{\theta}^{(r)})\|^2 \le \frac{\mathbb{E}F(\bar{\theta}^{(0)}) - F^\star}{(K+1)\alpha} + \frac{\beta\sigma^2}{m}\alpha + 128\frac{\beta^2[\sigma^2 + \frac{6}{1-\tilde{\rho}}\delta^2]}{1-\tilde{\rho}}\alpha^2.$$

Now, we compare this with the non-asymptotic bound we obtain for non-convex models in Theorem 4.12, which is

$$\frac{w_1}{K+1}\sum_{r=0}^{K}\|\nabla F(\bar{\theta}^{(r)})\|^2 \le \frac{F(\bar{\theta}^{(0)}) - F^\star}{\alpha(K+1)} + \frac{1+\Gamma_3}{1-\frac{1}{\Gamma_1}}\frac{\beta^2}{m(1-\tilde{\rho})}\frac{\|\Theta^{(0)} - \mathbf{1}_m\bar{\theta}^{(0)}\|^2}{K+1}$$

$$+ \frac{1+\Gamma_3}{1-\frac{1}{\Gamma_1}}\frac{\beta^2(16\frac{1+\tilde{\rho}}{1-\tilde{\rho}}\delta^2 + \sigma^2)}{1-\tilde{\rho}}\alpha^2 + (1+\Gamma_3)(1-d_{min})\delta^2 + \frac{\beta\sigma^2}{2m}\alpha.$$

where we used the fact that $d_{max} \le 1$.

We can see how our bound compares with the one derived in Koloskova et al. (2020). The exact value of $w_1$ in our paper depends on the arbitrarily chosen scalars $\Gamma_0, ..., \Gamma_4$, but we have in general that $w_1 \le \frac{1}{2}$. For common terms that our analysis has with Koloskova et al. (2020), we see how their coefficients are also very similar. For example, for the term proportional to $\alpha^2$, we observe that both our bound and the one in Koloskova et al. (2020) are proportional to $\frac{\beta^2}{(1-\tilde{\rho})}$ and $(\sigma^2 + \frac{q}{1-\tilde{\rho}}\delta^2)$, with slight differences in the value of $q$. Furthermore, for the term proportional to $\alpha$, both our bound and Koloskova et al. (2020) are proportional to $\frac{\beta\sigma^2}{2m}$.

However, we can observe that our bound consists of two extra terms compared to Koloskova et al. (2020). The first one, $\frac{1+\Gamma_3}{1-\frac{1}{\Gamma_1}}\frac{\beta^2}{m(1-\tilde{\rho})}\frac{\|\Theta^{(0)} - \mathbf{1}_m\bar{\theta}^{(0)}\|^2}{K+1}$, is capturing the effect of different model initializations for the clients. The second one, $(1+\Gamma_3)(1-d_{min})\delta^2$ is capturing the effect of sporadic

SGDs in our `DSpodFL` framework. We observe that if all clients conduct SGDs at every iteration, i.e., $d_i^{(k)} = 1$ for all $i \in \mathcal{M}$ and $k \geq 0$, this term becomes equal to zero, giving us the bound for non-sporadic methods outlined in Koloskova et al. (2020). Therefore, as we have claimed in Sec. 4.5 of our paper, our convergence bounds recover well-known results in the literature in the degenerate case of $d_{min} = 1$.

