# OpenReview forum: "Decentralized Sporadic Federated Learning: A Unified Algorithmic Framework with Convergence Guarantees"
_ICLR.cc/2025/Conference — ICLR 2025 Spotlight_

### Official Review · Reviewer_VTdm · 2024-10-22

**Soundness:** 4
**Presentation:** 3
**Contribution:** 3
**Rating:** 8
**Confidence:** 4

**Summary:**

This paper develops a generalized algorithmic framework, DSpodFL, for decentralized federated learning (DFL), which effectively integrates the heterogeneity and dynamics of local stochastic gradient descents (SGDs) and aggregation processes. The convergence of DSpodFL is analyzed for both strongly convex and nonconvex loss functions. Additionally, numerical experiments on various datasets demonstrate the advantages of the proposed method.

**Strengths:**

1.	The investigated problem is novel, as the authors address the combined effects of sporadic local client computations and inter-client communication, an aspect previously unexplored in DFL.

2.	The presentation of the paper is clear. The authors provide rigorous convergence results for both strongly convex and nonconvex settings, which are well-justified.

3.	The numerical experiments are sufficient and effectively showcase the advantages of the proposed DSpodFL framework.

**Weaknesses:**

1.In Section 3.3, why the random matrix $P^k$ is doubly stochastic and symmetric? This is not immediately clear, and providing a proof and an illustrative example would enhance understanding.

2.In Assumption 4.1, the gradient diversity is measured by $ || \nabla F(\theta) - \nabla F_i(\theta) ||$, while in Assumption 4.2, it is measured by $ ||\nabla F_i(\theta) ||$.What is the rationale behind this distinction?

3.In the analysis with a diminishing learning rate, why does exact convergence require an increasing frequency of SGDs over time? Moreover, the dependence of SGD probabilities on the stepsize seems unrealistic. Why does this requirement not apply when using a constant stepsize?

**Questions:**

Perhaps $p_{ii}^k = 1 -\sum_{j \in M} r_{ij} \hat v_{ij}^k + r_{ii} \hat v_{ii}^k $ in equation (4)?

---

> ### Author Response · Authors · 2024-11-16
>
> Thank you for your positive assessment of our paper, including our convergence results and experiments. We also appreciate you commending the novelty of our explored idea and the clarity of our presentation. Below, we respond to each comment alongside discussing how we have addressed them in the revised manuscript.
>
> ### **[Weakness 1] Double stochasticity of $P^{(k)}$**
> We apologize for not explaining all the steps required to show that this is true. The transition matrix $P^{(k)}$ being doubly stochastic and symmetric follows from the fact that $\hat{v}\_{ij}^{(k)} = \hat{v}\_{ji}^{(k)}$ and $r_{ij} = r_{ji}$ for $i,j \in \mathcal{M}$ and $i \neq j$, and $\hat{v}_{ii}^{(k)} = 0$ for all $i \in \mathcal{M}$ and $k \geq 0$. We thank you for pointing this out, and we have added this information in Sec. 3.1 of our revised manuscript.
>
> Having these pieces at hand, it remains to show that $p_{ij}^{(k)} = p_{ji}^{(k)}$ for all $i \neq j$, and $\sum_{j \in \mathcal{M}}{p_{ij}^{(k)}} = 1$ (which will then imply that $\sum_{i \in \mathcal{M}}{p_{ij}^{(k)}} = 1$ as well). Since $p_{ij}^{(k)} = r_{ij} \hat{v}\_{ij}^{(k)}$, symmetry of $r_{ij}$ and $\hat{v}\_{ij}^{(k)}$ immediately imply the symmetry of $p_{ij}^{(k)}$. Then, we have defined $p_{ii}^{(k)} = 1 - \sum_{j \in \mathcal{M}}{r_{ij} \hat{v}\_{ij}^{(k)}}$ in such a way that we can get $\sum_{j \in \mathcal{M}}{p_{ij}^{(k)}} = 1$. Noting that $\hat{v}\_{ii}^{(k)} = 0$, this in fact implies that $p_{ii}^{(k)} = 1 - \sum_{j \in \mathcal{M}, j \neq i}{p_{ij}^{(k)}}$.

---

> ### Author Response · Authors · 2024-11-16
>
> ### **[Weakness 2] Gradient Diversity Assumptions for Convex and Non-Convex Models**
> We have opted to use these data heterogeneity assumptions for convex and non-convex models based on standard assumptions for each of these models that are used in other papers, e.g., see [1-4]. In the revised version of our paper, we have added a subsection in Appendix C to further elaborate on our choice of data heterogeneity assumption for convex and non-convex models, a summary of which is given below.
>
> We note that if we are dealing with strongly-convex models, i.e., we make Assumption 4.1-(b) for $\mu$-strongly convex models, the inequality in Assumption 4.2-(b) for data heterogeneity is a much stronger assumption than 4.1-(c). To show this, we first note that making the $\beta$-smoothness assumption (made in both 4.1-(a) and 4.2-(a)) implies that
> $$\\| \nabla{F}(\theta) \\| \le \beta \\| \theta - \theta^\star \\|,$$
> where $\theta^\star$ is a point where $\nabla{F}(\theta^\star) = 0$, i.e., a globally optimal point for convex models and a stationary point for non-convex models (note that we present and prove this result in Lemma D.1-(b)). Then, since we have $\\| \nabla{F}_i(\theta) \\| \le \delta_i + \zeta_i \\| \nabla{F}(\theta) \\|$ according to Assumption 4.2-(b), we can conclude that
> $$\\| \nabla{F}(\theta) - \nabla{F}_i(\theta) \\| \le \\| \nabla{F}(\theta) \\| + \\| \nabla{F}_i(\theta) \\| \le \delta_i + (1 + \zeta_i ) \\| \nabla{F}(\theta) \\| \le \delta_i + (1 + \zeta_i) \beta \\| \theta - \theta^\star \\|.$$
> We observe that defining $\delta’_i = \delta_i$ and $\zeta’_i = (1 + \zeta_i) \beta > 0$, Assumption 4.1-(c) is implied with parameters $\delta’_i$ and $\zeta’_i$.
>
> The converse is not necessarily true though, i.e., Assumption 4.1-(c) does not imply Assumption 4.2-(b).  Note that some existing works assume stricter assumptions than both 4.1-(c) and 4.2-(b), and they thus might make a common assumption among convex and non-convex models. These would be specific cases of our Assumptions 4.1-(c) and 4.2-(b). For instance, setting $\zeta_i = 0$ in our Assumption 4.1-(c) gives us the assumption made in [2, 5, 6], and setting $\zeta_i = 0$ in our Assumption 4.2-(b) provides us with the assumptions made in [7, 8].
>
> [1] Koloskova, Anastasia, Nicolas Loizou, Sadra Boreiri, Martin Jaggi, and Sebastian Stich. "A unified theory of decentralized sgd with changing topology and local updates." In International Conference on Machine Learning, pp. 5381-5393. PMLR, 2020.
>
> [2] Wang, Shiqiang, Tiffany Tuor, Theodoros Salonidis, Kin K. Leung, Christian Makaya, Ting He, and Kevin Chan. "Adaptive federated learning in resource constrained edge computing systems." IEEE journal on selected areas in communications 37, no. 6 (2019): 1205-1221.
>
> [3] Bottou, Léon, Frank E. Curtis, and Jorge Nocedal. "Optimization methods for large-scale machine learning." SIAM review 60, no. 2 (2018): 223-311.
>
> [4] Lin, Frank Po-Chen, Seyyedali Hosseinalipour, Sheikh Shams Azam, Christopher G. Brinton, and Nicolo Michelusi. "Semi-decentralized federated learning with cooperative D2D local model aggregations." IEEE Journal on Selected Areas in Communications 39, no. 12 (2021): 3851-3869.
>
> [5] Bornstein, Marco, Tahseen Rabbani, Evan Z. Wang, Amrit Bedi, and Furong Huang. "SWIFT: Rapid Decentralized Federated Learning via Wait-Free Model Communication." In The Eleventh International Conference on Learning Representations.
>
> [6] Chen, Yiming, Kun Yuan, Yingya Zhang, Pan Pan, Yinghui Xu, and Wotao Yin. "Accelerating gossip SGD with periodic global averaging." In International Conference on Machine Learning, pp. 1791-1802. PMLR, 2021.
>
> [7] Koloskova, Anastasiia, Tao Lin, Sebastian Urban Stich, and Martin Jaggi. "Decentralized deep learning with arbitrary communication compression." In Proceedings of the 8th International Conference on Learning Representations. 2019.
>
> [8] Sun, Tao, Dongsheng Li, and Bao Wang. "Decentralized federated averaging." IEEE Transactions on Pattern Analysis and Machine Intelligence 45, no. 4 (2022): 4289-4301.

---

> > ### Comment · Reviewer_VTdm · 2024-11-18
> >
> > Thank you for the response. I still have one question: Why does the author not use the gradient diversity assumption, as in Assumption 4.1(c), when the loss function is non-convex?
> >
> > In Assumption 4.2(b), if we set $\zeta_i = 0$, the assumption reduces to the bounded gradient assumption. In this case, $\delta_i$ does not directly represent gradient diversity, as a large gradient norm does not necessarily imply high gradient diversity. Therefore, it remains unclear how $\delta_i$ and $\zeta_i$ are influenced by data heterogeneity.

---

> ### Author Response · Authors · 2024-11-16
>
> ### **[Weakness 3] Frequency of SGDs under diminishing learning rate policy**
> If we do not increase the frequency of SGDs $d_i^{(k)}$ when employing a diminishing learning rate $\alpha^{(k)}$, the final model will still converge to a neighborhood of the optimal solution similar to when a constant learning rate $\alpha$ is employed, but not to the exact solution. The intuitive reasoning behind the necessity of increasing the frequency of SGDs $d_i^{(k)}$ is that any DGD-like algorithm designed to solve the optimization problem in Eq. (1) exactly, needs all clients to participate equally.  Because otherwise the dataset of a client $i$ with higher SGD frequency influences the final model much more than the dataset of a client $j$ with lower participation, i.e., $d_i^{(k)} > d_j^{(k)}$.
>
> The reason that we do not need such a requirement when a constant learning rate $\alpha$ is being employed, is that using a constant learning rate will already culminate in an inexact solution in the neighborhood of the optimal one. In other words, although an increasing SGD probability will help to reduce the size of this neighborhood, it cannot make it become zero. Thus, we have preferred to keep the algorithm more simple in the case of a constant learning rate.
>
> Regarding the dependence the SGD probabilities $d_i^{(k)}$ on the learning rate $\alpha^{(k)}$, note that in fact they can be any increasing function and do not need to necessarily depend on the learning rate. However, in order to get the conventional convergence rate $\mathcal{O}(\ln(k) / \sqrt{k})$ known in the literature, we assume it increases with the same rate that the learning rate decreases, and we assume a diminishing policy for the learning rate as $\alpha^{(k)} = \alpha^{(0)} / \sqrt{1 + k / \gamma}$. Thus, in general if we did not care about obtaining the theoretical convergence rate, it is not necessary in practice to relate $d_i^{(k)}$ and $\alpha^{(k)}$ together in any way, and exact convergence can be achieved as long as $d_i^{(k)}$ are increasing and $\alpha^{(k)}$ is decreasing. To be precise, we would only need $\alpha^{(k)}$ to satisfy the constraints in the setup of Lemma J.4, and $d_i^{(k)}$ to satisfy $\sum_{k=0}^\infty{\alpha^{(k)} (1 - d_i^{(k)})} < \infty$.
>
> ### **[Question 1] Definition of $p_{ii}^{(k)}$**
> We note that since $\hat{v}\_{ii}^{(k)} = 0$ for all $i \in \mathcal{M}$ and $k \geq 0$ in DSpodFL, the update suggested by reviewer reduces to what we already have in the paper. We have added an explanation in Sec. 3.1 of our revised manuscript on the symmetry of $r_{ij}$ and $\hat{v}\_{ij}^{(k)}$, and the fact that $\hat{v}\_{ii}^{(k)} = 0$. Note that in our response to **[Weakness 1] Double stochasticity of $P^{(k)}**, we also discussed about this matter.
>
> Again, thank you for your time and efforts in reviewing our paper. We would appreciate further opportunities to answer any remaining concerns you might have.

---

> ### Author Response · Authors · 2024-11-20
>
> Thank you for your follow up. We answer this question in two parts: (i) why we use a different data heterogeneity assumption for non-convex models, and (ii) how this assumption captures data heterogeneity/gradient diversity.
>
> **(i) Different data heterogeneity assumption for non-convex models:** While the conventional assumption in the literature for data heterogeneity of convex models is $\\| \nabla{F}(\theta) - \nabla{F}_i(\theta) \\| \le \delta_i$ [2, 5, 6] (as referenced in our response to [Weakness 2] previously), our Assumption 4.1-(c) for strongly-convex models includes an extra term, i.e., $\\| \nabla{F}(\theta) - \nabla{F}_i(\theta) \\| \le \delta_i + \zeta_i \\| \theta - \theta^\star \\|$. This is less strict than letting $\zeta_i = 0$, but uses the value of the globally optimal point $\theta^\star \in \mathbb{R}^n$ in its formulation, where $\nabla{F}(\theta^\star) = 0$. When dealing with non-convex models, $\nabla{F}(\theta^{‘\star}) = 0$ will in general hold over a set $\mathcal{J}$ of stationary points $\theta^{‘\star}$ with $\theta^{‘\star} \in \mathcal{J}$, and $|\mathcal{J}| > 1$. Consequently, the data heterogeneity assumption of 4.1-(c), which uses the value of the single globally optimal point $\theta^\star$ for strongly-convex models, will no longer be well-defined for non-convex models, since it is not guaranteed that there will only be one $\theta^\star$ in $\mathbb{R}^n$.
>
> For this reason, if we wanted to use the same assumption for both convex and non-convex models, we would have to make the stricter assumption of $\\| \nabla{F}(\theta) - \nabla{F}_i(\theta) \\| \le \delta_i$, i.e., $\zeta_i = 0$ in our Assumption 4.1-(c) as well. In fact, we can show that if $\zeta_i = 0$, Assumption 4.1-(c) implies Assumption 4.2-(b). Specifically,
> $$\\| \nabla{F}_i(\theta) \\| \le \\| \nabla{F}_i(\theta) - \nabla{F}(\theta) \\| + \\| \nabla{F}(\theta) \\| \le \delta_i + \\| \nabla{F}(\theta) \\| \le \delta_i + \zeta_i’ \\| \nabla{F}(\theta) \\|,$$
> where $\zeta_i’ \geq 1$. Consequently, the strict assumption of $\\| \nabla{F}(\theta) - \nabla{F}_i(\theta) \\| \le \delta_i$ for non-convex models used in [2, 5, 6] (as referenced in our response to [Weakness 2] previously), implies our Assumption 4.2-(b) for the values of $\zeta_i > 1$. Thus, by making this implication as our direct assumption in Assumption 4.2-(b), we are essentially making a less strict assumption. Furthermore, allowing $\zeta_i > 0$ to be flexible for each client $i$ and not restricting them to $\zeta_i \geq 1$ better captures heterogeneity across clients and gives us the possibility to arrive at more generalized results.
>
> **(ii) Assumption 4.2-(b) capturing data heterogeneity/gradient diversity:** Let us now make Assumption 4.2-(b), which is to say $\\| \nabla{F}_i(\theta) \\| \le \delta_i + \zeta_i \\| \nabla{F}(\theta) \\|$. Then, we can calculate a measure of data heterogeneity/gradient diversity in the usual form as
>
> $$\\| \nabla{F}(\theta) - \nabla{F}\_i(\theta) \\| = \\| \frac1m \sum_{j=1, j \neq i}^m{\nabla{F}\_j(\theta)} - (1 - \frac1m) \nabla{F}\_i(\theta) \\| \le \frac1m \sum_{j=1, j \neq i}^m{\\| \nabla{F}\_j(\theta) \\|} + (1 - \frac1m) \\| \nabla{F}\_i(\theta) \\|$$
> $$\le \frac1m \sum_{j=1, j \neq i}^m{(\delta_j + \zeta_j \\| \nabla{F}(\theta) \\|)} + (1 - \frac1m) (\delta_i + \zeta_i \\| \nabla{F}(\theta) \\|) \le 2 (1 - \frac1m) (\delta + \zeta \\| \nabla{F}(\theta) \\|).$$
>
> where we used the fact that $F(\theta) = \frac1m \sum_{i=1}^m{F_i(\theta)}$, $\delta = \max_{i \in \mathcal{M}}{\delta_i}$ and $\zeta = \max_{i \in \mathcal{M}}{\zeta_i}$. Hence, we observe how $\delta_i$ and $\zeta_i$ translate to measures of data heterogeneity/gradient diversity. We can again see that our Assumption 4.2-(b) implies a more relaxed assumption than $\\| \nabla{F}(\theta) - \nabla{F}_i(\theta) \\| \le \delta_i$, since we have $\zeta_i > 0$ in general . The reviewer is correct in saying that $\zeta_i = 0$ in our Assumption 4.2-(b) reduces to the bounded gradient assumptions of [7, 8] (as referenced in our response to [Weakness 2] previously), and in this case, we have essential made a stricter assumption which does not capture the data heterogeneity precisely. As seen above, having $\zeta_i \neq 0$ more generally allows us to capture it.
>
> Thank you again. Please let us know if you have any remaining concerns we can address.

---

> > ### Comment · Reviewer_VTdm · 2024-11-25
> >
> > Thank you for your response. I have no further concerns, and I will revise my score to 8.

---

> > > ### Author Response · Authors · 2024-11-25
> > >
> > > Thank you very much for your time and consideration.

---

### Official Review · Reviewer_aSDE · 2024-10-25

**Soundness:** 3
**Presentation:** 3
**Contribution:** 2
**Rating:** 8
**Confidence:** 4

**Summary:**

The author present DSpodFL, a unified framework of decentralized federated learning. DSpodFL allows each client to take local update and communication intermittently according to its available resources. And DSpodFL framework also involves some existing decentralized federated learning algorithms. The authors provided the convergence analysis of DSpodFL in convex and nonconvex settings.

**Strengths:**

1. A unified algorithm framework.

2. Making clients able to take local update and communication according to its available resources.

3. Convergence analysis in both convex and nonconvex scenarios.

**Weaknesses:**

See questions

**Questions:**

1. The proposed frame can unify existing algorithms including DGD, DFedAvg and RG. If we can obtaining the convergence analysis of these existing analysis **directly** from Theorem 4.11 and 4.12? And if the convergence rates of the existing algorithms obtained from Theorem 4.11 and 4.12 **match** the existing convergence analysis? Please present the detailed steps.

2. Figure 2(b) only show the convergence curves when $\tau\in[0,10000]$, and DSpodFL only achieve the test accuracy of about **0.6** during this period. So can the authors provide convergence curves of these algorithms with a larger $\tau$ such as 20000 or 30000? Similarly, can the authors provide the convergence curves of Figure 2(c) with a larger $\tau$?

3. How do the algorithms including DSpodFL perform in test accuracy with respect to wall-clock time?

I will update the scord according to the response during the discussion period.

---

> ### Author Response · Authors · 2024-11-20
>
> We appreciate your comments on our paper. Thank you for acknowledging the strength of our framework and theoretical guarantees. Please find our responses below.
>
> ### **[Weakness 1] Matching Existing Convergence Rates Using DSpodFL**
> * First, we note that as outlined in Table 1 of Sec. 1, we provide convergence for last iterates of model parameters when dealing with strongly-convex models in our paper, in contrast to the majority of existing literature which only provide convergence for average iterates. Therefore when dealing with strongly-convex models, we can compare our theoretical results only for the DGD baseline because the DGD-like algorithms given in [1, 2] are among the few to show convergence for the last iterates of model parameters as well. According to Theorems 3.6, 5.5, 5.7 and D.1 in [1] and Theorems 3.5, 4.5 and B.2 in [2], the convergence rate of the algorithms ProxSkip, Decentralized Scaffnew, SplitSkip [1], GradSkip, GradSkip+ and VR-GradSkip+ [2] are all geometric, i.e., $\mathcal{O}(\rho^K)$ with $0 < \rho < 1$ (note that most of these algorithms are FL algorithms, and not decentralized FL algorithms). This rate agrees with the rate we provide in Eq. (10) of Theorem 4.11.
> \
> \
> When dealing with non-convex models, we can compare our rate with all the baselines, i.e., [3] for DGD, [4] for RG and [5] for DFedAvg. Since DGD is a special case of RG with no sporadicity in aggregations, i.e., $b_{ij} = 1$ for all $(i,j) \in \mathcal{E}$, we will compare our work with the more recent paper [4]. For DGD and RG, Lemma 17 of [4] shows the convergence upper bound before tuning the constant learning rate $\alpha$, which is
> $$\mathcal{O}{\left( \frac{\mathbb{E}{[F{( \bar{\theta}^{(0)} )}] - F^\star}}{\alpha (K+1)} \right)} + \mathcal{O}{(\alpha)} + \mathcal{O}{(\alpha^2)}.$$
> For DFedAvg, Theorem 1 in [5] with a zero momentum ($\theta = 0$) obtains a less tight bound as follows
> $$\mathcal{O}{\left( \frac{\mathbb{E}{[F{( \bar{\theta}^{(0)} )}] - F^\star}}{\alpha (K+1)} \right)} + \mathcal{O}{(\alpha)} + \mathcal{O}{(\alpha^2)} + \mathcal{O}{(\alpha^3)}.$$
> We observe that by setting $d_{min} = 1$ in Theorem 4.12 of our paper, these convergence rates are recovered. Recall that as discussed in Fig. 1 of our paper, all of these baseline algorithms fit within the general framework of DSpodFL with $d_i = 1$ for all clients $i \in \mathcal{M}$. That is why we can substitute $d_{min} = 1$ in Theorem 4.12 to compare our analytical results with the ones provided in [3-5].
>
> * For a step-by-step comparison of our bounds with the one given in [4] for non-convex models, let us examine the bound derived in the proof of Lemma 16 in [4] before tuning the learning rate, i.e.,
> $$\frac1{2(T+1)} \sum_{t=0}^T{{\\| \nabla{f}(\bar{x}^{(t)}) \\|}_2^2} \le \frac{\mathbb{E}{f(\bar{x}^{(0)})} - f^\star}{(T+1) \eta} + \frac{L \hat{\sigma}^2}{n}\eta + 64 \frac{L^2 [2 \hat{\sigma}^2 + 2(\frac{6 \tau}{p} + M) \hat{\zeta}^2] \tau}{p} \eta^2.$$
>
> Translating these parameters to the setup of our paper, we have
> $$T \to K, t \to r, f \to F, x \to \theta, \eta \to \alpha, L \to \beta, \hat{\sigma} \to \sigma, n \to m, \tau \to 1, p \to 1 - \tilde{\rho}, M \to 0.$$
>
> As a consequence, the bound in [4] using the notation and setup of our paper becomes
> $$\frac1{2(K+1)} \sum_{r=0}^K{{\\| \nabla{F}(\bar{\theta}^{(r)}) \\|}^2} \le \frac{\mathbb{E}{F(\bar{\theta}^{(0)})} - F^\star}{(K+1) \alpha} + \frac{\beta \sigma^2}{m} \alpha + 128 \frac{\beta^2 [\sigma^2 + \frac{6}{1 - \tilde{\rho}} \delta^2]}{1-\tilde{\rho}} \alpha^2.$$
>
> Now, we compare this with the non-asymptotic bound we obtain for non-convex models in Theorem 4.12, which is
> $$\frac{w_1}{K+1} \sum_{r=0}^K{{\\| \nabla{F}(\bar{\theta}^{(r)}) \\|}^2} \le \frac{F(\bar{\theta}^{(0)}) - F^\star}{\alpha (K+1)} + \frac{1 + \Gamma_3}{1-\frac1{\Gamma_1}} \frac{\beta^2}{m (1 - \tilde{\rho})} \frac{{\\| \mathbf{\Theta}^{(0)} - \mathbf{1}\_m \bar{\mathbf{\theta}}^{(0)} \\|}^2}{K+1} + \frac{1 + \Gamma_3}{1-\frac1{\Gamma_1}} \frac{\beta^2 (16 \frac{1+\tilde{\rho}}{1 - \tilde{\rho}} \delta^2 + \sigma^2)}{1 - \tilde{\rho}} \alpha^2 + (1+\Gamma_3) (1-d_{min}) \delta^2 + \frac{\beta \sigma^2}{2m} \alpha.$$
> where we used the fact that $d_{max} \le 1$.
> \
> \
> We can see how our bound compares with the one derived in [4]. The exact value of $w_1$ in our paper depends on the arbitrarily chosen scalars $\Gamma_0, …, \Gamma_4$, but we have in general that $w_1 \le \frac12$. For common terms that our analysis has with [4], we see how their coefficients are also very similar. For example, for the term proportional to $\alpha^2$, we observe that both our bound and the one in [4] are proportional to $\frac{\beta^2}{(1-\tilde{\rho})}$ and $(\sigma^2 +  \frac{q}{1-\tilde{\rho}} \delta^2)$, with slight differences in the value of $q$. Furthermore, for the term proportional to $\alpha$, both our bound and [4] are proportional to $\frac{\beta \sigma^2}{2m}$.

---

> ### Author Response · Authors · 2024-11-20
>
> However, we can observe that our bound consists of two extra terms compared to [4]. The first one, $\frac{1 + \Gamma_3}{1-\frac1{\Gamma_1}} \frac{\beta^2}{m (1 - \tilde{\rho})} \frac{{\\| \mathbf{\Theta}^{(0)} - \mathbf{1}\_m \bar{\mathbf{\theta}}^{(0)} \\|}^2}{K+1}$, is capturing the effect of different model initializations for the clients. The second one, $(1+\Gamma_3) (1-d_{min}) \delta^2$ is capturing the effect of sporadic SGDs in our DSpodFL framework. We observe that if all clients conduct SGDs at every iteration, i.e., $d_i^{(k)} = 1$ for all $i \in \mathcal{M}$ and $k \geq 0$, this term becomes equal to zero, giving us the bound for non-sporadic methods outlined in [4]. Therefore, as we have claimed in Sec. 4.5 of our paper, our convergence bounds recover well-known results in the literature in the degenerate case of $d_{min} = 1$.
>
> [1] Mishchenko, Konstantin, Grigory Malinovsky, Sebastian Stich, and Peter Richtárik. "Proxskip: Yes! local gradient steps provably lead to communication acceleration! finally!." In International Conference on Machine Learning, pp. 15750-15769. PMLR, 2022.
>
> [2] Maranjyan, Artavazd, Mher Safaryan, and Peter Richtárik. "Gradskip: Communication-accelerated local gradient methods with better computational complexity." arXiv preprint arXiv:2210.16402 (2022).
>
> [3] Nedic, Angelia, and Asuman Ozdaglar. "Distributed subgradient methods for multi-agent optimization." IEEE Transactions on Automatic Control 54, no. 1 (2009): 48-61.
>
> [4] Koloskova, Anastasia, Nicolas Loizou, Sadra Boreiri, Martin Jaggi, and Sebastian Stich. "A unified theory of decentralized sgd with changing topology and local updates." In International Conference on Machine Learning, pp. 5381-5393. PMLR, 2020.
>
> [5] Sun, Tao, Dongsheng Li, and Bao Wang. "Decentralized federated averaging." IEEE Transactions on Pattern Analysis and Machine Intelligence 45, no. 4 (2022): 4289-4301.
>
> ### **[Weakness 2] Experiments with higher $\tau$**
> We conducted new experiments during the rebuttal period, by running all the algorithms discussed in our paper for more epochs than what was initially reported on our original manuscript. As requested by the reviewer, in the tables below we present the comparison between DSpodFL and the baselines for the case of FMNIST dataset with non-IID data distribution, and also CIFAR10 with IID distribution. We ran each experiment 4 times, and the information provided in the tables below are the average of those results.
>
> FMNIST Non-IID:
> Algorithm | $\tau = 10000$ | $\tau = 20000$ | $\tau = 30000$ | $\tau = 40000$
> - | - | - | - | -
> DGD | 0.31 | 0.39 | 0.42 | 0.44
> DFedAvg | 0.40 | 0.47 | 0.50 | 0.57
> RG | 0.53 | 0.60 | 0.65 | 0.66
> Sporadic SGDs | 0.30 | 0.36 | 0.40 | 0.42
> DSpodFL | **0.60** | **0.63** | **0.67** | **0.70**
>
>
> CIFAR IID:
> Algorithm | $\tau = 10000$ | $\tau = 20000$ | $\tau = 30000$ | $\tau = 40000$
> - | - | - | - | -
> DGD | 0.45 | 0.57 | 0.65 | 0.68
> DFedAvg | 0.42 | 0.57 | 0.64 | 0.65
> RG | 0.62 | 0.68 | 0.74 | 0.75
> Sporadic SGDs | 0.42 | 0.55 | 0.60 | 0.65
> DSpodFL | **0.68** | **0.73** | **0.75** | **0.76**
>
> We observe that even for higher values of total delay $\tau$, our DSpodFL algorithm is able to outperform the baselines.

---

> ### Author Response · Authors · 2024-11-20
>
> ### **[Weakness 3] Wall-clock time**
> We provide experimental results for accuracy vs. wall-clock time per the reviewer’s request. However, we believe that these wall-clock time results are not informative in our setup, since we are simulating clients as different threads on the same computing cluster, as is typical in federated learning research. Therefore, resource heterogeneity does not naturally exist in this setup, and the stochastic gradient computations of all clients take roughly the same amount of wall-clock time to finish. Moreover, physical communication delays between nodes are not present in this single-cluster setup, and they do not significantly contribute to the overall wall-clock time. Hence, all of the algorithms we consider in our paper perform more or less the same in terms of accuracy vs. wall-clock time:
>
>
> FMNIST IID:
> Algorithm | Wall-clock time = 0.5 sec | Wall-clock time = 1.0 sec | Wall-clock time = 1.5 sec
> - | - | - | -
> DGD | 0.71 | 0.73 | 0.76
> DFedAvg | 0.57 | 0.63 | 0.64
> RG | 0.67 | 0.72 | 0.74
> Sporadic SGDs | 0.68 | 0.73 | 0.75
> DSpodFL | 0.71 | 0.75 | 0.76
>
>
> FMNIST Non-IID:
> Algorithm | Wall-clock time = 3 sec | Wall-clock time = 6 sec | Wall-clock time = 9 sec
> - | - | - | -
> DGD | 0.66 | 0.69 | 0.70
> DFedAvg | 0.37 | 0.41 | 0.44
> RG | 0.52 | 0.55 | 0.58
> Sporadic SGDs | 0.64 | 0.69 | 0.71
> DSpodFL | 0.60 | 0.63 | 0.66
>
>
> CIFAR10 IID:
> Algorithm | Wall-clock time = 10 sec | Wall-clock time = 20 sec | Wall-clock time = 30 sec
> - | - | - | -
> DGD | 0.69 | 0.73 | 0.74
> DFedAvg | 0.66 | 0.74 | 0.75
> RG | 0.66 | 0.71 | 0.74
> Sporadic SGDs | 0.63 | 0.69 | 0.72
> DSpodFL | 0.59 | 0.66 | 0.72
>
>
> CIFAR10 Non-IID:
> Algorithm | Wall-clock time = 30 sec | Wall-clock time = 60 sec | Wall-clock time = 90 sec
> - | - | - | -
> DGD | 0.67 | 0.77 | 0.80
> DFedAvg | 0.74 | 0.78 | 0.80
> RG | 0.67 | 0.76 | 0.79
> Sporadic SGDs | 0.68 | 0.77 | 0.80
> DSpodFL | 0.62 | 0.73 | 0.79
>
>
> In order to have more informative wall-clock time results using a single computing cluster, we can use the probability of SGDs, i.e., $d_i$’s, to model resource heterogeneity in the computational capabilities of clients. Specifically, we repeat the experiment above, this time calculating a weighted average of the wall-clock time of clients via $1 / d_i$. The following table contains some sample points from the accuracy vs. wall-clock time plots. We observe that our DSpodFL outperforms other baselines when resource heterogeneity is taken into account, i.e., when it takes heterogeneous clients different wall-clock times to finish their computations based on their resource availability. As a final note, we observe that the Sporadic SGDs algorithm is performing similarly to our DSpodFL when we measure wall-clock time scaled inversely by probability of SGDs, i.e., $1 / d_i$. This is because there are no wall-clock time delays due to communications on the single cluster setup, which is necessary in order to differentiate between DSpodFL and Sporadic SGDs. Note that the reported wall-clock times are in seconds, but scaled with $1 / d_i$ for each client.
>
>
> FMNIST IID:
> Algorithm | Scaled wall-clock time = 15 sec | Scaled wall-clock time = 30 sec | Scaled wall-clock time = 45 sec
> - | - | - | -
> DGD | 0.66 | 0.68 | 0.71
> DFedAvg | 0.50 | 0.53 | 0.54
> RG | 0.62 | 0.67 | 0.69
> Sporadic SGDs | 0.71 | 0.75 | 0.77
> DSpodFL | 0.70 | 0.76 | 0.75
>
>
> FMNIST Non-IID:
> Algorithm | Scaled wall-clock time = 3 sec | Scaled wall-clock time = 6 sec | Scaled wall-clock time = 9 sec
> - | - | - | -
> DGD | 0.40 | 0.50 | 0.54
> DFedAvg | 0.21 | 0.25 | 0.32
> RG | 0.35 | 0.42 | 0.47
> Sporadic SGDs | 0.62 | 0.65 | 0.66
> DSpodFL | 0.54 | 0.62 | 0.62
>
>
> CIFAR10 IID:
> Algorithm | Scaled wall-clock time = 75 sec | Scaled wall-clock time = 150 sec | Scaled wall-clock time = 225 sec
> - | - | - | -
> DGD | 0.51 | 0.61 | 0.69
> DFedAvg | 0.59 | 0.63 | 0.73
> RG | 0.52 | 0.64 | 0.69
> Sporadic SGDs | 0.66 | 0.73 | 0.74
> DSpodFL | 0.65 | 0.74 | 0.74
>
>
> CIFAR10 Non-IID:
> Algorithm | Scaled wall-clock time = 150 sec | Scaled wall-clock time = 300 sec | Scaled wall-clock time = 450 sec
> - | - | - | -
> DGD | 0.56 | 0.67 | 0.65
> DFedAvg | 0.34 | 0.45 | 0.57
> RG | 0.54 | 0.64 | 0.69
> Sporadic SGDs | 0.70 | 0.76 | 0.79
> DSpodFL | 0.74 | 0.79 | 0.80
>
> Again, thank you for your time and efforts in reviewing our paper. We would appreciate further opportunities to answer any remaining concerns you might have.

---

> ### Comment · Reviewer_aSDE · 2024-11-22
>
> Thank you for the additional analysis and experiments. The reviewer decides to update the score.

---

> > ### Author Response · Authors · 2024-11-22
> >
> > We are grateful for raising your score.

---

### Official Review · Reviewer_soQ2 · 2024-11-04

**Soundness:** 3
**Presentation:** 3
**Contribution:** 2
**Rating:** 8
**Confidence:** 4

**Summary:**

This paper studies decentralized federated learning setting, where a central server is lacking and the agents may perform a stochastic number of local iterations. Furthermore, the peer-to-peer communication also happen in a random fashion. The authors propose a method called DSpodFL, which combines consensus and local gradient descent while allowing stochasticity. The main contribution lies in the algorithm setup/flexibility, convergence analysis using indicator random variable to model stochasticity and allowing for milder assumptions for both convex and nonconvex cases.

**Strengths:**

The algorithm allows for a very general stochasticity capturing varying communication and computation cost and allows for heterogeneity in agents' capabilities. The algorithm is also written in a very clean fashion. The proofs appear to be mostly right.

**Weaknesses:**

The proofs in the appendix lacks explanation for the steps.

My main complaint is that while this is a good first step towards allowing different kinds of stochasticity, the proposed algorithm misses the mark in the sense that the value it converges to is not the original optimal solution. There has been a long line of literature in distributed optimization tackling the same setup (without multiple local steps) by including additional error correction terms, (almost) exact convergence can be achieved. A simple counting of the activation frequencies as in the following paper could help reduce the error term in the limit. Because of this, I'm not sure how useful the proposed method would be for any real applications, especially with large amount of heterogeneity in the activations. I'm marginally willing to accept the paper due to its innovative computation framework.

Ram, S. Sundhar, A. Nedić, and Venugopal V. Veeravalli. "Asynchronous gossip algorithms for stochastic optimization." Proceedings of the 48h IEEE Conference on Decision and Control (CDC) held jointly with 2009 28th Chinese Control Conference. IEEE, 2009.

**Questions:**

For the expectation calculations, some of them seems to be conditioned on past realizations, a filtration should be setup to rigorously justify the steps. If this is not the case, please explain.

How much information is necessary by each of the agents to calculate the learning rates?

Why should all agents use the same learning rates?

---

> ### Author Response · Authors · 2024-11-16
>
> We are grateful for your positive assessment of our paper. Please find below our responses to your individual comments, and how we have clarified them in the revision of our paper.
>
> ### **[Weakness 1] Proofs in the appendix**
> We apologize for some missing explanation in the proofs given in the appendix. In the revision of our manuscript, we have explained the steps of some of the proofs more clearly now. We will continue to add more explanation to the proofs in the appendix during the remainder of the rebuttal phase.
>
> ### **[Weakness 2] Failing to converge to the original optimal solution**
> Thank you for referring us to the paper on asynchronous stochastic optimization [1], which introduces an error correction term to the updates of clients by counting the number of times they have done local updates. Below, please see our response to this concern from two perspectives.
>
> #### **Justification of the current formulation of our methodology**
> * First, we would like to point out that our analytical bounds indeed capture the suboptimality that can be caused by sporadic gradient calculations. We observe in Theorems 4.11 and 4.12 that there exists a term in the asymptotic bounds proportional to $(1 - d_{min}) \delta^2$, where $d_{min} = \min_{0 \le k \le K, i \in \mathcal{M}}{d_i^{(k)}}$ is the minimum participation probability among clients and $|\nabla{F}\_i(\theta^\star)| \le \delta$ indicates that $\delta$ is a bound for local gradients when the globally optimal solution has been reached for strongly-convex models (or a stationary point for non-convex models). Therefore, if the level of heterogeneity among clients is high, i.e., bigger $\delta$, lower participation of some clients, i.e., lower $d_{min}$, will analytically result in the difference of final solution from the globally optimal one be more pronounced. We observe how this would not be an issue if all clients participate equally ($d_{min} = 1$) and/or the data distribution among clients is IID ($\delta = 0$), which is aligned with our intuition.
> \
> \
> Our motivation in introducing sporadic computations in the current formulation of DSpodFL follows some common setups of federated learning, in which although there is a heterogeneity in the data distribution of clients, there may be multiple sets of clients which have similarities in their distributions. For example, in our experiments given in Sec. 5, whenever 50 clients are present in the system and we divide the CIFAR-10 or FMNIST datasets among them in a way that each client would get data points belonging to only 1 class out of 10 classes in the datasets, naturally for each class there will be 5 clients in the system to share a common data distribution. Thus, the motivation behind DSpodFL’s current formulation was to allow clients with higher processing capabilities to make up for clients with lower participation if sharing a common data distribution. Our experimental results in Fig. 3-(c) shows that this is in fact true, i.e., when a higher number of clients are present in the system, better convergence results can be achieved.
>
> * Second, we agree with the reviewer that indeed there are multiple existing methods and future directions that can be implemented to guide the clients models not to drift from the original solution over the training process, e.g., the mentioned activation counting method for adaptation, momentum-based approaches, gradient tracking etc. Particularly in the case of the activation counting method, we see that it is essentially an approach to employ uncoordinated step sizes among clients. Meanwhile, It is widely explored in the uncoordinated step sizes literature that for conventional DGD-like algorithms (especially non gradient tracking methods), the results depend on some notion of step size heterogeneity [2-5]. Furthermore, If the uncoordinated step sizes are allowed to be time-varying, the coordination among the step sizes have to be increasing over time to reach an (almost) exact convergence [6].
> \
> \
> The methodology we have developed for DSpodFL can also be viewed as a decentralized optimization technique with uncoordinated step sizes $\alpha^{(k)} v_i^{(k)}$ for each client, where depending on the value of $v_i^{(k)} = 0$ or $v_i^{(k)} = 1$, client $i$ may use a step size of $0$ or $\alpha^{(k)}$ at iteration $k$, respectively. Similar to [5], we show in Appendices J and K that in order to reach an (almost) exact convergence, we would need both a decreasing step size $\alpha^{(k)}$ and an increasing $d_i^{(k)}$ for all clients, simultaneously. An increasing $d_i^{(k)}$ corresponds to more coordination among clients when choosing their step sizes as the training proceeds.

---

> > ### Author Response · Authors · 2024-11-16
> >
> > * Finally, allowing both uncoordinated step sizes and incorporating gradient tracking to DSpodFL are interesting future directions to our work, which we are currently considering. As in this paper, our main goal was to propose a generalized and elegant framework for sporadic decentralized federated learning based on indicator random variables $v_i^{(k)}$ and $\hat{v}_{ij}^{(k)}$, we focused on the implications of this formulation and left more advanced techniques for future work. In the case of the activating counting method that results in a more stable solution, a sequel to our work should consider very general uncoordinated step sizes $\alpha_i^{(k)}$ for each client in Eq. (2), which changes Eq. (3) to $\Theta^{(k+1)} = P^{(k)} \Theta^{(k)} - A^{(k)} V^{(k)} G^{(k)}$, where $A^{(k)}$ is a diagonal matrix similar to $V^{(k)}$ having the uncoordinated step sizes $\alpha_i^{(k)}$ on its main diagonal. The analysis continues on from there and would be similar to what we currently have in our submission, and would require some more algebra to deduce convergence guarantees under the new formulation.
> >
> > [1] Ram, S. Sundhar, A. Nedić, and Venugopal V. Veeravalli. "Asynchronous gossip algorithms for stochastic optimization." In Proceedings of the 48h IEEE Conference on Decision and Control (CDC) held jointly with 2009 28th Chinese Control Conference, pp. 3581-3586. IEEE, 2009.
> >
> > [2] Nedić, Angelia, Alex Olshevsky, Wei Shi, and César A. Uribe. "Geometrically convergent distributed optimization with uncoordinated step-sizes." In 2017 American Control Conference (ACC), pp. 3950-3955. IEEE, 2017.
> >
> > [3] Xu, Jinming, Shanying Zhu, Yeng Chai Soh, and Lihua Xie. "Convergence of asynchronous distributed gradient methods over stochastic networks." IEEE Transactions on Automatic Control 63, no. 2 (2017): 434-448.
> >
> > [4] Lü, Qingguo, Huaqing Li, and Dawen Xia. "Geometrical convergence rate for distributed optimization with time-varying directed graphs and uncoordinated step-sizes." Information Sciences 422 (2018): 516-530.
> >
> > [5] Xu, Jinming, Shanying Zhu, Yeng Chai Soh, and Lihua Xie. "Augmented distributed gradient methods for multi-agent optimization under uncoordinated constant stepsizes." In 2015 54th IEEE Conference on Decision and Control (CDC), pp. 2055-2060. IEEE, 2015.
> >
> > [6] Wang, Yongqiang, and Angelia Nedić. "Decentralized gradient methods with time-varying uncoordinated stepsizes: Convergence analysis and privacy design." IEEE Transactions on Automatic Control (2023).
> >
> > #### **Generalized framework of DSpodFL**
> > First, note that several other existing works in FL [7] and DFL [8] (which are the special cases of our framework) which make an attempt at improving the resource efficiency of FL/DFL methods, do not converge to the initially optimal point as well due to heterogeneous participation of clients. Since we consider a more generic framework that subsumes many existing frameworks, guaranteeing the exact convergence is much more challenging than existing works in decentralized FL. Despite this, our sporadicity terms to tackle communication/computation resource heterogeneity lead to much better performance in practice where delay and resource consumptions are crucial, as seen in Sec. 5 of our manuscript.
> >
> > However, we note that in very particular special cases of our DSpodFL framework,  i.e., when the participation levels of all clients $d_i^{(k)}$ are set to being equal, our methodology can be made to converge to the solution of the initial problem. This comes at the cost of excessive delay and resource consumptions, and if that is acceptable in our use case, we can set the parameters of DSpodFL such that it converges to the solution of the initial problem, i.e., without taking resource heterogeneity into account. However, we are mainly focusing on scenarios where resource consumptions and delay are also critical factors, and  in such cases, our approach provides significant advantages as shown in our empirical results in Sec. 5.
> >
> > [7] Yang, Haibo, Xin Zhang, Prashant Khanduri, and Jia Liu. "Anarchic federated learning." In International Conference on Machine Learning, pp. 25331-25363. PMLR, 2022.
> >
> > [8] Maranjyan, Artavazd, Mher Safaryan, and Peter Richtárik. "Gradskip: Communication-accelerated local gradient methods with better computational complexity." arXiv preprint arXiv:2210.16402 (2022).

---

> ### Author Response · Authors · 2024-11-16
>
> ### **[Question 1] Filtration for expectation calculations**
> We appreciate your insightful comment, and apologize for not being clear about this. As our end goal in Lemmas 4.7 and 4.8, and thus Definition 4.9, is to take the full expectation of the error terms, we have slightly abused notation and omitted the middle steps where filtration is done first and then the full expectation is calculated. In the revised manuscript, we have explicitly mentioned that this filtration is implicitly being done in our notation in Sec. 4.1.
>
> ### **[Question 2] Choosing the learning rate**
> In this paper, we assume that all clients start from an initial common learning rate $\alpha^{(0)}$. If a choice of constant learning rate policy is opted, they all keep using the same initial learning rate for all iterations, which is what we have done in our experiments. If a diminishing learning rate policy is employed, each client would use a predetermined decay rule and its local clock time to decrease the value of the learning rate over the iterations, so no information from its neighbors are needed to be shared in our work.
>
> To determine what value of $\alpha^{(0)}$ satisfies the constraints, e.g., in Propositions 4.10 & G.6, we observe that full topology knowledge is required, among other parameters like smoothness, data heterogeneity and convexity conditions etc. This is similar to existing works in decentralized ML [9-11] where the topology affects the learning rate condition. However, since the topology and other variables are often not available in practice for each client in decentralized system, a standard procedure is to choose a small enough learning rate (e.g., 0.01, as we have done) to ensure convergence, and validate the effectiveness of the algorithm empirically. Nonetheless, the step size constraints, alongside other upper bounds, give us better theoretical intuitions of how different system parameters affect training behavior. For instance, Propositions 4.10 & G.6 show that a poorly connected graph topology, i.e., one with higher $\tilde{\rho}$ (closer to 1), will limit our choice of step size to a smaller range to guarantee convergence.
>
> [9] Pu, Shi, and Angelia Nedić. "Distributed stochastic gradient tracking methods." Mathematical Programming 187, no. 1 (2021): 409-457.
>
> [10] Koloskova, Anastasia, Nicolas Loizou, Sadra Boreiri, Martin Jaggi, and Sebastian Stich. "A unified theory of decentralized sgd with changing topology and local updates." In International Conference on Machine Learning, pp. 5381-5393. PMLR, 2020.
>
> [11] Xin, Ran, Usman A. Khan, and Soummya Kar. "An improved convergence analysis for decentralized online stochastic non-convex optimization." IEEE Transactions on Signal Processing 69 (2021): 1842-1858.
>
> ### **[Question 3] Common learning rate for all clients**
>
> As we discussed in our response to the **[Weakness 2] Failing to converge to the original optimal solution** section, our goal in developing DSpodFL was to test out the idea of sporadicity in both computations and communications in DGD-like methods to see how much resource efficiency can be gained from this idea. Therefore, we focused on an initial setup where all clients use the same learning rate. As also explained in aforementioned section, the generalization of our analysis to uncoordinated step sizes should be straightforward using the formulation $\Theta^{(k+1)} = P^{(k)} \Theta^{(k)} - A^{(k)} V^{(k)} G^{(k)}$.
>
> Furthermore, we would like to emphasize that it is also a standard procedure in the decentralized FL literature to adopt a common learning rate for all clients. Please see [8-12] as examples ([8-11] are referenced in our responses above).
>
> [12] Bornstein, Marco, Tahseen Rabbani, Evan Z. Wang, Amrit Bedi, and Furong Huang. "SWIFT: Rapid Decentralized Federated Learning via Wait-Free Model Communication." In The Eleventh International Conference on Learning Representations.
>
> We are grateful for your time and efforts in reviewing our paper, and would be glad to answer any remaining concerns you might have.

---

> > ### Comment · Reviewer_soQ2 · 2024-11-22
> >
> > Thank you for the detailed comments and modifications of the proofs. I think this paper serves as a good first step to unify many versions of stochastic methods in the federated space and proposes a clean solution. My main concern about inexact convergence remains, however no paper can address all of the difficulties at the same time. I raised my score in recognition for the authors attempt to provide a clean and unified analytical framework.

---

> > > ### Author Response · Authors · 2024-11-22
> > >
> > > Thank you very much for your consideration.

---

### Official Review · Reviewer_ryLC · 2024-11-04

**Soundness:** 2
**Presentation:** 2
**Contribution:** 2
**Rating:** 5
**Confidence:** 4

**Summary:**

The paper considers decentralized optimization with nodes having varying communication time and computation resources that might also change in time. The paper analyzes convergence rate of proposed framework for strongly convex and non-convex functions and evaluates effectiveness of the framework in experiments.

**Strengths:**

The paper proposed framework is general and allows for varying resource availability. The paper compares its framework with the baselines.

**Weaknesses:**

1. The proposed framework does now allow for the slow gradient computation, as gradients cannot be delayed. This is because the framework does not allow for computation and communication sporadicity being correlated (per assumption 4.3.), and according to (2) each gradient has to be computed at the latest local model $\theta_i$. This means that if some node computed the gradient for a long time, and in the meantime local model is being updated due to the communication on this node occurring, the node has to drop its previous computations and start computing its gradient from scratch on the new model so that it can return the gradient at the latest model.

    There are two possible solutions to this: either allow for communication and computation being correlated (i.e. while node computes, no communication happens), or allow for the gradients to be delayed.

    To me this is a big limitation of the framework.

3. Only asymptotic convergence is proven (i.e. when the number of iterations K goes to infinity), while previous frameworks such as (Even 2024) give convergence bound for any number of iterations K.


4. Paper has several flows in presentation: e.g. first it states that framework the paper analyzes is (2), but then actual analyzed algorithm is in equations (3)-(4) that is not equivalent to (2) (difference comes from re-normalization of the diagonal element p_{ii}^k).

5. Convergence rate derived for non-convex functions is hard to understand, as it depends on some parameters $w_1, w_2, w_3, w_4, w_5$ that are defined only in appendix.

6. Paper does not tune the learning rate, but uses the same learning rate for different algorithms - that might give advantage to some algorithms.

7. Experimental comparison to (Even 2024) is missing. Is there a specific reason why you excluded that baseline from comparison?

**Questions:**

1. Paper mentions that (Even et al 2024) did not consider time variations in resource heterogeneity and I would like to ask you to elaborate on this. In my understanding asynchronous communication allows for resource heterogeneity.

2. What if in proposition 4.10, d_{max}^{k} = 0 for some k? I think it is allowed by the framework and it would mean that you have to divide by zero?

3. in line 360 you state that lim_{k ->0} \Phi(k:0) = 0 is enough to prove that consensus error and average error converges to zero, however even in this case, it depends on convergence of $\Psi^{(k)}$ that might be non zero or might even diverge. Can you clarify on that?

4. Why does the matrix R have to be doubly stochastic if you anyways renormalize the matrix P later to be doubly stochastic?

5. I did not understand how exactly result (10) recovers (Koloskova (2020)) (comment in lines 386 and 387), could you provide a more detailed derivation of this result? also, (Koloskova (2020)) allows for local steps in decentralized SGD, meaning that d_{min} = 0 at those steps, could you compare your result with (Koloskova (2020)) for that local decentralized SGD? Same for the non-convex functions.

6. How does $w_1, w_2, w_3, w_4, w_5$ depend on the key parameters of the problem (parameters from Assumptions 4.2-4.4). Is there a way to simplify the convergence rate, e.g. by using O-notation?

7. What does “average total delay occurred ip to iteration k" mean? Is it total simulated time?

8. In experimental comparison, how exactly are the baseline algorithms implemented under the proposed heterogeneity of communication and computation? How does d_i and b_{ij} translate to the speed of the node?

---

> ### Author Response · Authors · 2024-11-17
>
> We appreciate your time in reading our paper in detail and providing us with your extensive comments. Below please find our responses to each of them, and how we have addressed them in the revised manuscript.
>
> ### **[Weakness 1] Correlation between computations and communications**
> Thank you for your detailed evaluation of our methodology. In this response, we demonstrate how in fact we allow for communications and computations to be correlated, which in turn is a solution to the reviewer’s concern. In our DSpodFL methodology, we indeed allow for slow gradient computations for clients in between aggregations, so that the latest model a client is using to calculate its gradients is not interrupted.
>
> In our paper, we actually do not need to make the assumption that the computation and communication decisions of each client are uncorrelated in Assumption 4.3. Formally, we do not need $\mathbb{E}{[v_i^{(k)} \hat{v}\_{ij}^{(k)}]} = \mathbb{E}{[v_i^{(k)}]} \mathbb{E}{[\hat{v}\_{ij}^{(k)}]}$ for each $i \in \mathcal{M}$ and $j \in \mathcal{N}_i$. Our proofs only assume that the indicator variables $v_i^{(k)}$ and $\hat{v}\_{ij}^{(k)}$ are uncorrelated across the clients and across the network links, respectively. Formally, $\mathbb{E}{[v_i^{(k)} v_j^{(k)}]} = \mathbb{E}{[v_i^{(k)}]} \mathbb{E}{[v_j^{(k)}]}$ and $\mathbb{E}{[\hat{v}\_{ij}^{(k)} \hat{v}\_{lq}^{(k)}]} = \mathbb{E}{[\hat{v}\_{ij}^{(k)}]} \mathbb{E}{[\hat{v}\_{lq}^{(k)}]}$, for all $i \neq j$ and $(i,j) \neq (l,q)$, respectively. These assumptions essentially mean that the decision to conduct SGDs for a client $i$ at any iteration $k$ ($v_i^{(k)} \in \\{ 0,1 \\}$), is uncorrelated to the decision of a different client $j$ in doing so. Similarly, the decision for clients $(i,j)$ to utilize a link between them to communicate with each other ($\hat{v}\_{ij}^{(k)} \in \\{ 0,1 \\}$) is not correlated with the decision of a different pair of clients $(l,q)$ in doing so. Moreover, within each client, we in fact only need to assume that SGD noises $\epsilon_i^{(k)}$ are uncorrelated with the decision to conduct SGD, i.e., $v_i^{(k)}$, and no further assumptions are needed to be made on the communication decisions $\hat{v}\_{ij}^{(k)}$.
>
> We apologize for this confusion, which was introduced by our attempt to present Assumption 4.3 as succinctly as possible. We have now rewritten the assumption to make this clear. Additionally, we have provided the precise formulation of our assumption in Appendix C.
>
> ### **[Weakness 2] Non-asymptotic convergence rates**
> We agree with the reviewer that obtaining exact convergence rates in contrast to just asymptotic convergence behavior carries a higher analytical value. In fact, a big part of our convergence analysis has gone into trying to obtain such **exact convergence rates** for any given iteration $K$. Eq. (9) in Theorem 4.11 shows that the convergence rate of DSpodFL under strongly-convex models is geometric (sometimes also called linear) when a constant learning rate $\alpha$ is used, i.e., $\mathcal{O}{({\rho(\Phi)}^K)}$ where $\rho(\Phi) < 1$. We have also obtained the exact convergence rate for non-convex models, as outlined in Theorem 4.12, which is sub-linear $\mathcal{O}{(1 / K)}$ when a constant learning rate $\alpha$ is used. We would like to emphasize that we only let $K \to \infty$ after obtaining the exact convergence rates, in order to also investigate the asymptotic behavior of DSpodFL.
>
> Moreover, Theorems J.5 and L.5 in the Appendix show that when a diminishing learning rate policy of $\alpha^{(k)} = \alpha^{(0)} / \sqrt{1 + k/\gamma}$ is used, exact convergence can be achieved with a sub-linear rate of $\mathcal{O}{(\ln{K} / \sqrt{K})}$, which is standard in literature [1-3].
>
> [1] Nedić, Angelia, and Alex Olshevsky. "Stochastic gradient-push for strongly convex functions on time-varying directed graphs." IEEE Transactions on Automatic Control 61, no. 12 (2016): 3936-3947.
>
> [2] Makhdoumi, Ali, and Asuman Ozdaglar. "Graph balancing for distributed subgradient methods over directed graphs." In 2015 54th IEEE Conference on Decision and Control (CDC), pp. 1364-1371. IEEE, 2015.
>
> [3] Xi, Chenguang, and Usman A. Khan. "Distributed subgradient projection algorithm over directed graphs." IEEE Transactions on Automatic Control 62, no. 8 (2016): 3986-3992.

---

> > ### Comment · Reviewer_ryLC · 2024-11-18
> > **Reply to rebuttal for Weakness 1 & 2**
> >
> > Dear Authors,
> >
> > [weakness 1]: Thanks so much for your reply. If I understand correctly your proof, you still need to assume the independence of $v_{i}^{(k)}$ to all the previous $v_{ij}^{(k - j)}$ for $j > 0$. That is because for example when taking expectation on lines 1046-1049 (since you assume independence of $\theta_k$ and $v_{i}^{(k)}$). Which basically leads to the same problem as I explained above, of not allowing gradients to be delayed for more than 1 iteration.
> >
> > [weakness 2]: Could you compare non asymptotic rates that you get with previous works? do you get similar dependence on graph spectral gap as well as on the other problem parameters (such as smoothness, etc)?

---

> ### Author Response · Authors · 2024-11-17
>
> ### **[Weakness 3] Equivalence of Eq. (2) and (3)**
> We apologize for omitting the details due to page constraints. We would like to clarify that Eq. (3) is equivalent to Eq. (2). Please find below why this is the case using the re-definition of the transition weights given in Eq. (4). We have
> $$\theta_i^{(k+1)} = \theta_i^{(k)} + \sum_{j \in \mathcal{M}}{r_{ij} \left( \theta_j^{(k)} - \theta_i^{(k)} \right) \hat{v}\_{ij}^{(k)}} - \alpha^{(k)} g_i^{(k)} v_i^{(k)} = \theta_i^{(k)} (1 - \sum_{j \in \mathcal{M}}{r_{ij} \hat{v}\_{ij}^{(k)}}) + \sum_{j \in \mathcal{M}}{r_{ij} \hat{v}\_{ij}^{(k)} \theta_j^{(k)}} - \alpha^{(k)} g_i^{(k)} v_i^{(k)}.$$
> Now, using the definition of $p_{ij}^{(k)}$ and $p_{ii}^{(k)}$ as given in Eq. (4) and combining the vectors $\theta_i^{(k)}$ and $g_i^{(k)}$ into rows of the matrices $\Theta^{(k)}$ and $G^{(k)}$, respectively, and forming the doubly-stochastic symmetric matrix $P^{(k)}$ by using the elements $p_{ij}^{(k)}$ and also forming the diagonal matrix $V^{(k)}$ by elements $v_i^{(k)}$, we will arrive at Eq. (3).
>
> ### **[Weakness 4] Convergence rate for non-convex models**
> We apologize for not being able to present the constant scalars $w_1, …, w_5$ in the main text due to space constraints, and we can see how excluding them from Theorem 4.12 has made it harder to fully grasp. In the revision of our paper, we have directly written them down in Theorem 4.12.
>
> In the original presentation of Theorem 4.12, our main goal was to directly show the influence of the learning rate $\alpha$ and minimum SGD probability $d_{min}$, to emphasize how our bound generalizes the results of existing literature under the addition of sporadic computations. As we have also discussed in the paper, the bound in Theorem 4.12 includes all terms present in the bounds of works such as [4, 5], i.e., $\mathcal{O}{\left( \frac{F{( \bar{\theta}^{(0)} )} - F^\star}{\alpha (K+1)} \right)}$, $\mathcal{O}{\left( \frac{{\\| \mathbf{\Theta}^{(0)} - \mathbf{1}\_m \bar{\mathbf{\theta}}^{(0)} \\|}^2}{K+1} \right)}$, $\mathcal{O}{( \alpha^2 )}$ and $\mathcal{O}{(\alpha)}$, with the addition of an extra term $\mathcal{O}{(1 - d_{min})}$ due to sporadic computations. We observe how this extra term in fact becomes zero if we remove the sporadic computations component of DSpodFL, i.e., choose $d_i^{(k)} = 1$ for all $i \in \mathcal{M}$ so that $d_{\min} = 1$.
>
> Therefore, we have delegated all other parameters, including $\delta$ for data heterogeneity, $\beta$ for smoothness, $m$ for number of clients, $\sigma^2$ for the SGD noise variance and $\tilde{\rho}$ for the spectral radius of the mixing matrix to the constant scalars $w_1, …, w_5$. In our response to [Question 6], please see how each of these constant scalars are dependent on our problem-related parameters.
>
> [4] Koloskova, Anastasia, Nicolas Loizou, Sadra Boreiri, Martin Jaggi, and Sebastian Stich. "A unified theory of decentralized sgd with changing topology and local updates." In International Conference on Machine Learning, pp. 5381-5393. PMLR, 2020.
>
> [5] Sun, Tao, Dongsheng Li, and Bao Wang. "Decentralized federated averaging." IEEE Transactions on Pattern Analysis and Machine Intelligence 45, no. 4 (2022): 4289-4301.
>
> ### **[Weakness 5] Common learning rate for all clients**
> The choice for the learning rate $\alpha = 0.01$ that we have made for the baselines DGD [4], RG [4] and DFedAvg [5], comes from the original papers. For this reason, we believe the comparison we have made between our DSpodFL algorithm and the baselines, is a fair comparison since we have utilized the same learning rate reported in [4, 5] to properly work in training their algorithms.
>
> On another note, our choice of using the same learning for all baselines, and not individually tuning it for each of them, follows existing research in the decentralized FL literature [4, 6, 7] (where [4] is referenced in our response to [Weakness 4]).
>
> [6] Pu, Shi, and Angelia Nedić. "Distributed stochastic gradient tracking methods." Mathematical Programming 187, no. 1 (2021): 409-457.
>
> [7] Bornstein, Marco, Tahseen Rabbani, Evan Z. Wang, Amrit Bedi, and Furong Huang. "SWIFT: Rapid Decentralized Federated Learning via Wait-Free Model Communication." In The Eleventh International Conference on Learning Representations.

---

> > ### Comment · Reviewer_ryLC · 2024-11-19
> >
> > Thanks for explaining weakness 3, I have no further comments.
> >
> > [weakness 4]: see reply to weakness 2.
> >
> > [weakness 5]: I believe that [4] did not compare different algorithms, and run their algorithm only on artificially generated data, mentioning that "stepsize = 0.01 is chosen for illustration purposes", rather than being the best stepsize choice. As I understand changing the dataset might change the smoothness constant, making the optimal stepsize to change. Moreover, [7] has the stepsize equal to 0.1 rather than 0.01. I believe that tuning the stepsize separately for each algorithm will allow for a more fair comparison. Imagine I want to compare the two algorithms: $x_{t + 1} = x_t - \gamma \nabla f(x_t)$  and $x_{t + 1} = x_t - 0.01 \gamma \nabla f(x_t)$. Choosing the same stepsize for both of these algorithms would not allow for a fair comparison.

---

> ### Author Response · Authors · 2024-11-17
>
> ### **[Weakness 6] Numerical comparison with Even et al., 2024 [8]**
>
> The reason that we have not compared our DSpodFL framework to [8] directly, is that [8] does not provide any numerical experiments using the methodology developed within. Without any specific parameter values presented in the paper, we found it hard to make a fair comparison between our method and [8]. In other words, [8] is mostly a theoretical contribution to the field of decentralized optimization, which we have referenced in our work.
>
> [8] Even, Mathieu, Anastasia Koloskova, and Laurent Massoulié. "Asynchronous sgd on graphs: a unified framework for asynchronous decentralized and federated optimization." In International Conference on Artificial Intelligence and Statistics, pp. 64-72. PMLR, 2024.
>
> ### **[Question 1] Time-invariant resources in Even et al., 2024 [8]**
>
> We have stated that [8] (as referenced in our response to [Weakness 6]) does not allow time variations in resource heterogeneity, as they have assumed that resource availability of clients stay the same across the training process. Indeed, they allow for resource heterogeneity in their formulation as the reviewer has correctly mentioned, it is just that any initial resource considered for a client $i \in \mathcal{M}$ will remain fixed throughout the training. In contrast, through our formulation of DSpodFL using simple yet elegant indicator variables $v_i^{(k)}$ and $\hat{v}\_{ij}^{(k)}$ and their expected values $\mathbb{E}{[v_i^{(k)}]} = d_i^{(k)}$ and $\mathbb{E}{[\hat{v}\_{ij}^{(k)}]} = b_{ij}^{(k)}$, we allow for time-varying $d_i^{(k)}$ and $b_{ij}^{(k)}$ in DSpodFL. This in turn corresponds to dynamic resource heterogeneity in the system, which is not considered in [8]. Allowing for $d_i^{(k)}$ and $b_{ij}^{(k)}$ to be time-varying makes the theoretical analysis more challenging, e.g., see Lemma D.4.
>
> Also note that in Appendix P.4, we have conducted several experiments under the setup where both computational and communication resources are time-varying over the training process. We observe that DSpodFL is able to outperform the baselines even in these scenarios (some more than the other).
>
> ### **[Question 2] Clarification on $d_{max}^{(k)} = 0$ in Proposition 4.10**
> We can definitely substitute $d_{max}^{(k)} = 0$ in the constraint on $\alpha^{(k)}$ given in Proposition 4.10, and since this constraint is the minimum of three values, we will get that $\alpha^{(k)} < \min\\{ 1/\mu, \infty, \infty \\} = 1/\mu$. However, we also note that $d_{max}^{(k)} = 0$ essentially means that we have $d_i^{(k)} = 0$ for all $i \in \mathcal{M}$ for a particular iteration $k$. Therefore, none of the clients will compute stochastic gradients at that iteration, and thus there is no need to choose any learning rate $\alpha^{(k)}$ to begin with.
>
> ### **[Question 3] Clarification on $\lim_{k \to 0}{\Phi^{(k:0)}} = 0$ in Proposition 4.10**
> We thank you again for your thorough examination of our paper. It is correct that the values that the consensus error and average model error converge to, depend on $\Psi^{(k)}$. However, as we have stated in the paper, $\lim_{k \to 0}{\Phi^{(k:0)}} = 0$ is a sufficient condition just for **convergence**, and the exact values that $\nu^{(k)}$ will converge to will depend on $\Psi^{(k)}$. Since the elements of $\Psi^{(k)}$, i.e., $\psi_1^{(k)}$ and $\psi_2^{(k)}$, are bounded values as seen in their definitions in Lemmas 4.7 and 4.8, we do not need to worry about divergence of $\nu^{(k)}$ as long as $\lim_{k \to 0}{\Phi^{(k:0)}} = 0$. However, convergence of $\nu^{(k)}$ to non-zero values is indeed possible, and occurs in Theorems 4.11 and 4.12 due to using a constant learning rate. Achieving zero optimality gap requires DSpodFL to use diminishing learning rate, which is standard in literature, and is presented in detail in Appendices J and L.

---

> ### Author Response · Authors · 2024-11-17
>
> ### **[Question 4] Double stochasticity of matrices $R$ and $P^{(k)}$**
> The adjustment of transition weights $p_{ii}^{(k)}$ to make the matrix $P^{(k)}$ doubly stochastic, is primarily to account for the sporadicity of communications at each iteration $k$ of DSpodFL. In other words, since we have $p_{ij}^{(k)} = r_{ij} \hat{v}\_{ij}^{(k)}$ and $\hat{v}\_{ij}^{(k)} \in \\{ 0,1 \\}$, we essentially zero out some of the elements at each row of the matrix $R$ to form the off-diagonal elements of $P^{(k)}$, and thus need to adjust the diagonal elements to retain double stochasticity. However, note that the guarantee that $0 < p_{ii}^{(k)} \le 1$ relies on double stochasticity of $R$ and Metropolis-Hastings weights that are used to design its entries $r_{ij}$. Therefore, although the reviewer is correct in that we do not use the diagonal values $r_{ii}$ of $R$ directly and therefore they can have any arbitrary value, but we still need off-diagonal elements of each row in matrix $R$ to satisfy $0 \le \sum_{j \in \mathcal{M}, j \neq i}{r_{ij}} < 1$.
>
> For coherence in moving from $R$ to $P^{(k)}$, we have opted to complete the constraint $0 \le \sum_{j \in \mathcal{M}, j \neq i}{r_{ij}} < 1$ by defining $r_{ii}$ such that the matrix $R$ would become doubly-stochastic. This is also because if for a client $i \in \mathcal{M}$, we have that it is connected to all other clients in the network, i.e., $\mathcal{N}\_i = \mathcal{M} \setminus \\{ i \\}$, then if at some iteration it communicates with all of those clients, i.e., $\hat{v}\_{ij}^{(k)} = 1$ for all $j \in \mathcal{N}\_i$, then $p_{ii}^{(k)} = r_{ii}$.
>
> ### **[Question 5] Theoretical comparison with Koloskova et al., 2020 [4]**
>
> First, we note that as outlined in Table 1 of Sec. 1, we provide convergence for last iterates of model parameters when dealing with strongly-convex models in our paper, in contrast to the majority of existing literature which only provide convergence for average iterates. Therefore when dealing with strongly-convex models, we can compare our theoretical results only for the DGD baseline because the DGD-like algorithms given in [9, 10] are among the few to show convergence for the last iterates of model parameters as well. We apologize for any confusion that we might have caused by citing the work by Koloskova et al., 2020 [4] (referenced in our response to [Weakness 4]) in lines 386 and 387 of our original manuscript, as our goal was to cite this work as a paper which presents a unified framework for DGD methods. However, we will be able to compare our results in the non-convex case with [4], which we will talk about later in this response. In the revised manuscript  when presenting our results for convex models, we have changed the paper we cite on DGD-like methods to [10], which also shows convergence for last iterates and we are able to better compare our analytical bounds with it when dealing with strongly-convex models.
>
> * For strongly-convex models, according to Theorems 3.6, 5.5, 5.7 and D.1 in [9] and Theorems 3.5, 4.5 and B.2 in [10], the convergence rate of the algorithms ProxSkip, Decentralized Scaffnew, SplitSkip [9], GradSkip, GradSkip+ and VR-GradSkip+ [10] are all geometric, i.e., $\mathcal{O}(\rho^K)$ with $0 < \rho < 1$ (note that most of these algorithms are FL algorithms, and not decentralized FL algorithms). This rate agrees with the rate we provide in Eq. (10) of Theorem 4.11.
>
> * When dealing with non-convex models, we can compare our rate with all the baselines, i.e., [11] for DGD, [4] for RG and [5] for DFedAvg (referenced in our response to [Weakness 4]). Since DGD is a special case of RG with no sporadicity in aggregations, i.e., $b_{ij} = 1$ for all $(i,j) \in \mathcal{E}$, we will compare our work with the more recent paper [4]. For DGD and RG, Lemma 17 of [4] shows the convergence upper bound before tuning the constant learning rate $\alpha$, which is
> $$\mathcal{O}{\left( \frac{\mathbb{E}{[F{( \bar{\theta}^{(0)} )}] - F^\star}}{\alpha (K+1)} \right)} + \mathcal{O}{(\alpha)} + \mathcal{O}{(\alpha^2)}.$$
> For DFedAvg, Theorem 1 in [5] with a zero momentum ($\theta = 0$) obtains a less tight bound as follows
> $$\mathcal{O}{\left( \frac{\mathbb{E}{[F{( \bar{\theta}^{(0)} )}] - F^\star}}{\alpha (K+1)} \right)} + \mathcal{O}{(\alpha)} + \mathcal{O}{(\alpha^2)} + \mathcal{O}{(\alpha^3)}.$$
> We observe that by setting $d_{min} = 1$ in Theorem 4.12 of our paper, these convergence rates are recovered. Recall that as discussed in Fig. 1 of our paper, all of these baseline algorithms fit within the general framework of DSpodFL with $d_i = 1$ for all clients $i \in \mathcal{M}$. That is why we can substitute $d_{min} = 1$ in Theorem 4.12 to compare our analytical results with the ones provided in [3-5].

---

> ### Author Response · Authors · 2024-11-17
>
> Finally, regarding papers that allow multiple local steps in between aggregations in decentralized FL, like [4, 5, 10], we note that we will in fact have $d_i^{(k)} = 1$ for all $i \in \mathcal{M}$ in those iterations, resulting in $d_{min} = 1$, but instead have $b_{ij}^{(k)} = 0$. Looking at the bounds in Theorems 4.11 and 4.12, we see that substituting $d_{min} = 1$ simplifies the bounds and gives us the ones provided in [4, 5, 10], up to differences in constant scalars.
>
> [9] Mishchenko, Konstantin, Grigory Malinovsky, Sebastian Stich, and Peter Richtárik. "Proxskip: Yes! local gradient steps provably lead to communication acceleration! finally!." In International Conference on Machine Learning, pp. 15750-15769. PMLR, 2022.
>
> [10] Maranjyan, Artavazd, Mher Safaryan, and Peter Richtárik. "Gradskip: Communication-accelerated local gradient methods with better computational complexity." arXiv preprint arXiv:2210.16402 (2022).
>
> [11] Nedic, Angelia, and Asuman Ozdaglar. "Distributed subgradient methods for multi-agent optimization." IEEE Transactions on Automatic Control 54, no. 1 (2009): 48-61.
>
> ### **[Question 6] The constant scalars $w_1, …, w_5$**
> The values of these scalars which show up in Theorem 4.12 are as follows
> $$w_1 = \frac12 ( 1 - \frac1{\Gamma_0} ) ( 1 - \frac1{\Gamma_2} ) ( 1 - \frac1{\Gamma_4^2} ), w_2 = \frac{\beta^2 d_{\max} ( 1 + \Gamma_3 )}{m ( 1 - \tilde{\rho} ) \left( 1 - \frac1{\Gamma_1} \right)}, w_3 = m d_{\max} ( 16 \frac{1 + \tilde{\rho}}{1 - \tilde{\rho}} \delta^2 + \sigma^2), w_4 = ( 1 + \Gamma_3 ) \delta^2, w_5 = \frac{\beta d_{\max} \sigma^2}{2m}.$$
> Note that the constant scalars $\Gamma_i$ with $0 \le i \le 4$ are arbitrary which only need to satisfy the conditions $\Gamma_0, \Gamma_1, \Gamma_2, \Gamma_4 > 1$ and $\Gamma_3 > 0$.
>
> To simplify the convergence upper bounds in Theorem 4.12 further using O-notation, we can assume a specific choice of constant learning rate $\alpha$ and minimum SGD probability $d_{min}$. If we run DSpodFL for $K$ iterations using non-convex models, let us choose $\alpha \sim 1 / \sqrt{K}$ and $d_{min} = 1 - 1/K$. Then, we would have that the convergence upper bound diminishes with the rate
> $$\mathcal{O}{\left( \frac1K \left[ {\\| \mathbf{\Theta}^{(0)} - \mathbf{1}\_m \bar{\mathbf{\theta}}^{(0)} \\|}^2 w_2 + w_2 w_3 + w_4 \right] \right)} + \mathcal{O}{\left( \frac1{\sqrt{K}} \left[ (F{( \bar{\theta}^{(0)} )} - F^\star) + w_5 \right] \right)},$$
> where we have absorbed the constant scalar $w_1$ in the O-notation as it does not depend on problem-related parameters. Substituting the constant scalars $w_2, …, w_5$, we get
> $$\mathcal{O}{\left( \frac1K \left[ \frac{\beta^2 d_{\max} ( 1 + \Gamma_3 )}{m ( 1 - \tilde{\rho} ) \left( 1 - \frac1{\Gamma_1} \right)} \left( {\\| \mathbf{\Theta}^{(0)} - \mathbf{1}\_m \bar{\mathbf{\theta}}^{(0)} \\|}^2 + \frac{\beta^2 d_{\max} ( 1 + \Gamma_3 )}{m ( 1 - \tilde{\rho} ) \left( 1 - \frac1{\Gamma_1} \right)} m d_{\max} ( 16 \frac{1 + \tilde{\rho}}{1 - \tilde{\rho}} \delta^2 + \sigma^2) \right) + ( 1 + \Gamma_3 ) \delta^2 \right] \right)} + \mathcal{O}{\left( \frac1{\sqrt{K}} \left[ (F{( \bar{\theta}^{(0)} )} - F^\star) + \frac{\beta d_{\max} \sigma^2}{2m} \right] \right)}$$
> for the non-convex convergence rate
>
> ### **[Question 7] Comparison metric $\tau$**
> As outlined in Sec. 5.1, the average total delay $\tau_{total}^{(k)}$ at iteration $k$ is defined as the sum of average processing delay $\tau_{proc}^{(k)}$ and average transmission delay $\tau_{trans}^{(k)}$ up to iteration $k$. At each iteration, depending on whether a client $i$ computed SGDs or not, i.e., $v_i^{(k)} \in \\{ 0,1 \\}$, there will be a processing delay incurred to finish the computation proportional to $1/d_i^{(k)}$. Similarly, depending on $\hat{v}\_{ij}^{(k)} \in \\{0,1\\}$, some of the links $(i,j)$ in the network graph will be utilized for communications, incurring a transmission delay proportional to $1/b_{ij}^{(k)}$ for that link. In order to obtain comparable result regardless of the size, connectivity and the topology of the underlying graph, we normalize (i.e., average) both processing and transmission delay to obtain $\tau_{proc}^{(k)}$, $\tau_{trans}^{(k)}$. Formally, these are defined as $\tau_{trans}^{(k)} = [ \sum_{i=1}^m{( 1 / | \mathcal{N}\_i | ) \sum_j{\hat{v}\_{ij}^{(k)} / b_{ij}}} ] / [ \sum_{i=1}^m{( 1 / | \mathcal{N}\_i | ) \sum_j{1 / b_{ij}}} ]$ and $\tau_{proc}^{(k)} = [ \sum_{i=1}^m{v_i^{(k)} / d_i}] / [\sum_{i=1}^m{1 / d_i}]$. Finally, we define $\tau_{total}^{(k)} = \tau_{trans}^{(k)} + \tau_{proc}^{(k)}$ as the average total delay as a metric to compare DSpodFL with the baselines.

---

> ### Author Response · Authors · 2024-11-17
>
> ### **[Question 8] Effect of $d_i$/$b_{ij}$ on speed of a client/link**
> Having explained the average total delay $\tau_{total}^{(k)} = \tau_{trans}^{(k)} + \tau_{proc}^{(k)}$ in the answer to [Question 7], we can now see what happens to different baselines under our proposed heterogeneity framework, and how $d_i$ and $b_{ij}$ translate to the speed of a client and a link, respectively.
>
> In DGD, we have that computations and communications occur at every iteration, i.e., $v_i^{(k)} = \hat{v}\_{ij}^{(k)} = 1$ for all $i \in \mathcal{M}$, $(i,j) \in \mathcal{E}$ and $k \geq 0$. Thus, we will have that $\tau_{trans}^{(k)} = [ \sum_{i=1}^m{( 1 / | \mathcal{N}\_i | ) \sum_j{1 / b_{ij}}} ] / [ \sum_{i=1}^m{( 1 / | \mathcal{N}\_i | ) \sum_j{1 / b_{ij}}} ] = 1$ and $\tau_{proc}^{(k)} = [ \sum_{i=1}^m{1 / d_i}] / [\sum_{i=1}^m{1 / d_i}] = 1$, and thus each iteration of training will incur a total average delay of $\tau_{total}^{(k)} = 2$ for this baseline. For DFedAvg, we still have $\tau_{proc}^{(k)} = 1$ as SGDs occur at every iteration. But since aggregations occur every $D$ iterations, i.e., clients conduct $D$ consecutive iterations of local SGD steps before communications, we will have $\tau_{trans}^{(k)} = 0$ for $D$ iterations and then $\tau_{trans}^{(k)} = 1$ for the next iteration. This deterministic cycle will then continue for DFedAvg.
>
> In RG and the Sporadic SGDs baselines, we only have one of these operations occurring at every iteration, i.e., computations and communications, respectively. This means that in RG, we have $v_i^{(k)} = 1$ is deterministic but $\hat{v}\_{ij}^{(k)}$ is stochastic, and for Sporadic SGDs, we have deterministic $\hat{v}\_{ij}^{(k)} =1$ but stochastic $v_i^{(k)}$. Thus, for RG we will have $\tau_{total}^{(k)} = \tau_{trans}^{(k)} + 1 < 2$ implied by $\tau_{proc}^{(k)} = 1$, and for Sporadic SGDs, we will have that $\tau_{total}^{(k)} = 1 + \tau_{proc}^{(k)} < 2$ implied by $\tau_{trans}^{(k)} = 1$.
>
> However, note that in DSpodFL, both computation and communication operations are carried out in a stochastic way, and thus each iteration of training requires less delay incurred on the whole decentralized system to start the next round of training. Note however, that less computation and communication might come at the cost of losing performance at each iteration, but our motivation in DSpodFL was to prove that in fact if we evaluate these algorithms based on their total delay, it can outperform existing baselines. The intuition is that due to the data distribution of clients available in the network, their processing capabilities, the graph topology and the link bandwidth capabilities, it is not necessary to over utilize all of these resources at every iteration for fast convergence. In fact, a resource-aware approach like DSpodFL, takes a step at optimally utilizing the resources to achieve the same final solutions in a shorter amount of time.
>
> We would like to thank you again for your time and efforts in providing a comprehensive review of our paper. We have revised our manuscript accordingly, and would appreciate the opportunity to respond to any remaining concerns that you might have.

---

> ### Author Response · Authors · 2024-11-20
>
> We are grateful for your follow up questions regarding our paper. Below, please find our responses to each comment.
>
> ### **Follow-up to [Weakness 1]**:
> It is correct that our formulation allows for $v_i^{(k)}$ and $\hat{v}_{ij}^{(k)}$ to be uncorrelated only at iteration $k$, and our expectation calculations at any iteration $k$ in fact depend on conditioning over past realizations of random variables involved in our paper. However, we will demonstrate that in our formulation of DSpodFL, it is sufficient for those indicator random variables to be uncorrelated for only one iteration. This is mainly because of the way we define an iteration of training in our work, which is elaborated in several steps as follows:
>
> (i) Start iteration $k=0$ at time $t=0$.
>
> (ii) Let $T_i(t)$ be the processing time it takes for client $i$ to finish its gradient computation at time $t$. Then, start all inter-client communications at time $t’ = t + \max_{i \in \mathcal{M}}{T_i(t)}$ after all gradient computations have finished. This essentially means having correlated $v_i$ and $\hat{v}_{ij}$ for a single iteration, because we are delaying the communications until all gradients have been computed.
>
> (iii) Define $T_{(i,j)}'(t')$ be the total transmission time it takes for clients $i, j$ to use the link $(i, j)$ to transmit their models to each other at time $t'$. Then wait until $t'' = t' + \max_{(i,j) \in \mathcal{E}}{T_{(i,j)}'(t')}$ before starting any other gradient computation for the clients. This again is possible due to $v_i$ and $\hat{v}_{ij}$ being allowed to be correlated for a single iteration.
>
> (iv) Start iteration $k+1$ at time $t = t'' = t + \max_{i \in \mathcal{M}}{T_i(t)} + \max_{(i,j) \in \mathcal{E}}{T_{(i,j)}'(t')}$, and start over from step (ii).
>
> This shows that in DSpodFL, the time in between the iterations can vary. In other words, the actual time in seconds it takes for DSpodFL to get from iteration $k_1$ to $k_2$, is not necessarily equal to the time it takes to get from $k_3$ to $k_4$, with $k_4 - k_3 = k_2 - k_1$, $k_2 > k_1$ and $k_4 > k_3$. In fact, this is how DSpodFL is able to outperform the baselines because according to the resource availability of clients and network links at a certain time, it adjusts the participation of clients in the DFL process to reduce the delay in between the iterations for faster convergence. Hence, it does not require the same amount of work from stragglers as much as it does from resource-abundant clients.
>
> ### **Follow-up to [Weaknesses 2 & 4]**:
> Let us examine the bound derived in the proof of Lemma 16 in [4] before tuning the learning rate, i.e.,
> $$\frac1{2(T+1)} \sum_{t=0}^T{{\\| \nabla{f}(\bar{x}^{(t)}) \\|}_2^2} \le \frac{\mathbb{E}{f(\bar{x}^{(0)})} - f^\star}{(T+1) \eta} + \frac{L \hat{\sigma}^2}{n}\eta + 64 \frac{L^2 [2 \hat{\sigma}^2 + 2(\frac{6 \tau}{p} + M) \hat{\zeta}^2] \tau}{p} \eta^2.$$
>
> Translating these parameters to the setup of our paper, we have
> $$T \to K, t \to r, f \to F, x \to \theta, \eta \to \alpha, L \to \beta, \hat{\sigma} \to \sigma, n \to m, \tau \to 1, p \to 1 - \tilde{\rho}, M \to 0.$$
>
> As a consequence, the bound in [4] using the notation and setup of our paper becomes
> $$\frac1{2(K+1)} \sum_{r=0}^K{{\\| \nabla{F}(\bar{\theta}^{(r)}) \\|}^2} \le \frac{\mathbb{E}{F(\bar{\theta}^{(0)})} - F^\star}{(K+1) \alpha} + \frac{\beta \sigma^2}{m} \alpha + 128 \frac{\beta^2 [\sigma^2 + \frac{6}{1 - \tilde{\rho}} \delta^2]}{1-\tilde{\rho}} \alpha^2.$$
>
> Now, we compare this with the non-asymptotic bound we obtain for non-convex models in Theorem 4.12, which is
> $$\frac{w_1}{K+1} \sum_{r=0}^K{{\\| \nabla{F}(\bar{\theta}^{(r)}) \\|}^2} \le \frac{F(\bar{\theta}^{(0)}) - F^\star}{\alpha (K+1)} + \frac{1 + \Gamma_3}{1-\frac1{\Gamma_1}} \frac{\beta^2}{m (1 - \tilde{\rho})} \frac{{\\| \mathbf{\Theta}^{(0)} - \mathbf{1}\_m \bar{\mathbf{\theta}}^{(0)} \\|}^2}{K+1} + \frac{1 + \Gamma_3}{1-\frac1{\Gamma_1}} \frac{\beta^2 (16 \frac{1+\tilde{\rho}}{1 - \tilde{\rho}} \delta^2 + \sigma^2)}{1 - \tilde{\rho}} \alpha^2 + (1+\Gamma_3) (1-d_{min}) \delta^2 + \frac{\beta \sigma^2}{2m} \alpha.$$
> where we used the fact that $d_{max} \le 1$.
>
> We can see how our bound compares with the one derived in [4]. The exact value of $w_1$ in our paper depends on the arbitrarily chosen scalars $\Gamma_0, …, \Gamma_4$, but we have in general that $w_1 \le \frac12$. For common terms that our analysis has with [4], we see how their coefficients are also very similar. For example, for the term proportional to $\alpha^2$, we observe that both our bound and the one in [4] are proportional to $\frac{\beta^2}{(1-\tilde{\rho})}$ and $(\sigma^2 +  \frac{q}{1-\tilde{\rho}} \delta^2)$, with slight differences in the value of $q$. Furthermore, for the term proportional to $\alpha$, both our bound and [4] are proportional to $\frac{\beta \sigma^2}{2m}$.

---

> ### Author Response · Authors · 2024-11-20
>
> However, we can observe that our bound consists of two extra terms compared to [4]. The first one, $\frac{1 + \Gamma_3}{1-\frac1{\Gamma_1}} \frac{\beta^2}{m (1 - \tilde{\rho})} \frac{{\\| \mathbf{\Theta}^{(0)} - \mathbf{1}\_m \bar{\mathbf{\theta}}^{(0)} \\|}^2}{K+1}$, is capturing the effect of different model initializations for the clients. The second one, $(1+\Gamma_3) (1-d_{min}) \delta^2$ is capturing the effect of sporadic SGDs in our DSpodFL framework. We observe that if all clients conduct SGDs at every iteration, i.e., $d_i^{(k)} = 1$ for all $i \in \mathcal{M}$ and $k \geq 0$, this term becomes equal to zero, giving us the bound for non-sporadic methods outlined in [4]. Therefore, as we have claimed in Sec. 4.5 of our paper, our convergence bounds recover well-known results in the literature in the degenerate case of $d_{min} = 1$.
>
> We included these discussions on the comparison of DSpodFL's convergence rate with the ones provided in related works, in Appendix P.4 of our revised manuscript.
>
> ### **Follow-up to [Weakness 5]**:
> It is true that [7] did not use a step size of $0.01$, and we would like to emphasize that we did not claim this in our initial response. It is also true that the authors in [7] use a step size of $0.1$ (and $0.8$ in some cases), but they are utilizing the same step size for all baselines that they are comparing their method against. There are a total of five algorithms and their variants experimented with in [7], i.e., SWIFT, D-SGD, LD-SGD, PA-SGD and AD-PSGD, and in each particular experiment the same step size of $0.1$ (or $0.8$) is employed for all of these algorithms. Please see Table 8 in [7].
>
> Similarly, in [6], there are two sets of experiments, one set run with a step size of $0.05$ and the other one with $0.005$. However, the value of the step size has been chosen to be the same across all the algorithms for a given set of experiments. There are a total of six algorithms experimented with in [6], which are DSGT, GSGT, CSG, DSG, EXTRA and DLM, and as seen in Fig. 1 of [6], the value of step size is chosen to be the same for all of these algorithms in a given experiment.
>
> Our goal in our initial response referring to [4, 6, 7] was to point out that it is a common practice in the literature to employ the same learning rate for all algorithms being compared against each other in a given experiment.
>
> For the example provided by the reviewer, i.e., a method with learning rate $\gamma$ compared against another one with implicit effective learning rate of $0.01 \gamma$, we agree that a choice of learning rate will clearly affect the performance of these two approaches. However, as we have argued in Sec 3.2 and Fig. 1 of our paper, our DSpodFL framework captures the baselines that we are comparing our method against, i.e., DGD, RG, Sporadic SGDs and DFedAvg. Therefore, the update rule for all of these algorithms is similar and follows Eq. (2) of our paper, making the use of a common learning rate for all of them a fair comparison. Furthermore, employing the same learning rate for all of these algorithms allows us to understand the effects of introducing sporadicity to DFL methods much better, as we can isolate its effect by keeping all other parameters fixed across the experiments.
>
> Alg. 1 in Appendix B also illustrates how different baselines that we are comparing our methodology against fall under the generalized DSpodFL framework, with particular choices of
> $$\\{ v_i^{(k)} \\}_{i \in \mathcal{M}, 0 \le k \le K}$$
> and
>
> $$\\{ \hat{v}\_{ij}^{(k)} \\}_{(i,j) \in \mathcal{E}^{(k)}, 0 \le k \le K}.$$
>
> Recall that $v_i^{(k)} = 1$ for DGD and RG, $\hat{v}\_{ij}^{(k)} = 1$ for DGD and Sporadic SGDs, $v_i^{(k)} = 1$ with probability $d_i^{(k)}$ for DSpodFL and Sporadic SGDs, and $\hat{v}\_{ij}^{(k)} = 1$ with probability $b_{ij}^{(k)}$ for DSpodFL and RG. For DFedAvg, we have that $v_i^{(k)} = 1$ if $k \mod D \neq 0$ and $\hat{v}\_{ij}^{(k)} = 1$ if $k \mod D = 0$, where $D$ is the number of local SGD updates. Therefore, it is meaningful to use the same learning rate for all of these methodologies.
>
> Finally, in scenarios where hyperparameter tuning is challenging as in FL settings (which require significant communication burden to try different hyperparameter variations), achieving better performance with the same parameters is a significant advantage.
>
> We are grateful for your questions and feedback, and would be glad to answer any further concerns you might have.

---

> ### Author Response · Authors · 2024-11-22
>
> ### **New experiments for [Weakness 5]**
> In addition to our response above, we conducted new experiments during the rebuttal period, by varying the learning rate in the range $\alpha \in \\{ 0.0001, 0.0005, 0.001, 0.005, 0.01, 0.05, 0.1 \\}$ for each of the algorithms as suggested by the reviewer. We ran this experiment using both the FMNIST and CIFAR10 datasets, under both IID and non-IID data distributions among clients. For each choice of learning rate, we consider the testing accuracy achieved by our DSpodFL algorithm and the baselines upon reaching a total delay of $\tau = 500$ for FMNIST and $\tau=5000$ for CIFAR10. Below, we report performance achieved by each algorithm on their best choice of step size:
>
> FMNIST IID
> Algorithm | Best achieved accuracy
> - | -
> DGD | 0.24
> DFedAvg | 0.39
> RG | 0.47
> Sporadic SGDs | 0.22
> DSpodFL | **0.64**
>
> FMNIST Non-IID
> Algorithm | Best achieved accuracy
> - | -
> DGD | 0.17
> DFedAvg | 0.14
> RG | 0.28
> Sporadic SGDs | 0.16
> DSpodFL | **0.36**
>
> CIFAR10 IID
> Algorithm | Best achieved accuracy
> - | -
> DGD | 0.14
> DFedAvg | 0.21
> RG | 0.43
> Sporadic SGDs | 0.16
> DSpodFL | **0.61**
>
> CIFAR10 Non-IID
> Algorithm | Best achieved accuracy
> - | -
> DGD | 0.18
> DFedAvg | 0.24
> RG | 0.41
> Sporadic SGDs | 0.15
> DSpodFL | **0.52**
>
> We observe that the best performance of DSpodFL is still higher than all other baselines, even when we tune the step size separately for each algorithm. The complete results are reported in the following tables, which we also added as plots to Appendix O.9 of our revised manuscript:
>
> FMNIST IID:
> Algorithm | $\alpha = 0.0001$ | $\alpha = 0.0005$ | $\alpha = 0.001$ | $\alpha = 0.005$ | $\alpha = 0.01$ | $\alpha = 0.05$ | $\alpha = 0.1$
> - | - | - | - | - | - | - | -
> DGD | 0.07 | 0.07 | 0.09 | 0.15 | 0.21 | 0.24 | 0.24
> DFedAvg | 0.08 | 0.1 | 0.16 | 0.36 | 0.37 | 0.39 | 0.39
> RG | 0.08 | 0.16 | 0.24 | 0.42 | 0.47 | 0.47 | 0.47
> Sporadic SGDs | 0.07 | 0.07 | 0.08 | 0.14 | 0.15 | 0.21 | 0.22
> DSpodFL | 0.09 | 0.25 | 0.37 | **0.58** | **0.64** | **0.64** | **0.64**
>
> FMNIST Non-IID:
> Algorithm | $\alpha = 0.0001$ | $\alpha = 0.0005$ | $\alpha = 0.001$ | $\alpha = 0.005$ | $\alpha = 0.01$ | $\alpha = 0.05$ | $\alpha = 0.1$
> - | - | - | - | - | - | - | -
> DGD | 0.08 | 0.11 | 0.11 | 0.16 | 0.16 | 0.17 | 0.17
> DFedAvg | 0.08 | 0.13 | 0.13 | 0.13 | 0.13 | 0.14 | 0.14
> RG | 0.09 | 0.14 | 0.15 | 0.16 | 0.23 | 0.28 | 0.28
> Sporadic SGDs | 0.08 | 0.11 | 0.15 | 0.15 | 0.15 | 0.17 | 0.16
> DSpodFL | 0.09 | 0.18 | 0.22 | **0.32** | **0.33** | **0.36** | **0.36**
>
> CIFAR10 IID:
> Algorithm | $\alpha = 0.0001$ | $\alpha = 0.0005$ | $\alpha = 0.001$ | $\alpha = 0.005$ | $\alpha = 0.01$ | $\alpha = 0.05$ | $\alpha = 0.1$
> - | - | - | - | - | - | - | -
> DGD | 0.03 | 0.08 | 0.1 | 0.13 | 0.14 | 0.1 | 0.1
> DFedAvg | 0.04 | 0.17 | 0.19 | 0.20 | 0.21 | 0.1 | 0.1
> RG | 0.13 | 0.25 | 0.24 | 0.39 | 0.43 | 0.1 | 0.11
> Sporadic SGDs | 0.09 | 0.08 | 0.08 | 0.14 | 0.16 | 0.09 | 0.09
> DSpodFL | 0.18 | 0.36 | **0.5** | **0.57** | **0.61** | 0.09 | 0.11
>
> CIFAR10 Non-IID:
> Algorithm | $\alpha = 0.0001$ | $\alpha = 0.0005$ | $\alpha = 0.001$ | $\alpha = 0.005$ | $\alpha = 0.01$ | $\alpha = 0.05$ | $\alpha = 0.1$
> - | - | - | - | - | - | - | -
> DGD | 0.04 | 0.08 | 0.14 | 0.16 | 0.18 | 0.1 | 0.1
> DFedAvg | 0.05 | 0.18 | 0.23 | 0.24 | 0.24 | 0.1 | 0.1
> RG | 0.13 | 0.23 | 0.36 | 0.38 | 0.41 | 0.1 | 0.11
> Sporadic SGDs | 0.04 | 0.09 | 0.15 | 0.15 | 0.15 | 0.08 | 0.09
> DSpodFL | 0.17 | 0.33 | **0.44** | **0.45** | **0.52** | 0.1 | 0.11
>
> For FMNIST, we can observe that even the best performance of the baselines with optimized learning rates is lower than our DSpodFL algorithm with learning rates  $\alpha = 0.005, 0.01, 0.05, 0.1$. In the non-IID case, DSpodFL’s performance even at $\alpha=0.001$ (suboptimal) is better than the best performance of the baselines at $\alpha=0.1$ (optimized) with the exception of RG. For CIFAR10, we make a similar observation where the best performance of the baselines is lower than DSpodFL with learning rates $\alpha=0.001, 0.005, 0.01$. Also, DSpodFL's performance even at $\alpha=0.0005$ (suboptimal) is better than the best performance of the baselines at $\alpha=0.01$ (optimized) with the exception of RG, for both IID and non-IID cases.
>
> Again, we appreciate your time and effort. If you have any remaining concerns, we would be grateful if you could share those with us so that we can address them.

---

> ### Author Response · Authors · 2024-11-27
>
> ### **New experiments for [Weakness 6]**
> Over the past few days, we worked on implementing realizations of the AGRAF SGD algorithm proposed in Even et al., 2024 [8]. Note this is based on our own code as we are unaware of any implementation of [8]. Our results indicate that DSpodFL obtains performance improvements over AGRAF SGD. Below, we will first describe our implementation of [8], and then present the results.
>
> To implement asynchronous SGDs, we use a similar notion of sporadic SGDs from DSpodFL with $\mathbb{E}[v_i^{(k)}]=d_i^{(k)}$. To allow delayed gradients for each client, we start the computation of new gradients $\nabla{F}\_i(\theta_i^{(k)})$ at each iteration, if $v_i^{(k)}=1$. Then, we only utilize those gradients to do gradient descent at a later iteration $k’>k$ if we again have $v_i^{(k’)}=1$. Setting $k=k’-L_i^{(k’)}$, where $L_i^{(k’)}$ is the number of iterations at $k’$ that have passed since client $i$’s last gradient computation, and doing a change of variables, we can write the update rule for AGRAF SGD with respect to Eq. (2) of our paper as
> $$
> \theta_i^{(k)} = \theta_i^{(k)} + \sum_{j \in \mathcal{M}}{r_{ij} (\theta_j^{(k)} - \theta_i^{(k)}) \hat{v}\_{ij}^{(k)}} - \alpha^{(k)} g_i^{(k-L_i^{(k)})} v_i^{(k)},
> $$
> where $v_i^{(k-L_i^{(k)})} = 1$ but $v_i^{(k-\ell)} = 0$ for all $1 \le \ell \le L_i^{(k)}-1$. As such, whenever a client decides to participate in the training process, i.e., $v_i^{(k)} = v_i^{(k-L_i^{(k)})} = 1$, it indicates that it has finished computing one set of gradients and has freed up enough computation resources to start a new gradient calculation again. At this point, it can use the delayed gradients to conduct an iteration of gradient descent and start calculating new gradients, and repeat this cycle. This implies that no processing delays will be incurred.
>
> For the $\hat{v}\_{ij}^{(k)}$ indicator variables, we have two potential choices for AGRAF SGD. One choice is $\hat{v}\_{ij}^{(k)}=1$, which seems to be aligned with Algorithm 1 in [8] since it is stated that communications are done in parallel while new gradients are being computed. However, we found that this leads to poor performance of AGRAF SGD compared to DSpodFL, since it ends up over utilizing communication resources and only optimizing processing time. To see this, below, we report the test accuracy results of this approach vs. our DSpodFL algorithm for two different values of overall delay $\tau$ for the FMNIST dataset using the SVM model:
>
> FMNIST Non-IID with $\mathbb{E}[\hat{v}\_{ij}^{(k)}] = 1$
> Algorithm | $\tau = 1000$ | $\tau = 10000$
> - | - | -
> AGRAF SGD | 0.08 | 0.15
> DSpodFL | **0.40** | **0.55**
>
> The second choice is $\mathbb{E}[\hat{v}\_{ij}^{(k)}] = b_{ij}^{(k)}$, i.e., adding a probabilistic approach to communications into Algorithm 1 of [8]. This allows us to isolate the effect of asynchrony. We present results below in this case for the CIFAR10 dataset using a VGG11 model. We observe that, despite the processing time savings of AGRAF SGDs due to delayed gradients, it still has lower performance than DSpodFL.
>
> CIFAR10 Non-IID with $\mathbb{E}[\hat{v}\_{ij}^{(k)}] = b_{ij}^{(k)}$
> Algorithm | $\tau = 1000$ | $\tau = 10000$
> - | - | -
> AGRAF SGD | 0.17 | 0.45
> DSpodFL | **0.23** | **0.52**
>
> We would like to make a disclaimer that **we do not claim we have the best possible implementation of AGRAF SGD**. Algorithm 1 in [8] is written in a general form, which required us to make the assumptions above in translating it to code. We also checked other papers which have cited [8] to see if any implementation of this algorithm exists, but there does not seem to be any online implementation of it available currently. A few subsequent works by the same authors [12,13] (focused on privacy-preserving learning) also do not seem to have compared their new methodologies against [8] numerically, instead focusing on theoretical aspects. Nonetheless, we have made our best effort to compare the resource efficiency of this algorithm with DSpodFL as requested by the reviewer, using a realization that fits within DSpodFL’s framework which employs indicator variables $v_i^{(k)}$ and $\hat{v}_{ij}^{(k)}$. Given the points above, though, we have hesitated to add these AGRAF SGD results to the paper. We are happy to do so if the reviewer requests it.
>
> We again thank you for providing very helpful comments. We would be happy to answer any remaining concerns you might have.
>
> [12] Biswas, Sayan, Mathieu Even, Laurent Massoulié, Anne-Marie Kermarrec, Rafael Pereira Pires, Rishi Sharma, and Martijn de Vos. "Noiseless privacy-preserving decentralized learning." In The 25th Privacy Enhancing Tech. Symp., vol. 2025, no. 1, pp. 824-844. Privacy Enhancing Tech. Symp. Advisory Board, 2024.
>
> [13] Biswas, Sayan, Mathieu Even, Anne-Marie Kermarrec, Laurent Massoulie, Rafael Pires, Rishi Sharma, and Martijn de Vos. "Beyond Noise: Privacy-Preserving Decentralized Learning with Virtual Nodes." arXiv preprint arXiv:2404.09536 (2024).

---

> > ### Author Response · Authors · 2024-12-01
> >
> > Dear Reviewer ryLC,
> >
> > Thank you again for your time spent reviewing our paper. We just wanted to follow up and see whether you had a chance to review our responses above, and if you have any further questions. We would be happy to address any remaining concerns you have prior to the end of the rebuttal period on Dec. 3.
> >
> > Sincerely,
> > Authors

---

### Official Review · Reviewer_PFSw · 2024-11-05

**Soundness:** 3
**Presentation:** 3
**Contribution:** 3
**Rating:** 8
**Confidence:** 3

**Summary:**

This work proposes a generalized framework for analyzing the challenges of federated learning in a decentralized setting. Such challenges include heterogeneity in the updates of individual clients, and communication constraints for achieving consensus in a decentralized setup (which is more challenging that the typical "star" topology with a central PS). The term "sporadicity" in the title refers to varying frequencies of both -- local updates at clients (limited by computational capacity), and inter-client communication (limited by network capacity). The work proposes a unifying framework for analyzing the effect of this sporadicity on the convergence rate.

**Strengths:**

The algorithm framework proposed in eq (2) that considers the effect of both kinds of sporadicity as mentioned above is one of the primary contributions of the paper. This is further formulated as a matrix update in eq. (3), and the consensus weights are chosen the mixing matrix is doubly stochastic.

The unifying approach to analyze computation and communication sporadicity is definitely an important step towards analyzing decentralized algorithm. The work has been placed well (and compared with) in the context of existing literature. The convergence analysis is quite comprehensive, and aligns with the standard analysis techniques of works in this line. The analytical contribution of the paper is definitely commendable.

**Weaknesses:**

I have some important concerns which I list here in the "Weaknesses" section. Other minor concerns or questions are mentioned in the next section. I would appreciate it if the authors could clarify the following concerns:

1. My major concern is that it is not really clear where the gain of DSpodFL is coming from? The work shows improvements in numerical simulations, in which the X-axis in the total latency. Is the primary intuition behind the improved performance of DSpodFL just the fact that frequent communications are not necessary, which other decentralized algorithms do? Is the benefit of DSpodFL over other algorithms quantified/discussed anywhere in the paper?

2. Are the weights $r_{ij}$ need to be known beforehand? For example, in Metropolis-Hastings mixing, the future topology of the graph needs to be known -- an assumption that may not hold true for federated learning when the network is dynamically changing (for mobile clients, for example), and it is not easy to know the union of all edge sets and determine the weight. How are they chosen in the simulations?

**Questions:**

I have a few concerns, and I would be very glad if these are addressed:

1. Will taking into account the correlation between inter-client communication improve the guarantee, instead of each client independently making a decision for itself. In general, sporadic communications are often correlated.

2. Referring to Def. 4.6, why are the indicator variables not defined as Bernoulli random variables? It seems like they are modeled as Bernoulli variables, but it is not explicitly mentioned anywhere.

3. In the aggregation step, the clients need to know which nodes communicate to it? In other words, is a blind aggregation where the identity of the participating clients is anonymized, possible?

Also, some recent works in communication sporadicity is missing: For example, in this work: https://ieeexplore.ieee.org/document/10705313, communication sporadicity is considered in conjunction with privacy constraints, although in a semi-decentralized setting. Privacy is an important aspect of federated learning, and so the paper will benefit with some discussions on the privacy implications of the current algorithm.

**Details Of Ethics Concerns:**

None needed.

---

> ### Author Response · Authors · 2024-11-16
>
> We would first like to thank you for acknowledging our analytical contribution towards the unifying framework of joint computation and communication sporadicity in decentralized federated learning. Please find our responses to your other comments below, including how we have addressed them in the revised manuscript.
>
> ### **[Weakness 1] Performance superiority of DSpodFL**
> To be precise, both overly frequent or infrequent computation/communication can result in suboptimal performance from an accuracy-latency tradeoff standpoint, depending on each client’s resource availability. The performance improvements in DSpodFL come from allowing heterogeneous and time-varying decentralized aggregation periods and local SGDs, unlike existing works that focus on fixed aggregation intervals and local SGDs. In decentralized FL, resource availability is often heterogeneous and dynamic, as discussed in Sec. 1: there are (i) variations in computation capabilities at clients, causing bottlenecks in consistent participation in SGD computations, and (ii) variations in link bandwidth, causing bottlenecks during model aggregations. DSpodFL overcomes these limitations by enabling sporadicity in both communication and computations, and tying device decisions on these operations to the clients’’  local resource availability. Referring to Algorithm 1 in Appendix B:
>
> * Client $i$ conducts an SGD in iteration $k$ only if $v_i^{(k)}=1$ (line 7), which has a probability $d_i$ proportional to its processing availability. Consider that multiple clients may possess overlapping data distributions. In such scenarios, the system can benefit from more frequent SGD updates from the clients with the highest resource availability, as they provide information representative of multiple clients and finish iterations faster.
>
> * Link $(i,j)$ is used for model sharing in iteration $k$ only if $v_{ij}^{(k)}=1$ (line 12), which has a probability $b_{ij}$ proportional to its bandwidth availability. DSpodFL takes advantage of the fact that model information can propagate through the system rapidly over the fastest links in the graph. For example, if link $(i,j)$ has low bandwidth, our method will make client $i$’s relevant local update information reach client $j$ more rapidly through a series of other high bandwidth links.
>
> Unlike DSpodFL, the baselines evaluated in Sec. 5 work under (a) fixed SGDs and/or (b) fixed aggregations. DGD assumes local SGDs and aggregations at every iteration, RG employs constant SGDs (but sporadic aggregations), Sporadic SGDs assumes constant aggregations, and DFedAvg is DGD with decreased communication frequencies.
> We discussed the above intuitions in Sec. 3.2 of our original manuscript under the paragraph “interpreting sporadicity”, but we had to present a brief version due to space constraints. In the revised manuscript, we have added Appendix Q.1 explaining these remarks to further clarify the motivation behind DSpodFL’s development and why it is more resource-efficient compared to other baselines.

---

> ### Author Response · Authors · 2024-11-16
>
> ### **[Weakness 2] The weights $r_{ij}$**
> Thanks for your insightful question. When DSpodFL is deployed in practice, each client $i$ neither needs to have information about the future of the network graph, nor does it need to know the weights $r_{ij}$ beforehand. Instead, it only needs to know the degree of its one-hop neighbors at each iteration $k$, and when it communicates to a neighbor $j$, it can calculate the Metropolis-Hastings weights as $1 / (1 + \max\{ |\mathcal{N}_i^{(k)}, \mathcal{N}_j^{(k)}| \})$. This way, a doubly-stochastic mixing matrix can be designed at each iteration of training on the fly.
>
> In the theoretical analysis paper, we have differentiated between two types of graphs: (i) the dynamic communication graph $\mathcal{G}^{(k)}$ which is not necessarily connected at each iteration, and (ii) the underlying connected physical graph $\mathcal{G}$, which can be also represented as $\mathcal{G} = \lim_{K \to \infty}{\cup_{k=0}^K{\mathcal{G}^{(k)}}}$. Indeed $\mathcal{G}$ connects the clients together and the weights $r_{ij}$ are assigned over its links. However, at each training iteration of DSpodFL, only a subset of these links are utilized for inter-client communications, represented by $\mathcal{G}^{(k)}$, for which the link weights $r_{ij} \hat{v}_{ij}^{(k)}$ are assigned. In conclusion, during convergence analysis, we have assumed that the graph over time is given to provide theoretical insights.  We have made sure to clarify the distinction between these two types of graphs better in Sec. 3.1 of the revised manuscript.
>
> We would also like to emphasize that it is standard in the decentralized learning literature to assume such connectivity requirements for the underlying graph. Please see [1-4] as examples.
>
> [1] Koloskova, Anastasia, Nicolas Loizou, Sadra Boreiri, Martin Jaggi, and Sebastian Stich. "A unified theory of decentralized sgd with changing topology and local updates." In International Conference on Machine Learning, pp. 5381-5393. PMLR, 2020.
>
> [2] Nedic, Angelia, and Asuman Ozdaglar. "Distributed subgradient methods for multi-agent optimization." IEEE Transactions on Automatic Control 54, no. 1 (2009): 48-61.
>
> [3] Sun, Tao, Dongsheng Li, and Bao Wang. "Decentralized federated averaging." IEEE Transactions on Pattern Analysis and Machine Intelligence 45, no. 4 (2022): 4289-4301.
>
> [4] Xin, Ran, Usman A. Khan, and Soummya Kar. "An improved convergence analysis for decentralized online stochastic non-convex optimization." IEEE Transactions on Signal Processing 69 (2021): 1842-1858.

---

> ### Author Response · Authors · 2024-11-16
>
> ### **[Question 1] Correlation of inter-client communications**
> We agree with the reviewer that in certain cases, carefully designed correlations among inter-client communications might lead to faster convergence. However, this would need full knowledge of all parameters involved in DFL, and is not feasible to solve in a decentralized way as we elaborate further below.
>
> In a fully decentralized setting, the standing assumption is that each client is oblivious of the condition of the network graph beyond its one-hop neighbors, e.g., it does not know the total number of clients. Therefore, if our goal was to introduce correlations to find an optimal way of setting $\hat{v}\_{ij}^{(k)}$ for all $i,j \in \mathcal{M}$ and all $0 \le k \le K$, where $K$ is the total number of iterations we run the algorithm for, this would require an overlay dynamic optimization problem to be solved to choose values for $\hat{v}\_{ij}^{(k)}$. Note that the solution to this allocation would need to incorporate (i) client data distributions, (ii) their processing resources, (iii) the structure of the underlying graph network and (iv) the bandwidths of graph links. Furthermore, since each client performs stochastic gradient descent rather than full-batch gradient descent, any chosen solution for $\hat{v}\_{ij}^{(k)}$ would cease to be the optimal choice after several iterations of stochastic training. Therefore, the overlay optimization problem to choose values for $\hat{v}_{ij}^{(k)}$ would have to be solved again iteratively after every/every few iterations. Note that this is in contrast to conventional federated learning, where a centralized server exists in the system to coordinate the clients and make more intelligent decisions as it has full/almost full knowledge of the network.
>
> We can again see that such an auxiliary optimization problem would be difficult to solve in a decentralized setup. Thus, we have opted to use a more elegant choice of uncorrelated decisions for inter-client communications, which is a practical solution for real-world scenarios by making localized decisions at each client. In other words, in DSpodFL, each communication link is activated, i.e., $\hat{v}\_{ij}^{(k)} = 1$, by either one of nodes $i$ or $j$ only, and it does not affect any other $\hat{v}_{i’j’}$ for all $i’,j’ \in \mathcal{M}$. Using this method, DSpodFL immediately starts to solve the main downstream optimization problem, which is the image classification task in our paper. Moreover, note that even if we were to make the communications correlated and try to come up with a better strategy that uncorrelated decisions, it would incur a high amount of communication overhead for the training process, which goes against the main motivation of DSpodFL which is to speed up the training.
>
> ### **[Question 2] Bernoulli random variables**
> We again would like to thank you for closely reading our paper, and we appreciate the meticulous questions and comments that you have raised. The indicator variables $v_i^{(k)}$ and $\hat{v}\_{ij}^{(k)}$ are indeed Bernoulli random variables for each $k$. We refrained from explicitly mentioning it because the probability parameter of these indicator variables can change over time, i.e., $\mathbb{E}{[v_i^{(k)}]} = d_i^{(k)}$ and $\mathbb{E}{[\hat{v}\_{ij}^{(k)}]} = b_{ij}^{(k)}$. In the revised version of our manuscript, we have made a footnote in Definition 4.6 given in Sec. 4.1 about the fact that these random variables can be thought of as Bernoulli with dynamic parameters.
>
> ### **[Question 3] Blind aggregation**
> The way we have developed DSpodFL, each client $i$ does not need to know the identity of the neighbor $j$ it is communicating with, i.e., the communications can be anonymous. However, at each iteration of training $k$, it still needs to do some kind of handshake with that neighbor so that they can exchange their model parameters $w_j^{(k)}$ and the degrees $|\mathcal{N}\_j|$. Please see [2] (referenced in our response to [Weakness 2] The weights $r_{ij}$) for a discussion on simultaneous information exchange for detailed discussion on this.
>
> However, we note that even when a client does not know its neighbors during transmission, it can still broadcast its model through the wireless channel so that their neighbors can detect the model. On the other side, its neighbors can constantly keep sensing their wireless channels for possible reception of models from the network. See [5] for an example of a DFL algorithm which does this.
>
> [5] George, Jemin, and Prudhvi Gurram. "Distributed stochastic gradient descent with event-triggered communication." In Proceedings of the AAAI Conference on Artificial Intelligence, vol. 34, no. 05, pp. 7169-7178. 2020.

---

> ### Author Response · Authors · 2024-11-16
>
> ### **[Question 4] Privacy constraints**
> Thank you for referring us to the work on privacy-preserving semi-decentralized optimization [6]. Indeed, sporadic aggregations is a very effective method, and an increasing number of papers are using it for efficient algorithms in decentralized optimization literature. In our current manuscript, DSpodFL only enjoys the privacy benefits of regular FL algorithms, where the datasets of clients are not shared among each other, and instead they communicate their model parameters. An interesting future direction to our work would be to incorporate more advanced privacy-preserving techniques to the sporadic framework of DSpodFL, like differential privacy, similar to what some recent papers in the DFL community have done [7]. In the revised version of our paper, we have cited this paper among the works that do sporadic aggregations in Sec. 2, and mentioned the importance of privacy preservation in the field.
>
> [6] R. Saha, M. Seif, M. Yemini, A. J. Goldsmith and H. V. Poor, "Privacy Preserving Semi-Decentralized Mean Estimation over Intermittently-Connected Networks," in IEEE Transactions on Signal Processing, doi: 10.1109/TSP.2024.3473939.
>
> [7] Chen, Shuzhen, Dongxiao Yu, Yifei Zou, Jiguo Yu, and Xiuzhen Cheng. "Decentralized wireless federated learning with differential privacy." IEEE Transactions on Industrial Informatics 18, no. 9 (2022): 6273-6282.
>
> Again, thank you for your time and efforts in reviewing our paper. We would appreciate further opportunities to answer any remaining concerns you might have.

---

> > ### Author Response · Authors · 2024-11-29
> >
> > Dear Reviewer PFSw,
> >
> > We appreciate your time and effort reviewing our paper. We just wanted to reach out again and ask if you have had a chance to review our response? We have carefully considered your comments and revised our manuscript accordingly, especially clarifying (i) the gain of DSpodFL compared to existing DFL algorithms and (ii) the weights $r_{ij}$ in our paper. We are happy to answer any remaining concerns that you might have during the remainder of the Author-Reviewer discussion period, which ends in 4 days.
> >
> > Best,
> >
> > Authors of Paper 5391

---

> > > ### Comment · Reviewer_PFSw · 2024-11-30
> > > **Ack**
> > >
> > > Thank you for your rebuttal. I appreciate that the authors put in the effort to make the contributions of the paper more apparent, and for the added discussions. I have revised my score to 8.

---

> > > > ### Author Response · Authors · 2024-11-30
> > > >
> > > > Thank you very much for your consideration.

---

### Meta-Review · Area_Chair_3nWc · 2024-12-20

**Metareview:**

This paper studies the decentralized learning setting, where clients are connected via a network topology and perform gradient updates based on their local data, followed by averaging with their neighboring nodes in the network. A key contribution of this work is the proposal of a general framework, DSpodFL, for analyzing optimization methods with sporadicity in both local gradient computations and aggregation processes.

A major strength of this framework lies in its flexibility, as it employs indicator random variables to model stochasticity and allows for milder assumptions in both convex and non-convex cases. This represents a meaningful advancement in the literature on decentralized optimization, and the paper is likely to be of interest to the ICLR community.

However, a potential weakness, which stems from the generality of the framework, is that it remains unclear in this version of the paper whether the convergence results obtained using the DSpodFL framework can match the prior analyses of earlier work for specific settings. The discussion in the newly added Section P.2 of the appendix remains somewhat vague in this regard, as the big-O notation obscures not only constants but also problem-specific parameters. I hope this can be further clarified in the final version.

**Additional Comments On Reviewer Discussion:**

The discussion between the authors and the reviewers addressed many technical questions but also highlighted concerns about the methodology raised by reviewer ryLC. While some of reviewer ryLC's concerns regarding the class of algorithms covered by the assumptions in DSpodFL remain unresolved, the positive votes from the other reviewers have led me to recommend acceptance of this work.

---

### Decision · Program_Chairs · 2025-01-22

Accept (Spotlight)